# Clustering by measuring local direction centrality for data with heterogeneous density and weak connectivity

**Dehua Peng** [1,2,3,4], **Zhipeng Gui** [2,3,4] ✉, **Dehe Wang** [5,6], **Yuncheng Ma**[2,3], **Zichen Huang**[2,3], **Yu Zhou** [5,6] **& Huayi Wu**[1,3,4]

Clustering is a powerful machine learning method for discovering similar patterns according to the proximity of elements in feature space. It is widely used in computer science, bioscience, geoscience, and economics. Although the state-of-the-art partition-based and connectivity-based clustering methods have been developed, weak connectivity and heterogeneous density in data impede their effectiveness. In this work, we propose a boundary-seeking Clustering algorithm using the local Direction Centrality (CDC). It adopts a density-independent metric based on the distribution of K-nearest neighbors (KNNs) to distinguish between internal and boundary points. The boundary points generate enclosed cages to bind the connections of internal points, thereby preventing cross-cluster connections and separating weakly-connected clusters. We demonstrate the validity of CDC by detecting complex structured clusters in challenging synthetic datasets, identifying cell types from single-cell RNA sequencing (scRNA-seq) and mass cytometry (CyTOF) data, recognizing speakers on voice corpuses, and testifying on various types of real-world benchmarks.

Heterogeneous density and weak connectivity in point distributions are challenging in cluster analysis. As a powerful machine learning method, clustering explores similar patterns lurking in data[1]. It aims to find an optimized partition to group independent points to clusters by maximizing the intra-cluster similarity and the inter-cluster difference. Identification of arbitrary shapes, adaptability to the high dimensionality and elimination of noisy instances are universal problems that have been studied extensively in cluster analysis. However, the heterogeneous density and weak connectivity also affect the clustering quality significantly, and should receive more attention. Heterogeneous density means that a cluster with uneven density tends to be separated into parts and the sparse clusters are easy to be misidentified as noise, while weak connectivity causes nearby clusters difficult to separate. Although numerous clustering techniques based

on diverse principles have been developed[2], it is still insufficient to tackle abovementioned challenges effectively using the proximity of physical distance or density alone.

Partition-based and connectivity-based clustering are two commonly used methods to associate independent points. Partition-based clustering finds cluster centers and assigns the points to their nearest cluster centers using distance measurements. Conventional algorithms, K-means[3] and K-medoids[4], determine the optimal cluster centers by constantly modifying the centroid of each cluster. However, these algorithms cannot identify non-ellipsoidal clusters and have a weak robustness to the noise. Clustering by finding Density Peaks (CDP)[5] improves the search strategy of cluster centers based on the idea that cluster centers are characterized by high density locally and large distance from the points with higher densities. CDP enables the

[1]State Key Laboratory of Information Engineering in Surveying, Mapping and Remote Sensing, Wuhan University, Wuhan, China. [2]School of Remote Sensing and Information Engineering, Wuhan University, Wuhan, China. [3]Collaborative Innovation Center of Geospatial Technology, Wuhan University, Wuhan, China. [4]Hubei Luojia Laboratory, Wuhan, China. [5]State Key Laboratory of Virology, Modern Virology Research Center, College of Life Sciences, Wuhan University, Wuhan, China. [6]Frontier Science Center for Immunology and Metabolism, Wuhan University, Wuhan, China. ✉e-mail: zhipeng.gui@whu.edu.cn

identification of arbitrarily shaped clusters and noise points, but the association rule may cause incorrect assignments of boundary points when clusters with large differences in size are closely distributed[6].

Connectivity-based clustering accurately identifies arbitrary shapes by continuously aggregating high-density points with a growth approach. Typically, Density-based Spatial Clustering of Applications with Noise (DBSCAN)[7] clusters circular neighborhoods of connected points whose densities are greater than the threshold. Although this condition preserves the local details of a cluster shape, it easily mis-identifies sparse clusters as noise and even splits an entire cluster when the points are unevenly distributed. WaveCluster[8] and CLIQUE[9] map the original points to a grid and merge the connected grid cells to generate clusters. Connecting points with a grid representation pro-motes time efficiency, but weakly-connected clusters cannot be separated and low-density cluster boundaries tend to be detected as noise points[2]. As a boundary-seeking approach, local gravitation clus-tering (LGC)[10] proposes two mean-shift-based metrics, centrality (CE) and coordination (CO), to measure the consistency between the local attractive forces and mean-shift directions of neighbors. Accordingly, it is capable to distinguish internal and boundary points of clusters, and forms clusters by connecting boundary and unlabeled points from internal points with a damping connecting capability. However, internal points in sparse clusters are difficult to detect since the mean shifts tend to move towards the dense regions. Density-based metrics such as Reverse K-Nearest Neighbors (RKNN)[11] have been also utilized to detect the boundary points of cluster. It queries the number of objects that consider a given point as the membership of their KNNs, but might fail to seek the boundaries with low-density densities.

In this work, we propose a clustering algorithm named CDC by measuring direction centrality locally, which contributes to handling data with heterogeneous density and weak connectivity. The core idea is to detect the boundary points of clusters firstly, and then connect the internal points within the enclosed cages generated by surround-ing boundary points. Specifically, an internal point of clusters tends to be surrounded by its KNNs in all directions, while a boundary point only includes neighboring points within a certain directional range. Taking advantage of this difference, we measure the local centrality by calculating the directional uniformity of KNNs to distinguish internal and boundary points. Hence, CDC can avoid the cross-cluster con-nections and separate weakly-connected clusters effectively. Mean-while, it can preserve the completeness of sparse clusters, since it utilizes KNN to search the neighboring points that is irrelevant to the point density. To validate the effectiveness, we compared CDC with totally 38 specialized and versatile baselines on 47 datasets derived from different fields, including 15 scRNA-seq, two CyTOF, two speaker corpuses, eight UCI, one handwritten image, one face image and 17 synthetic datasets. Results demonstrated that CDC attains superior clustering accuracy and robust outcomes in a time efficient manner, and presented its great potentials in various applications. Moreover, we investigated the dimension expansion and noise elimination methods, analyzed the parameter sensitivity, and designed adaptive methods for parameter settings.

## Results

### Comparison with clustering baselines on synthetic datasets

In this experiment, we selected three synthetic datasets (DS1-DS3) with different shaped clusters and compared the results with four typical clustering algorithms (i.e., K-means, CDP, DBSCAN, and LGC) as shown in Fig. 1. In general, CDC outperformed the baselines on the three datasets. A total of ten spherical clusters are contained in DS1, with two nearby clusters in the upper right. K-means and CDP effectively iden-tified spherical clusters based on the nearest partitioning principle. However, several boundary points were assigned to the incorrect clusters by K-means, since it selects the centroids as the cluster centers that is affected by the cluster sizes. DBSCAN merged the two weakly-

connected clusters as a whole, unless the between-cluster points are removed as noise. Both LGC and CDC obtained the accurate clustering results in DS1. It demonstrates that the centrality metrics proposed by the two algorithms can efficiently contribute to handling spherical clusters with uniform densities.

Clusters in DS2[12] are more challenging in distributions, which contains two weakly-connected spherical clusters, a non-spherical dense cluster, and a sparse cluster. Meanwhile, a ring cluster surrounds a spherical cluster, appearing as an island distribution. The results illustrate that both of K-means and CDP can separate the two weakly-connected clusters, but cannot accurately detect the island distributed clusters. Due to the density dependence, CDP cannot find the sparse cluster. It is difficult for DBSCAN to balance the tasks of separating nearby clusters and detecting sparse clusters. LGC failed to recognize the complete sparse cluster. Points belonging to the sparse cluster were wrongly assigned to the left dense ring-shaped cluster due to the reliance on the mean-shift directions. The metrics in CDC are irrelevant to density, which depicts the local centrality by measuring the dis-tribution uniformity of neighboring points in the surrounding direc-tions. Both the points in the dense or sparse clusters conform to this law. Thus, the clusters with complex structures in DS2 can be identified accurately by CDC.

DS3[13] contains six dense clusters in the shape of letters sur-rounded by a ring-shaped cluster with a significant density difference. Similar to the results for DS2, the ring-shaped cluster was split to multiple pieces by K-means and CDP due to the distance association rule. Connectivity-based approaches retain the integrity of cluster shapes when handling the clusters with complex structures. Thus, DBSCAN can identify a ring cluster using a coarse-granular radius, but can lead to a misconnection of the "O" and "R" as a whole. As LGC is sensitive to point density, the significant difference of point density between the ring and the letter clusters incurred errors to detect the internal and boundary points, thereby affecting the clustering quality. To further validate the robustness of CDC, we applied it to other six synthetic datasets, i.e., DS4-DS9[14,15] and the performance can be seen in Supplementary Fig. 3.

### Application of cell type identification on scRNA-seq and CyTOF datasets

Cluster analysis on gene-expression profiles of single cells contributes to revealing the types and cell-to-cell heterogeneity[16]. scRNA-seq and CyTOF are revolutionary techniques to detect the expression levels of single cells and has been extensively used in the cell type identification[17,18]. To evaluate the applicability of our algorithm, we applied CDC on multiple scRNA-seq and CyTOF datasets with different number of cells and features by comparing with specialized biological and versatile baselines in both clustering accuracy and time efficiency.

The applicability analysis of CDC was performed on nine pub-lished and annotated scRNA-seq datasets, i.e., Baron-human (BH), Baron-mouse (BM)[19], Muraro[20], Segerstolpe[21], Xin[22], Allen Mouse Brain (AMB)[23], Anterior Lateral Motor (ALM)[23], Primary Visual Cortex (VISp)[23] and Tabula Muris (TM)[24] (see details in Supplementary Table 1). Seven biological baselines, i.e., Seurat v3[25], monocle3[26], SC3[27], dropClust[28], MetaCell[29], Shared-Nearest-Neighbor-Walktrap (SNN-Walktrap)[30], SNN-Louvain[31], and seven versatile clustering baselines, i.e., AGNES[4], DIANA[32], hclust[33], DBCSAN, K-means, C-means[34], CLARA[4], were selec-ted for comparison. The standard preprocessing pipeline of scRNA-seq clustering is presented in Fig. 2a[35], and more details can be seen in Supplementary Note 4. Principal Component Analysis (PCA)[36] was used to select the first 50 principal components from the preprocessed data, and Uniform Manifold Approximation and Projection (UMAP)[37] further reduced the dimensions in the PCA space. The SNN-based and classical clustering methods were conducted both in multi-dimensional PCA (i.e., 2D-50D, denoted as -PCA) and 2D UMAP spaces (denoted as -U2), while CDC was carried out in 2D-5D UMAP

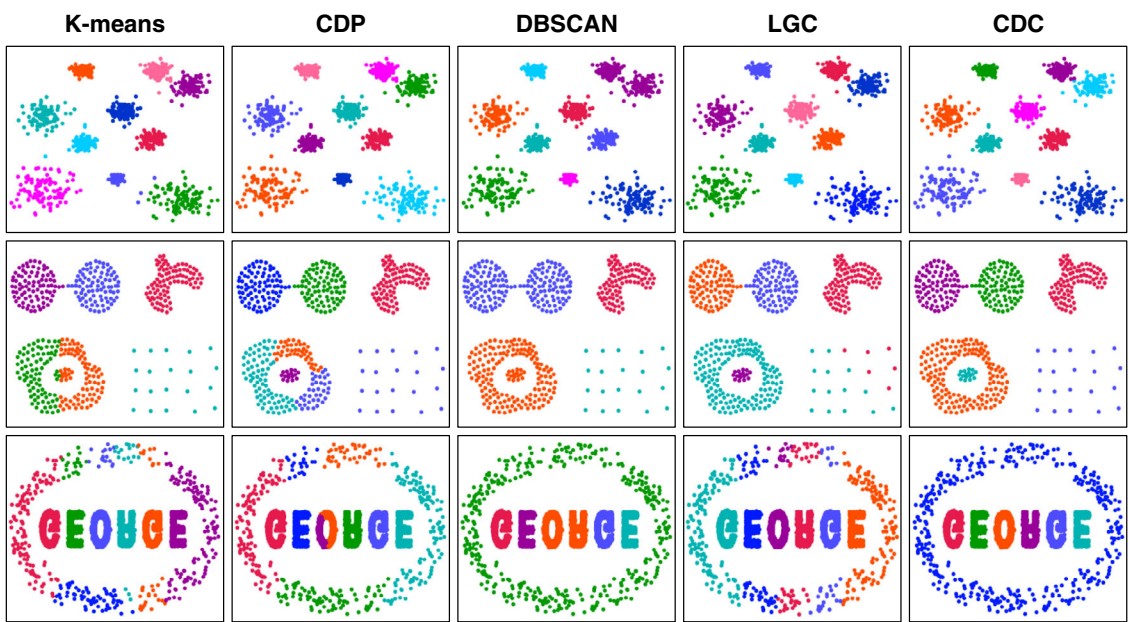

**Fig. 1 | Comparison with four typical clustering algorithms on three synthetic datasets (DS1-DS3).** The three datasets shown in the three rows contain 999, 459 and 7247 points respectively. Each color represents a cluster.

space (denoted as CDC-U2, -U3, -U4 and -U5). To evaluate the clustering accuracy quantitatively, we calculated the Adjusted Rand Index (ARI)[38] between assignment and true cell annotation.

According to the ARIs reported in Fig. 2b and Supplementary Table 2, CDC-U2 outperforms the biological and versatile baselines in both clustering accuracy and stability, and CDC-U3, -U4 and -U5 also achieved high accuracies on more than half of the datasets. CDC-U2 obtained promising max ARI scores (BH: 0.9532, BM: 0.9683, Muraro: 0.9235, Segerstolpe: 0.9734, Xin: 0.9727, AMB: 0.8873, ALM: 0.7186, VISp: 0.8569, TM: 0.8384), where seven of the nine datasets (except Muraro and Xin) are the highest among all algorithms (if does not consider CDC-U3, -U4 and -U5 in Supplementary Table 2) and having distinct advantages on AMB, ALM and VISp. As shown in the Sankey and t-SNE plots (Fig. 2c, d), the predicted results of CDC-U2 are almost perfectly matched with the true cell type annotation. Compared to the other three baselines (SC3, SNN-Louvain-U2, Kmeans-U2), CDC-U2 achieved higher identification accuracy especially on the GABAergic class of ALM (Fig. 2e). Meanwhile, CDC-U2 has the most robust outcomes and obtained the highest average ARIs on eight datasets (except VISp), which are larger than the second places 0.220, 0.264, 0.005, 0.216, 0.122, 0.016, 0.126, 0.099 respectively. In terms of the average rank, CDC-U2 achieved 1.8 on the max ARIs and 1.2 on the average, which are significantly superior to the second rank of 5.3 and 4.2 respectively. Furtherly, most of the baselines performed worse on the last four scRNA-seq datasets (AMB, ALM, VISp and TM), which have a larger data volume and multi-level types of cells (e.g., class and subclass) with complex manifold distributions, i.e., data dimension is lower than the feature dimension. Hence, the heterogeneous densities and weak connectivity may occur after the preprocessing, and makes it is hard to identify all cell types accurately. However, by utilizing local direction centrality, CDC can effectively extract boundary points of clusters in proper embedded dimension (To be noted, 2D UMAP space may not always the most appropriate to represent the data distribution, and users are free to specify appropriate dimension according to their needs), where the boundary points can bind the internal connections in all directions. Therefore, CDC achieved better performance than the baselines on complex structured scRNA-seq datasets. We also verified the robustness of CDC on four mouse retina scRNA-seq datasets in Supplementary Fig. 4.

The validity of CDC-U2 was also evaluated on two CyTOF datasets, i.e., Levine[39] and Samusik[40] (Supplementary Table 1), by comparing with 15 clustering baselines including two classical algorithms (K-means and MeanShift) and 13 dedicated algorithms for the cell population detection. We followed the data preprocessing and parameter settings in[18] (Supplementary Note 5). The ARI scores and runtimes are shown in Fig. 2f. CDC-U2 achieved the highest ARI scores on both of the two datasets (Levine: 0.9628, Samusik: 0.8564). Meanwhile, it can be found that CDC-U2 has promising runtimes (the dimension reduction is included) and is more efficient than eight and ten baselines on the two datasets respectively. In comparison, ACCENSE, DensVM and flowMerge, require subsampling to make them computable due to the excessive time complexity on large data size. Moreover, CDC embraces parallel computing due to the nature of KNN-based calculation. It can be easily extended to parallel versions using GPGPU and distributed computing techniques such as Apache Spark[41], for performance acceleration.

To testify the scalability of CDC, we further adopted a large-scale scRNA-seq dataset collected from the adult Mouse Isocortex and HiPpocampal Formation (MIHPF)[42]. It contains 1,093,785 total cells along with 31,053 genes, which are assigned to three classes (Glutamatergic neurons, GABAergic neurons and Non-neuronal) and can be subdivided into 30, 7 and 6 subclasses respectively. The subclasses have been grouped into 8 neighborhoods, including 5 glutamatergic (DG/SUB/CA, L2/3 IT, L4/5/6 IT Car3, PT and NP/CT/L6b), 2 GABAergic (MGE and CGE), and one "other" neighborhood. We first performed a single-round clustering comparison with K-means and SNN-Louvain in 2D UMAP space, using the preprocessing steps in Supplementary Note 4. In general, CDC obtained higher accuracy than the two competitors at subclass level (Fig. 3d). Due to the heterogeneity in cluster size and density, it is hard to capture multi-level differences of gene expression and identify all cell types in a fine granularity at one time. However, the confusion matrix of the CDC result with a high recall score (Fig. 3a) indicates that the identified clusters are in accord with the proximity reflected in the transcriptomic taxonomy tree and in line with the cell development. 14 subclasses have been detected, where only two small subclasses of them, CR and SMC-Peri (including 268 and 198 cells respectively) have relatively low accuracies. While, the rest subclasses have been assigned to three relatively complete

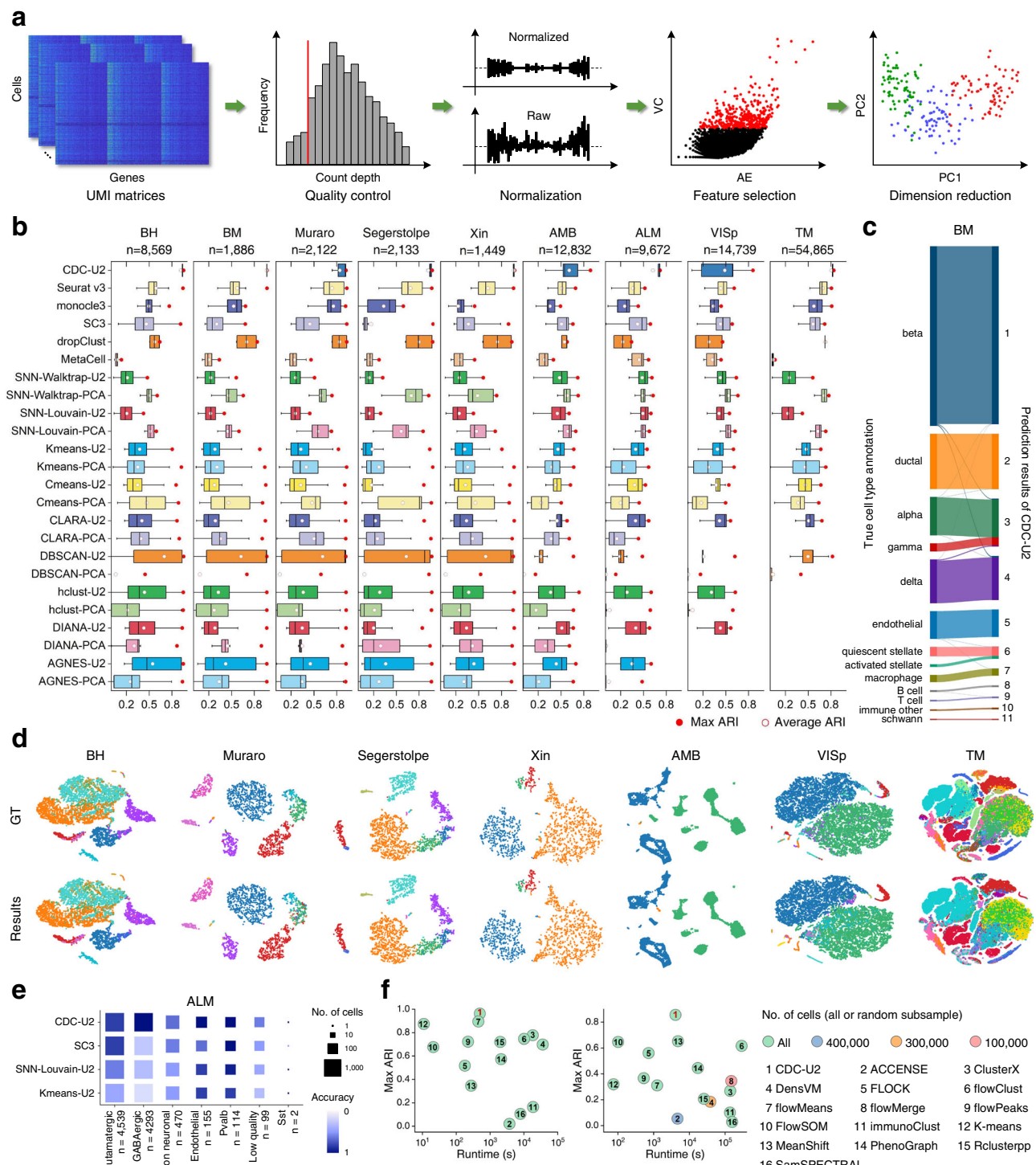

**Fig. 2 | Clustering performances on scRNA-seq and CyTOF datasets. a** Overview of the standard preprocessing pipeline for scRNA-seq clustering[35]. First, the raw Unique Molecular Identified (UMI) matrices are input, and the unreliable cells are removed through the quality control. The feature expression of each cell is then normalized to correct the differences of read counts. Next, the Highly Variable Genes (HVG) are selected according to their Average Expression (AE) and Variable Coefficient (CV). PCA is used to reduce the feature dimensions. **b** Clustering accuracy reported by ARI of 15 clustering algorithms conducted in PCA and UMAP spaces on nine scRNA-seq datasets, where the red points denote the max ARI scores and the white ones refer to the average ARI scores. CLARA-PCA, DIANA-PCA, AGNES-U2, AGNES-PCA are not applicable on VISp and TM. hclust-U2, hclust-PCA

and DIANA-U2 are not applicable on TM. Boxes show the median and the 25–75% range, while whiskers refer to the 1.5 times interquartile range. **c** A Sankey diagram shows the match between the CDC-U2 results and the published cell type annotation on BM. **d** The t-distributed Stochastic Neighbor Embedding (t-SNE)[63] plots present that the best CDC-U2 results are almost identical to the ground truth (GT). **e** The identification accuracy of different cell types of ALM by CDC-U2, SC3, SNN-Louvain-U2, Kmeans-U2. **f** Performances of 16 clustering algorithms on two CyTOF datasets, Levine[39] and Samusik[40] datasets, including CDC-U2, ACCENSE[64], ClusterX[65], DensVM[66], FLOCK[67], flowClust[68], flowMeans[69], flowMerge[70], flowPeaks[71], flowSOM[72], immunoClust[73], K-means, MeanShift[74], PhenoGraph[39], Rclusterpp[75], and SamSPECTRAL[76], where flowMerge is not applicable on Levine.

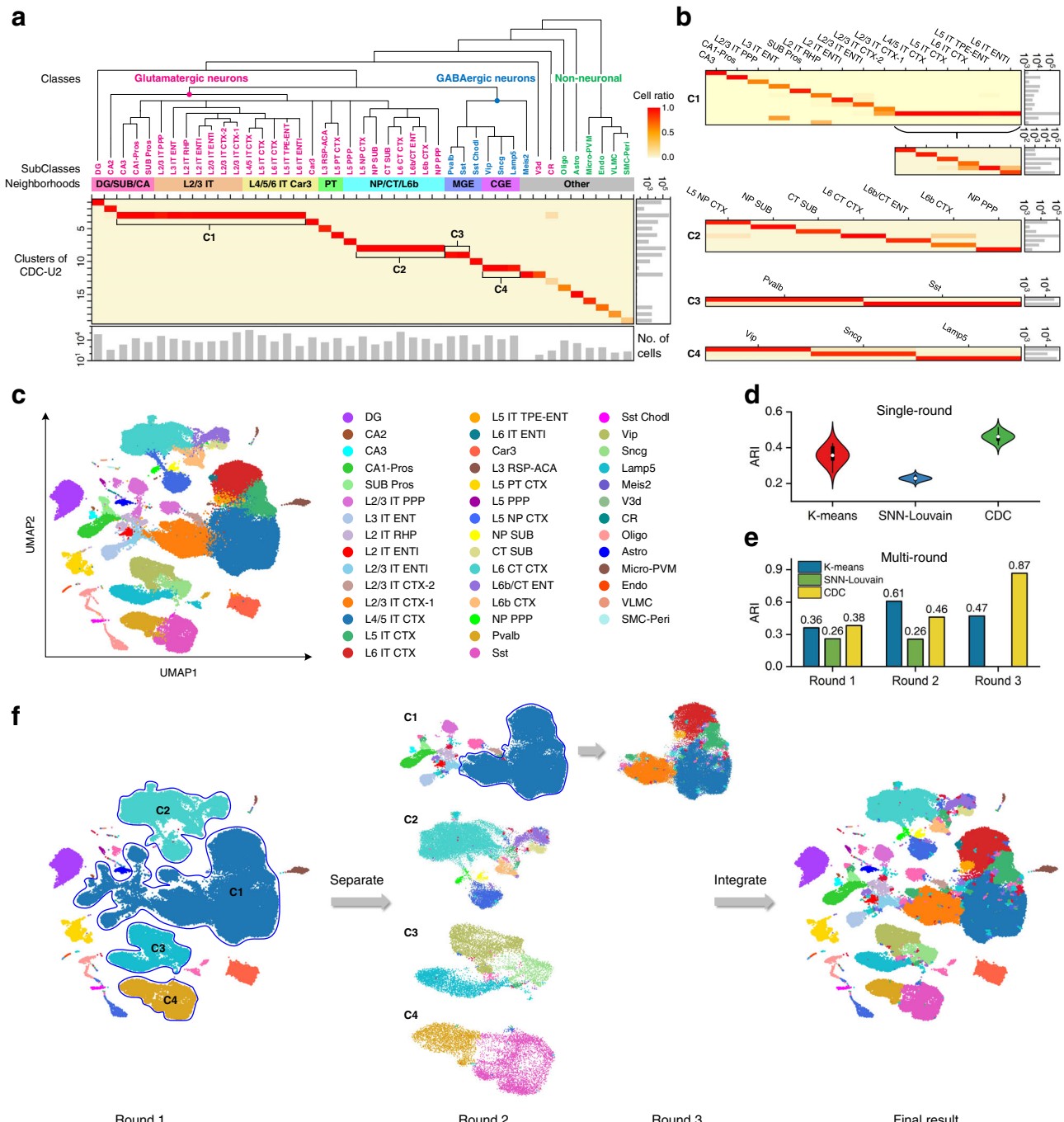

**Fig. 3 | Multi-round cluster analysis on MIHPF dataset composed of more than one million cells. a** Confusion matrix of the first-round CDC result with high recall score ($k = 30$, $ratio = 0.99$), where the transcriptomic taxonomy tree and labels of classes and subclasses are from Yao et al. 2021[42]. **b** Confusion matrices of the CDC results through the second and third rounds of clustering optimization. **c** The published cell type annotation of MIHPF. **d**, **e** Clustering accuracies reported by ARI score of K-means, SNN-Louvain and CDC using single-round clustering and three rounds of clustering optimization respectively, where SNN-Louvain only conduct clustering two rounds because of the iteration criterion. Boxes show the median and the 25–75% range, while whiskers refer to the 1.5 times interquartile range. **f** Clustering results of CDC in 2D UMAP space through three rounds of clustering optimization.

neighborhoods (NP/CT/L6b, MGE and CGE), except Meis2 (only having one cell) and V3d being mixed together. The integrity of clusters identified by CDC are the bases for conducting further clustering to explore fine-grained subclasses.

We adopted a multi-round clustering strategy for further improving the clustering accuracy. To ensure high integrity, clusters with high recall score are selected as the initial result (Supplementary Fig. 5a). The criterion for carrying out the next round clustering is that there are clusters are connected in the confusion matrix[43] or multiple

significant density peaks[5] in the generated clusters (Supplementary Fig. 5b), otherwise clustering terminates (Supplementary Fig. 5c). Through three rounds of clustering optimization, CDC extracted more fine-grained subclasses from the initial clusters C1-C4 (Fig. 3b), and the integrated final result (Fig. 3f) almost identical to the true cell annotation (Fig. 3c). The ARI score of CDC has been improved from 0.38 to 0.87, which strongly outperforms that of K-means and SNN-Louvain following the same criterion (Fig. 3e). While K-means and SNN-Louvain split partial subclasses into smaller clusters and cannot keep the

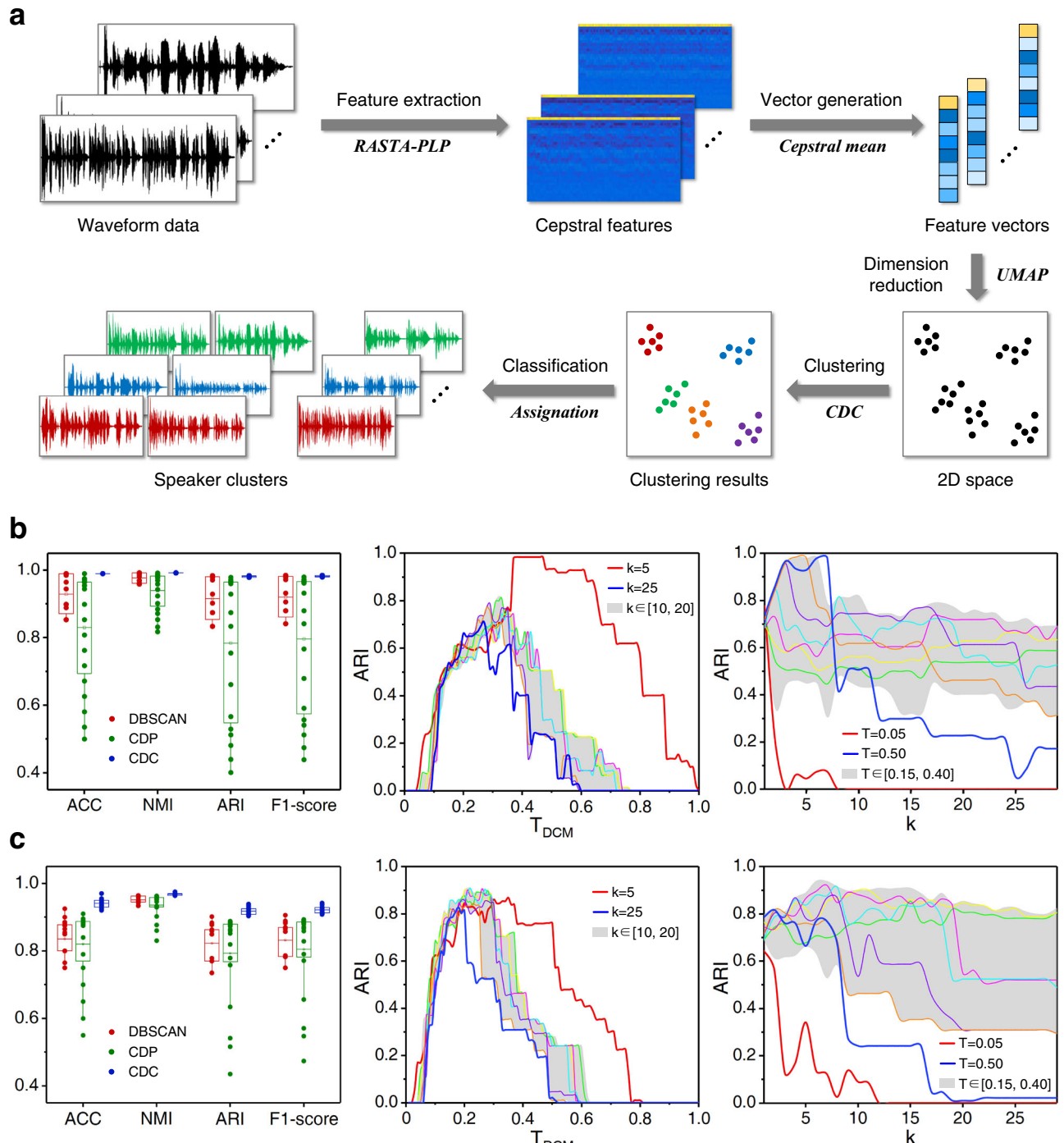

**Fig. 4 | Speaker recognition on ELSDSR and MSLT datasets. a** Workflow of speaker recognition using CDC. **b, c** The first column represents the validity indexes of the top 20 clustering results of DBSCAN, CDP and CDC in 2D space transformed by UMAP (from cepstral feature vectors with 31 dimensions). Boxes show the mean and the 25–75% range, while whiskers refer to the 1.5 times interquartile range. The second and third columns represent the ARI curves obtained by varying $T_{DCM}$ under fixed $k$ and varying $k$ under fixed $T_{DCM}$ respectively, where the gray bands represent the ARI ranges when $k$ and $T_{DCM}$ are in the ranges of [10,20] and [0.15, 0.40], and the curves falling in the bands were sampled with fixed intervals of 2 and 0.05.

accurate structures of subclasses as shown in Supplementary Figs. 6 and 7.

**Application of speaker recognition on corpus datasets**
Voice is a distinctive biometric characteristic and physiological modality of every human being. Speaker recognition aims to perform a classification of unlabeled speech samples from the voice data. However, a speech signal is susceptible to external or internal distorting factors (e.g., environment, recording device, tone, etc). To verify the ability of our algorithm to cope with distorting effects in speaker recognition, we benchmarked CDC using the English Language Speech Database for Speaker Recognition (ELSDSR)[44] and the Microsoft Speech Language Translation (MSLT)[45] corpus datasets. ELSDSR contains English speech data collected from 22 speakers composed of 12 males and 10 females. The speech was recorded in a noise-free chamber. The volunteers read the same sentences in a declarative

voice. MSLT corpus consists of tens of thousands of conversational utterances in multiple languages. The audio data were captured from the conversations held by the authorized volunteers in communication software. The contents of conversations were not predefined and the environments were random. A quantity of modal particles and emotional utterances were contained in the conversations. We selected 200 segmented audio files of the utterances of 20 speakers in Chinese including 12 males and 8 females.

The clustering workflow is illustrated in Fig. 4a. Cepstral features are commonly used to characterize speech signals. We leveraged RelAtive SpecTrAl-Perceptual Linear Prediction (RASTA-PLP)[46] to extract the cepstral features from the raw waveform data. To limit the influence of the noisy frames, we converted the cepstrum to feature vectors by calculating the mean of all the column vectors. UMAP was then used to reduce the dimensions of the original feature vectors. Based on the embedded data distributions, CDC was adopted to recognize the different speakers without any pre-training process.

The clustering results confirmed the effectiveness of CDC on speaker recognition through comparison with DBSCAN and CDP. Their max score reported by ACCuracy (ACC), Normalized Mutual Information (NMI), ARI and F1-score[47] (Supplementary Note 3) are similar on the ELSDSR dataset (Fig. 4b), but the advantage of CDC is more pronounced on MSLT (Fig. 4c). Rising and falling intonations usually appear in the real conversations collected by MSLT, while the recording environment of ELSDSR is quieter without ambient noise and reverberation. This reason causes data in MSLT corpus are more dispersedly distributed and generates more weakly-connected clusters than ELSDSR after dimension reduction. With the ability to handle weak connectivity, CDC yielded more accurate and stable outcomes than DBSCAN and CDP. In comparison, the performance of CDP is significantly influenced by the decisions of cluster centers. To further evaluate the robustness of CDC in parameter adjustment, we present the ARI curves under different combinations of input parameters $k$ and $T_{DCM}$ (see Methods). ARI fluctuated sensitively as $T_{DCM}$ changes, while it floated steadily from 0.3 to 1 when $k$ varies in the range [0.15, 0.4] of $T_{DCM}$. This pattern is relevant to the parameter sensitivity which is further analyzed in Supplementary Table 6.

### Dimension expansion for high-dimensional data

The core of CDC is leveraging Direction Centrality Metric (DCM) to distinguish internal and boundary points. DCM calculation requires to map the KNNs onto the unit spherical surface drawn by their center point firstly, then conducts spherical surface subdivision and angle measurement of the subdivision units. It is intuitive to understand the definition of DCM in 2D space that measures the variance of angles formed by KNNs. While in high-dimensional space, we construct the convex complex of KNNs for subdividing the spherical surface and calculate the volume of each simplex in the complex (Supplementary Fig. 8). Then, the volume of each subdivision unit is measured based on the simplex volumes (see Methods).

Since CDC is able to handle high-dimensional data and is not limited to 2D space, its applicability is of general significance. For the sake of the validation of CDC in high-dimensional space, we applied it to various datatypes in other fields, including UCI datasets, and handwritten and face images. Specifically, we collected eight UCI and one handwritten digits datasets, i.e., Iris, Seeds, Breast-Cancer, Wine, PenDigits, Dermatology, Control, Digits[48], MNIST10k[49] (details in Supplementary Table 3), and compared CDC with ten state-of-the-art clustering baselines, i.e., K-means, DBSCAN, CDP, AGNES, MeanShift, Rcut[50], Ncut[51], densityCut[52], Robust Continuous Clustering (RCC)[53], Graph Clustering with simultaneous Spectral Embedding and Discretization (GCSED)[54]. We normalized the features of all datasets and performed CDC under 2 to 5 UMAP dimensions, and the parameter settings can be found in Supplementary Table 7. ACC, NMI and ARI are calculated for clustering accuracy evaluation (Fig. 5a,

Supplementary Tables 4 and 5). In general, CDC obtained the highest scores on eight datasets, and CDC-U2 to CDC-U5 occupied the top four places of the average rank in all the three evaluation metrics. In terms of the time efficiency measured by overall runtimes, CDC-U2 ranked in the forefront, and achieved the second rank on the largest dataset MNIST10k with 784 features (Fig. 5b). Meanwhile, as the dimension increases, the runtime of CDC-U2 grows slower than densityCut, GCSED and much slower than RCC (Fig. 5c).

Moreover, we assessed the adaptability of CDC for handling image features on ORL face datasets. This dataset contains 40 distinct individuals and each one has ten images with different lighting, accessories, and facial expressions. All 400 face images are grayscale with 92×112 pixels in size. We extracted Gabor features as the input of clustering. In addition to K-means, three cutting-edge multi-view subspace clustering algorithms are selected for comparison, i.e., Co-Regularized multi-view subspace clustering (Co-Reg)[55], Pairwise/Centroid Multi-view Low-Rank Sparse Subspace Clustering (PMLRSSC/CMLRSSC)[56]. As the ACC, NMI and ARI scores illustrated in Fig. 5d, CDC has the highest and most stable accuracies, and achieve up to 0.9202 ARI score. The best CDC result is displayed in Fig. 5e, where the faces of 28 individuals were correctly recognized.

### Noise elimination using KNN-based methods

Noise points are ubiquitous and negatively impact clustering quality. KNN-based noise disposal methods can be integrated with CDC to handle data with noise as a data preprocessing step. This integration can be seamless by sharing the same $k$. Here, we adopted three KNN-based methods, i.e., Inverse Distance Metric (IDM), RKNN, and Local Outlier Factor (LOF)[57] to evaluate the effectiveness of noise elimination. Considering that the noise points are usually isolated and far away from the clusters, IDM detects noise by calculating the inverse of the distance sum of the KNNs. RKNN measures the number of points that treat the given point as a KNN object. LOF is formulized as the mean relative reachable density between the center point and its KNN. The formulas are illustrated in Supplementary Fig. 9.

We benchmarked the integrated methods on four synthetic datasets (i.e., DS10-DS13) with clusters of various shapes and noise points. As shown in Supplementary Fig. 10, all the three methods can detect the noise effectively in general, but performed slightly differently in the boundary areas of the clusters. In face of the clusters with uniform densities in DS10[2], IDM, RKNN and LOF can remove the noise and preserve the cluster points accurately. However, RKNN and LOF performed slightly worse than IDM in detecting noise points near the weakly-connected clusters in DS11[58], and misidentified low-density boundary points of the spindle-shaped clusters as noise in DS13[5].

### Parameter sensitivity analysis

Parameter sensitivity analysis aims to measure the degree to which the dependent variable is affected by the perturbation of the parameters, which facilitates to control the granularity of parameter tuning. In this paper, the clustering accuracy is the dependent variable, and we assessed the impact of the two input parameters of CDC on it, i.e., $k$ and $T_{DCM}$ (see Methods). We conducted sensitivity analysis using stratified sampling and random perturbation, which is modified from Latin-Hypercube One-factor-At-a-Time (LH-OAT)[59] in hydrology. This method divides the two-parameter space into multiple grid cells of the same size and samples one point in each cell. It then randomly changes each sampling point with a fixed perturbation, usually from a predefined constant, and finally calculates the sensitivity indexes to measure the effects on the results. In the simulations, we selected six synthetic datasets (DS4-DS9) with different numbers of points and clusters. We divided the value range of each parameter into 10 equal intervals to generate a 10 by 10 grid. Five sizes of perturbations for each parameter were specified in advance and ARI was treated as the dependent variable to evaluate the clustering accuracy. Each group in

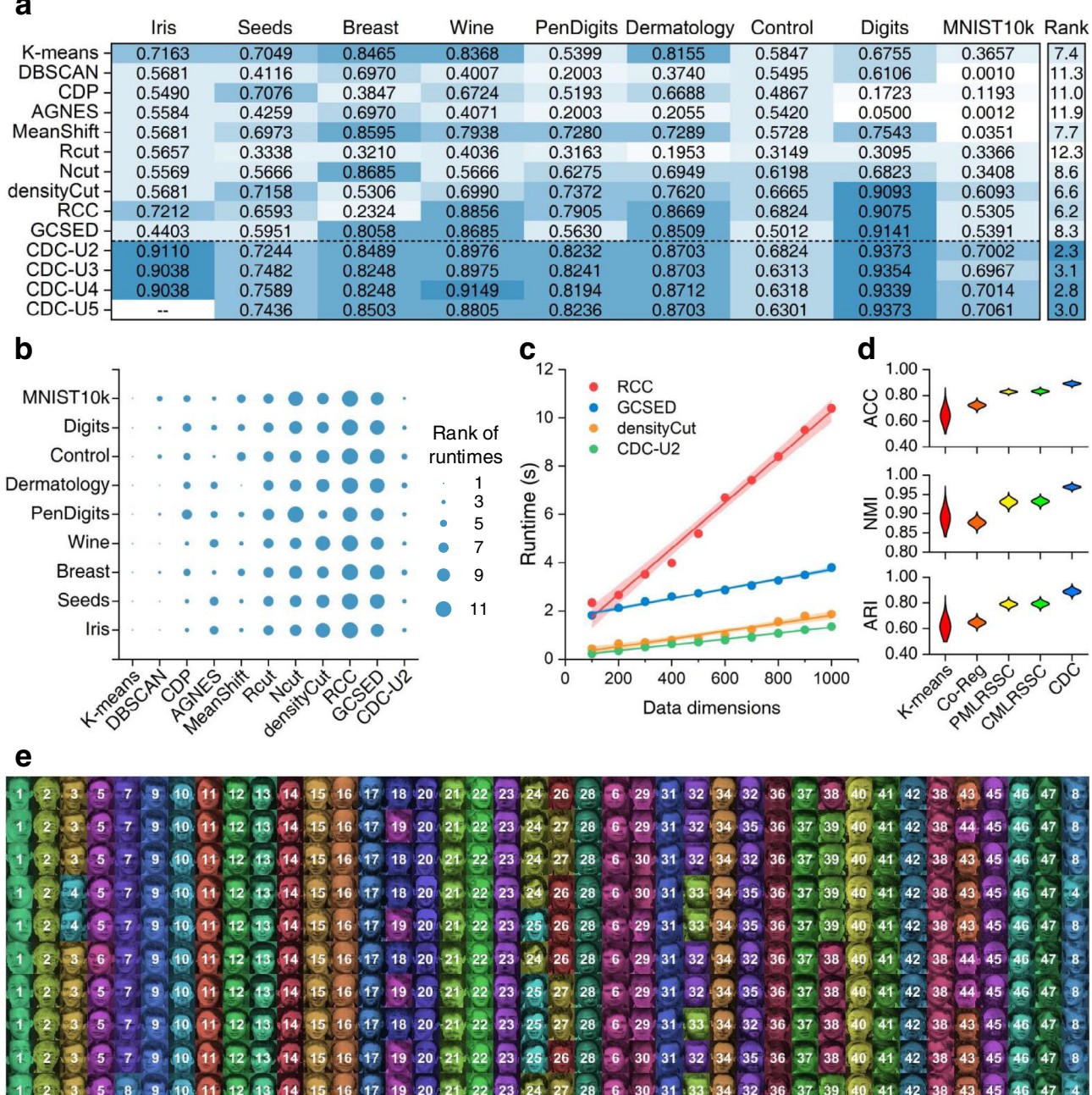

**Fig. 5 | Performances on UCI, MNIST10k, and ORL face datasets. a** Heatmap for the ARI scores of 11 clustering algorithms. **b** Comparison of runtimes among 11 clustering algorithms on eight UCI datasets and MNIST10k, where the runtime of CDC-U2 includes dimension reduction using UMAP and CDC clustering. All algorithms were implemented on a commodity desktop computer with an 8-core Intel i7 processor and 64 GB RAM (the smaller circle, the better time performance). **c** Runtimes of RCC, GCSED, densityCut and CDC-U2 (includes UMAP runtime as well) on simulated datasets with 1,000 samples and different number of dimensions. Shadow represents 95% confidence band. **d** Clustering accuracies reported by ACC, NMI and ARI of K-means, Co-Reg, PMLRSSC, CMLRSSC and CDC on ORL face dataset. **e** Pictorial representation of the best CDC cluster assignations, where each column contains ten images of one person and each color denotes a cluster.

the experiment was averaged five times to avoid the randomness in the simulations (detailed in Supplementary Note 7).

The results shown in Supplementary Table 6 indicate that $T_{DCM}$ is more sensitive to clustering accuracy than $k$. With the increase of perturbation, both of the two sensitivity indexes $S_k$ and $S_T$ grow. $k$ only reached the mild sensitive level when the perturbation is 20, since the local centrality of most points is stable as $k$ varies within a certain range. However, $T_{DCM}$ tended to reach the hypersensitive level when the perturbation is >0.15. *DCM* ranges from 0 to 1 theoretically, but most of *DCMs* commonly lie in a small range. A slight adjustment of

$T_{DCM}$ may cause many points to be converted from boundary to internal points or vice versa. This conversion affects the recognition of boundary points and their binding force for internal points, which in turn influences the clustering quality. In practice, we adopt a more stable parameter *ratio* to replace $T_{DCM}$ (see Methods). The sensitivity analysis for *ratio* was conducted in its entire value range of [0, 1]. The results show that the sensitivity indexes of *ratio* are significantly lower than that of $T_{DCM}$, which reveals its mild sensitivity. Therefore, $k$ and *ratio* can generate relatively stable outcomes when vary in a certain range, which in turn contributes to the robustness of CDC.

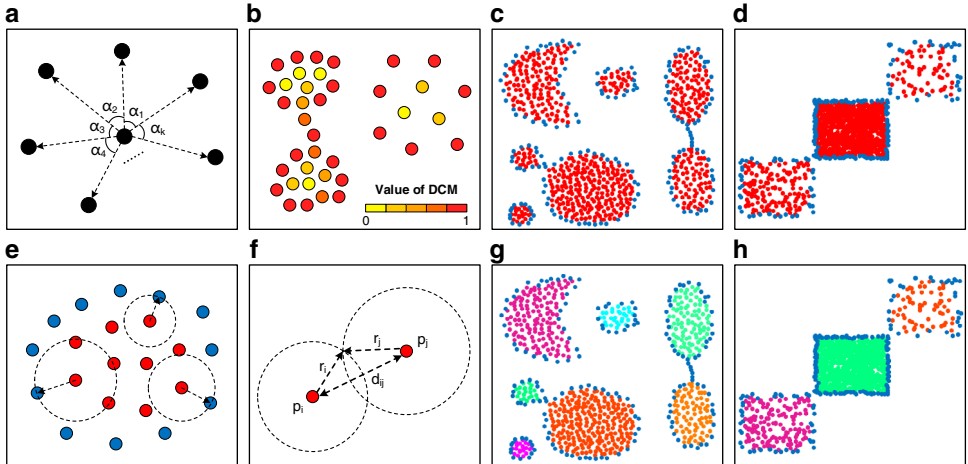

**Fig. 6 | Illustration of algorithm and intermediate results of CDC in 2D space.**
**a** Central angles formed by KNNs of a center point. **b** *DCM* calculation results of
sample data. **c**, **d** Division results of internal and boundary points on two synthetic
datasets, with $k = 10$ and $T_{DCM} = 0.1$ for DS5, and $k = 30$ and $T_{DCM} = 0.1$ for DS7. The
red points denote the internal points and the blue points denote the boundary
points. **e** Reachable distances of internal points. **f** Association rule for connecting
internal points. **g**, **h** Connection results of internal points on DS5 and DS7.

## Validation of the adaptive methods

Parameter tuning is a labor-intensive task that requires constant
trial and error in cluster analysis. To alleviate this issue, we pro-
pose two adaptive methods to determine the appropriate para-
meters. $T_{DCM}$ (or *ratio*) is estimated through graph theory analysis
on a Triangulated Irregular Network (TIN) in 2D space. Com-
monly, boundary points tend to have lower centrality (i.e., higher
*DCM*) than internal points. Thus, we sort all *DCMs* in a descend
order and the optimal $T_{DCM}$ (or *ratio*) can be searched if the
number of boundary points is given. Based on the graph theory
and 2D Euler's formula, it can be found that the boundary points
are associated with the vertexes, intra-cluster triangles and the
number of clusters (see Methods). With the estimation of the
cumulative number of vertexes, and intra-cluster triangles in the
multiple disconnected subgraphs, the number of boundary point
can be approximately determined using Eq. (15). We testified the
TIN-based adaptive method for $T_{DCM}$ on four synthetic datasets,
i.e., DS14-DS17 (Supplementary Fig. 11a–c), on which all clusters
can be identified accurately under the estimated parameters
(Supplementary Fig. 11c). The blue-colored cross-cluster triangles
were detected by the judge rule in Eq. (16) from the whole TIN
networks (Supplementary Fig. 11a). Although a few intra-cluster
triangles at the boundaries are misidentified as cross-cluster tri-
angles and several cross-triangles are undetected due to the close
proximity between clusters, these two biases can be offset
partially.

In practice, a rough estimation of $k$ would likely yield satisfac-
tory clustering results due to its insensitivity. $k$ is relevant to the
number of points $n$, and a linear or logarithmic relation between
them has been validated in the KNN-based cluster analysis[10,52].
According to the existed studies, we generalize an empirical model
to set $k$ with a continuous piecewise function in Eq. (9) (see Meth-
ods). It is formulized as a range of $k$ and ensures the growth speed to
flatten out when $k$ is >1000. We collated the optimal $k$ specified on
17 synthetic (DS1-DS17) and nine real-world datasets (Supplementary
Table 3). All the optimal $k$ lie within the range determined by Eq. (9)
and demonstrates the validity of the proposed empirical model
(Supplementary Fig. 11d). CDC equipped with the two adaptive
methods was further assessed on the real-world datasets. Results
show that it can achieve top 25% clustering accuracy on all datasets,
and even got the highest ARI on Wine, PenDigits, Control and Digits
(Supplementary Fig. 11e).

## Discussion

In many real-world applications, the data distribution in the feature
space tends to be heterogeneous and complex. Association rules
based on density or proximity of physical distance alone are difficult to
identify the clusters in an effective and stable manner. We utilize the
local centrality to distinguish the internal and boundary points. The
determined boundary points generate enclosed cages to prevent the
cross-cluster connection of internal points and enable the accurate
extraction of cluster shapes. In the constraint of the generated cages,
weakly-connected clusters can be separated. Moreover, the local
centrality depends on the directional uniformity of KNNs rather than
the density of the center point. Therefore, CDC is competent to
identify sparse clusters with low density.

CDC has strong robustness to heterogeneous distributions and
algorithm parameters. When handling clusters with clear shapes and
uniform density, it can accurately detect the boundary points. Actually,
the heterogeneous density would cause some wrong divisions of
internal and boundary points. Nonetheless, if the determined bound-
ary points are enough to avoid the cross-cluster connections, and only
a few internal points are misidentified, the intra-cluster connections
would not be cut off, thus not affecting the global assignments.

Beyond identifying the cell types, recognizing speaker voices and
face images, CDC has more potentials. It can be a promising technique
to segment the cell images, explore the spatial living patterns of spe-
cies, and reveal the aggregation distributions of geographic objectives.
However, CDC may be invalid to handle data with manifold structure
directly, since the detected boundary points cannot constrain the
internal connections in all directions in the feature space. Utilizing
dimension reduction techniques such as UMAP to embed the data to a
proper dimension can broaden the application of CDC. Consequently,
in some extreme cases, parameter setting should be more careful since
dimension reduction may cause "crowding problem" and affect clus-
tering accuracy. In addition, the *DCM* calculation would generate
increasing number of simplices as the dimension increases. More
effective *DCM* calculation could be further investigated.

## Methods
### Procedure of the clustering algorithm
The core idea behind CDC is to distinguish boundary and internal
points of clusters based on the distribution of KNNs. The boundary
points outline the shape of clusters and generate cages to bind the
connections of internal points. The internal points of clusters tend to

be surrounded by their neighboring points in all directions, while boundary points only include neighboring points within a certain directional range. To measure such differences in the directional distribution, we define the variance of the angles formed by the KNNs in 2D space as the local Direction Centrality Metric (DCM):

$$DCM = \frac{1}{k}\sum_{i=1}^{k}\left(\alpha_i - \frac{2\pi}{k}\right)^2 \quad (1)$$

KNNs of the center point can form $k$ angles $\alpha_1, \alpha_2...\alpha_k$ (Fig. 6a). For 2D angles, the condition $\sum_{i=1}^{k}\alpha_i = 2\pi$ holds. DCM reaches the minimum 0 if and only if all the angles are equal. This condition means that the KNNs of the center point are evenly distributed in all directions. It can be maximized as $\frac{4(k-1)\pi^2}{k^2}$ when one of these angles is $2\pi$ and the remaining are 0 (see Supplementary Note 1). Such an extreme situation happens when the KNNs are distributed in the same direction. According to the extrema, DCM can be normalized to the range [0, 1] as follows:

$$DCM = \frac{k}{4(k-1)\pi^2}\sum_{i=1}^{k}\left(\alpha_i - \frac{2\pi}{k}\right)^2 \quad (2)$$

A sample result of DCM calculation reveals that the internal points of clusters have relatively low DCMs and the boundary points have higher values (Fig. 6b). Thus, internal and boundary points can be divided by a threshold $T_{DCM}$. The division results of two synthetic datasets DS5 and DS7 validate the effectiveness (Fig. 6c, d).

To ensure that the internal points $p_1, p_2, ...,p_m$ connect to each other within the area restricted by the surrounding boundary points $q_1$, $q_2, ..., q_{n-m}$, we define the minimum distance between the internal point $p_i$ and all boundary points as its reachable distance:

$$r_i = \min_{j=1}^{n-m}d(p_i, q_j) \quad (3)$$

where $d(p_i, q_j)$ is the distance between the two points $p_i$ and $q_j$ (Fig. 6e). Two internal points can be connected as the same cluster if the following association rule is guaranteed:

$$d\left(p_i, p_j\right) \leq r_i + r_j \quad (4)$$

where $r_i$ and $r_j$ are the reachable distances of internal points $p_i$ and $p_j$, respectively (Fig. 6f). On the premise of correct identification of boundary points (except for extreme cases in the situation when the boundary points are identified incompletely), the connections of internal points are constrained in the area defined by the boundary points. If a cross-cluster connection exists between two internal points, there will be boundary points contained in the range defined by their reachable distances, which conflicts with the definition of the reachable distance. Therefore, the internal points of the same cluster can be trapped in the same external contour consisting of boundary points, and the cross-cluster connections will be avoided based on this association rule. The connection results of DS5 and DS7 are generated by applying the rule to the division results (Fig. 6g, h). Although couples of clusters are connected weakly in DS5 and DS7, and the three clusters in DS7 differ greatly in density, all the clusters have been identified accurately.

After calculating the DCM and connecting internal points, we finish the procedure by assigning each boundary point to the cluster to which its nearest internal point belongs. CDC contains two controllable parameters, $k$ and $T_{DCM}$. $k$ adjusts the number of nearest neighbors, and $T_{DCM}$ determines the division of internal and boundary points. The pseudocode of CDC is detailed in Supplementary Note 2. In practice, considering $T_{DCM}$ varies with data distributions, we adopt a percentile ratio of internal points to determine $T_{DCM}$ as the

$[n•(1-ratio)]th$ DCM sorted in a descending order (Supplementary Fig. 11b). The parameter ratio has intuitive physical meaning and better stability (see Supplementary Table 6), which makes it easier to specify than $T_{DCM}$. According to our experiments, 70%-99% internal points are the suggested default parameter range of ratio for promising clustering results. Nevertheless, when clusters are mixed up with each other, more boundary points (lower ratio) are necessary to separate the close clusters.

## Time complexity analysis

To assess the computational efficiency of CDC, the time complexity is analyzed. Runtime of CDC can be decomposed as:

$$T_{CDC} = T_1 + T_2 + T_3 \quad (5)$$

where $T_1$, $T_2$, $T_3$ denote the runtime of the division of internal and boundary points, internal connection, and boundary assignment respectively. As described in the procedure, the step of division includes measuring the distances, searching the KNNs, and calculating the DCMs. So, the time complexity of $T_1$ in 2D space can be represented as:

$$O(T_1) = O\left(n^2 + n\left(\sum_{i=1}^{k} n - i\right) + nk\right) = O(kn^2) \quad (6)$$

where $n$ refers to the total number of points. The internal connection is composed of the calculation of reachable distances and point-wise association:

$$O(T_2) = O(m(n-m) + m^2) = O(mn) \quad (7)$$

where $m$ refers to the number of identified internal points. Then, the search of the nearest internal point for each boundary point and assigning the corresponding labels require the time complexity:

$$O(T_3) = O(m(n-m) + (n-m)) = O(mn) \quad (8)$$

The specified parameters $k$ and $m$ are much $<n$ commonly, hence the total time complexity of CDC is approximately $O(n^2)$. The search of KNN can be refined using indexes techniques[41], such as K-D tree. It can avoid the calculation of pairwise distances and improves time efficiency to $O(n\log n)$.

## Empirical estimation method of $k$

Through the analysis of parameter sensitivity and existing studies[10,52], we know that $k$ is an insensitive parameter and relates to the number of points $n$ in dataset. Thus, we propose an empirical method by formulizing the relation between $k$ and $n$ as:

$$k = \begin{cases} \lceil\frac{n}{50}\rceil \sim \lceil\frac{n}{20}\rceil & \text{if } 100 \leq n \leq 1000 \\ \lceil\log_2(n)+10\rceil \sim 5\lceil\log_2(n)\rceil & \text{if } n \geq 1000 \end{cases} \quad (9)$$

where $\lceil\cdot\rceil$ denotes to the nearest integer upwards. This empirical model is represented as a continuous piecewise function that depicts a growth trend of $k$ as $n$ increases.

## Estimation of the number of boundary points for determining $T_{DCM}$

As shown in Supplementary Fig. 12a, we constructed a Triangulated Irregular Network (TIN) to connect all points. In graph theory, the degree of a vertex is defined as the number of edges incident to the vertex and each edge connects two vertexes. Based on this law, we can

obtain:

$$\sum_{i=1}^{V} \deg(v_i) = 2E \tag{10}$$

where $\deg(v_i)$ represents the degree of vertex $v_i$, $V$ denotes the total number of vertexes, and $E$ represents the total number of edges. In a graph, each triangle has three edges and each edge is shared by two triangles except the outermost edges. Actually, for a TIN that has a single connected component, the total number of boundary points is equal to that of the outermost edges, since all the outermost edges are connected end to end by boundary points and form a closed polygon. This law can be summarized as:

$$\sum_{i=1}^{V} \deg(v_i) = 3F + B \tag{11}$$

where $F$ and $B$ refer to the total number of triangles and boundary points respectively. Meanwhile, 2D Euler's formula can be considered as follows:

$$V + F - E = 1 \tag{12}$$

By combining these formulas, we can infer the solution of $B$ as follows:

$$B = 2V - F - 2 \tag{13}$$

However, the number of initial boundary points in the whole TIN is not equal to the total number of boundary points in the separated clusters. To conduct an accurate estimation, the whole TIN should be treated as multiple sub-networks (Supplementary Fig. 12b). Given $C$ clusters, the number of boundary points in clusters can be solved as follows:

$$\sum_{i=1}^{m} B_i = 2\sum_{i=1}^{m} V_i - \sum_{i=1}^{m} F_i - 2C \tag{14}$$

$$B = 2V - F - 2C \tag{15}$$

where $F$ is the total number of intra-cluster triangles in the multiple separated networks. $V$ is known in a given dataset (i.e., $n$), but $F$ and $C$ are not. The initial $F$ is the total number of triangles in the whole TIN, which includes the triangles connecting different clusters, i.e., cross-cluster triangles whose three vertices are not all in the same cluster (otherwise is intra-cluster triangle). Using the excessive number of triangles would make the number of boundary points $B$ smaller than the true value. To identify the cross-cluster triangles, we set a judgment rule:

$$\sum_{i=1}^{3} \sum_{j=1, j \neq i}^{3} \sigma(v_i, v_j) < 3 \tag{16}$$

where $v_1$, $v_2$, $v_3$ are the three vertices of a triangle, and $\sigma(v_i, v_j)$ is an indicator function:

$$\sigma(v_i, v_j) = \begin{cases} 0 & \text{if } v_j \notin KNN(v_i) \\ 1 & \text{if } v_j \in KNN(v_i) \end{cases} \tag{17}$$

Equation 16. considers the proximity of the vertices in an intra-cluster triangle. The final $F$ can be calculated as the initial $F$ minus the number of cross-cluster triangles that satisfies Eq. (16) (Supplementary

Fig. 12c). In terms of the number of clusters $C$, it is significantly $<V$ and $F$ usually, which has a trivial effect on the estimation of $B$. Moreover, CDC is robustness to the $DCM$ threshold as mentioned in Discussion. Thus, $C$ can be treated as 1, when it is vague or difficult to determine.

In an actual implementation, we adopt Delaunay triangulation algorithm to construct the initial TIN, which can prevent the connection of internal points far apart within the same cluster. The time complexity to construct the TIN can be represented as $O(n \log n)$, which is same as CDC. Moreover, it should be noted that using the intermediate results after removing the cross-cluster triangles as the final clustering results is inappropriate. Because Eq. (16) cannot guarantee to remove the cross-cluster triangles in an accurate and complete manner (Supplementary Fig. 12a), thereby causing the clusters in close proximity to be merged as a whole. The pseudocode of adaptive method to determine $T_{DCM}$ is illustrated in Supplementary Note 2.

## DCM calculation in high-dimensional space

$DCM$ calculation requires to map the KNNs onto the unit hyperspherical surface drawn by their center point firstly, then subdivides the hyperspherical surface and measures the generalized angles of each subdivision unit. In 2D space, KNNs are mapped onto a unit circle. They subdivide the circle into multiple arcs and each arc corresponds to a central angle. $DCM$ measures the variance of these angles. While in 3D space, KNNs are mapped onto a unit spherical surface. They connect neighboring points to form a spherical triangulation and $DCM$ is extended as the variance of the solid angles of the triangles. For subdividing a hyperspherical surface in a higher-dimensional space, we adopt Qhull algorithm[60] to construct the convex complex of KNNs. Since all the KNNs have been mapped onto the hyperspherical surface, they are guaranteed to be the vertices of the convex complex (Repeat KNNs are not included). In $d$-dimensional space, each facet of the convex complex is a $(d\text{-}1)$-simplex and corresponds to a subdivision unit (Simplex here denotes the simplest figure that contains $d+1$ given points in $d$-dimensional space and that does not lie in a space of lower dimension). For instances, a 2D and 3D convex complex consists of multiple line segments (1-simplex) and triangles (2-simplex), which subdivides the circle and spherical surface into arcs and spherical triangles respectively (Supplementary Fig. 8).

After subdividing, the generalized angles could be measured. Natively, the angles are equivalent to volumes of the corresponding subdivision units (e.g., arc length in 2D circle, area of the spherical triangle in 3D sphere). However, it is difficult to calculate the volumes of subdivision units in high-dimensional space due to the computational complexity of multiple integral. Therefore, we measure the volume of each simplices and then allocate the global volume error to each subdivision unit evenly for an approximate calculation. Although there are errors between the true and calculated volumes of subdivision units, the $DCM$ sort orders based on the two kinds of volumes are the same, since the volumes of subdivision units increase monotonically with the corresponding simplices. Thus, a $DCM$ threshold can be searched to distinguish the internal and boundary points effectively.

Specifically, we suppose a simplex $s$ in $d$-dimensional space is composed of $d$ KNN points $p_1, p_2, \ldots, p_d$ which have been mapped onto the hypersphere, where $p_i = (x_i^1, x_i^2, \ldots, x_i^d)$. Vectors $\overrightarrow{p_d p_1}$, $\overrightarrow{p_d p_2}$, ..., $\overrightarrow{p_d p_{d-1}}$ with the same origin $p_d$ can determine a $(d\text{-}1)$-dimensional parallelepiped $P$ in vector space $\mathbb{R}^d$. Let $\varepsilon_i = \overrightarrow{p_d p_i} = (x_i^1 - x_d^1, x_i^2 - x_d^2, \ldots, x_i^d - x_d^d)^T$ and $A = (\varepsilon_1, \varepsilon_2, \ldots, \varepsilon_{d-1})^T$, then we have the volume of $P$[61]:

$$vol(P) = \sqrt{\det(AA^T)} \tag{18}$$

Simplex *s* is a hypertetrahedron embedded in the parallelepiped *P* and they share *d*-1 edges $p_d p_1, p_d p_2, ..., p_d p_{d-1}$. So, the volume of *s* is[62]:

$$vol(s) = \frac{vol(P)}{\Gamma(d-1)} = \frac{\sqrt{\det\left(\mathbf{A}\mathbf{A}^T\right)}}{(d-1)!} \tag{19}$$

The detailed volume calculations of parallelepiped *P* and simplex *s* are presented in Supplementary Note 6. After measuring the simplex volume, we calculate the deviation between the volume sum of simplices and the hyperspherical surface area, and further assign the global volume error equally to each subdivision unit. The assignment guarantees the volume sum of subdivision units under different subdivisions is constant in the same dimension. The generalized surface area of a *d*-dimensional unit sphere is:

$$S = \frac{2\pi^{\frac{d}{2}}}{\Gamma(\frac{d}{2})} \tag{20}$$

We suppose that the convex complex consists of *f* simplices $s_1, s_2, ..., s_f$ and subdivides the hypersphere into *f* units $u_1, u_2, ..., u_f$ accordingly. The volume of subdivision unit $u_i$ can be solved as:

$$vol(u_i) = vol(s_i) + \frac{S - \sum_{i=1}^{f} vol(s_i)}{f} \tag{21}$$

*DCM* measures the variance of the volumes of all subdivision units:

$$DCM = \frac{1}{f}\sum_{i=1}^{f}\left(vol(u_i) - \frac{S}{f}\right)^2 \tag{22}$$

Like *DCM* in 2D space, it can also be normalized as:

$$DCM = \frac{f}{(f-1)S^2}\sum_{i=1}^{f}\left(vol(u_i) - \frac{S}{f}\right)^2 \tag{23}$$

**Reporting summary**

Further information on research design is available in the Nature Research Reporting Summary linked to this article.

## Data availability

The synthetic datasets used in this study have been deposited at https://github.com/ZPGuiGroupWhu/ClusteringDirectionCentrality/tree/master/Toolkit/Synthetic%20Data%20Analysis/SyntheticDatasets. The scRNA-seq, CyTOF, corpus and other real-world datasets used in this study are available publicly: BH, BM, Muraro, Segerstolpe, Xin, AMB and TM (https://zenodo.org/record/2877646#.YjBPGGXpByUl), ALM and VISp (https://portal.brain-map.org/atlases-and-data/rnaseq/mouse-v1-and-alm-smart-seq), WT_R1, WT_R2, NdpKO_R1 and NdpKo_R2 (GSE125708), MIHPF (https://portal.brain-map.org/atlases-and-data/rnaseq/mouse-whole-cortex-and-hippocampus-10x), Levine and Samusik (FlowRepository: FR-FCM-ZZPH), ELSDSR (http://www2.imm.dtu.dk/~lfen/elsdsr/), MSLT (https://www.microsoft.com/en-us/download/details.aspx?id=55951), Iris (http://archive.ics.uci.edu/ml/datasets/Iris), Seeds (http://archive.ics.uci.edu/ml/datasets/seeds), Breast-Cancer (http://archive.ics.uci.edu/ml/datasets/Breast+Cancer+Wisconsin+%28Original%29), Wine (http://archive.ics.uci.edu/ml/datasets/Wine), PenDigits (http://archive.ics.uci.edu/ml/datasets/Pen-Based+Recognition+of+Handwritten+Digits), Dermatology (http://archive.ics.uci.edu/ml/datasets/Dermatology), Control (http://archive.ics.uci.edu/ml/datasets/Synthetic+Control+Chart+Time+Series), Digits (https://archive.ics.uci.edu/ml/datasets/Optical+Recognition+of+Handwritten+Digits), MNIST10k (http://yann.lecun.com/exdb/mnist/), ORL face dataset (https://www.kaggle.com/datasets/jagadeeshkasaraneni/orlfaces).

## Code availability

The code of CDC in MATLAB, R and Python, and the toolkit with six applications can be downloaded at https://github.com/ZPGuiGroupWhu/ClusteringDirectionCentrality and https://zenodo.org/record/7029720#.YwuFsuxByZw. Digital Object Identifier https://doi.org/10.5281/zenodo.7029720.

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

## Acknowledgements

We thank S.C. McClure for language and S. Chen for technical discussion. This paper is supported by National Natural Science Foundation of China (No. 42090010, No. 41971349 and No. 41930107), National Key R&D Program of China (No. 2021YFE0117000, No. 2018YFC0809806 and No. 2017YFB0503704) and Zhizhuo Research Fund on Spatial-Temporal Artificial Intelligence (No.ZZJJ202201). Part of computation in this work was done on the supercomputing system in the Supercomputing Center of Wuhan University.

## Author contributions

Z.G. and H.W. envisioned the study. D.P. and Z.G. designed the algorithm and collected datasets. D.P., Z.G., D.W., Y.M. and Z.H. conducted experiments and performed the analysis. D.P. and Z.G. wrote the manuscript. H.W. and Y.Z. provided advice in analysis.

## Competing interests

The authors declare no competing interests.
