## [Peer Review File · Nature Communications]

Reviewers' Comments:

Reviewer #1:

Remarks to the Author:

The proposed clustering algorithm measures Direction Centrality locally (CDC) based on the distribution character of internal and boundary points. The idea seems quite intuitive. CDC shows its robustness to outliers and different kinds of data distribution on several synthetic datasets. Besides, the authors evaluate CDC on three real-world application including scRNA-seq, CyTOF, and voice corpuses. I would to point out some of my concerns and questions to this work:

1. The observation that "the internal points of clusters tend to be surrounded by their neighboring points in all directions, while boundary points only include neighboring points within a certain directional range" seems novel. However, it is a bit violent to simply reduce the dimension of raw data to two for the DCM computation since such a dimensionality reduction could lead to severe information loss. Is there any way to conduct DCM in high dimensional spaces?
2. Other clustering metrics such as NMI and ACC should be reported while comparing CDC to other methods such as recent contrastive clustering; robust continuous clustering, etc. In addition, the currently used metrics are kind of similar to each other.
3. In Equation 3, the definition of the reachable distances r should be formally given. And more explanations should be made on how the cross-cluster connections are avoided because it seems that every pair of data satisfies the connection requirement.
4. What is the estimated $T_{\{DCM\}}$ on datasets in Table 1? Extra experiments are needed to prove the effectiveness of strategy for determining the $T_{\{DCM\}}$ parameter since it is very sensitive. Does the estimation strategy works well on all real-world datasets?
5. The definition of sensitivity used in Table 1 should be formally given.
6. Does the noise robustness of CDC rely on the Local Outlier Factor (LOF) algorithm or itself is noise robust?
7. To further prove the effectiveness of CDC with simply reducing the dimension to two on more complex data, could you conduct CDC on commonly used image dataset like MNIST and CIFAR-10? It could make the method more convincing.

Reviewer #2:

Remarks to the Author:

Summary

The authors present a new clustering algorithm called CDC designed for data with heterogeneous density and low connectivity. CDC is based on the observation that the directions from boundary points to their k nearest neighbours (k -NNs) vary much less than the directions from inner points to their k -NNs. To exploit this observation, the normalised Direction Centrality Measure (DCM) is defined, which, for each node, returns a value close to 0 if the k -NNs are distributed in very diverse directions and a value close to 1 if they are distributed in few directions. Subsequently, boundary points are defined as data points with DCM larger than a threshold T and inner points are connected into clusters via the association rule $p_i \sim p_j \Leftrightarrow d_{\{ij\}} \leq r_i + r_j$, where r_i is the distance from p_i to the closest boundary point. Finally, each boundary point is assigned to the cluster of the nearest inner point. Strategies for automatically selecting the two hyper-parameters k and T are presented, and CDC is compared against existing approaches on various datasets.

Overall evaluation

The idea behind the algorithm is simple and elegant and the results are very promising. However, I feel that the both the algorithm itself and the way it is presented in the paper could still be improved considerably (see comments below). Moreover, the classical "Introduction", "Results", "Discussion", "Methods" foreseen by Nature Communications doesn't really fit the paper: The main contribution/result of the paper - namely the clustering algorithm itself - is now presented in the "Methods" section, which, if accepted, would be typeset in small font at the very end of the paper. On the other hand, the experimental results reported in the "Results" section are really only proofs of concept that the algorithm might be valuable for the community. Of course, this is not really to

blame on the authors, since they only followed the Nature Communications guidelines. However, it indicates that dedicated computer science venues (eg., KDD, AAAI, ICML, Pattern Recognition, TPAMI) might be better fits for the manuscript.

Comment 1

The main drawback of CDC is that it is designed for 2-dim data only. It would be a much more valuable contribution if it could also handle n-dim data. My intuition is that the algorithm could be generalised to the n-dim case. Two things would have to be changed: (1) The definition of DCM. (2) The automated parameter selection routine for the threshold T. For (1), the authors would have to come up with a k-NN direction diversity measure that generalizes to n-dim data. Here's an idea: Let $n_1, \dots, n_j, \dots, n_k$ be the k-NNs of a point p_i and $d_j = p_i - r_j$ be the corresponding directions. Why not defining the diversity measure as the mean correlation coefficient across all pairs of directions (i.e., as $\frac{1}{\binom{k}{2}} \sum_{j=1}^{k-1} \sum_{l=j+1}^k \frac{d_j^T d_l}{\|d_j\| \|d_l\|}$)? Of course, this doesn't really generalize DCM to the n-dim case, but it formalizes the same underlying idea. W.r.t. (2), the only thing I can think of at the moment is to determine T empirically via large-scale benchmarking on tons of datasets. But there might be a cleverer way.

Comment 2

Since CDC can only cope with 2-dim data, the authors transformed the real-world datasets to 2-dim space via dimensionality reduction techniques before comparing CDC to other clustering algorithms. This, however, is not a fair experimental setup, because the competitors are not restricted to 2-dim data. So while the experiments indeed show that CDC outperforms the competitors on the 2-dim datasets, it might be the case that the competitors perform better on the original datasets or higher dimensional transformations. I hence suggest reconsider the setup of their experiments and run the competitors also on higher dimensional versions of the datasets.

Comment 3

Here's how I understood the routine for selecting the parameter T: (1) Compute Delaunay triangulation on all data points. (2) Remove cross-cluster triangles that satisfy a certain condition (see next comment for an issue here). (3) Determine number b of boundary points in the resulting partial triangulation. (4) Set T to DCMS[b], where DCMS is the array of DCM values sorted in non-decreasing order. If this understanding is correct, I am wondering how the clustering induced by the partial triangulation obtained after removing cross-cluster triangles relates to the clustering computed by CMC. Does CMC merely reproduce the partial triangulation clustering? If so, why not directly using the partial triangulation clustering? If not, how exactly does CDC improve the partial triangulation clustering? It's also possible that I misunderstood something here. If so, please clarify the respective paragraphs in the manuscript.

Comment 4

What does it mean to remove a cross-cluster triangle? My guess: removing all three edges. But this is not stated explicitly in the paper. Also: Equation (15) does not make sense, since j is not defined and $v_i \in \text{KNN}(v_j)$ is a truth-value and therefore has no cardinality. Please provide a sound characterisation of cross-triangles, e.g., something like this: "A triangle $\{\{p_1, p_2\}, \{p_2, p_3\}, \{p_1, p_3\}\}$ is called cross-triangle if and only if <some condition involving the k-NNs of the points $p_1, p_2,$ and p_3 >".

Comment 5

Since the parameter T has no intuitive interpretation, it is very difficult to set manually. Therefore, users will most probably use the automated selection routine. Consequently, I think that this routine should be regarded as an integral part of the proposed algorithm and therefore be considered both in the theoretical complexity analysis and in the runtime evaluation.

Comment 6

Equation (8) used to automatically select the parameter k is somewhat adhoc. How exactly did you come up with this equation and how did you select the involved constants?

Comment 7

Since the main contribution of the paper is the algorithm CDC itself, it would be of great help for the reader if both the main algorithm and the routine for automatically selecting the parameter T were summarised via pseudo-code.

Comment 8

In the past years, Python has more and more developed into the quasi-standard language for data science applications. To increase the likelihood that the CDC is actually used by the targeted community, I therefore strongly suggest that the authors provide a Python implementation in addition to the existing MATLAB implementation.

David Blumenthal

Reviewer #3:

Remarks to the Author:

Summary

The authors present an interesting new clustering algorithm, and demonstrate performance on single-cell RNA sequencing (scRNA-seq), mass cytometry (CyTOF), and speech data. The authors' evaluations show good performance of the algorithm. However, to provide users with more information about whether this algorithm is useful in practice for analyzing scRNA-seq and CyTOF data, these evaluations need to be expanded and compared against standard analysis pipelines for these data types. More details are provided below.

Major comments

1) The authors' performance evaluations for scRNA-seq and CyTOF data rely on (i) an initial dimension reduction step (using either t-SNE or UMAP) to two dimensions, followed by (ii) clustering in the 2-dimensional space using the authors' algorithm. This is quite different to standard analysis pipelines for these data types (e.g. for scRNA-seq: initial dimension reduction to 50 dimensions using PCA on a subset of highly variable genes, followed by clustering in 50-dimensional PCA space using graph-based clustering; and for CyTOF: clustering in the full-dimensional space using algorithms such as FlowSOM / PhenoGraph).

To provide the most useful information for readers, the authors should extend these comparisons to include the full standard clustering pipelines for scRNA-seq and CyTOF data.

2) For the clustering of CyTOF and scRNA-seq data, much more detail is required for how the authors ran their clustering algorithm on these datasets – in particular preprocessing steps. For example, for the CyTOF datasets (Supplementary Figure 2), no details are provided for whether “unclassified” points – i.e. cells without ground truth labels – were excluded prior to clustering, which would create an unfair comparison.

3) For the scalability comparisons for scRNA-seq and CyTOF data (Figure 3), it is not clear whether the runtimes shown here are only after the initial t-SNE or UMAP dimension reduction. In this case, these results would be difficult for users to interpret, since clustering in 2-dimensional t-SNE or UMAP space is not a recommended analysis pipeline for scRNA-seq or CyTOF data, and previously published runtime/scalability estimates start from the full high-dimensional data. These results

should be clarified, and presented in a way that enables comparisons of runtime/scalability starting from the full high-dimensional data.

4) The evaluations for scRNA-seq and CyTOF data also rely on quite old and small datasets by modern standards. To be most useful for readers, the analyses should be compared on modern full-sized datasets (especially in terms of number of cells for scRNA-seq data).

Minor comments

In several locations (e.g. Abstract; Introduction lines 30 and 34), the authors use the terminology "spatial distribution" to refer to the distribution of points in feature space. This is somewhat confusing, since readers may interpret this as referring to physical spatial data (e.g. geographical data). Suggest replacing this with clearer terminology such as "feature space" or "high-dimensional space".

Wording and / or figure captions (e.g. Figure 1) for scRNA-seq and CyTOF data should make clearer when this is discussing only 2-dimensional data, since this is a much easier clustering task than typical (high-dimensional) scRNA-seq or CyTOF data.

Introduction, line 37: this line states that k-means and k-medoids can only identify spherical clusters. This is not correct – these algorithms can identify ellipsoidal clusters. This wording should be replaced with "ellipsoidal" or similar.

Reviewer #4:

Remarks to the Author:

In this paper, the authors introduce a density-based clustering algorithm (CDC) by first searching for the boundary points between clusters, based on the observations that the k-NN of boundary points are from a small angle range, while for internal points, their k-NNs are from a much wider angle range. The authors used both simulated and real datasets to evaluate their method.

Clustering analysis is an important exploratory data analysis technique, and density-based clustering is appealing because of the natural definition of clusters based on density peaks. However, these methods may be sensitive to parameter setting. I have some major comments about the method itself, its connection with existing methods, the experimental setting (robust to parameters), and the results, which need to be addressed.

1, Exploring the k-NNs of data points to identify boundary points have been used implicitly or explicitly for density-based clustering. For example, in (Ding et al, densityCut, Bioinformatics, 2016), the authors used directed k-NN graphs for their density-based clustering algorithm, and boundary points have zero or few 'in-neighbors' while the non-boundary points tend to have more 'in-neighbors'. The definition of boundary points based on 'in-neighbors' is much easier than angles as used in this study because the number of neighbors is independent of the dimensional of the data. Will the incorporation of this 'in-neighbor' definition of boundary points improve CDC? How CDC performs compared to densityCut?

2, Mean-shift (Chen YZ, Mean-shift, PAMI, 1995) is a widely used density-based clustering algorithm based on density model-seeking. This 'model-seeking' approach is complementary to the 'boundary-seeking' used by CDC. The authors need to add mean-shift to discussions and comparisons. It seems that the 'model-seeking' approach is an easier and more elegant solution to density-based clustering as CDC is sensitive to the definition of the boundary points.

3, Compared to Mean-shift, which has only one parameter, the kernel density estimation bandwidth, and densityCut, which only has one parameter, the number of neighbors, CDC needs both the number of neighbors and also the threshold to define boundary points. As the authors discussed, CDC is sensitive to these parameters. The authors need to show that the added

complexity improves performance compared to the simpler methods.

4, In Figure 1a, the authors compared CDC with baselines using synthetic datasets. Although useful, the problem is that these algorithms, e.g., DBSCAN, are very sensitive to parameters, and the performance is largely determined by the parameters used. For a fair comparison, the authors need to tune the parameters the same way as for CDC, or at least for all methods (including CDC) used the default setting.

5, Similarly, in Figure 2, the authors showed that CDC performed favorably compared to other methods. It's hard to know if this is a parameter setting problem. Could the authors show the results of using different combinations of parameters?

6, Why the authors only reported ARI in Figure 3? Why not reporting FI-score as in Figure 1. Also, normalized mutual information is a widely used metric. In addition, why the authors only used 4 datasets for the comparison in Supp. Fig. 2 instead of all the 13 datasets? Supp.Fig.2 is a very informative figure, but again, it's not clear how the parameters were chosen for CDC. The authors should do the same parameter selection for all methods for a fair comparison.

7, The claim that CDC is insensitive to k is not generally true, as demonstrated in Figure 4d.

8, Figure 6 legend, 'blue represents intra-cluster triangles ...' should be 'inter-cluster'?

REVIEWER COMMENTS

Reviewer #1 (Clustering and ML):

The proposed clustering algorithm measures Direction Centrality locally (CDC) based on the distribution character of internal and boundary points. The idea seems quite intuitive. CDC shows its robustness to outliers and different kinds of data distribution on several synthetic datasets. Besides, the authors evaluate CDC on three real-world application including scRNA-seq, CyTOF, and voice corpuses. I would to point out some of my concerns and questions to this work:

Thank you.

Comment 1

The observation that "the internal points of clusters tend to be surrounded by their neighboring points in all directions, while boundary points only include neighboring points within a certain directional range" seems novel. However, it is a bit violent to simply reduce the dimension of raw data to two for the DCM computation since such a dimensionality reduction could lead to severe information loss. Is there any way to conduct DCM in high dimensional spaces?

Thank you for your positive comment and invaluable suggestion. **The idea about the direction centrality also applies to higher-dimensional data**, e.g., DCM depicts the variance of all solid angles or corresponding spherical areas in 3D space (R1-Fig. 1a), Nonetheless, generalizing the calculation of DCM (i.e., the angle variance) to hyperspace directly is challenging, because **the subdivision criterion for high-dimensional data is difficult to be defined, and may causes variations in the subdivision of spheres**. So, in this paper, we utilize t-SNE and UMAP to tackle this issue by transforming high-dimensional raw data into 2D. **Although these two dimensionality reduction techniques will lead to some degree of information loss, the data distribution in high-dimensional feature space could be well-maintained actually** (see experiment results below and existing studies(Maaten and Hinton, 2008; McInnes et al., 2018)).

According to your suggestion, **we have also proposed three alternative approaches**, i.e., Vector direction decomposition (VDD), Space division counting (SDC), and Orthogonal plane projection (OPP), **to achieve the calculation of DCM in hyperspace by following the same underlying idea of CDC**. To be noted, although the three methods achieve KNN search and DCM calculation in hyperspace, they actually transform the variance of high-dimensional angles to low-dimensional substitutes using projection and subspaces division. Specifically, to measure DCM, SDC counts the number of points (0D) in divided subspaces for each dimension; VDD utilizes the 1D distribution of projection points in each dimension; while OPP uses angle variances in projected 2D planes. **These transformation processes may introduce bias in DCM calculation comparing with conducting it in original high-dimensional space directly**. The methodology is detailed in Supplementary Note 5 and the experimental results are in Supplementary Table 7:

➤ Vector direction decomposition (VDD)

We decompose the directions of all unit vectors going from the center point to the KNNs onto each axis and measure the uniformity of the distribution of the projection points in each dimension. An example in 3D space is shown in R1-Fig. 1b, the red lines with arrows are the unit vectors in the direction from the center point to KNNs. We decompose the unit vectors onto X, Y, Z axes that each vector will project three points on the axes. If the center point is an internal point, it will be surrounded by its KNNs in all directions. Meanwhile, their projection points formed by corresponding unit vectors are distributed uniformly on each axis. Otherwise, the uneven distribution of KNNs will be reflected in that of the projection points on some axes.

We assume that $p_0(x_0^1, x_0^2, \dots, x_0^D) \in \mathbb{R}^D$ is a center point that has D -dimensional features, and its KNNs are p_1, p_2, \dots, p_k . The i th unit vector $v_i \in \mathbb{R}^D$ can be calculated:

$$v_i = \frac{1}{\sum_{j=1}^D (x_i^j - x_0^j)^2} (x_i^1 - x_0^1, x_i^2 - x_0^2, \dots, x_i^D - x_0^D)$$

Thus, we can obtain k ($k > 1$) projection points on the s th axis (the s th dimension) which locate between -1 and 1, and their coordinates are:

$$\left\{ \frac{x_1^s - x_0^s}{\sum_{j=1}^D (x_1^j - x_0^j)^2}, \frac{x_2^s - x_0^s}{\sum_{j=1}^D (x_2^j - x_0^j)^2}, \dots, \frac{x_k^s - x_0^s}{\sum_{j=1}^D (x_k^j - x_0^j)^2} \right\}$$

We sort the coordinates of these k projection points and can obtain $k - 1$ distances between two adjacent points, i.e., $d_1^s, d_2^s, \dots, d_{k-1}^s$ on the s th axis. We define the DCM as:

$$y^s = \frac{1}{k-1} \sum_{j=1}^{k-1} \left(d_j^s - \frac{2}{k-1} \right)^2$$

When $d_1^s, d_2^s, \dots, d_{k-1}^s$ are all equal to $\frac{2}{k-1}$, y^s can reach the minimum 0. Meanwhile, the maximum of DCM on the s th axis can be solved by the following derivation:

$$\begin{aligned} y^s &= \frac{1}{k-1} \sum_{j=1}^{k-1} \left(d_j^s - \frac{2}{k-1} \right)^2 \\ &= \frac{1}{k-1} \left(\sum_{j=1}^{k-1} (d_j^s)^2 - \frac{4}{k-1} \sum_{j=1}^{k-1} d_j^s + \frac{4}{k-1} \right) \\ &\leq \frac{1}{k-1} \left(\left(\sum_{j=1}^{k-1} d_j^s \right)^2 - \frac{4}{k-1} \sum_{j=1}^{k-1} d_j^s + \frac{4}{k-1} \right) \end{aligned}$$

$$= \frac{1}{k-1} \left(\sum_{j=1}^{k-1} d_j^s - \frac{2}{k-1} \right)^2 + \frac{4k-8}{(k-1)^3}$$

Because $\min \sum_{j=1}^{k-1} d_j^s = 0$, $\max \sum_{j=1}^{k-1} d_j^s = 2$, and $k \geq 3$ in a common dataset, we can obtain the maximum:

$$\max y^s = \frac{1}{k-1} \left(2 - \frac{2}{k-1} \right)^2 + \frac{4k-8}{(k-1)^3} = \frac{4k-8}{(k-1)^2}$$

where $\sum_{j=1}^{k-1} d_j^s = 2$ and one of the distances is equal to 2 and the remaining are 0.

Thus, we can normalize y^s as:

$$\widehat{y}^s = \frac{y^s - \min y^s}{\max y^s - \min y^s} = \frac{k-1}{4k-8} \sum_{j=1}^{k-1} \left(d_j^s - \frac{2}{k-1} \right)^2$$

We select the maximum \widehat{y}^s among all the dimensions as the ultimate DCM:

$$DCM = \max\{\widehat{y}^1, \widehat{y}^2, \dots, \widehat{y}^D\}$$

➤ Space division counting (SDC)

In SDC, the original hyperspace is divided into two subspaces in each dimension by taking the center point as the cut point. As shown in R1-Fig. 1c, X, Y and Z axis can be divided into negative and positive parts respectively in 3D space. The proximity of the numbers of points in the two subspaces for each dimension reflect the uniformity of KNNs in the original space. In the i th dimension, we count the number of points in the negative and positive subspaces as n_i^- and n_i^+ . The DCM can be defined as:

$$DCM = \frac{1}{kD} \sum_{i=1}^D |n_i^+ - n_i^-|$$

DCM reaches the minimum 0 when $n_i^+ = n_i^-$ in each dimension, and reaches maximum 1 when $|n_i^+ - n_i^-| = k$ in each dimension.

➤ Orthogonal plane projection (OPP)

We project D -dimensional data onto $\binom{2}{D} = \frac{D(D-1)}{2}$ orthogonal 2D planes. R1-Fig. 1d gives an example in 3D space that can be projected to three orthogonal 2D planes (X-Y, Y-Z and Z-X planes). The data distributions vary in different planes. If a point has a high centrality, it will have a high DCM in each projective plane. By contrast, a point located in the boundary areas of clusters tends to have a low DCM in one or several projective planes. We calculate the DCM in each 2D orthogonal plane:

$$y^i = \frac{k}{4(k-1)\pi^2} \sum_{j=1}^k \left(\alpha_j^i - \frac{2\pi}{k} \right)^2$$

where y^i and α_j^i refer to the DCM and j th angles in the i th 2D planes. The maximum DCM in all planes is selected as the ultimate DCM of the center point:

$$DCM = \max\left\{y^1, y^2, \dots, y^{\frac{D(D-1)}{2}}\right\}$$

R1-Fig. 1. Dimension expansion of CDC and three approaches to calculate DCM in hyperspace by taking 3D as an example. (a) Mapping KNNs to a sphere surface and subdividing the sphere into a network of spherical triangulation that forms multiple solid angles. (b) Illustration of Vector direction decomposition (VDD). Decomposing each vector from the center point to its KNNs onto the orthogonal directions and measuring the uniformity of the projection points. (c) Illustration of Space division counting (SDC). Dividing the whole hyperspace into multiple subspaces via orthogonal 2D planes and counting the number of points in each subspace. (d) Illustration of Orthogonal plane projection (OPP). Projecting KNNs of the center point and itself onto multiple orthogonal 2D planes.

To evaluate the performance of the above three approaches, we compared them to CDC equipped with t-SNE or UMAP on five real-world datasets (Supplementary Table 2). The results show that the highest accuracy of VDD, SDC and OPP are close to that of CDC-tSNE and CDC-UMAP on datasets with relatively low dimensions (R1-Table 1), i.e., Iris, Seeds and Wine. **However, the increase of dimensions weakens their abilities to depict the centrality of points, while CDC-tSNE and CDC-UMAP maintain robust. It reveals that the information loss caused by t-SNE and UMAP is less than the biased measurement using VDD, SDC and OPP.** Therefore, CDC combined with dimension reduction techniques is a feasible strategy to handle high-dimensional datasets.

R1-Table 1. The average validity indexes of the top 20 results obtained by VDD, SDC, OPP, CDC-tSNE and CDC-UMAP to handle high-dimensional datasets.

Iris Dataset (d = 4)						
	ACC	NMI	ARI	F1-score	JI	RI
VDD	0.6910	0.6861	0.6364	0.7336	0.5810	0.8513
SDC	0.7670	0.6596	0.5689	0.7239	0.5682	0.7987
OPP	0.7913	0.7054	0.6555	0.7619	0.6162	0.8519
CDC-tSNE	0.8780	0.8127	0.8161	0.8718	0.7740	0.9218
CDC-UMAP	0.8833	0.7831	0.7351	0.8226	0.6987	0.8828
Seeds Dataset (d = 7)						
	ACC	NMI	ARI	F1-score	JI	RI
VDD	0.6902	0.5792	0.5181	0.6513	0.4870	0.7999
SDC	0.8848	0.7055	0.7348	0.8205	0.6957	0.8839
OPP	0.8164	0.6905	0.6896	0.7815	0.6444	0.8690
CDC-tSNE	0.8900	0.7141	0.7331	0.8206	0.6960	0.8823
CDC-UMAP	0.9071	0.7328	0.7700	0.8446	0.7311	0.8992
Wine Dataset (d = 13)						
	ACC	NMI	ARI	F1-score	JI	RI
VDD	0.6480	0.4362	0.3501	0.6224	0.4542	0.6628
SDC	0.7008	0.5847	0.5043	0.7095	0.5545	0.7455
OPP	0.8885	0.7666	0.7825	0.8522	0.7445	0.9049
CDC-tSNE	0.8980	0.7133	0.7288	0.8181	0.6924	0.8801
CDC-UMAP	0.9298	0.7910	0.7963	0.8645	0.7613	0.9093
PenDigits Dataset (d = 16)						
	ACC	NMI	ARI	F1-score	JI	RI
VDD	0.4779	0.5638	0.3654	0.4353	0.2799	0.8644
SDC	0.5481	0.6826	0.4718	0.5150	0.3474	0.9167
OPP	0.1350	0.0691	0.0051	0.1844	0.1016	0.1706
CDC-tSNE	0.7957	0.8620	0.8048	0.8214	0.6970	0.9688
CDC-UMAP	0.8689	0.8717	0.8201	0.8373	0.7201	0.9689
Digits Dataset (d = 64)						
	ACC	NMI	ARI	F1-score	JI	RI

VDD	0.1181	0.0351	0.0022	0.1827	0.1005	0.1307
SDC	0.2445	0.2124	0.0886	0.2425	0.1381	0.4999
OPP	0.1043	0.0066	0.0000	0.1815	0.0998	0.1072
CDC-tSNE	0.9195	0.9144	0.9050	0.9139	0.8414	0.9838
CDC-UMAP	0.9426	0.9365	0.9304	0.9370	0.8815	0.9879

Reference:

van der Maaten, L. & Hinton, G. Visualizing data using t-SNE. *J. Mach. Learn. Res.* 9, 2579-2605 (2008).

McInnes, L., Healy, J., Saul, N. & Großberger, L. UMAP: uniform manifold approximation and projection. *J. Open Source Softw.* 3, 861 (2018).

According to your suggestion, we have revised the manuscript (line 313-325, page 16-17) as well as the Supplementary Information (Supplement Note 5; Supplement Figure 1; Supplement Table 7).

Comment 2

Other clustering metrics such as NMI and ACC should be reported while comparing CDC to other methods such as recent contrastive clustering; robust continuous clustering, etc. In addition, the currently used metrics are kind of similar to each other.

Thank you for your invaluable suggestion. **We have added the results of NMI and ACC in all the experiments** and complements the equations of six validity indexes used in this paper in Supplementary Note 3 as follows:

To evaluate the clustering performance quantitatively, we adopt six validity indexes Accuracy (ACC), Normalized Mutual Information (NMI), Adjusted Rand Index (ARI), F1-score, Rand Index (RI) and Jaccard Index (JI).

ACC refers to the accuracy rate of the clustering results compared with the true labels. We set the true label vector and the predicted label vector as $\mathbf{l} = (l_1, l_2, \dots, l_n) \in \mathbb{R}^n$ and $\mathbf{r} = (r_1, r_2, \dots, r_n) \in \mathbb{R}^n$ respectively, and the ACC can be defined as

$$ACC = \frac{\sum_{i=1}^n \delta(l_i, \text{map}(r_i))}{n}$$

where $\delta(\cdot)$ denote an indicator function

$$\delta(x, y) = \begin{cases} 1 & \text{if } x = y \\ 0 & \text{otherwise} \end{cases}$$

$\text{map}(\cdot)$ is a mapping function that maps each predicted label to one of the true cluster label. Commonly, the best mapping can be found by using the Kuhn-Munkres or Hungarian Algorithm.

NMI measures the agreement of the predict and true assignments, ignoring permutations. It can be defined as

$$NMI = \frac{\sum_{i=1}^{|L|} \sum_{j=1}^{|R|} |L_i \cap R_j| \log \frac{n |L_i \cap R_j|}{|L_i| |R_j|}}{\sqrt{\left(\sum_{i=1}^{|L|} |L_i| \log \frac{|L_i|}{n} \right) \left(\sum_{j=1}^{|R|} |R_j| \log \frac{|R_j|}{n} \right)}}$$

where L_i denote the point set of the i th cluster that predicted by the algorithm, while R_j denotes the j th cluster of the true labels.

We define the number of point pairs that belong to the same clusters in both of the true and predicted labels as TP , that belong to the different clusters in both of the true and predicted labels as TN , that belong to the same clusters in the true labels but not in the predicted labels as FP , that belong to the same clusters in the predicted labels but not in the true labels as FN . Then we have

$$ARI = \frac{2(TP \cdot TN - FP \cdot FN)}{2(TP \cdot TN - FP \cdot FN) + (FP + FN)(TP + TN + FP + FN)}$$

$$F1\text{-score} = \frac{2TP}{2TP + FP + FN}$$

$$RI = \frac{TP + TN}{TP + TN + FP + FN}$$

$$JI = \frac{TP}{TP + FP + FN}$$

In addition, **we have added the robust continuous clustering (RCC) as a comparative baseline**. We did not select contrastive clustering (CC) since it needs to train the deep neural network model and learn the features from massive natural images. CC is not a generic clustering method but is oriented to image classification exclusively, which is hard to apply to non-image datasets directly. **Nonetheless, we selected two image datasets, i.e., Digits and MNIST, to evaluate the performance of CDC on image clustering.**

In this experiment, we compared CDC with eight baselines, i.e., K-means, DBSCAN, CDP, LGC, Meanshift, Ncut, DensityCut, RCC, on seven high-dimensional datasets as illustrated in R1-Table 2:

R1-Table 2. Description of seven real-world datasets.

Datasets	Type	No. of samples	No. of dimensions	No. of classes
Iris	UCI	150	4	3
Seeds	UCI	210	7	3
Wine	UCI	178	13	3
PenDigits	UCI	10,992	16	10
Digits	Image	5,620	8×8	10
MNIST-train	Image	60,000	28×28	10
Baron-mouse1	scRNA-seq	822	14,878	13

We conducted each clustering algorithm on the high-dimensional (HD) and 2D data after dimensionality reduction by t-SNE and UMAP respectively. The complete results of six

validity indexes, i.e., ACC, NMI, ARI, F1-score, RI and JI are calculated and provided in the supplementary Excel file and the average of the first four indexes are listed in R1-Table 3-6. The results show that **CDC combined with t-SNE and UMAP obtained the top 2 average rank on all of the four indexes and outperformed other baselines with a strong robustness**. Although t-SNE and UMAP caused some degree of information loss, **they did not reduce the accuracy severely and even improved the clustering quality of the baselines comparing with that in HD space** (e.g., K-means, DBSCAN, CDP, Meanshift, DensityCut). **Dimension reduction is actually beneficial when the number of data features is large due to the curse of dimensionality**. Features have a significant difference in value range, which impacts the measurement of similarity (e.g., Euclidean distance) even if the data have been normalized. **t-SNE and UMAP can well preserve the distribution characteristics in HD space when embedding the raw data to 2D space** (in R1-Fig. 2). Although the density heterogeneity and weak connectivity of data distribution after embedding occur in 2D space, CDC is able to tackle the two issues properly. Therefore, we think CDC combined with t-SNE and UMAP are the feasible strategies to handle high-dimensional datasets. **This experiment is also a complementary analysis to your acute insight in comment 1 (“dimensionality reduction could lead to severe information loss”)**.

We implemented all the clustering algorithms on an ordinary desktop computer with a 8-core Intel i7 processor and 64 GB RAM. We presented the results of ACC and counted the running time of each algorithm in HD and 2D space (including the time cost of t-SNE and UMAP) in R1-Fig. 3-4. It demonstrates that CDC can obtain a promising accuracy in an efficient manner and even runs fast than K-means, LGC, Ncut and RCC that handles high-dimensional data directly.

R1-Table 3. The average ACC (\pm standard deviation) of the top 20 clustering results obtained by nine clustering algorithms on seven high-dimensional real-world datasets. (NA means not applicable)

		Iris	Seeds	Wine	PenDigits	Digits	MNIST-train	Baron-mouse1	Rank
K-means	HD	0.437(\pm 0.180)	0.433(\pm 0.192)	0.466(\pm 0.209)	0.601(\pm 0.090)	0.644(\pm 0.116)	0.442(\pm 0.034)	0.392(\pm 0.056)	19.3
	t-SNE	0.402(\pm 0.206)	0.396(\pm 0.220)	0.406(\pm 0.202)	0.608(\pm 0.096)	0.697(\pm 0.160)	0.584(\pm 0.112)	0.435(\pm 0.101)	19.3
	UMAP	0.398(\pm 0.203)	0.400(\pm 0.213)	0.410(\pm 0.216)	0.698(\pm 0.125)	0.726(\pm 0.151)	0.630(\pm 0.130)	0.430(\pm 0.100)	17.6
DBSCAN	HD	0.676(\pm 0.031)	0.570(\pm 0.013)	0.572(\pm 0.008)	0.304(\pm 0.112)	0.180(\pm 0.103)	0.010(\pm 0.000)	0.339(\pm 0.050)	20.0
	t-SNE	0.692(\pm 0.045)	0.413(\pm 0.278)	0.786(\pm 0.085)	0.722(\pm 0.133)	0.754(\pm 0.102)	0.489(\pm 0.223)	0.816(\pm 0.053)	11.4
	UMAP	0.742(\pm 0.101)	0.603(\pm 0.051)	0.787(\pm 0.097)	0.651(\pm 0.109)	0.618(\pm 0.137)	0.553(\pm 0.088)	0.766(\pm 0.041)	12.4
CDP	HD	0.656(\pm 0.097)	0.661(\pm 0.131)	0.620(\pm 0.018)	0.483(\pm 0.022)	0.407(\pm 0.009)	NA	0.379(\pm 0.011)	19.0
	t-SNE	0.437(\pm 0.176)	0.456(\pm 0.185)	0.507(\pm 0.176)	0.767(\pm 0.044)	0.878(\pm 0.047)	NA	0.508(\pm 0.006)	15.1
	UMAP	0.475(\pm 0.147)	0.476(\pm 0.184)	0.522(\pm 0.162)	0.772(\pm 0.046)	0.869(\pm 0.030)	NA	0.482(\pm 0.064)	14.9
LGC	HD	0.667(\pm 0.000)	0.691(\pm 0.068)	0.927(\pm 0.009)	0.685(\pm 0.047)	0.613(\pm 0.117)	NA	0.474(\pm 0.044)	13.7
	t-SNE	0.763(\pm 0.084)	0.919(\pm 0.006)	0.881(\pm 0.025)	0.690(\pm 0.077)	0.890(\pm 0.040)	NA	0.578(\pm 0.023)	9.3
	UMAP	0.820(\pm 0.045)	0.686(\pm 0.046)	0.935(\pm 0.003)	0.522(\pm 0.073)	0.834(\pm 0.055)	NA	0.442(\pm 0.046)	12.3
Meanshift	HD	0.655(\pm 0.043)	0.766(\pm 0.093)	0.862(\pm 0.028)	0.644(\pm 0.077)	0.752(\pm 0.059)	0.125(\pm 0.030)	0.360(\pm 0.044)	14.7
	t-SNE	0.827(\pm 0.078)	0.735(\pm 0.083)	0.892(\pm 0.023)	0.604(\pm 0.118)	0.771(\pm 0.180)	0.684(\pm 0.055)	0.690(\pm 0.063)	9.4
	UMAP	0.788(\pm 0.096)	0.801(\pm 0.111)	0.875(\pm 0.099)	0.777(\pm 0.095)	0.796(\pm 0.104)	0.706(\pm 0.088)	0.402(\pm 0.012)	9.1
Ncut	HD	0.442(\pm 0.182)	0.422(\pm 0.179)	0.509(\pm 0.181)	0.112(\pm 0.001)	0.473(\pm 0.142)	NA	0.418(\pm 0.006)	21.0
	t-SNE	0.410(\pm 0.195)	0.405(\pm 0.211)	0.391(\pm 0.193)	0.689(\pm 0.103)	0.720(\pm 0.155)	NA	0.472(\pm 0.081)	18.9
	UMAP	0.404(\pm 0.204)	0.388(\pm 0.214)	0.393(\pm 0.205)	0.663(\pm 0.134)	0.716(\pm 0.155)	NA	0.416(\pm 0.099)	20.6
DensityCut	HD	0.667(\pm 0.000)	0.800(\pm 0.089)	0.657(\pm 0.054)	0.812(\pm 0.012)	0.894(\pm 0.009)	0.440(\pm 0.000)	0.719(\pm 0.047)	9.3
	t-SNE	0.667(\pm 0.000)	0.921(\pm 0.005)	0.785(\pm 0.141)	0.813(\pm 0.003)	0.908(\pm 0.001)	0.827(\pm0.022)	0.787(\pm 0.000)	5.7
	UMAP	0.729(\pm 0.088)	0.928(\pm0.007)	0.938(\pm0.004)	0.821(\pm 0.022)	0.922(\pm 0.000)	0.716(\pm 0.022)	0.692(\pm 0.000)	4.1
RCC	HD	0.872(\pm 0.146)	0.719(\pm 0.120)	0.613(\pm 0.062)	0.847(\pm 0.045)	0.622(\pm 0.066)	0.113(\pm 0.225)	0.767(\pm 0.094)	10.0
	t-SNE	0.843(\pm 0.096)	0.639(\pm 0.109)	0.593(\pm 0.113)	0.200(\pm 0.022)	0.325(\pm 0.017)	0.123(\pm 0.039)	0.447(\pm 0.045)	16.7
	UMAP	0.865(\pm 0.168)	0.695(\pm 0.130)	0.734(\pm 0.182)	0.417(\pm 0.038)	0.547(\pm 0.026)	0.614(\pm 0.054)	0.306(\pm 0.018)	15.3
CDC	t-SNE	0.878(\pm 0.049)	0.890(\pm 0.029)	0.898(\pm 0.014)	0.796(\pm 0.003)	0.920(\pm 0.005)	0.803(\pm 0.006)	0.820(\pm0.063)	3.4
	UMAP	0.883(\pm0.026)	0.907(\pm 0.007)	0.930(\pm 0.005)	0.869(\pm0.004)	0.943(\pm0.002)	0.697(\pm 0.001)	0.761(\pm 0.046)	3.0

R1-Table 4. The average NMI (\pm standard deviation) of the top 20 clustering results obtained by nine clustering algorithms on seven high-dimensional real-world datasets. (NA means not applicable)

		Iris	Seeds	Wine	PenDigits	Digits	MNIST-train	Baron-mouse1	Rank
K-means	HD	0.602(\pm 0.074)	0.522(\pm 0.067)	0.595(\pm 0.106)	0.704(\pm 0.028)	0.692(\pm 0.038)	0.432(\pm 0.021)	0.468 (\pm 0.042)	19.3
	t-SNE	0.601(\pm 0.094)	0.532(\pm 0.077)	0.528(\pm 0.081)	0.764(\pm 0.043)	0.852(\pm 0.041)	0.680(\pm 0.034)	0.582(\pm 0.035)	15.7
	UMAP	0.602(\pm 0.089)	0.536(\pm 0.073)	0.564(\pm 0.087)	0.807(\pm 0.033)	0.874(\pm 0.043)	0.728(\pm 0.031)	0.577(\pm 0.037)	13.4
DBSCAN	HD	0.729(\pm 0.011)	0.492(\pm 0.010)	0.490(\pm 0.006)	0.408(\pm 0.201)	0.129(\pm 0.149)	0.499(\pm 0.000)	0.332(\pm 0.021)	21.1
	t-SNE	0.727(\pm 0.11)	0.482(\pm 0.095)	0.644(\pm 0.044)	0.813(\pm 0.095)	0.855(\pm 0.046)	0.524(\pm 0.251)	0.722(\pm 0.044)	13.1
	UMAP	0.761(\pm 0.039)	0.548(\pm 0.020)	0.655(\pm 0.054)	0.719(\pm 0.174)	0.768(\pm 0.095)	0.650(\pm 0.044)	0.662(\pm 0.052)	12.3
CDP	HD	0.682(\pm 0.045)	0.571(\pm 0.063)	0.539(\pm 0.028)	0.632(\pm 0.016)	0.461(\pm 0.043)	NA	0.132(\pm 0.024)	20.3
	t-SNE	0.599(\pm 0.069)	0.528(\pm 0.070)	0.543(\pm 0.067)	0.833(\pm 0.039)	0.907(\pm 0.021)	NA	0.580(\pm 0.037)	14.9
	UMAP	0.618(\pm 0.071)	0.544(\pm 0.074)	0.591(\pm 0.084)	0.826(\pm 0.012)	0.910(\pm 0.013)	NA	0.580(\pm 0.030)	12.7
LGC	HD	0.734(\pm 0.000)	0.605(\pm 0.032)	0.772(\pm 0.022)	0.721(\pm 0.029)	0.590(\pm 0.118)	NA	0.167(\pm 0.059)	15.3
	t-SNE	0.735(\pm 0.016)	0.738(\pm 0.007)	0.707(\pm 0.028)	0.822(\pm 0.027)	0.907(\pm 0.019)	NA	0.663(\pm 0.058)	8.1
	UMAP	0.733(\pm 0.023)	0.624(\pm 0.019)	0.800(\pm 0.006)	0.761(\pm 0.023)	0.890(\pm 0.022)	NA	0.564(\pm 0.017)	11.6
Meanshift	HD	0.717(\pm 0.035)	0.607(\pm 0.059)	0.712(\pm 0.039)	0.707(\pm 0.057)	0.729(\pm 0.041)	0.028(\pm 0.065)	0.317(\pm 0.034)	16.3
	t-SNE	0.768(\pm 0.049)	0.632(\pm 0.038)	0.705(\pm 0.031)	0.714(\pm 0.088)	0.841(\pm 0.078)	0.728(\pm 0.026)	0.510(\pm 0.022)	11.9
	UMAP	0.747(\pm 0.030)	0.659(\pm 0.058)	0.739(\pm 0.094)	0.812(\pm 0.047)	0.876(\pm 0.041)	0.799(\pm 0.034)	0.573(\pm 0.012)	7.9
Ncut	HD	0.598(\pm 0.080)	0.512(\pm 0.067)	0.578(\pm 0.112)	0.020(\pm 0.005)	0.584(\pm 0.110)	NA	0.038(\pm 0.021)	22.6
	t-SNE	0.595(\pm 0.086)	0.533(\pm 0.074)	0.515(\pm 0.078)	0.826(\pm 0.047)	0.866(\pm 0.043)	NA	0.596(\pm 0.049)	15.9
	UMAP	0.602(\pm 0.088)	0.532(\pm 0.074)	0.554(\pm 0.082)	0.802(\pm 0.026)	0.869(\pm 0.044)	NA	0.542(\pm 0.037)	16.6
DensityCut	HD	0.734(\pm 0.000)	0.621(\pm 0.072)	0.359(\pm 0.073)	0.800(\pm 0.009)	0.881(\pm 0.009)	0.480(\pm 0.000)	0.571(\pm 0.066)	13.6
	t-SNE	0.734(\pm 0.000)	0.737(\pm 0.009)	0.688(\pm 0.072)	0.876(\pm0.004)	0.927(\pm 0.001)	0.855(\pm0.014)	0.667(\pm 0.000)	4.6
	UMAP	0.733(\pm 0.019)	0.752(\pm0.016)	0.807(\pm0.009)	0.864(\pm 0.010)	0.935(\pm 0.000)	0.825(\pm 0.016)	0.517(\pm 0.000)	5.7
RCC	HD	0.855(\pm0.091)	0.573(\pm 0.065)	0.422(\pm 0.043)	0.868(\pm 0.021)	0.788(\pm 0.050)	0.000(\pm 0.313)	0.038(\pm 0.004)	14.4
	t-SNE	0.773(\pm 0.046)	0.531(\pm 0.032)	0.496(\pm 0.028)	0.523(\pm 0.008)	0.643(\pm 0.006)	0.432(\pm 0.013)	0.568(\pm 0.024)	17.6
	UMAP	0.831(\pm 0.097)	0.642(\pm 0.040)	0.683(\pm 0.076)	0.713(\pm 0.018)	0.757(\pm 0.009)	0.723(\pm 0.019)	0.512(\pm 0.004)	12.1
CDC	t-SNE	0.813(\pm 0.031)	0.714(\pm 0.017)	0.713(\pm 0.015)	0.862(\pm 0.002)	0.914(\pm 0.005)	0.754(\pm 0.006)	0.744(\pm0.042)	4.0
	UMAP	0.783(\pm 0.017)	0.733(\pm 0.008)	0.791(\pm 0.010)	0.872(\pm 0.002)	0.937(\pm0.001)	0.751(\pm 0.001)	0.679(\pm 0.020)	3.1

R1-Table 5. The average ARI (\pm standard deviation) of the top 20 clustering results obtained by nine clustering algorithms on seven high-dimensional real-world datasets. (NA means not applicable)

		Iris	Seeds	Wine	PenDigits	Digits	MNIST-train	Baron-mouse1	Rank
K-means	HD	0.376(\pm 0.159)	0.331(\pm 0.157)	0.400(\pm 0.203)	0.540(\pm 0.053)	0.616(\pm 0.074)	0.312(\pm 0.011)	0.182(\pm 0.022)	17.9
	t-SNE	0.358(\pm 0.188)	0.327(\pm 0.179)	0.322(\pm 0.169)	0.596(\pm 0.067)	0.718(\pm 0.118)	0.562(\pm 0.063)	0.321(\pm 0.075)	17.1
	UMAP	0.355(\pm 0.179)	0.333(\pm 0.174)	0.340(\pm 0.183)	0.670(\pm 0.086)	0.756(\pm 0.118)	0.636(\pm 0.094)	0.307(\pm 0.083)	15.7
DBSCAN	HD	0.574(\pm 0.030)	0.392(\pm 0.011)	0.384(\pm 0.006)	0.137(\pm 0.088)	0.051(\pm 0.085)	0.000(\pm 0.000)	0.117(\pm 0.013)	19.7
	t-SNE	0.597(\pm 0.047)	0.283(\pm 0.222)	0.629(\pm 0.092)	0.682(\pm 0.187)	0.721(\pm 0.108)	0.396(\pm 0.286)	0.719(\pm 0.022)	12.4
	UMAP	0.654(\pm 0.112)	0.440(\pm 0.058)	0.629(\pm 0.101)	0.524(\pm 0.261)	0.481(\pm 0.223)	0.477(\pm 0.106)	0.639(\pm 0.073)	12.9
CDP	HD	0.577(\pm 0.080)	0.492(\pm 0.117)	0.429(\pm 0.028)	0.397(\pm 0.010)	0.198(\pm 0.021)	NA	-0.005(\pm 0.004)	19.0
	t-SNE	0.370(\pm 0.138)	0.344(\pm 0.158)	0.380(\pm 0.148)	0.738(\pm 0.080)	0.871(\pm 0.039)	NA	0.344(\pm 0.010)	14.6
	UMAP	0.392(\pm 0.141)	0.377(\pm 0.162)	0.417(\pm 0.155)	0.742(\pm 0.017)	0.880(\pm 0.020)	NA	0.318(\pm 0.050)	13.6
LGC	HD	0.568(\pm 0.000)	0.529(\pm 0.063)	0.786(\pm 0.025)	0.594(\pm 0.033)	0.512(\pm 0.147)	NA	0.040(\pm 0.017)	15.7
	t-SNE	0.631(\pm 0.054)	0.783(\pm 0.007)	0.710(\pm 0.042)	0.692(\pm 0.074)	0.877(\pm 0.037)	NA	0.481(\pm 0.024)	9.0
	UMAP	0.663(\pm 0.038)	0.560(\pm 0.039)	0.808(\pm 0.007)	0.529(\pm 0.074)	0.840(\pm 0.045)	NA	0.284(\pm 0.041)	11.6
Meanshift	HD	0.561(\pm 0.042)	0.571(\pm 0.088)	0.727(\pm 0.054)	0.521(\pm 0.099)	0.653(\pm 0.080)	0.005(\pm 0.012)	0.115(\pm 0.021)	15.4
	t-SNE	0.735(\pm 0.075)	0.594(\pm 0.074)	0.704(\pm 0.039)	0.533(\pm 0.116)	0.726(\pm 0.170)	0.599(\pm 0.049)	0.507(\pm 0.058)	9.6
	UMAP	0.659(\pm 0.085)	0.635(\pm 0.111)	0.721(\pm 0.135)	0.698(\pm 0.096)	0.764(\pm 0.114)	0.662(\pm 0.073)	0.283(\pm 0.013)	8.7
Ncut	HD	0.366(\pm 0.155)	0.316(\pm 0.146)	0.419(\pm 0.192)	0.000(\pm 0.000)	0.311(\pm 0.100)	NA	0.002(\pm 0.005)	22.1
	t-SNE	0.353(\pm 0.173)	0.329(\pm 0.177)	0.306(\pm 0.163)	0.686(\pm 0.103)	0.736(\pm 0.126)	NA	0.337(\pm 0.055)	17.7
	UMAP	0.353(\pm 0.178)	0.317(\pm 0.173)	0.325(\pm 0.178)	0.638(\pm 0.087)	0.741(\pm 0.124)	NA	0.266(\pm 0.077)	19.0
DensityCut	HD	0.568(\pm 0.000)	0.564(\pm 0.104)	0.302(\pm 0.069)	0.699(\pm 0.017)	0.831(\pm 0.020)	0.305(\pm 0.000)	0.555(\pm 0.083)	12.3
	t-SNE	0.568(\pm 0.000)	0.781(\pm 0.012)	0.622(\pm 0.154)	0.818(\pm 0.005)	0.908(\pm 0.001)	0.806(\pm0.020)	0.680(\pm 0.005)	5.6
	UMAP	0.597(\pm 0.049)	0.797(\pm0.018)	0.816(\pm0.011)	0.791(\pm 0.015)	0.919(\pm 0.000)	0.670(\pm 0.018)	0.502(\pm 0.000)	4.3
RCC	HD	0.812(\pm 0.168)	0.538(\pm 0.116)	0.293(\pm 0.049)	0.790(\pm 0.059)	0.463(\pm 0.132)	0.000(\pm 0.236)	0.101(\pm 0.007)	15.1
	t-SNE	0.738(\pm 0.082)	0.492(\pm 0.091)	0.398(\pm 0.081)	0.168(\pm 0.019)	0.320(\pm 0.017)	0.100(\pm 0.037)	0.311(\pm 0.062)	15.9
	UMAP	0.806(\pm 0.169)	0.574(\pm 0.099)	0.616(\pm 0.158)	0.445(\pm 0.049)	0.558(\pm 0.028)	0.528(\pm 0.057)	0.203(\pm 0.014)	13.3
CDC	t-SNE	0.816(\pm0.043)	0.733(\pm 0.022)	0.729(\pm 0.016)	0.805(\pm 0.001)	0.905(\pm 0.005)	0.782(\pm 0.006)	0.769(\pm0.058)	3.0
	UMAP	0.735(\pm 0.010)	0.770(\pm 0.006)	0.796(\pm 0.012)	0.820(\pm0.005)	0.930(\pm0.001)	0.651(\pm 0.000)	0.635(\pm 0.062)	3.4

R1-Table 6. The average F1-score (\pm standard deviation) of the top 20 clustering results obtained by nine clustering algorithms on seven high-dimensional real-world datasets. (NA means not applicable)

		Iris	Seeds	Wine	PenDigits	Digits	MNIST-train	Baron-mouse1	Rank
K-means	HD	0.476(\pm 0.172)	0.438(\pm 0.175)	0.499(\pm 0.200)	0.579(\pm 0.051)	0.648(\pm 0.069)	0.371(\pm 0.017)	0.357(\pm 0.051)	19.4
	t-SNE	0.452(\pm 0.196)	0.427(\pm 0.192)	0.428(\pm 0.184)	0.628(\pm 0.063)	0.740(\pm 0.112)	0.596(\pm 0.062)	0.428(\pm 0.100)	18.0
	UMAP	0.449(\pm 0.193)	0.434(\pm 0.189)	0.438(\pm 0.196)	0.698(\pm 0.081)	0.776(\pm 0.111)	0.667(\pm 0.092)	0.421(\pm 0.116)	16.6
DBSCAN	HD	0.747(\pm 0.009)	0.603(\pm 0.050)	0.604(\pm 0.009)	0.283(\pm 0.065)	0.218(\pm 0.062)	0.000(\pm 0.000)	0.396(\pm 0.051)	19.1
	t-SNE	0.750(\pm 0.010)	0.365(\pm 0.276)	0.745(\pm 0.053)	0.716(\pm 0.158)	0.752(\pm 0.098)	0.480(\pm 0.234)	0.807(\pm 0.100)	12.3
	UMAP	0.787(\pm 0.056)	0.633(\pm 0.028)	0.754(\pm 0.064)	0.591(\pm 0.210)	0.559(\pm 0.182)	0.552(\pm 0.087)	0.755(\pm 0.050)	12.1
CDP	HD	0.693(\pm 0.081)	0.609(\pm 0.121)	0.651(\pm 0.022)	0.481(\pm 0.008)	0.328(\pm 0.015)	NA	0.414(\pm 0.005)	19.0
	t-SNE	0.476(\pm 0.156)	0.461(\pm 0.166)	0.498(\pm 0.159)	0.763(\pm 0.068)	0.883(\pm 0.036)	NA	0.460(\pm 0.002)	14.9
	UMAP	0.495(\pm 0.152)	0.486(\pm 0.167)	0.530(\pm 0.154)	0.765(\pm 0.016)	0.891(\pm 0.019)	NA	0.428(\pm 0.083)	14.3
LGC	HD	0.746(\pm 0.000)	0.714(\pm 0.020)	0.858(\pm 0.017)	0.633(\pm 0.030)	0.566(\pm 0.113)	NA	0.438(\pm 0.021)	13.1
	t-SNE	0.746(\pm 0.022)	0.854(\pm 0.005)	0.804(\pm 0.029)	0.716(\pm 0.071)	0.888(\pm 0.034)	NA	0.577(\pm 0.019)	9.6
	UMAP	0.775(\pm 0.023)	0.680(\pm 0.032)	0.873(\pm 0.005)	0.557(\pm 0.073)	0.854(\pm 0.042)	NA	0.406(\pm 0.063)	13.1
Meanshift	HD	0.725(\pm 0.060)	0.715(\pm 0.055)	0.810(\pm 0.037)	0.580(\pm 0.081)	0.693(\pm 0.067)	0.186(\pm 0.009)	0.415(\pm 0.040)	14.3
	t-SNE	0.816(\pm 0.054)	0.703(\pm 0.064)	0.803(\pm 0.026)	0.578(\pm 0.102)	0.752(\pm 0.157)	0.636(\pm 0.044)	0.683(\pm 0.070)	10.6
	UMAP	0.782(\pm 0.044)	0.752(\pm 0.077)	0.821(\pm 0.079)	0.730(\pm 0.084)	0.791(\pm 0.099)	0.697(\pm 0.070)	0.391(\pm 0.016)	9.3
Ncut	HD	0.469(\pm 0.171)	0.431(\pm 0.163)	0.529(\pm 0.180)	0.182(\pm 0.000)	0.419(\pm 0.078)	NA	0.430(\pm 0.005)	20.7
	t-SNE	0.450(\pm 0.188)	0.430(\pm 0.191)	0.416(\pm 0.175)	0.714(\pm 0.091)	0.758(\pm 0.118)	NA	0.452(\pm 0.077)	18.3
	UMAP	0.448(\pm 0.194)	0.420(\pm 0.189)	0.425(\pm 0.194)	0.669(\pm 0.084)	0.762(\pm 0.117)	NA	0.397(\pm 0.110)	20.3
DensityCut	HD	0.746(\pm 0.000)	0.720(\pm 0.047)	0.560(\pm 0.025)	0.730(\pm 0.015)	0.848(\pm 0.018)	0.417(\pm 0.000)	0.708(\pm 0.051)	10.4
	t-SNE	0.746(\pm 0.000)	0.853(\pm 0.008)	0.773(\pm 0.077)	0.834(\pm 0.004)	0.917(\pm 0.000)	0.824(\pm0.024)	0.784(\pm 0.003)	5.0
	UMAP	0.756(\pm 0.022)	0.864(\pm0.012)	0.878(\pm0.007)	0.811(\pm 0.014)	0.927(\pm 0.000)	0.712(\pm 0.020)	0.679(\pm 0.000)	4.3
RCC	HD	0.880(\pm0.103)	0.699(\pm 0.070)	0.572(\pm 0.072)	0.811(\pm 0.051)	0.542(\pm 0.108)	0.182(\pm 0.183)	0.761(\pm 0.107)	10.4
	t-SNE	0.818(\pm 0.064)	0.626(\pm 0.098)	0.550(\pm 0.097)	0.193(\pm 0.022)	0.352(\pm 0.018)	0.119(\pm 0.033)	0.416(\pm 0.072)	17.3
	UMAP	0.857(\pm 0.137)	0.682(\pm 0.091)	0.711(\pm 0.141)	0.473(\pm 0.049)	0.586(\pm 0.027)	0.583(\pm 0.048)	0.287(\pm 0.019)	15.0
CDC	t-SNE	0.872(\pm 0.031)	0.821(\pm 0.016)	0.818(\pm 0.011)	0.821(\pm 0.001)	0.914(\pm 0.005)	0.801(\pm 0.006)	0.840(\pm0.072)	3.3
	UMAP	0.823(\pm 0.009)	0.845(\pm 0.005)	0.865(\pm 0.008)	0.837(\pm0.004)	0.937(\pm0.001)	0.695(\pm 0.000)	0.746(\pm 0.048)	3.4

R1-Fig. 2. The clustering results of CDC combined with t-SNE and UMAP on seven real-world datasets.

R1-Fig. 3. Comparison of nine clustering algorithms in ACC and running time on PenDigits and Digits datasets.

R1-Fig. 4. Comparison of nine clustering algorithms in ACC and running time on MNIST-train and Baron-mouse1 datasets.

According to your suggestion, we have revised the manuscript (line 287-312, page 15-16) as well as the Supplementary Information (Supplement Note 3; Supplement Figure 9-11; Supplement Table 2-6).

Comment 3

In Equation 3, the definition of the reachable distances r should be formally given. And more explanations should be made on how the cross-cluster connections are avoided because it seems that every pair of data satisfies the connection requirement.

Thank you for your constructive suggestion, and we have revised the description in the manuscript accordingly (line 478-491, page 25). A supplementary illustration of the reachable distance and connection rule is shown in R1-Fig. 5.

To ensure that the internal points p_1, p_2, \dots, p_s connect to each other within the area restricted by the surrounding boundary points q_1, q_2, \dots, q_{n-s} , we define the minimum distance between the internal point p_i and all boundary points as its reachable distance:

$$r_i = \min_{j=1}^{n-s} d(p_i, q_j)$$

where $d(p_i, q_j)$ is the distance between the two points p_i and q_j (Fig. 5e). Two internal points can be connected as the same cluster if the following association rule is guaranteed:

$$d(p_i, p_j) \leq r_i + r_j$$

where r_i and r_j are the reachable distances of internal points p_i and p_j , respectively (Fig. 5f). On the premise of correct identification of boundary points (except for extreme cases in the situation when the boundary points are identified incompletely), the connections of internal points are constrained in the area defined by the boundary points. If a cross-cluster connection exists between two internal points, there will be boundary points contained in the range defined by their reachable distances, which conflicts with the definition of the reachable distance. Therefore, the internal points of the same cluster can be trapped in the same external contour consisting of boundary points, and the cross-cluster connections will be avoided based on this association rule.

R1-Fig. 5. Illustration of the reachable distance and connection rule.

Comment 4

What is the estimated T_{DCM} on datasets in Table 1? Extra experiments are needed to prove the effectiveness of strategy for determining the T_{DCM} parameter since it is very sensitive. Does the estimation strategy works well on all real-world datasets?

Thank you. **There is no fixed estimated T_{DCM} in Table 1. k and T_{DCM} are generated through a random stratified sampling and vary at different simulations. Meanwhile, we further evaluate the performance of the estimation strategy of T_{DCM} on more**

real-world datasets according to your suggestion.

We previously analyzed the sensitivity using Latin-Hypercube One-factor-At-a-Time (LH-OAT) method, however, the sampling points generated by LH are insufficient to depict the complete range of the two parameters, since LH divides the two-parameter space into $m \times m$ grid and only samples one points in each row or column. Meanwhile, the previous sensitivity indexes in Eq. (1) and (2) are affected by the proportion of the perturbation size to the whole range. Because the potential value range of k is from 0 to n and that of T_{DCM} is from 0 to 1, while the appropriate value ranges of them are $[0, 0.1n]$ and $[0, 0.5]$ respectively according to our experiments. **To improve the accuracy and stability of the analysis approach, we have modified the original LH-OAT method and exerted perturbations with finer granularities.**

$$S_k = \frac{1}{m} \sum_{i=1}^m \left| \frac{\text{ARI}(k_i + \Delta k, T_i) - \text{ARI}(k_i, T_i)}{\text{ARI}(k_i + \Delta k, T_i) + \text{ARI}(k_i, T_i)} / \frac{\Delta k}{2k_i + \Delta k} \right| \quad (1)$$

$$S_T = \frac{1}{m} \sum_{i=1}^m \left| \frac{\text{ARI}(k_i, T_i + \Delta T) - \text{ARI}(k_i, T_i)}{\text{ARI}(k_i, T_i + \Delta T) + \text{ARI}(k_i, T_i)} / \frac{\Delta T}{2T_i + \Delta T} \right| \quad (2)$$

Specifically, as shown in R1-Fig. 6, we firstly divide the value range of the two parameters on average, which generates hundreds of grid cells of the same size. Then we select one sampling point in each cell and exert a positive or negative perturbation to each point. The sizes of perturbations are specified in advance. Only one parameter is varied at a time and the other is fixed. Finally, we calculate the sensitivity indexes that measures the relative change of the clustering quality before and after perturbations.

R1-Fig. 6. Illustration of sensitivity analysis using stratified sampling and random perturbation. (a) Sampling points generated by stratified sampling. (b) Operating random perturbations that could be either positive or negative to each sampling point.

In this experiment, we conduct a stratified sampling that divided the range of each

parameter into 10 equal intervals to generate a 10 by 10 grid and sampled 100 points randomly from the 100 cells. We specify five fixed perturbations for each parameter and conducted each group of experiment five times to avoid randomness in the simulations. We adopted the Adjusted Rand Index (ARI) as the objective function, since it can assess the clustering quality strictly. To ensure that ARI is positive ranging from 0 to 1, we conducted the sampling in the range of k from 0 to $0.1n$ and that of T_{DCM} from 0 to 0.5. Because it makes no sense for the clustering obtained by most of sampling points when k is greater than $0.1n$ or T_{DCM} surpass 0.5 (e.g., all the points are identified as internal points that form a single cluster), which would generate excess invalid results (ARI = 0) that affect the effectiveness of the sensitivity evolution. The estimation results are shown in R1-Table 1:

R1-Table 1. Sensitivity analysis of parameters in CDC using stratified sampling and random perturbation on six synthetic datasets. The shades of blue represent the sensitivity levels and a darker color indicates stronger sensitivity.

Parameter	Perturbation	DS4 n=442	DS5 n=788	DS6 n=1000	DS7 n=1500	DS8 n=3000	DS9 n=6847
K	1	0.0389	0.0363	0.0308	0.0973	0.0338	0.0050
	5	0.0893	0.0719	0.0400	0.1187	0.0723	0.0174
	10	0.1790	0.1864	0.1036	0.1988	0.1678	0.0282
	15	0.2608	0.2341	0.1347	0.2403	0.2177	0.0320
	20	0.3584	0.2865	0.1744	0.2821	0.2734	0.0491
T_{DCM}	0.01	0.0580	0.1198	0.0844	0.2403	0.0916	0.0704
	0.05	0.1836	0.2577	0.1888	0.4276	0.5224	0.2525
	0.10	0.2622	0.3809	0.3209	0.6034	0.8385	0.3629
	0.15	0.3921	0.6968	0.4342	0.9683	0.9286	0.4495
	0.20	0.5428	0.7812	0.5021	0.9878	0.9813	0.5644

Sensitivity levels

According to your suggestion, **we further evaluate the performance of the TIN-based estimation strategy of T_{DCM} on seven real-world datasets** (see Supplementary Table 2). The clustering results and running time are illustrated in R1-Fig. 7. The blue scatter plot represents the top 200 clustering results by under a sampled parameter setting (i.e., k is set using Eq. (8) in manuscript and T_{DCM} is set from 0 to 1 with a 0.02 interval), and the red points refer to the ARI obtained by adaptive CDC (aCDC). It shows aCDC can reach top 25% clustering accuracy on all datasets, and even got the highest ARI on Iris, Wine, PenDigits, Digits. Besides, **the adaptive process to construct a TIN and estimate the T_{DCM} is relatively time-efficient, whose growth speed of running time is slower than that of CDC as the data size increases.** In general, the estimation strategy worked well on the real-world datasets and can provide valid reference to specify T_{DCM} .

R1-Fig. 7. Performance of the adaptive CDC (aCDC) method in clustering accuracy and time efficiency on seven real-world datasets.

According to your suggestion, we have revised the manuscript (line 350-362, page 18; line 364-369, 375-378, page 19; line 407-410, page 21; line 418-424, page 22) as well as the Supplementary Information (Supplement Note 6; Supplement Figure 2).

Comment 5

The definition of sensitivity used in Table 1 should be formally given.

Thank you for your helpful advice. We have added the formally definitions of the modified sensitivity indexes S_k , S_T in Supplement Note 6 as:

$$S_k = \frac{1}{100} \sum_{i=1}^{100} \left| \frac{\text{ARI}(k_i + \Delta k, T_i) - \text{ARI}(k_i, T_i)}{\text{ARI}(k_i + \Delta k, T_i) + \text{ARI}(k_i, T_i)} \right|$$

$$S_T = \frac{1}{100} \sum_{i=1}^{100} \left| \frac{\text{ARI}(k_i, T_i + \Delta T) - \text{ARI}(k_i, T_i)}{\text{ARI}(k_i, T_i + \Delta T) + \text{ARI}(k_i, T_i)} \right|$$

Comment 6

Does the noise robustness of CDC rely on the Local Outlier Factor (LOF) algorithm or itself is noise robust?

Thank you. **The noise robustness of CDC relies on the Local Outlier Factor (LOF) algorithm, and CDC cannot identify noise points by itself.** Actually, besides LOF, other KNN-based noise disposal methods can be integrated with CDC seamlessly to handle data with noise as well. Here, we adopted three KNN-based methods, i.e., Inverse Distance Metric (IDM), Reverse K-Nearest Neighbors (RKNN), and Local Outlier Factor (LOF) to eliminate the noise points as a step of data preprocessing. Since the noise points are usually isolated and far away from the clusters, IDM detects noise by calculating the inverse of the distance sum of the KNNs. RKNN queries the objects that consider a given point as the membership of their KNNs. We further normalize RKNN as a metric ranging from 0 to 1. LOF is formulized as the mean relative reachable density between the center point and its KNN. The formulas are illustrated in R1-Fig. 8.

R1-Fig. 8. Formulas of three KNN-based methods for noise elimination. (a) Inverse Distance Metric (IDM), (b) Reverse k Nearest Neighbors (RKNN), (c) Local Outlier Factor (LOF).

We benchmarked the integrated methods on four synthetic datasets (i.e., DS10-DS13) with clusters of various shapes and noise points. As shown in R1-Fig. 9, **all the three methods can detect the noise effectively in general, but performed slightly differently in the boundary areas of the clusters.** In face of the clusters with uniform densities in DS10, IDM, RKNN and LOF can remove the noise and preserve the cluster points accurately. However, RKNN and LOF performed slightly worse than IDM in detecting noise points near the weakly-connected clusters in DS11, and misidentified low-density boundary points of the spindle-shaped clusters as noise in DS13. **Although, noise elimination partially impacts the overall clustering accuracy, the abovementioned methods are feasible.**

R1-Fig. 9. Clustering results of CDC equipped with three noise elimination methods, (a) IDM, (b) RKNN, (c) LOF, on four synthetic datasets (DS4 to DS7) with noise.

According to your suggestion, we have revised the manuscript (line 326-347, page 17-18) as well as the Supplementary Information (Supplement Figure 6-7).

Comment 7

To further prove the effectiveness of CDC with simply reducing the dimension to two on more complex data, could you conduct CDC on commonly used image dataset like MNIST and CIFAR-10? It could make the method more convincing.

Thank you for your constructive suggestion. **In our opinion, as an unsupervised clustering algorithm, CDC is not dedicated to image classification and cannot capture the underlying features of the images by itself. Image understanding requires the model training process. Simply transforming a 2D image to a one-dimensional vector is unable to represent its underlying features and scenes. CDC can process the data whose features have been extracted, rather than the semi-structured or unstructured raw data.** To achieve promising outcomes in complex image classification task, it needs to combine with image feature extraction methods. Consequently, it is difficult to measure the respective contributions of CDC and feature extraction methods on clustering accuracy when adopting the extracted features as the inputs of CDC. Besides, **the enhanced experiments on cell datasets (i.e., scRNA-seq and CyTOF) have validated the robustness of CDC on handling high-dimensional data and the effectiveness of dimension reduction using UMAP and t-SNE.**

According to your suggestion, we have added two image datasets, i.e., MNIST (28×28) and Digits (8×8), but not CIFAR-10 to testify the performance of CDC combined with dimension reduction techniques. The CIFAR-10 dataset consists of 60000 32×32 colour images in 10 classes, containing natural objects with complex features. Conducting image classification on CIFAR-10 requires intensive model training to learn the underlying features. In contrast, MNIST and Digits are grayscale images of handwritten digits with simple structures and features. The shapes of handwritten digits can be represented by transforming MNIST and Digits to one-dimensional vectors. We used the one-dimensional vectors as the inputs of CDC in this experiment. The final clustering results are shown in R1-Table 3-6. **CDC combined with t-SNE and UMAP obtained promising results on MNIST and Digits. It got the highest clustering accuracy on Digits and second on MNIST, and both of them surpass 0.8.** Nonetheless, CDC may need to combine with other image feature extraction methods to handle complex images by embedding the features to a one-dimensional form. In general, **CDC is an effective and generic unsupervised method**, which can be applied to the cell type identification, speaker recognition, geographic object clustering and so on.

Reviewer #2 (Expertise: Clustering with experience relevant to the biosciences):

Summary

The authors present a new clustering algorithm called CDC designed for data with heterogeneous density and low connectivity. CDC is based on the observation that the directions from boundary points to their k nearest neighbours (k -NNs) vary much less than the directions from inner points to their k -NNs. To exploit this observation, the normalised Direction Centrality Measure (DCM) is defined, which, for each node, returns a value close to 0 if the k -NNs are distributed in very diverse directions and a value close to 1 if they are distributed in few directions. Subsequently, boundary points are defined as data points with DCM larger than a threshold T and inner points are connected into clusters via the association rule $p_i \sim p_j \Leftrightarrow d_{\{ij\}} \leq r_i + r_j$, where r_i is the distance from p_i to the closest boundary point. Finally, each boundary point is assigned to the cluster of the nearest inner point. Strategies for automatically selecting the two hyper-parameters k and T are presented, and CDC is compared against existing approaches on various datasets.

Thank you for your careful reviews and comments.

Overall evaluation

The idea behind the algorithm is simple and elegant and the results are very promising. However, I feel that the both the algorithm itself and the way it is presented in the paper could still be improved considerably (see comments below). Moreover, the classical "Introduction", "Results", "Discussion", "Methods" foreseen by Nature Communications doesn't really fit the paper: The main contribution/result of the paper - namely the clustering algorithm itself - is now presented in the "Methods" section, which, if accepted, would be typeset in small font at the very end of the paper. On the other hand, the experimental results reported in the "Results" section are really only proofs of concept that the algorithm might be valuable for the community. Of course, this is not really to blame on the authors, since they only followed the Nature Communications guidelines. However, it indicates that dedicated computer science venues (eg., KDD, AAAI, ICML, Pattern Recognition, TPAMI) might be better fits for the manuscript.

We respect your constructive comments and kindly suggestion. We have carefully revised the sections of Results, Methods, and Supplementary Information by adding more experiments, analysis and explanations accordingly. We believe this work is suitable for the aim and scope of Nature Communications. As a multidisciplinary journal, Nature Communications publishes high-quality research from all areas. Our algorithm CDC is a generic and novel method, which **can be applied to many fields**, e.g., scRNA-seq clustering, speaker recognition, species spatial pattern exploration and so on, besides computer science classical applications (e.g., computer vision, signal or image processing). So, we insist our choice for potential publication on Nature Communications.

Comment 1

The main drawback of CDC is that it is designed for 2-dim data only. It would be a much more valuable contribution if it could also handle n -dim data. My intuition is that the algorithm could be generalized to the n -dim case. Two things would have to be changed:

(1) The definition of DCM. (2) The automated parameter selection routine for the threshold T . For (1), the authors would have to come up with a k -NN direction diversity measure that generalizes to n -dim data. Here's an idea: Let $n_1, \dots, n_j, \dots, n_k$ be the k -NNs of a point p_i and $d_j = p_i - r_j$ be the corresponding directions. Why not defining the diversity measure as the mean correlation coefficient across all pairs of directions (i.e., as $\frac{1}{\binom{k}{2}} \sum_{j=1}^{k-1} \sum_{l=j+1}^k \frac{d_j^T d_l}{\|d_j\| \|d_l\|}$)? Of course, this doesn't really generalize DCM to the n -dim case, but it formalizes the same underlying idea. W.r.t. (2), the only thing I can think of at the moment is to determine T empirically via large-scale benchmarking on tons of datasets. But there might be a cleverer way.

Thank you for your kindly comment and suggestion. We have carefully considered your idea that can be formalized as:

$$DCM = \frac{1}{\binom{k}{2}} \sum_{j=1}^{k-1} \sum_{l=j+1}^k \frac{d_j^T d_l}{\|d_j\| \cdot \|d_l\|} \quad (1)$$

Eq. (1) measures the average of the total cosines of the angles between any two pairs of vectors. However, **we think that it is unable to depict the direction centrality and distinguish different distributions appropriately. Cosine correlation coefficient measures the similarity of two directional vectors in pairs, but cannot capture the distribution characteristics of a series of points as a whole in high-dimensional space** (i.e., agglomerate or disperse patterns). It may generate the same DCMs under different distributions. Two examples in 2D and 3D space are given in R2-Fig. 1. For the two different data distributions in 2D space, the centrality of the center point in R2-Fig. 1a is higher than that in R2-Fig. 1b, however, their DCMs are the same according to Eq. (1). The same situation occurs in the two different 3D data distributions (R1-Fig. 1c-d).

Furthermore, a valid centrality metric should be formalized as the deviation to an intuitive benchmark having a priori centrality (i.e., distribution). While the DCM via **mean correlation coefficient in Eq. (1) lacks such a benchmark for comparison, thus cannot evaluate the level of centrality**. Although the mean cosine is between -1 and 1, the minimum and maximum of Eq. (1) cannot reach to the two theoretical extremes. Its value range varies with k under an unexplored pattern. Besides, the highest centrality does not monotonically increase/decrease with the DCM in Eq. (1), i.e., a higher DCM does not necessarily indicate a higher/lower centrality. As a result, the DCM in Eq. (1) is hard to be normalized and specified by users.

In terms of the adaptive method for specifying the threshold T_{DCM} , we have **enhanced the explanation of its work mechanism and expanded the experiments on more real-world datasets to validate its effectiveness** (please see the response of your comment 5).

R2-Fig. 1. Examples of the direction centrality metric using mean correlation coefficient.

Nonetheless, we were inspired by your idea and tried our best to redefine DCM for calculating directional centrality in hyperspace. Here, we use the same underlying idea of CDC and propose three alternative approaches i.e., Vector direction decomposition (VDD), Space division counting (SDC), Orthogonal plane projection (OPP), that can handle high-dimensional data directly.

To evaluate the performance of the above three approaches, we compared them to CDC equipped with t-SNE or UMAP on five real-world datasets (Supplementary Table 2). The highest accuracy of VDD, SDC and OPP are close to that of CDC-tSNE and CDC-UMAP on datasets with relatively low dimensions (R2-Table 1), i.e., Iris, Seeds and Wine. However, the increase of dimensions weakens their abilities to depict the centrality of points, while CDC-tSNE and CDC-UMAP maintain robust. It reveals that the information loss caused by t-SNE and UMAP is less than the biased measurement using VDD, SDC and OPP. Therefore, CDC combined with dimension reduction techniques is a feasible strategy to handle high-dimensional datasets.

R2-Table 1: The average validity indexes of the top 20 results obtained by VDD, SDC, OPP, CDC-tSNE and CDC-UMAP to handle high-dimensional datasets.

Iris Dataset (d = 4)						
	ACC	NMI	ARI	F1-score	Jl	RI
VDD	0.6910	0.6861	0.6364	0.7336	0.5810	0.8513
SDC	0.7670	0.6596	0.5689	0.7239	0.5682	0.7987
OPP	0.7913	0.7054	0.6555	0.7619	0.6162	0.8519
CDC-tSNE	0.8780	0.8127	0.8161	0.8718	0.7740	0.9218
CDC-UMAP	0.8833	0.7831	0.7351	0.8226	0.6987	0.8828
Seeds Dataset (d = 7)						
	ACC	NMI	ARI	F1-score	Jl	RI
VDD	0.6902	0.5792	0.5181	0.6513	0.4870	0.7999
SDC	0.8848	0.7055	0.7348	0.8205	0.6957	0.8839
OPP	0.8164	0.6905	0.6896	0.7815	0.6444	0.8690
CDC-tSNE	0.8900	0.7141	0.7331	0.8206	0.6960	0.8823
CDC-UMAP	0.9071	0.7328	0.7700	0.8446	0.7311	0.8992
Wine Dataset (d = 13)						
	ACC	NMI	ARI	F1-score	Jl	RI
VDD	0.6480	0.4362	0.3501	0.6224	0.4542	0.6628
SDC	0.7008	0.5847	0.5043	0.7095	0.5545	0.7455
OPP	0.8885	0.7666	0.7825	0.8522	0.7445	0.9049
CDC-tSNE	0.8980	0.7133	0.7288	0.8181	0.6924	0.8801
CDC-UMAP	0.9298	0.7910	0.7963	0.8645	0.7613	0.9093
PenDigits Dataset (d = 16)						
	ACC	NMI	ARI	F1-score	Jl	RI
VDD	0.4779	0.5638	0.3654	0.4353	0.2799	0.8644
SDC	0.5481	0.6826	0.4718	0.5150	0.3474	0.9167
OPP	0.1350	0.0691	0.0051	0.1844	0.1016	0.1706
CDC-tSNE	0.7957	0.8620	0.8048	0.8214	0.6970	0.9688
CDC-UMAP	0.8689	0.8717	0.8201	0.8373	0.7201	0.9689
Digits Dataset (d = 64)						
	ACC	NMI	ARI	F1-score	Jl	RI
VDD	0.1181	0.0351	0.0022	0.1827	0.1005	0.1307
SDC	0.2445	0.2124	0.0886	0.2425	0.1381	0.4999
OPP	0.1043	0.0066	0.0000	0.1815	0.0998	0.1072
CDC-tSNE	0.9195	0.9144	0.9050	0.9139	0.8414	0.9838
CDC-UMAP	0.9426	0.9365	0.9304	0.9370	0.8815	0.9879

The details about the methodology and experimental results are added in Supplementary Note 5 as follows:

- Vector direction decomposition (VDD)

We decompose the directions of all unit vectors going from the center point to the KNNs onto each axis and measure the uniformity of the distribution of the projection points in each dimension. An example in 3D space is shown in R2-Fig. 2b, the red lines with arrows are the unit vectors in the direction from the center point to KNNs. We decompose the unit vectors onto X, Y, Z axes that each vector will project three points on the axes. If the center point is an internal point, it will be surrounded by its KNNs in all directions. Meanwhile, their projection points formed by corresponding unit vectors are distributed uniformly on each axis. Otherwise, the uneven distribution of KNNs will be reflected in that of the projection points on some axes.

We assume that $p_0(x_0^1, x_0^2, \dots, x_0^D) \in \mathbb{R}^D$ is a center point that has D -dimensional features, and its KNNs are p_1, p_2, \dots, p_k . The i th unit vector $v_i \in \mathbb{R}^D$ can be calculated:

$$v_i = \frac{1}{\sum_{j=1}^D (x_i^j - x_0^j)^2} (x_i^1 - x_0^1, x_i^2 - x_0^2, \dots, x_i^D - x_0^D)$$

Thus, we can obtain k ($k > 1$) projection points on the s th axis (the s th dimension) which locate between -1 and 1, and their coordinates are:

$$\left\{ \frac{x_1^s - x_0^s}{\sum_{j=1}^D (x_1^j - x_0^j)^2}, \frac{x_2^s - x_0^s}{\sum_{j=1}^D (x_2^j - x_0^j)^2}, \dots, \frac{x_k^s - x_0^s}{\sum_{j=1}^D (x_k^j - x_0^j)^2} \right\}$$

We sort the coordinates of these k projection points and can obtain $k - 1$ distances between two adjacent points, i.e., $d_1^s, d_2^s, \dots, d_{k-1}^s$ on the s th axis. We define the DCM as:

$$y^s = \frac{1}{k-1} \sum_{j=1}^{k-1} \left(d_j^s - \frac{2}{k-1} \right)^2$$

When $d_1^s, d_2^s, \dots, d_{k-1}^s$ are all equal to $\frac{2}{k-1}$, y^s can reach the minimum 0. Meanwhile, the maximum of DCM on the s th axis can be solved by the following derivation:

$$\begin{aligned} y^s &= \frac{1}{k-1} \sum_{j=1}^{k-1} \left(d_j^s - \frac{2}{k-1} \right)^2 \\ &= \frac{1}{k-1} \left(\sum_{j=1}^{k-1} (d_j^s)^2 - \frac{4}{k-1} \sum_{j=1}^{k-1} d_j^s + \frac{4}{k-1} \right) \\ &\leq \frac{1}{k-1} \left(\left(\sum_{j=1}^{k-1} d_j^s \right)^2 - \frac{4}{k-1} \sum_{j=1}^{k-1} d_j^s + \frac{4}{k-1} \right) \\ &= \frac{1}{k-1} \left(\sum_{j=1}^{k-1} d_j^s - \frac{2}{k-1} \right)^2 + \frac{4k-8}{(k-1)^3} \end{aligned}$$

Because $\min \sum_{j=1}^{k-1} d_j^s = 0$, $\max \sum_{j=1}^{k-1} d_j^s = 2$, and $k \geq 3$ in a common dataset, we can obtain the maximum:

$$\max y^s = \frac{1}{k-1} \left(2 - \frac{2}{k-1} \right)^2 + \frac{4k-8}{(k-1)^3} = \frac{4k-8}{(k-1)^2}$$

where $\sum_{j=1}^{k-1} d_j^s = 2$ and one of the distances is equal to 2 and the remaining are 0.

Thus, we can normalize y^s as:

$$\widehat{y}^s = \frac{y^s - \min y^s}{\max y^s - \min y^s} = \frac{k-1}{4k-8} \sum_{j=1}^{k-1} \left(d_j^s - \frac{2}{k-1} \right)^2$$

We select the maximum \widehat{y}^s among all the dimensions as the ultimate DCM:

$$DCM = \max\{\widehat{y}^1, \widehat{y}^2, \dots, \widehat{y}^D\}$$

➤ Space division counting (SDC)

In SDC, the original hyperspace is divided into two subspaces in each dimension by taking the center point as the cut point. As shown in R2-Fig. 2c, X, Y and Z axis can be divided into negative and positive parts respectively in 3D space. The proximity of the numbers of points in the two subspaces for each dimension reflect the uniformity of KNNs in the original space. In the i th dimension, we count the number of points in the negative and positive subspaces as n_i^- and n_i^+ . The DCM can be defined as:

$$DCM = \frac{1}{kD} \sum_{i=1}^D |n_i^+ - n_i^-|$$

DCM reaches the minimum 0 when $n_i^+ = n_i^-$ in each dimension, and reaches maximum 1 when $|n_i^+ - n_i^-| = k$ in each dimension.

➤ Orthogonal plane projection (OPP)

We project D -dimensional data onto $\binom{D}{2} = \frac{D(D-1)}{2}$ orthogonal 2D planes. R2-Fig. 2d gives an example in 3D space that can be projected to three orthogonal 2D planes (X-Y, Y-Z and Z-X planes). The data distributions vary in different planes. If a point has a high centrality, it will have a high DCM in each projective plane. By contrast, a point located in the boundary areas of clusters tends to have a low DCM in one or several projective planes. We calculate the DCM in each 2D orthogonal plane:

$$y^i = \frac{k}{4(k-1)\pi^2} \sum_{j=1}^k \left(\alpha_j^i - \frac{2\pi}{k} \right)^2$$

where y^i and α_j^i refer to the DCM and j th angles in the i th 2D planes. The maximum DCM in all planes is selected as the ultimate DCM of the center point:

$$DCM = \max\left\{y^1, y^2, \dots, y^{\frac{D(D-1)}{2}}\right\}$$

R2-Fig. 2. Dimension expansion of CDC and three approaches to calculate DCM in hyperspace by taking 3D as an example. (a) Mapping KNNs to a sphere surface and subdividing the sphere into a network of spherical triangulation that forms multiple solid angles. (b) Illustration of Vector direction decomposition (VDD). Decomposing each vector from the center point to its KNNs onto the orthogonal directions and measuring the uniformity of the projection points. (c) Illustration of Space division counting (SDC). Dividing the whole hyperspace into multiple subspaces via orthogonal 2D planes and counting the number of points in each subspace. (d) Illustration of Orthogonal plane projection (OPP). Projecting KNNs of the center point and itself onto multiple orthogonal 2D planes.

According to your suggestion, we have revised the manuscript (line 313-325, page 16-17) as well as the Supplementary Information (Supplement Note 5; Supplement Figure 1).

Comment 2

Since CDC can only cope with 2-dim data, the authors transformed the real-world datasets to 2-dim space via dimensionality reduction techniques before comparing CDC to other clustering algorithms. This, however, is not a fair experimental setup, because the competitors are not restricted to 2-dim data. So while the experiments indeed show that CDC outperforms the competitors on the 2-dim datasets, it might be the case that the competitors perform better on the original datasets or higher dimensional transformations. I hence suggest reconsider the setup of their experiments and run the competitors also on higher dimensional versions of the datasets.

Thank you for your constructive suggestion. We agree with you that it may lead to an unfair comparison without performing the competitors in the original high-dimensional space. Therefore, we redesigned the experiment by comparing CDC with eight baselines, i.e., K-means, DBSCAN, CDP, LGC, Meanshift, Ncut, DensityCut, RCC, on seven high-dimensional datasets as illustrated in R2-Table 2. **We conducted each clustering algorithm on the high-dimensional (HD) and 2D data after dimensionality reduction by t-SNE and UMAP respectively.**

R2-Table 2. Description of seven real-world datasets.

Datasets	Type	No. of samples	No. of dimensions	No. of classes
Iris	UCI	150	4	3
Seeds	UCI	210	7	3
Wine	UCI	178	13	3
PenDigits	UCI	10,992	16	10
Digits	Image	5,620	8×8	10
MNIST-train	Image	60,000	28×28	10
Baron-mouse1	scRNA-seq	822	14,878	13

The complete results of six validity indexes, i.e., ACC, NMI, ARI, F1-score, RI and JI are calculated and provided in the supplementary Excel file and the average of the first four indexes are listed in R2-Table 3-6. The results show that **CDC combined with t-SNE and UMAP obtained the top 2 average rank on all of the four indexes and outperformed other baselines with a strong robustness.** Although t-SNE and UMAP caused some degree of information loss, **they did not reduce the accuracy severely and even improved the clustering quality of the baselines comparing with that in HD space (e.g., K-means, DBSCAN, CDP, Meanshift, DensityCut).** **Dimension reduction is actually beneficial when the number of data features is large due to the curse of dimensionality.** Features have a significant difference in value range, which impacts the measurement of similarity (e.g., Euclidean distance) even if the data have been normalized. **t-SNE and UMAP can well preserve the distribution characteristics in HD space when embedding the raw data to 2D space (R2-Fig. 3).** Although the density heterogeneity and weak connectivity of data distribution after embedding occur in 2D space, CDC is able to tackle the two issues properly. Therefore, we think CDC combined with t-SNE and UMAP are the feasible strategies to handle high-dimensional datasets.

We implemented all the clustering algorithms on an ordinary desktop computer with a 8-core Intel i7 processor and 64 GB RAM. We presented the results of ACC and counted the running time of each algorithm in HD and 2D space (including the time cost of t-SNE and UMAP) in R2-Fig. 4-5. It demonstrates that CDC can obtain a promising accuracy in an efficient manner and even runs fast than K-means, LGC, Ncut and RCC that handles high-dimensional data directly.

R2-Table 3. The average ACC (\pm standard deviation) of the top 20 clustering results obtained by nine clustering algorithms on seven high-dimensional real-world datasets. (NA means not applicable)

		Iris	Seeds	Wine	PenDigits	Digits	MNIST-train	Baron-mouse1	Rank
K-means	HD	0.437(\pm 0.180)	0.433(\pm 0.192)	0.466(\pm 0.209)	0.601(\pm 0.090)	0.644(\pm 0.116)	0.442(\pm 0.034)	0.392(\pm 0.056)	19.3
	t-SNE	0.402(\pm 0.206)	0.396(\pm 0.220)	0.406(\pm 0.202)	0.608(\pm 0.096)	0.697(\pm 0.160)	0.584(\pm 0.112)	0.435(\pm 0.101)	19.3
	UMAP	0.398(\pm 0.203)	0.400(\pm 0.213)	0.410(\pm 0.216)	0.698(\pm 0.125)	0.726(\pm 0.151)	0.630(\pm 0.130)	0.430(\pm 0.100)	17.6
DBSCAN	HD	0.676(\pm 0.031)	0.570(\pm 0.013)	0.572(\pm 0.008)	0.304(\pm 0.112)	0.180(\pm 0.103)	0.010(\pm 0.000)	0.339(\pm 0.050)	20.0
	t-SNE	0.692(\pm 0.045)	0.413(\pm 0.278)	0.786(\pm 0.085)	0.722(\pm 0.133)	0.754(\pm 0.102)	0.489(\pm 0.223)	0.816(\pm 0.053)	11.4
	UMAP	0.742(\pm 0.101)	0.603(\pm 0.051)	0.787(\pm 0.097)	0.651(\pm 0.109)	0.618(\pm 0.137)	0.553(\pm 0.088)	0.766(\pm 0.041)	12.4
CDP	HD	0.656(\pm 0.097)	0.661(\pm 0.131)	0.620(\pm 0.018)	0.483(\pm 0.022)	0.407(\pm 0.009)	NA	0.379(\pm 0.011)	19.0
	t-SNE	0.437(\pm 0.176)	0.456(\pm 0.185)	0.507(\pm 0.176)	0.767(\pm 0.044)	0.878(\pm 0.047)	NA	0.508(\pm 0.006)	15.1
	UMAP	0.475(\pm 0.147)	0.476(\pm 0.184)	0.522(\pm 0.162)	0.772(\pm 0.046)	0.869(\pm 0.030)	NA	0.482(\pm 0.064)	14.9
LGC	HD	0.667(\pm 0.000)	0.691(\pm 0.068)	0.927(\pm 0.009)	0.685(\pm 0.047)	0.613(\pm 0.117)	NA	0.474(\pm 0.044)	13.7
	t-SNE	0.763(\pm 0.084)	0.919(\pm 0.006)	0.881(\pm 0.025)	0.690(\pm 0.077)	0.890(\pm 0.040)	NA	0.578(\pm 0.023)	9.3
	UMAP	0.820(\pm 0.045)	0.686(\pm 0.046)	0.935(\pm 0.003)	0.522(\pm 0.073)	0.834(\pm 0.055)	NA	0.442(\pm 0.046)	12.3
Meanshift	HD	0.655(\pm 0.043)	0.766(\pm 0.093)	0.862(\pm 0.028)	0.644(\pm 0.077)	0.752(\pm 0.059)	0.125(\pm 0.030)	0.360(\pm 0.044)	14.7
	t-SNE	0.827(\pm 0.078)	0.735(\pm 0.083)	0.892(\pm 0.023)	0.604(\pm 0.118)	0.771(\pm 0.180)	0.684(\pm 0.055)	0.690(\pm 0.063)	9.4
	UMAP	0.788(\pm 0.096)	0.801(\pm 0.111)	0.875(\pm 0.099)	0.777(\pm 0.095)	0.796(\pm 0.104)	0.706(\pm 0.088)	0.402(\pm 0.012)	9.1
Ncut	HD	0.442(\pm 0.182)	0.422(\pm 0.179)	0.509(\pm 0.181)	0.112(\pm 0.001)	0.473(\pm 0.142)	NA	0.418(\pm 0.006)	21.0
	t-SNE	0.410(\pm 0.195)	0.405(\pm 0.211)	0.391(\pm 0.193)	0.689(\pm 0.103)	0.720(\pm 0.155)	NA	0.472(\pm 0.081)	18.9
	UMAP	0.404(\pm 0.204)	0.388(\pm 0.214)	0.393(\pm 0.205)	0.663(\pm 0.134)	0.716(\pm 0.155)	NA	0.416(\pm 0.099)	20.6
DensityCut	HD	0.667(\pm 0.000)	0.800(\pm 0.089)	0.657(\pm 0.054)	0.812(\pm 0.012)	0.894(\pm 0.009)	0.440(\pm 0.000)	0.719(\pm 0.047)	9.3
	t-SNE	0.667(\pm 0.000)	0.921(\pm 0.005)	0.785(\pm 0.141)	0.813(\pm 0.003)	0.908(\pm 0.001)	0.827(\pm0.022)	0.787(\pm 0.000)	5.7
	UMAP	0.729(\pm 0.088)	0.928(\pm0.007)	0.938(\pm0.004)	0.821(\pm 0.022)	0.922(\pm 0.000)	0.716(\pm 0.022)	0.692(\pm 0.000)	4.1
RCC	HD	0.872(\pm 0.146)	0.719(\pm 0.120)	0.613(\pm 0.062)	0.847(\pm 0.045)	0.622(\pm 0.066)	0.113(\pm 0.225)	0.767(\pm 0.094)	10.0
	t-SNE	0.843(\pm 0.096)	0.639(\pm 0.109)	0.593(\pm 0.113)	0.200(\pm 0.022)	0.325(\pm 0.017)	0.123(\pm 0.039)	0.447(\pm 0.045)	16.7
	UMAP	0.865(\pm 0.168)	0.695(\pm 0.130)	0.734(\pm 0.182)	0.417(\pm 0.038)	0.547(\pm 0.026)	0.614(\pm 0.054)	0.306(\pm 0.018)	15.3
CDC	t-SNE	0.878(\pm 0.049)	0.890(\pm 0.029)	0.898(\pm 0.014)	0.796(\pm 0.003)	0.920(\pm 0.005)	0.803(\pm 0.006)	0.820(\pm0.063)	3.4
	UMAP	0.883(\pm0.026)	0.907(\pm 0.007)	0.930(\pm 0.005)	0.869(\pm0.004)	0.943(\pm0.002)	0.697(\pm 0.001)	0.761(\pm 0.046)	3.0

R2-Table 4. The average NMI (\pm standard deviation) of the top 20 clustering results obtained by nine clustering algorithms on seven high-dimensional real-world datasets. (NA means not applicable)

		Iris	Seeds	Wine	PenDigits	Digits	MNIST-train	Baron-mouse1	Rank
K-means	HD	0.602(\pm 0.074)	0.522(\pm 0.067)	0.595(\pm 0.106)	0.704(\pm 0.028)	0.692(\pm 0.038)	0.432(\pm 0.021)	0.468 (\pm 0.042)	19.3
	t-SNE	0.601(\pm 0.094)	0.532(\pm 0.077)	0.528(\pm 0.081)	0.764(\pm 0.043)	0.852(\pm 0.041)	0.680(\pm 0.034)	0.582(\pm 0.035)	15.7
	UMAP	0.602(\pm 0.089)	0.536(\pm 0.073)	0.564(\pm 0.087)	0.807(\pm 0.033)	0.874(\pm 0.043)	0.728(\pm 0.031)	0.577(\pm 0.037)	13.4
DBSCAN	HD	0.729(\pm 0.011)	0.492(\pm 0.010)	0.490(\pm 0.006)	0.408(\pm 0.201)	0.129(\pm 0.149)	0.499(\pm 0.000)	0.332(\pm 0.021)	21.1
	t-SNE	0.727(\pm 0.11)	0.482(\pm 0.095)	0.644(\pm 0.044)	0.813(\pm 0.095)	0.855(\pm 0.046)	0.524(\pm 0.251)	0.722(\pm 0.044)	13.1
	UMAP	0.761(\pm 0.039)	0.548(\pm 0.020)	0.655(\pm 0.054)	0.719(\pm 0.174)	0.768(\pm 0.095)	0.650(\pm 0.044)	0.662(\pm 0.052)	12.3
CDP	HD	0.682(\pm 0.045)	0.571(\pm 0.063)	0.539(\pm 0.028)	0.632(\pm 0.016)	0.461(\pm 0.043)	NA	0.132(\pm 0.024)	20.3
	t-SNE	0.599(\pm 0.069)	0.528(\pm 0.070)	0.543(\pm 0.067)	0.833(\pm 0.039)	0.907(\pm 0.021)	NA	0.580(\pm 0.037)	14.9
	UMAP	0.618(\pm 0.071)	0.544(\pm 0.074)	0.591(\pm 0.084)	0.826(\pm 0.012)	0.910(\pm 0.013)	NA	0.580(\pm 0.030)	12.7
LGC	HD	0.734(\pm 0.000)	0.605(\pm 0.032)	0.772(\pm 0.022)	0.721(\pm 0.029)	0.590(\pm 0.118)	NA	0.167(\pm 0.059)	15.3
	t-SNE	0.735(\pm 0.016)	0.738(\pm 0.007)	0.707(\pm 0.028)	0.822(\pm 0.027)	0.907(\pm 0.019)	NA	0.663(\pm 0.058)	8.1
	UMAP	0.733(\pm 0.023)	0.624(\pm 0.019)	0.800(\pm 0.006)	0.761(\pm 0.023)	0.890(\pm 0.022)	NA	0.564(\pm 0.017)	11.6
Meanshift	HD	0.717(\pm 0.035)	0.607(\pm 0.059)	0.712(\pm 0.039)	0.707(\pm 0.057)	0.729(\pm 0.041)	0.028(\pm 0.065)	0.317(\pm 0.034)	16.3
	t-SNE	0.768(\pm 0.049)	0.632(\pm 0.038)	0.705(\pm 0.031)	0.714(\pm 0.088)	0.841(\pm 0.078)	0.728(\pm 0.026)	0.510(\pm 0.022)	11.9
	UMAP	0.747(\pm 0.030)	0.659(\pm 0.058)	0.739(\pm 0.094)	0.812(\pm 0.047)	0.876(\pm 0.041)	0.799(\pm 0.034)	0.573(\pm 0.012)	7.9
Ncut	HD	0.598(\pm 0.080)	0.512(\pm 0.067)	0.578(\pm 0.112)	0.020(\pm 0.005)	0.584(\pm 0.110)	NA	0.038(\pm 0.021)	22.6
	t-SNE	0.595(\pm 0.086)	0.533(\pm 0.074)	0.515(\pm 0.078)	0.826(\pm 0.047)	0.866(\pm 0.043)	NA	0.596(\pm 0.049)	15.9
	UMAP	0.602(\pm 0.088)	0.532(\pm 0.074)	0.554(\pm 0.082)	0.802(\pm 0.026)	0.869(\pm 0.044)	NA	0.542(\pm 0.037)	16.6
DensityCut	HD	0.734(\pm 0.000)	0.621(\pm 0.072)	0.359(\pm 0.073)	0.800(\pm 0.009)	0.881(\pm 0.009)	0.480(\pm 0.000)	0.571(\pm 0.066)	13.6
	t-SNE	0.734(\pm 0.000)	0.737(\pm 0.009)	0.688(\pm 0.072)	0.876(\pm0.004)	0.927(\pm 0.001)	0.855(\pm0.014)	0.667(\pm 0.000)	4.6
	UMAP	0.733(\pm 0.019)	0.752(\pm0.016)	0.807(\pm0.009)	0.864(\pm 0.010)	0.935(\pm 0.000)	0.825(\pm 0.016)	0.517(\pm 0.000)	5.7
RCC	HD	0.855(\pm0.091)	0.573(\pm 0.065)	0.422(\pm 0.043)	0.868(\pm 0.021)	0.788(\pm 0.050)	0.000(\pm 0.313)	0.038(\pm 0.004)	14.4
	t-SNE	0.773(\pm 0.046)	0.531(\pm 0.032)	0.496(\pm 0.028)	0.523(\pm 0.008)	0.643(\pm 0.006)	0.432(\pm 0.013)	0.568(\pm 0.024)	17.6
	UMAP	0.831(\pm 0.097)	0.642(\pm 0.040)	0.683(\pm 0.076)	0.713(\pm 0.018)	0.757(\pm 0.009)	0.723(\pm 0.019)	0.512(\pm 0.004)	12.1
CDC	t-SNE	0.813(\pm 0.031)	0.714(\pm 0.017)	0.713(\pm 0.015)	0.862(\pm 0.002)	0.914(\pm 0.005)	0.754(\pm 0.006)	0.744(\pm0.042)	4.0
	UMAP	0.783(\pm 0.017)	0.733(\pm 0.008)	0.791(\pm 0.010)	0.872(\pm 0.002)	0.937(\pm0.001)	0.751(\pm 0.001)	0.679(\pm 0.020)	3.1

R2-Table 5. The average ARI (\pm standard deviation) of the top 20 clustering results obtained by nine clustering algorithms on seven high-dimensional real-world datasets. (NA means not applicable)

		Iris	Seeds	Wine	PenDigits	Digits	MNIST-train	Baron-mouse1	Rank
K-means	HD	0.376(\pm 0.159)	0.331(\pm 0.157)	0.400(\pm 0.203)	0.540(\pm 0.053)	0.616(\pm 0.074)	0.312(\pm 0.011)	0.182(\pm 0.022)	17.9
	t-SNE	0.358(\pm 0.188)	0.327(\pm 0.179)	0.322(\pm 0.169)	0.596(\pm 0.067)	0.718(\pm 0.118)	0.562(\pm 0.063)	0.321(\pm 0.075)	17.1
	UMAP	0.355(\pm 0.179)	0.333(\pm 0.174)	0.340(\pm 0.183)	0.670(\pm 0.086)	0.756(\pm 0.118)	0.636(\pm 0.094)	0.307(\pm 0.083)	15.7
DBSCAN	HD	0.574(\pm 0.030)	0.392(\pm 0.011)	0.384(\pm 0.006)	0.137(\pm 0.088)	0.051(\pm 0.085)	0.000(\pm 0.000)	0.117(\pm 0.013)	19.7
	t-SNE	0.597(\pm 0.047)	0.283(\pm 0.222)	0.629(\pm 0.092)	0.682(\pm 0.187)	0.721(\pm 0.108)	0.396(\pm 0.286)	0.719(\pm 0.022)	12.4
	UMAP	0.654(\pm 0.112)	0.440(\pm 0.058)	0.629(\pm 0.101)	0.524(\pm 0.261)	0.481(\pm 0.223)	0.477(\pm 0.106)	0.639(\pm 0.073)	12.9
CDP	HD	0.577(\pm 0.080)	0.492(\pm 0.117)	0.429(\pm 0.028)	0.397(\pm 0.010)	0.198(\pm 0.021)	NA	-0.005(\pm 0.004)	19.0
	t-SNE	0.370(\pm 0.138)	0.344(\pm 0.158)	0.380(\pm 0.148)	0.738(\pm 0.080)	0.871(\pm 0.039)	NA	0.344(\pm 0.010)	14.6
	UMAP	0.392(\pm 0.141)	0.377(\pm 0.162)	0.417(\pm 0.155)	0.742(\pm 0.017)	0.880(\pm 0.020)	NA	0.318(\pm 0.050)	13.6
LGC	HD	0.568(\pm 0.000)	0.529(\pm 0.063)	0.786(\pm 0.025)	0.594(\pm 0.033)	0.512(\pm 0.147)	NA	0.040(\pm 0.017)	15.7
	t-SNE	0.631(\pm 0.054)	0.783(\pm 0.007)	0.710(\pm 0.042)	0.692(\pm 0.074)	0.877(\pm 0.037)	NA	0.481(\pm 0.024)	9.0
	UMAP	0.663(\pm 0.038)	0.560(\pm 0.039)	0.808(\pm 0.007)	0.529(\pm 0.074)	0.840(\pm 0.045)	NA	0.284(\pm 0.041)	11.6
Meanshift	HD	0.561(\pm 0.042)	0.571(\pm 0.088)	0.727(\pm 0.054)	0.521(\pm 0.099)	0.653(\pm 0.080)	0.005(\pm 0.012)	0.115(\pm 0.021)	15.4
	t-SNE	0.735(\pm 0.075)	0.594(\pm 0.074)	0.704(\pm 0.039)	0.533(\pm 0.116)	0.726(\pm 0.170)	0.599(\pm 0.049)	0.507(\pm 0.058)	9.6
	UMAP	0.659(\pm 0.085)	0.635(\pm 0.111)	0.721(\pm 0.135)	0.698(\pm 0.096)	0.764(\pm 0.114)	0.662(\pm 0.073)	0.283(\pm 0.013)	8.7
Ncut	HD	0.366(\pm 0.155)	0.316(\pm 0.146)	0.419(\pm 0.192)	0.000(\pm 0.000)	0.311(\pm 0.100)	NA	0.002(\pm 0.005)	22.1
	t-SNE	0.353(\pm 0.173)	0.329(\pm 0.177)	0.306(\pm 0.163)	0.686(\pm 0.103)	0.736(\pm 0.126)	NA	0.337(\pm 0.055)	17.7
	UMAP	0.353(\pm 0.178)	0.317(\pm 0.173)	0.325(\pm 0.178)	0.638(\pm 0.087)	0.741(\pm 0.124)	NA	0.266(\pm 0.077)	19.0
DensityCut	HD	0.568(\pm 0.000)	0.564(\pm 0.104)	0.302(\pm 0.069)	0.699(\pm 0.017)	0.831(\pm 0.020)	0.305(\pm 0.000)	0.555(\pm 0.083)	12.3
	t-SNE	0.568(\pm 0.000)	0.781(\pm 0.012)	0.622(\pm 0.154)	0.818(\pm 0.005)	0.908(\pm 0.001)	0.806(\pm0.020)	0.680(\pm 0.005)	5.6
	UMAP	0.597(\pm 0.049)	0.797(\pm0.018)	0.816(\pm0.011)	0.791(\pm 0.015)	0.919(\pm 0.000)	0.670(\pm 0.018)	0.502(\pm 0.000)	4.3
RCC	HD	0.812(\pm 0.168)	0.538(\pm 0.116)	0.293(\pm 0.049)	0.790(\pm 0.059)	0.463(\pm 0.132)	0.000(\pm 0.236)	0.101(\pm 0.007)	15.1
	t-SNE	0.738(\pm 0.082)	0.492(\pm 0.091)	0.398(\pm 0.081)	0.168(\pm 0.019)	0.320(\pm 0.017)	0.100(\pm 0.037)	0.311(\pm 0.062)	15.9
	UMAP	0.806(\pm 0.169)	0.574(\pm 0.099)	0.616(\pm 0.158)	0.445(\pm 0.049)	0.558(\pm 0.028)	0.528(\pm 0.057)	0.203(\pm 0.014)	13.3
CDC	t-SNE	0.816(\pm0.043)	0.733(\pm 0.022)	0.729(\pm 0.016)	0.805(\pm 0.001)	0.905(\pm 0.005)	0.782(\pm 0.006)	0.769(\pm0.058)	3.0
	UMAP	0.735(\pm 0.010)	0.770(\pm 0.006)	0.796(\pm 0.012)	0.820(\pm0.005)	0.930(\pm0.001)	0.651(\pm 0.000)	0.635(\pm 0.062)	3.4

R2-Table 6. The average F1-score (\pm standard deviation) of the top 20 clustering results obtained by nine clustering algorithms on seven high-dimensional real-world datasets. (NA means not applicable)

		Iris	Seeds	Wine	PenDigits	Digits	MNIST-train	Baron-mouse1	Rank
K-means	HD	0.476(\pm 0.172)	0.438(\pm 0.175)	0.499(\pm 0.200)	0.579(\pm 0.051)	0.648(\pm 0.069)	0.371(\pm 0.017)	0.357(\pm 0.051)	19.4
	t-SNE	0.452(\pm 0.196)	0.427(\pm 0.192)	0.428(\pm 0.184)	0.628(\pm 0.063)	0.740(\pm 0.112)	0.596(\pm 0.062)	0.428(\pm 0.100)	18.0
	UMAP	0.449(\pm 0.193)	0.434(\pm 0.189)	0.438(\pm 0.196)	0.698(\pm 0.081)	0.776(\pm 0.111)	0.667(\pm 0.092)	0.421(\pm 0.116)	16.6
DBSCAN	HD	0.747(\pm 0.009)	0.603(\pm 0.050)	0.604(\pm 0.009)	0.283(\pm 0.065)	0.218(\pm 0.062)	0.000(\pm 0.000)	0.396(\pm 0.051)	19.1
	t-SNE	0.750(\pm 0.010)	0.365(\pm 0.276)	0.745(\pm 0.053)	0.716(\pm 0.158)	0.752(\pm 0.098)	0.480(\pm 0.234)	0.807(\pm 0.100)	12.3
	UMAP	0.787(\pm 0.056)	0.633(\pm 0.028)	0.754(\pm 0.064)	0.591(\pm 0.210)	0.559(\pm 0.182)	0.552(\pm 0.087)	0.755(\pm 0.050)	12.1
CDP	HD	0.693(\pm 0.081)	0.609(\pm 0.121)	0.651(\pm 0.022)	0.481(\pm 0.008)	0.328(\pm 0.015)	NA	0.414(\pm 0.005)	19.0
	t-SNE	0.476(\pm 0.156)	0.461(\pm 0.166)	0.498(\pm 0.159)	0.763(\pm 0.068)	0.883(\pm 0.036)	NA	0.460(\pm 0.002)	14.9
	UMAP	0.495(\pm 0.152)	0.486(\pm 0.167)	0.530(\pm 0.154)	0.765(\pm 0.016)	0.891(\pm 0.019)	NA	0.428(\pm 0.083)	14.3
LGC	HD	0.746(\pm 0.000)	0.714(\pm 0.020)	0.858(\pm 0.017)	0.633(\pm 0.030)	0.566(\pm 0.113)	NA	0.438(\pm 0.021)	13.1
	t-SNE	0.746(\pm 0.022)	0.854(\pm 0.005)	0.804(\pm 0.029)	0.716(\pm 0.071)	0.888(\pm 0.034)	NA	0.577(\pm 0.019)	9.6
	UMAP	0.775(\pm 0.023)	0.680(\pm 0.032)	0.873(\pm 0.005)	0.557(\pm 0.073)	0.854(\pm 0.042)	NA	0.406(\pm 0.063)	13.1
Meanshift	HD	0.725(\pm 0.060)	0.715(\pm 0.055)	0.810(\pm 0.037)	0.580(\pm 0.081)	0.693(\pm 0.067)	0.186(\pm 0.009)	0.415(\pm 0.040)	14.3
	t-SNE	0.816(\pm 0.054)	0.703(\pm 0.064)	0.803(\pm 0.026)	0.578(\pm 0.102)	0.752(\pm 0.157)	0.636(\pm 0.044)	0.683(\pm 0.070)	10.6
	UMAP	0.782(\pm 0.044)	0.752(\pm 0.077)	0.821(\pm 0.079)	0.730(\pm 0.084)	0.791(\pm 0.099)	0.697(\pm 0.070)	0.391(\pm 0.016)	9.3
Ncut	HD	0.469(\pm 0.171)	0.431(\pm 0.163)	0.529(\pm 0.180)	0.182(\pm 0.000)	0.419(\pm 0.078)	NA	0.430(\pm 0.005)	20.7
	t-SNE	0.450(\pm 0.188)	0.430(\pm 0.191)	0.416(\pm 0.175)	0.714(\pm 0.091)	0.758(\pm 0.118)	NA	0.452(\pm 0.077)	18.3
	UMAP	0.448(\pm 0.194)	0.420(\pm 0.189)	0.425(\pm 0.194)	0.669(\pm 0.084)	0.762(\pm 0.117)	NA	0.397(\pm 0.110)	20.3
DensityCut	HD	0.746(\pm 0.000)	0.720(\pm 0.047)	0.560(\pm 0.025)	0.730(\pm 0.015)	0.848(\pm 0.018)	0.417(\pm 0.000)	0.708(\pm 0.051)	10.4
	t-SNE	0.746(\pm 0.000)	0.853(\pm 0.008)	0.773(\pm 0.077)	0.834(\pm 0.004)	0.917(\pm 0.000)	0.824(\pm0.024)	0.784(\pm 0.003)	5.0
	UMAP	0.756(\pm 0.022)	0.864(\pm0.012)	0.878(\pm0.007)	0.811(\pm 0.014)	0.927(\pm 0.000)	0.712(\pm 0.020)	0.679(\pm 0.000)	4.3
RCC	HD	0.880(\pm0.103)	0.699(\pm 0.070)	0.572(\pm 0.072)	0.811(\pm 0.051)	0.542(\pm 0.108)	0.182(\pm 0.183)	0.761(\pm 0.107)	10.4
	t-SNE	0.818(\pm 0.064)	0.626(\pm 0.098)	0.550(\pm 0.097)	0.193(\pm 0.022)	0.352(\pm 0.018)	0.119(\pm 0.033)	0.416(\pm 0.072)	17.3
	UMAP	0.857(\pm 0.137)	0.682(\pm 0.091)	0.711(\pm 0.141)	0.473(\pm 0.049)	0.586(\pm 0.027)	0.583(\pm 0.048)	0.287(\pm 0.019)	15.0
CDC	t-SNE	0.872(\pm 0.031)	0.821(\pm 0.016)	0.818(\pm 0.011)	0.821(\pm 0.001)	0.914(\pm 0.005)	0.801(\pm 0.006)	0.840(\pm0.072)	3.3
	UMAP	0.823(\pm 0.009)	0.845(\pm 0.005)	0.865(\pm 0.008)	0.837(\pm0.004)	0.937(\pm0.001)	0.695(\pm 0.000)	0.746(\pm 0.048)	3.4

R2-Fig. 3. The clustering results of CDC combined with t-SNE and UMAP on seven real-world datasets.

R2-Fig. 4. Comparison of nine clustering algorithms in ACC and running time on PenDigits and Digits datasets.

R2-Fig. 5. Comparison of nine clustering algorithms in ACC and running time on MNIST-train and Baron-mouse1 datasets.

According to your suggestion, we have revised the manuscript (line 287-312, page 15-16) as well as the Supplementary Information (Supplement Figure 9-11; Supplement Table 2-6).

Comment 3

Here's how I understood the routine for selecting the parameter T: (1) Compute Delaunay triangulation on all data points. (2) Remove cross-cluster triangles that satisfy a certain condition (see next comment for an issue here). (3) Determine number b of boundary points

in the resulting partial triangulation. (4) Set T to $DCMS[b]$, where $DCMS$ is the array of DCM values sorted in non-decreasing order. If this understanding is correct, I am wondering how the clustering induced by the partial triangulation obtained after removing cross-cluster triangles relates to the clustering computed by CMC . Does CMC merely reproduce the partial triangulation clustering? If so, why not directly using the partial triangulation clustering? If not, how exactly does CDC improve the partial triangulation clustering? It's also possible that I misunderstood something here. If so, please clarify the respective paragraphs in the manuscript.

Thank you for your comment. Your understanding of T_{DCM} determination workflow is basically correct, **but there is no clustering result generated after identifying the cross-cluster triangles**. Identifying the cross-cluster triangles is for better estimating the number of intra-cluster triangles F in the triangular network, and further calculating the number of boundary points B using Eq. (2).

$$B = 2V - F - 2C \quad (2)$$

where V denotes the total number of vertexes, C denotes the number of clusters. V is known in a given dataset (i.e., n), but F and C are not. The initial F is the total number of triangles in the whole TIN, which includes the triangles connecting different clusters, i.e., cross-cluster triangles whose three vertices are not all in the same cluster (otherwise is intra-cluster triangle). Using the excessive number of triangles would make the number of boundary points B smaller than the true value. To identify the cross-cluster triangles, we set a judgment rule:

$$\sum_{i=1, i \neq j}^3 |v_i \in KNN(v_j)| < 3 \quad (3)$$

Using the intermediate results after removing the cross-cluster triangles as the clustering results is inappropriate. Because Eq. (3) cannot guarantee to remove the cross-cluster triangles in an accurate and complete manner, thereby causing the clusters in close proximity to be merged as a whole. As shown in R2-Fig. 6a, d, the green rectangles indicate the unrecognized cross-triangles. Clustering at this step would connects different clusters as an individual one (in R2-Fig. 6b, e).

R2-Fig. 6. Clustering results obtained by directly using the intermediate results after removing the cross-cluster triangles (b, e) and the adaptive CDC (c, f) respectively.

In comparison, the adaptive CDC by determining the threshold of T_{DCM} is more robust to the misidentification of cross-cluster triangles. Although a few intra-cluster triangles at the boundaries are misidentified as cross-cluster triangles and several cross-cluster triangles are undetected due to the close proximity between clusters in R2-Fig. 6a, d, these two biases can be offset partially. Moreover, T_{DCM} is extracted from the DCM curve according to the number of boundary points B . A slight fluctuation of the number B has subtle impact on the clustering quality. Thus, all the clusters are identified accurately using adaptive CDC (R2-Fig. 6c, f).

According to your suggestion, we have revised the manuscript (line 564-577, page 29; line 580-586, page 29-30) as well as the Supplementary Information (Supplement Note 2; Supplement Figure 8).

Comment 4

What does it mean to remove a cross-cluster triangle? My guess: removing all three edges. But this is not stated explicitly in the paper. Also: Equation (15) does not make sense, since j is not defined and $v_i \in \text{KNN}(v_j)$ is a truth-value and therefore has no cardinality. Please provide a sound characterisation of cross-triangles, e.g., something like this: "A triangle $\{\{p_1, p_2\}, \{p_2, p_3\}, \{p_1, p_3\}\}$ is called cross-triangle if and only if <some condition involving the k-NNs of the points p_1 , p_2 , and p_3 >".

Thank you for your comment, we are sorry for misleading you by the inappropriate expression "remove a cross-cluster triangle". Actually, **we just count the number of cross-cluster triangles rather than removing the edges explicitly.** The purpose to count the cross-cluster triangles is to estimate F accurately, thus can calculate the number of boundary points using Eq. (14) in manuscript. According to your suggestion, **we have added the definition and judge rule of cross-cluster triangle, and more**

explanations in manuscript as follows.

However, the number of initial boundary points in the whole TIN is not equal to the total number of boundary points in the separated clusters. To conduct an accurate estimation, the whole TIN should be treated as multiple sub-networks (Supplementary Figure 8c). Given C clusters, the number of boundary points in clusters can be solved as follows:

$$\sum_{i=1}^m B_i = 2 \sum_{i=1}^m V_i - \sum_{i=1}^m F_i - 2C$$

$$B = 2V - F - 2C$$

where F is the total number of intra-cluster triangles in the multiple separated networks. V is known in a given dataset (i.e., n), but F and C are not.

The initial F is the total number of triangles in the whole TIN, which includes the triangles connecting different clusters, i.e., **cross-cluster triangles whose three vertices are not all in the same cluster (otherwise is intra-cluster triangle)**. Using the excessive number of triangles would make the number of boundary points B smaller than the true value. To identify the cross-cluster triangles, we set a judgment rule:

$$\sum_{i=1}^3 \sum_{j=1, j \neq i}^3 \sigma(v_i, v_j) < 3$$

where v_1, v_2, v_3 are the three vertices of a triangle, and $\sigma(v_i, v_j)$ is an indicator function:

$$\sigma(v_i, v_j) = \begin{cases} 0 & \text{if } v_j \notin \text{KNN}(v_i) \\ 1 & \text{if } v_j \in \text{KNN}(v_i) \end{cases}$$

Eq. (15) considers the proximity of the vertices in an intra-cluster triangle. The final F is equal to initial F minus the number of cross-cluster triangles that satisfies Eq. (15). In terms of the number of clusters C , it is significantly smaller than that of points usually, which has a trivial effect on the estimation of B . When C is vague or difficult to determine, it can be neglected.

According to your suggestion, we have revised the manuscript (line 562-575, page 29).

Comment 5

Since the parameter T has no intuitive interpretation, it is very difficult to set manually. Therefore, users will most probably use the automated selection routine. Consequently, I think that this routine should be regarded as an integral part of the proposed algorithm and therefore be considered both in the theoretical complexity analysis and in the runtime evaluation.

Thank you for your invaluable comments. I agree that the DCM threshold T_{DCM} has no intuitive interpretation, however, DCM has been normalized and ranges from 0 to 1. According to our experiments on different datasets, we found that the optimal threshold T_{DCM} is always between 0 and 0.5, which can be a reference for users to set T_{DCM} effectively. **The adaptive method to determine T_{DCM} definitely ease the setting of the parameter, but we do not think it should be integrated with the CDC mandatorily.**

Like many other clustering algorithms, exploratory parameter setting in a manual manner is necessary to explore the diverse clustering patterns (e.g., the damping factor α in DensityCut (Ding et al., 2016)). Meanwhile, the adaptive method might not always guarantee the highest accuracy. **It is better to adopt the adaptive method as a free option of CDC for the users with different backgrounds and domain knowledge, e.g., experts and novices. Hence, we only analyzed its theoretical time complexity, but not count its running time as a part of CDC in manuscript.**

Although adopting the adaptive method does lengthen the overall running time, it does not increase the time complexity. The procedure of this adaptive method is composed of the construction of Triangulated Irregular Network (TIN) and detection of the cross-cluster triangles. The time complexity can be represented as $O(n \log n + u)$, where u denotes the number of triangles in the TIN. Commonly, u ranges from $n - 2$ to $2n - 5$ ($n \geq 3$), so the time complexity can be simplified as $O(n \log n)$.

We tested the performance of adaptive CDC (aCDC) on seven real-world datasets and the results are illustrated in R2-Fig. 7. The blue scatter plot represents the top 200 clustering results by under a sampled parameter setting (i.e., k is set using Eq. (8) in manuscript and T_{DCM} is set from 0 to 1 with a 0.02 interval), and the red points refer to the ARI obtained by aCDC. It shows aCDC can reach top 25% clustering accuracy on all datasets, and even got the highest ARI on Iris, Wine, PenDigits, Digits. Besides, **the adaptive process to construct a TIN and estimate the T_{DCM} is relatively time-efficient, whose growth speed of running time is slower than that of CDC as the data size increases, which demonstrates its validity on the real-world datasets.**

R2-Fig. 7. Performance of the adaptive CDC (aCDC) method in clustering accuracy and time efficiency on seven real-world datasets.

According to your suggestion, we have revised the manuscript (line 407-410, page 21; line 418-424, page 22; line 580-581, page 30).

Comment 6

Equation (8) used to automatically select the parameter k is somewhat adhoc. How exactly did you come up with this equation and how did you select the involved constants?

Thank you for the constructive comment. Eq. (8) presents a continuous piecewise function, which was summarized through empirical experiments on synthetic datasets. However, we found that it has weak robustness to some datasets. Therefore, we redesigned it by carefully investigating KNN-relevant studies. Ding et al. (2016) found that k should be data-dependent and consider that $\log_2(n)$ is the most appropriate k through a theoretical analysis in (Von Luxburg, 2007) and an empirical study. Wang et al. (2017) suggest that k should be set to around 0.5%–1.5% of the total number of points. Through the investigation and extensive tests on the synthetic and real-world datasets, we have modified the function as follows:

$$k = \begin{cases} \left\lceil \frac{n}{50} \right\rceil \sim \left\lceil \frac{n}{20} \right\rceil & \text{if } 100 \leq n \leq 1,000 \\ \lceil \log_2(n) + 10 \rceil \sim 5 \lceil \log_2(n) \rceil & \text{if } n \geq 1000 \end{cases} \quad (4)$$

where n refers to the total number of points and $\lceil \cdot \rceil$ denotes to the nearest integer upwards. This equation provides a reasonable range to set k , which is more robust than the previous version. The estimation function is formalized as $\log_2(n)$ inspired by DensityCut when k is larger than 1,000. It ensures that the growth speed flattens out as k increases. The upper and lower bounds present a linear growth when k is smaller than 1,000. Meanwhile, this equation is continuous at $k = 1000$.

In order to validate its effectiveness, we collated the optimal k specified on 24 datasets used in this paper including 17 synthetic and 7 real-world datasets. From the presented result in R2-Fig. 8, we can find that all optimal values are between the upper and lower bounds determined by Eq. (4), which verified the validity of the refined piecewise function.

R2-Fig. 8. The optimal k of CDC specified on all datasets used in this paper.

According to your suggestion, we have revised the manuscript (line 405-407, page 21; line 418-420, page 22; line 532-537, page 27).

Comment 7

Since the main contribution of the paper is the algorithm CDC itself, it would be of great help for the reader if both the main algorithm and the routine for automatically selecting the parameter T were summarised via pseudo-code.

Thank you for your helpful suggestion. We added the pseudo codes of these two algorithms in the Supplementary Note 2 as follows:

Algorithm 1 Clustering by Measuring Local Direction Centrality (CDC)

Input: the dataset $X(x_1, x_2 \dots x_n)$, k for KNN and DCM threshold T_{DCM}

```
1: for each point  $x_i$ 
2:   Search the  $k$  nearest neighbors of point  $x_i$ ;
3:   Calculate the  $k$  angles formed by its KNNs;
4:   Calculate the normalized  $DCM_i$  of point  $x_i$ ;
5:   if  $DCM_i > T_{DCM}$ 
6:     Add the point  $x_i$  to the set of boundary points  $B$ ;
7:   else
8:     Add the point  $x_i$  to the set of internal points  $I$ ;
9:   end if
10: end for
11: for each internal point  $x_i$  in  $I$ 
12:   Calculate the distances between  $x_i$  and all the points in  $B$ ;
13:   Select the minimum as the reachable distance  $r_i$  of  $x_i$ ;
14: end for
15: Initialize the point labels  $C(c_1, c_2 \dots c_n)$  as a zero vector (i.e., unlabeled);
16: Set  $temp = 1$ ;
17: for each internal point  $x_i$  in  $I$  and  $c_i == 0$ 
18:   for another internal point  $x_j$  in  $I$ 
19:      $c_i = temp$ ;
20:     Calculate the distance  $d_{ij}$  between  $x_i$  and  $x_j$ ;
21:     if  $d_{ij} \leq r_i + r_j$  and  $c_j == 0$ 
22:        $c_j = c_i$ ;
23:     else if  $d_{ij} \leq r_i + r_j$  and  $c_j > 0$ 
24:       Assign  $c_i$  to all points whose labels equal to  $c_j$ ;
25:     end if
26:   end for
27:    $temp = temp + 1$ ;
28: end for
29: for each boundary point  $x_i$  in  $B$ 
30:   Assign the label of the nearest internal point to  $x_i$ ;
31: end for
32: return the cluster labels  $C$ ;
```

Algorithm 2 Adaptive Method for T_{DCM}

Input: the dataset $X(x_1, x_2 \dots x_V)$, k for KNN, the number of clusters C

```
1: for each point  $x_i$ 
2:   Search the  $k$  nearest neighbors of point  $x_i$ ;
3:   Calculate the  $k$  angles formed by its KNNs;
4:   Calculate the normalized  $DCM_i$  of point  $x_i$ ;
10: end for
11: Sort the DCMs in a descend order as  $DCM_1, DCM_2, \dots, DCM_V$ ;
12: Construct the initial TIN using Delaunay algorithm;
13: Count the number of triangles  $F$  in the TIN;
14: for each triangle  $\mathcal{T}$ 
15:   if  $\mathcal{T}$  is a cross-cluster triangle
16:      $F = F - 1$ ;
17:   end if
18: end for
19: Calculate the number of boundary points  $B = 2V - F - 2C$ ;
20: return  $DCM_B$ ;
```

According to your suggestion, we have revised the manuscript (line 502-503, page 26; line 585-586, page 30) as well as the Supplementary Information (Supplement Note 2).

Comment 8

In the past years, Python has more and more developed into the quasi-standard language for data science applications. To increase the likelihood that the CDC is actually used by the targeted community, I therefore strongly suggest that the authors provide a Python implementation in addition to the existing MATLAB implementation.

Thank you for your kindly suggestion. We have added a preliminary version of Python-based CDC in our GitHub platform:

<https://github.com/ZPGuiGroupWhu/ClusteringDirectionCentrality>

David Blumenthal

Reviewer #3 (Expertise: Clustering with experience relevant to CyTOF and scRNASeq data):

Summary

The authors present an interesting new clustering algorithm, and demonstrate performance on single-cell RNA sequencing (scRNA-seq), mass cytometry (CyTOF), and speech data. The authors' evaluations show good performance of the algorithm. However, to provide users with more information about whether this algorithm is useful in practice for analyzing scRNA-seq and CyTOF data, these evaluations need to be expanded and compared against standard analysis pipelines for these data types. More details are provided below. Thank you for your comment. We have expanded the evaluations by comparing with standard analysis pipeline accordingly.

Major comments

Comment 1

The authors' performance evaluations for scRNA-seq and CyTOF data rely on (i) an initial dimension reduction step (using either t-SNE or UMAP) to two dimensions, followed by (ii) clustering in the 2-dimensional space using the authors' algorithm. This is quite different to standard analysis pipelines for these data types (e.g. for scRNA-seq: initial dimension reduction to 50 dimensions using PCA on a subset of highly variable genes, followed by clustering in 50-dimensional PCA space using graph-based clustering; and for CyTOF: clustering in the full-dimensional space using algorithms such as FlowSOM / PhenoGraph).

To provide the most useful information for readers, the authors should extend these comparisons to include the full standard clustering pipelines for scRNA-seq and CyTOF data.

Thank you for the invaluable suggestions. **According to your advice, we have added the comparison with standard analysis pipelines for both the scRNA-seq and CyTOF datasets. The results demonstrate the effectiveness of the proposed algorithm CDC.**

For scRNA-seq datasets, we have added two pipelines, i.e., flowSOM and PhenoGraph in the cluster analysis for comparison purpose. We conducted the two algorithms in the 50-dimensional PCA space reduced from the raw data (i.e., flowSOM-PCA50, PhenoGraph-PCA50). While other four typical baselines, K-means, CDP, DBSCAN and LGC were performed on the 2D results of UMAP (i.e., Kmeans-UMAP, CDP-UMAP, DBSCAN-UMAP and LGC-UMAP), since they perform better in 2D space than high-dimensional space (see details in the response of Comment 2 of Reviewer 2). For the CyTOF datasets, we have redesigned the experiment by comparing CDC combined with UMAP (CDC-UMAP) to other 16 typical clustering algorithms (including pipelines for CyTOF datasets and classical clustering baselines) in the full-dimensional space.

The results of ARI are presented in R3-Fig. 1 and the rest five validity indexes (NMI, ACC, F1-score, RI, JI) can be seen in Supplementary Result 1. The curves show the average,

maximum and minimum ARIs of the seven algorithms (R3-Fig. 1f-l). CDP and LGC are not applicable on TM dataset ($n = 54,865$) due to the problem of memory overflow. CDC-UMAP outperform baselines and have stronger robustness. **All the five algorithms equipped with UMAP obtained higher clustering accuracy than flowSOM and PhenoGraph in general. Among the algorithms combined with UMAP, CDC obtained the highest and most stable accuracy.** We also applied CDC-UMAP and PhenoGraph-PCA50 on seven simulated scRNA-seq datasets with 10,000 genes and the increasing number of cells (R3-Fig. 1m). **The running times demonstrate that CDC-UMAP is more efficient than the pipeline PhenoGraph-PCA50.** In contrast, PCA for the pipeline is more likely to incur memory overflow due to the eigen-decomposition of matrix on large datasets during our experiments.

For CyTOF datasets (R3-Fig. 1n-o), **CDC-UMAP has a high and stable accuracy, and a promising time efficiency above the average.** CDC-UMAP got the highest ARI on Samusik-1 and the 5th on Levine-13dim, and both surpassing 0.8. Although CDC-UMAP has a lower ARI than PhenoGraph, clusterX, DensVM, DensityCut on Levine-13dim, it is more efficient than them on both datasets (50s on Samusik-1 and 72s on Levine-13dim).

R3-Fig. 1. Clustering accuracy and time efficiency on scRNA-seq and CyTOF datasets. (a)-(d) Comparison among the ground truth, and the optimal clustering results of DBSCAN, CDP and CDC, in 2D space transformed by t-SNE on Pan-Cancer dataset. The rectangles in each sub-figure highlight the differences between the ground truths and clustering results. (e) Division result of the internal and boundary points by CDC. (f)-(l) ARIs of the top 20 results obtained by seven clustering algorithms on seven scRNA-seq datasets, where the top, middle and bottom curves represent the highest, average, and lowest ARIs respectively. (m) Running time of CDC, CDC-UMAP and PhenoGraph-PCA50 on the simulated scRNA-seq datasets with an increasing number of cells up to 100,000 and a fixed number of genes (10,000). (n)-(o) Performances of 17 clustering algorithms on Samusik-1 and Levine-13dim datasets, including K-means, DBSCAN, Meanshift, DensityCut, Robust Continuous Clustering (RCC), PhenoGraph, flowSOM, flowSOM_pre, flowPeaks, flowMeans, flowMerge, Rclusterpp, immunoClust, samSpectral, DensVM, clusterX, and CDC.

According to your suggestion, we have revised the manuscript (line 152-186, page 8-9; line 192-227, page 10-12) as well as the Supplementary Information (Supplement Table 1).

Comment 2

For the clustering of CyTOF and scRNA-seq data, much more detail is required for how the authors ran their clustering algorithm on these datasets – in particular preprocessing steps. For example, for the CyTOF datasets (Supplementary Figure 2), no details are provided for whether “unclassified” points – i.e. cells without ground truth labels – were excluded prior to clustering, which would create an unfair comparison.

Thank you for your comment. We are sorry for the unclear statements of the preprocessing steps. **For a fair comparison, we have run all the clustering baselines on a same computer, and compared them on the same preprocessed scRNA-seq and CyTOF datasets in the revision.** We have also merged the Supplementary Figure 2 with Fig. 2-3 in the previous version into a new figure (see R3-Fig. 1), and the detailed preprocessing steps are presented in Supplementary Note 4 as follows:

We removed the cells with unclassified labels in the gene-level read count values of ALM and VISp datasets. All the clustering algorithms were implemented on a commodity desktop computer with a 8-core Intel i7 processor and 64 GB RAM. Details of the codes are listed in Supplementary Table 9 and the parameter settings are available in Supplementary Parameter Settings. In particular, to automatically select the cluster centers in CDP and obtain the top 20 results, we set fixed weights of the density and distance as the substitution of manual selection. Moreover, CDP and LGC are not applicable on TM dataset. Because caching large distance matrix would lead to memory overflow on a commodity computer when the data size increases (around $n > 50,000$).

We downloaded the pre-processed CyTOF datasets from FlowRepository (repository FR-FCM-ZZPH) (Spidlen et al., 2012). As described in (Weber et al., 2016), “the data pre-

processing included the application of an arcsinh transformation with a standard cofactor of 5 (CyTOF data) or 150 (flow cytometry data). For the flow cytometry datasets, pre-gating to exclude doublets, debris, and dead cells was also required. The clustering algorithms were run on all remaining single, live cells and no additional pre-gating was performed, since our aim is to evaluate performance in maximally automated settings. In addition, we did not perform any standardization of individual protein marker dimensions. This was unnecessary since the arcsinh already transforms all dimensions to comparable scales. More importantly, standardization of dimensions that do not contain a true signal could amplify the effect of noise and outliers, adversely affecting clustering performance.” In addition, we removed the cells without manually gated population labels.

For each clustering method, we adjusted the input parameters deliberately in order to obtain an approximately highest accuracy. In terms of the number of clusters (denoted as C) required by the following methods (e.g., K-means, flowSOM, flowMeans, Rclusterpp, etc), it can be specified according to the number of manually gated populations. For example, we set C=1:1:40 (ranging from 1 to 40 with a step size of 1) for K-means when the true number of clusters is 24 on Samusik-1 dataset. But for the parameters in other algorithms, we tuned the combinations and intervals of input parameters extensively to achieve a high accuracy. Finally, we calculate the top 20 ARIs of the clustering results obtained by each method. The details about the parameter settings can be available in Supplementary Parameter Settings.

References:

Spidlen, J., Breuer, K., Rosenberg, C., Kotecha, N. & Brinkman, R. R. FlowRepository: A resource of annotated flow cytometry datasets associated with peer-reviewed publications. *Cytom. Part A* 81A, 727-731 (2012).

Weber, L. M. & Robinson, M. D. Comparison of Clustering Methods for High-Dimensional Single-Cell Flow and Mass Cytometry Data. *Cytom. Part A* 89, 1084-1096 (2016).

According to your suggestion, we have revised the Supplementary Information (Supplement Note 4).

Comment 3

For the scalability comparisons for scRNA-seq and CyTOF data (Figure 3), it is not clear whether the runtimes shown here are only after the initial t-SNE or UMAP dimension reduction. In this case, these results would be difficult for users to interpret, since clustering in 2-dimensional t-SNE or UMAP space is not a recommended analysis pipeline for scRNA-seq or CyTOF data, and previously published runtime/scalability estimates start from the full high-dimensional data. These results should be clarified, and presented in a way that enables comparisons of runtime/scalability starting from the full high-dimensional data.

Thank you for your invaluable comment. According to your suggestion, **we have added the running time of UMAP onto CDC as that of CDC-UMAP** in R3-Fig. 11-o. In this experiment, the running times of all algorithms were counted starting from the full high-

dimensional data (before the dimension reduction). In general, **CDC-UMAP conducts the cluster analysis in a time efficient manner, which has lower time cost than PhenoGraph-PCA50 on full high-dimensional scRNA-seq datasets and is above average among the 16 algorithms** (i.e., No. 5 on Samusik-1, No. 8 on Levine-13dim). Because CDC has a relatively low time complexity of $O(n \log n)$, and that of UMAP is empirically approximately $O(n^{1.14})$, which is lower than that of t-SNE and PCA (McInnes et al., 2018). In addition, the time complexity of UMAP can be further optimized to $O(kn)$ (k is the parameter of KNN) using probabilistic edge sampling and negative sampling.

Reference:

McInnes, L., Healy, J., Saul, N. & Großberger, L. UMAP: uniform manifold approximation and projection. *J. Open Source Softw.* 3, 861 (2018).

According to your suggestion, we have revised the manuscript (line 159-165, page 8; line 211-215, 221-222, page 11).

Comment 4

The evaluations for scRNA-seq and CyTOF data also rely on quite old and small datasets by modern standards. To be most useful for readers, the analyses should be compared on modern full-sized datasets (especially in terms of number of cells for scRNA-seq data).

Thank you for your comment. **According to your suggestion, we have added two scRNA-seq datasets with larger size**, i.e., Primary Visual Cortex (VISp, $n = 14,739$ cells, $d = 45,768$ genes) and Tabula Muris (TM, $n = 54,862$ cells, $d = 19,791$ genes) (in R3-Table 1). Besides, **to further evaluate the time efficiency of CDC-UMAP, we counted the running time on the simulated scRNA-seq datasets with a fixed number of genes ($d = 10,000$) and the increasing number of cells up to 100,000** (R3-Fig. 1m). Comparing with the recent publication on Nature Communications, Li et al. (2020) (the maximum scRNA-seq dataset has 100,000 cells and 500 genes), we believe our experiments follow the modern standard in data size.

Limited by the memory of a commodity desktop computer (memory overflow problem of the dimension reduction for PCA, t-SNE and UMAP), we could not further test CDC-UMAP in a scRNA-seq dataset with millions of cells and ten thousand of genes. Nonetheless, we found that **CDC-UMAP has the comparable time efficiency with pipelines (e.g., DensityCut, flowMeans, Rclusterpp, etc) and is faster than PhenoGraph, flowMerge, DensVM, clusterX, etc**, according to the experimental results.

R3-Table 1. Description of the two newly added scRNA-seq datasets

Dataset	Type	No. of cells	No. of dimensions	No. of cell populations	Description	Protocol
VISp	scRNA-seq	14,739	45,768	5	Primary Visual Cortex	inDrop

TM	scRNA-seq	54,862	19,791	55	Whole Mus musculus	SMART-Seq
----	-----------	--------	--------	----	-----------------------	-----------

Reference:

Li, C. et al. SciBet as a portable and fast single cell type identifier. Nat. Commun. 11, 1818 (2020).

According to your suggestion, we have revised the Supplementary Information (Supplement Table 1).

Minor comments

Comment 5

In several locations (e.g. Abstract; Introduction lines 30 and 34), the authors use the terminology “spatial distribution” to refer to the distribution of points in feature space. This is somewhat confusing, since readers may interpret this as referring to physical spatial data (e.g. geographical data). Suggest replacing this with clearer terminology such as “feature space” or “high-dimensional space”.

Thank you. According to your kindly comment, we have modified the terminology “spatial distribution” as follows:

Line 11-12, Page 1

This algorithm distinguishes between internal and boundary points based on the distribution of K-nearest neighbors (KNNs).

Line 62-63, Page 3

We consider the KNNs (11) of internal points to be distributed differently from that of boundary points in the feature space.

Line 424-425, Page 22

In many real-world applications, the data distribution in the feature space tends to be heterogeneous and complex.

Line 458-459, Page 24

The core idea behind CDC is to distinguish boundary and internal points of clusters based on the distribution of KNNs.

Comment 6

Wording and / or figure captions (e.g. Figure 1) for scRNA-seq and CyTOF data should make clearer when this is discussing only 2-dimensional data, since this is a much easier clustering task than typical (high-dimensional) scRNA-seq or CyTOF data.

Thank you for your comment. **We have redesigned the experiments and introduced more pipelines for comparing clustering accuracy and time efficiency from the**

original high-dimensional datasets. Specifically, on the scRNA-seq datasets, flowSOM and PhenoGraph were conducted in 50-dimensional space transformed from the full-dimensional raw data by PCA (denoted as flowSOM-PCA50 and PhenoGraph-PCA50). While, the rest five algorithms (K-means, CDP, DBSCAN, LGC and CDC) were equipped with UMAP, and performed them in 2D space (denoted as Kmeans-UMAP, CDP-UMAP, DBSCAN-UMAP, LGC-UMAP and CDC-UMAP). On the CyTOF datasets, we conducted CDC in 2D space using UMAP, while others on the full-dimensional raw data. The refined experiments demonstrate that CDC-UMAP can obtain the highest accuracy and most robust outcomes in an efficient manner.

According to your suggestion, we have revised the manuscript (line 159-165, page 8; line 221-222, page 11).

Comment 7

Introduction, line 37: this line states that k-means and k-medoids can only identify spherical clusters. This is not correct – these algorithms can identify ellipsoidal clusters. This wording should be replaced with “ellipsoidal” or similar.

Thank you. According to your suggestion, we have modified the statement as follows:

Line 36-37, Page 2

However, these algorithms cannot identify non-ellipsoidal clusters and have a weak robustness to the noise.

Reviewer #4 (Expertise: Clustering with experience in the biosciences):

In this paper, the authors introduce a density-based clustering algorithm (CDC) by first searching for the boundary points between clusters, based on the observations that the k-NN of boundary points are from a small angle range, while for internal points, their k-NNs are from a much wider angle range. The authors used both simulated and real datasets to evaluate their method.

Clustering analysis is an important exploratory data analysis technique, and density-based clustering is appealing because of the natural definition of clusters based on density peaks. However, these methods may be sensitive to parameter setting. I have some major comments about the method itself, its connection with existing methods, the experimental setting (robust to parameters), and the results, which need to be addressed.

Thank you for your comments and the valuable suggestions below. According to your advice and that of other reviewers, **we have made a comprehensive revision to the manuscript by adding the verification experiments, supplementing the parameter settings and sensitivity analysis, verifying the effectiveness and robustness of the algorithm.**

Besides, we think that there are some misunderstandings of our algorithm. According to the working mechanism, CDC actually belongs to a connectivity-based approach rather than density-based. It divides all points into boundary and internal points by calculating the angle variance formed by KNNs. Then, the internal points are connected according to their reachable ranges. Finally, the boundary points are assigned to the clusters of its nearest internal point.

Comment 1

Exploring the k-NNs of data points to identify boundary points have been used implicitly or explicitly for density-based clustering. For example, in (Ding et al, densityCut, Bioinformatics, 2016), the authors used directed k-NN graphs for their density-based clustering algorithm, and boundary points have zero or few 'in-neighbors' while the non-boundary points tend to have more 'in-neighbors'. The definition of boundary points based on 'in-neighbors' is much easier than angles as used in this study because the number of neighbors is independent of the dimensional of the data. Will the incorporation of this 'in-neighbor' definition of boundary points improve CDC? How CDC performs compared to densityCut?

Thank you for your comment and constructive idea. We consider that the concept of 'in-neighbor' in a KNN graph is similar to Reversed K-Nearest Neighbors (RKNN) (actually, the concept of 'in-neighbor' is not introduced in the paper of DensityCut). Both of the two concepts refer to the objects that consider a given point as the membership of their KNNs. We can adopt 'in-neighbors' to distinguish boundary and internal points, since there is difference between them in the number of 'in-neighbors' (local density). **However, it is inaccurate to handle datasets with heterogeneous density and cluster size (i.e., the number of points in a cluster).** Because the internal points in sparse clusters have few

'in-neighbors', and the points in clusters of small sizes also have a small number of 'in-neighbors' especially when they are far away from the other clusters. **A global threshold is difficult to distinguish them from the boundary points.** Hence, we consider that **introducing 'in-neighbors' will not improve the clustering quality of that using the angle variance of CDC.**

According to your suggestion, **we compared CDC with DensityCut on seven high-dimensional datasets** as illustrated in R4-Table 1. In addition, we also selected other **seven baselines, i.e., K-means, DBSCAN, CDP, LGC, Meanshift, Ncut, RCC** for comparison. In this experiment, we conducted each clustering algorithm on the high-dimensional (HD) and 2D data after dimensionality reduction by t-SNE and UMAP respectively. The complete results of six validity indexes, i.e., ACC, NMI, ARI, F1-score, RI and JI have been calculated and provided in the Supplementary Result 2 and the average of the first four indexes are listed in R4-Table 2-5. The results show that **CDC combined with t-SNE and UMAP obtained the top 2 average rank on all of the four validity indexes and they outperformed other baselines with a strong robustness. DensityCut slightly worse than CDC, but outperformed other seven baselines.**

In this experiment, we further analyzed the necessity of dimension reduction. Although t-SNE and UMAP caused some degree of information loss, they did not reduce the accuracy severely and even improved the clustering quality of the baselines comparing with that in HD space (e.g., K-means, DBSCAN, CDP, Meanshift, DensityCut). Dimension reduction is actually beneficial when the number of data features is large due to the curse of dimensionality. Features have a significant difference in value range, which impacts the measurement of similarity (e.g., Euclidean distance) even if the data have been normalized. t-SNE and UMAP can well preserve the distribution characteristics in HD space when embedding the raw data to 2D space (R4-Fig. 2). Although the density heterogeneity and weak connectivity of data distribution occur after being transformed to 2D space, CDC is able to tackle the two issues properly.

In terms of time efficiency, we counted the running times of all the algorithms on PenDigits, Digits, MNIST-train and Baron-mouse1 datasets (R4-Fig. 2-3). They were implemented on an ordinary desktop computer with a 8-core Intel i7 processor and 64 GB RAM. **It demonstrates that CDC can obtain a promising ACC in an efficient manner and even runs fast than K-means, LGC, Ncut and RCC that handles high-dimensional data directly. CDC and DensityCut have the close time cost due to the same time complexity $O(n \log n)$.**

R4-Table 1. Description of seven real-world datasets.

Datasets	Type	No. of samples	No. of dimensions	No. of classes
Iris	UCI	150	4	3
Seeds	UCI	210	7	3
Wine	UCI	178	13	3
PenDigits	UCI	10,992	16	10
Digits	Image	5,620	8×8	10
MNIST-train	Image	60,000	28×28	10
Baron-mouse1	scRNA-seq	822	14,878	13

R4-Table 2. The average ACC (\pm standard deviation) of the top 20 clustering results obtained by multiple clustering algorithms on seven high-dimensional real-world datasets. (NA means not applicable)

		Iris	Seeds	Wine	PenDigits	Digits	MNIST-train	Baron-mouse1	Rank
K-means	HD	0.437(\pm 0.180)	0.433(\pm 0.192)	0.466(\pm 0.209)	0.601(\pm 0.090)	0.644(\pm 0.116)	0.442(\pm 0.034)	0.392(\pm 0.056)	19.3
	t-SNE	0.402(\pm 0.206)	0.396(\pm 0.220)	0.406(\pm 0.202)	0.608(\pm 0.096)	0.697(\pm 0.160)	0.584(\pm 0.112)	0.435(\pm 0.101)	19.3
	UMAP	0.398(\pm 0.203)	0.400(\pm 0.213)	0.410(\pm 0.216)	0.698(\pm 0.125)	0.726(\pm 0.151)	0.630(\pm 0.130)	0.430(\pm 0.100)	17.6
DBSCAN	HD	0.676(\pm 0.031)	0.570(\pm 0.013)	0.572(\pm 0.008)	0.304(\pm 0.112)	0.180(\pm 0.103)	0.010(\pm 0.000)	0.339(\pm 0.050)	20.0
	t-SNE	0.692(\pm 0.045)	0.413(\pm 0.278)	0.786(\pm 0.085)	0.722(\pm 0.133)	0.754(\pm 0.102)	0.489(\pm 0.223)	0.816(\pm 0.053)	11.4
	UMAP	0.742(\pm 0.101)	0.603(\pm 0.051)	0.787(\pm 0.097)	0.651(\pm 0.109)	0.618(\pm 0.137)	0.553(\pm 0.088)	0.766(\pm 0.041)	12.4
CDP	HD	0.656(\pm 0.097)	0.661(\pm 0.131)	0.620(\pm 0.018)	0.483(\pm 0.022)	0.407(\pm 0.009)	NA	0.379(\pm 0.011)	19.0
	t-SNE	0.437(\pm 0.176)	0.456(\pm 0.185)	0.507(\pm 0.176)	0.767(\pm 0.044)	0.878(\pm 0.047)	NA	0.508(\pm 0.006)	15.1
	UMAP	0.475(\pm 0.147)	0.476(\pm 0.184)	0.522(\pm 0.162)	0.772(\pm 0.046)	0.869(\pm 0.030)	NA	0.482(\pm 0.064)	14.9
LGC	HD	0.667(\pm 0.000)	0.691(\pm 0.068)	0.927(\pm 0.009)	0.685(\pm 0.047)	0.613(\pm 0.117)	NA	0.474(\pm 0.044)	13.7
	t-SNE	0.763(\pm 0.084)	0.919(\pm 0.006)	0.881(\pm 0.025)	0.690(\pm 0.077)	0.890(\pm 0.040)	NA	0.578(\pm 0.023)	9.3
	UMAP	0.820(\pm 0.045)	0.686(\pm 0.046)	0.935(\pm 0.003)	0.522(\pm 0.073)	0.834(\pm 0.055)	NA	0.442(\pm 0.046)	12.3
Meanshift	HD	0.655(\pm 0.043)	0.766(\pm 0.093)	0.862(\pm 0.028)	0.644(\pm 0.077)	0.752(\pm 0.059)	0.125(\pm 0.030)	0.360(\pm 0.044)	14.7
	t-SNE	0.827(\pm 0.078)	0.735(\pm 0.083)	0.892(\pm 0.023)	0.604(\pm 0.118)	0.771(\pm 0.180)	0.684(\pm 0.055)	0.690(\pm 0.063)	9.4
	UMAP	0.788(\pm 0.096)	0.801(\pm 0.111)	0.875(\pm 0.099)	0.777(\pm 0.095)	0.796(\pm 0.104)	0.706(\pm 0.088)	0.402(\pm 0.012)	9.1
Ncut	HD	0.442(\pm 0.182)	0.422(\pm 0.179)	0.509(\pm 0.181)	0.112(\pm 0.001)	0.473(\pm 0.142)	NA	0.418(\pm 0.006)	21.0
	t-SNE	0.410(\pm 0.195)	0.405(\pm 0.211)	0.391(\pm 0.193)	0.689(\pm 0.103)	0.720(\pm 0.155)	NA	0.472(\pm 0.081)	18.9
	UMAP	0.404(\pm 0.204)	0.388(\pm 0.214)	0.393(\pm 0.205)	0.663(\pm 0.134)	0.716(\pm 0.155)	NA	0.416(\pm 0.099)	20.6
DensityCut	HD	0.667(\pm 0.000)	0.800(\pm 0.089)	0.657(\pm 0.054)	0.812(\pm 0.012)	0.894(\pm 0.009)	0.440(\pm 0.000)	0.719(\pm 0.047)	9.3
	t-SNE	0.667(\pm 0.000)	0.921(\pm 0.005)	0.785(\pm 0.141)	0.813(\pm 0.003)	0.908(\pm 0.001)	0.827(\pm0.022)	0.787(\pm 0.000)	5.7
	UMAP	0.729(\pm 0.088)	0.928(\pm0.007)	0.938(\pm0.004)	0.821(\pm 0.022)	0.922(\pm 0.000)	0.716(\pm 0.022)	0.692(\pm 0.000)	4.1
RCC	HD	0.872(\pm 0.146)	0.719(\pm 0.120)	0.613(\pm 0.062)	0.847(\pm 0.045)	0.622(\pm 0.066)	0.113(\pm 0.225)	0.767(\pm 0.094)	10.0
	t-SNE	0.843(\pm 0.096)	0.639(\pm 0.109)	0.593(\pm 0.113)	0.200(\pm 0.022)	0.325(\pm 0.017)	0.123(\pm 0.039)	0.447(\pm 0.045)	16.7
	UMAP	0.865(\pm 0.168)	0.695(\pm 0.130)	0.734(\pm 0.182)	0.417(\pm 0.038)	0.547(\pm 0.026)	0.614(\pm 0.054)	0.306(\pm 0.018)	15.3
CDC	t-SNE	0.878(\pm 0.049)	0.890(\pm 0.029)	0.898(\pm 0.014)	0.796(\pm 0.003)	0.920(\pm 0.005)	0.803(\pm 0.006)	0.820(\pm0.063)	3.4
	UMAP	0.883(\pm0.026)	0.907(\pm 0.007)	0.930(\pm 0.005)	0.869(\pm0.004)	0.943(\pm0.002)	0.697(\pm 0.001)	0.761(\pm 0.046)	3.0

R4-Table 3. The average NMI (\pm standard deviation) of the top 20 clustering results obtained by multiple clustering algorithms on seven high-dimensional real-world datasets. (NA means not applicable)

		Iris	Seeds	Wine	PenDigits	Digits	MNIST-train	Baron-mouse1	Rank
K-means	HD	0.602(\pm 0.074)	0.522(\pm 0.067)	0.595(\pm 0.106)	0.704(\pm 0.028)	0.692(\pm 0.038)	0.432(\pm 0.021)	0.468 (\pm 0.042)	19.3
	t-SNE	0.601(\pm 0.094)	0.532(\pm 0.077)	0.528(\pm 0.081)	0.764(\pm 0.043)	0.852(\pm 0.041)	0.680(\pm 0.034)	0.582(\pm 0.035)	15.7
	UMAP	0.602(\pm 0.089)	0.536(\pm 0.073)	0.564(\pm 0.087)	0.807(\pm 0.033)	0.874(\pm 0.043)	0.728(\pm 0.031)	0.577(\pm 0.037)	13.4
DBSCAN	HD	0.729(\pm 0.011)	0.492(\pm 0.010)	0.490(\pm 0.006)	0.408(\pm 0.201)	0.129(\pm 0.149)	0.499(\pm 0.000)	0.332(\pm 0.021)	21.1
	t-SNE	0.727(\pm 0.11)	0.482(\pm 0.095)	0.644(\pm 0.044)	0.813(\pm 0.095)	0.855(\pm 0.046)	0.524(\pm 0.251)	0.722(\pm 0.044)	13.1
	UMAP	0.761(\pm 0.039)	0.548(\pm 0.020)	0.655(\pm 0.054)	0.719(\pm 0.174)	0.768(\pm 0.095)	0.650(\pm 0.044)	0.662(\pm 0.052)	12.3
CDP	HD	0.682(\pm 0.045)	0.571(\pm 0.063)	0.539(\pm 0.028)	0.632(\pm 0.016)	0.461(\pm 0.043)	NA	0.132(\pm 0.024)	20.3
	t-SNE	0.599(\pm 0.069)	0.528(\pm 0.070)	0.543(\pm 0.067)	0.833(\pm 0.039)	0.907(\pm 0.021)	NA	0.580(\pm 0.037)	14.9
	UMAP	0.618(\pm 0.071)	0.544(\pm 0.074)	0.591(\pm 0.084)	0.826(\pm 0.012)	0.910(\pm 0.013)	NA	0.580(\pm 0.030)	12.7
LGC	HD	0.734(\pm 0.000)	0.605(\pm 0.032)	0.772(\pm 0.022)	0.721(\pm 0.029)	0.590(\pm 0.118)	NA	0.167(\pm 0.059)	15.3
	t-SNE	0.735(\pm 0.016)	0.738(\pm 0.007)	0.707(\pm 0.028)	0.822(\pm 0.027)	0.907(\pm 0.019)	NA	0.663(\pm 0.058)	8.1
	UMAP	0.733(\pm 0.023)	0.624(\pm 0.019)	0.800(\pm 0.006)	0.761(\pm 0.023)	0.890(\pm 0.022)	NA	0.564(\pm 0.017)	11.6
Meanshift	HD	0.717(\pm 0.035)	0.607(\pm 0.059)	0.712(\pm 0.039)	0.707(\pm 0.057)	0.729(\pm 0.041)	0.028(\pm 0.065)	0.317(\pm 0.034)	16.3
	t-SNE	0.768(\pm 0.049)	0.632(\pm 0.038)	0.705(\pm 0.031)	0.714(\pm 0.088)	0.841(\pm 0.078)	0.728(\pm 0.026)	0.510(\pm 0.022)	11.9
	UMAP	0.747(\pm 0.030)	0.659(\pm 0.058)	0.739(\pm 0.094)	0.812(\pm 0.047)	0.876(\pm 0.041)	0.799(\pm 0.034)	0.573(\pm 0.012)	7.9
Ncut	HD	0.598(\pm 0.080)	0.512(\pm 0.067)	0.578(\pm 0.112)	0.020(\pm 0.005)	0.584(\pm 0.110)	NA	0.038(\pm 0.021)	22.6
	t-SNE	0.595(\pm 0.086)	0.533(\pm 0.074)	0.515(\pm 0.078)	0.826(\pm 0.047)	0.866(\pm 0.043)	NA	0.596(\pm 0.049)	15.9
	UMAP	0.602(\pm 0.088)	0.532(\pm 0.074)	0.554(\pm 0.082)	0.802(\pm 0.026)	0.869(\pm 0.044)	NA	0.542(\pm 0.037)	16.6
DensityCut	HD	0.734(\pm 0.000)	0.621(\pm 0.072)	0.359(\pm 0.073)	0.800(\pm 0.009)	0.881(\pm 0.009)	0.480(\pm 0.000)	0.571(\pm 0.066)	13.6
	t-SNE	0.734(\pm 0.000)	0.737(\pm 0.009)	0.688(\pm 0.072)	0.876(\pm0.004)	0.927(\pm 0.001)	0.855(\pm0.014)	0.667(\pm 0.000)	4.6
	UMAP	0.733(\pm 0.019)	0.752(\pm0.016)	0.807(\pm0.009)	0.864(\pm 0.010)	0.935(\pm 0.000)	0.825(\pm 0.016)	0.517(\pm 0.000)	5.7
RCC	HD	0.855(\pm0.091)	0.573(\pm 0.065)	0.422(\pm 0.043)	0.868(\pm 0.021)	0.788(\pm 0.050)	0.000(\pm 0.313)	0.038(\pm 0.004)	14.4
	t-SNE	0.773(\pm 0.046)	0.531(\pm 0.032)	0.496(\pm 0.028)	0.523(\pm 0.008)	0.643(\pm 0.006)	0.432(\pm 0.013)	0.568(\pm 0.024)	17.6
	UMAP	0.831(\pm 0.097)	0.642(\pm 0.040)	0.683(\pm 0.076)	0.713(\pm 0.018)	0.757(\pm 0.009)	0.723(\pm 0.019)	0.512(\pm 0.004)	12.1
CDC	t-SNE	0.813(\pm 0.031)	0.714(\pm 0.017)	0.713(\pm 0.015)	0.862(\pm 0.002)	0.914(\pm 0.005)	0.754(\pm 0.006)	0.744(\pm0.042)	4.0
	UMAP	0.783(\pm 0.017)	0.733(\pm 0.008)	0.791(\pm 0.010)	0.872(\pm 0.002)	0.937(\pm0.001)	0.751(\pm 0.001)	0.679(\pm 0.020)	3.1

R4-Table 4. The average ARI (\pm standard deviation) of the top 20 clustering results obtained by multiple clustering algorithms on seven high-dimensional real-world datasets. (NA means not applicable)

		Iris	Seeds	Wine	PenDigits	Digits	MNIST-train	Baron-mouse1	Rank
K-means	HD	0.376(\pm 0.159)	0.331(\pm 0.157)	0.400(\pm 0.203)	0.540(\pm 0.053)	0.616(\pm 0.074)	0.312(\pm 0.011)	0.182(\pm 0.022)	17.9
	t-SNE	0.358(\pm 0.188)	0.327(\pm 0.179)	0.322(\pm 0.169)	0.596(\pm 0.067)	0.718(\pm 0.118)	0.562(\pm 0.063)	0.321(\pm 0.075)	17.1
	UMAP	0.355(\pm 0.179)	0.333(\pm 0.174)	0.340(\pm 0.183)	0.670(\pm 0.086)	0.756(\pm 0.118)	0.636(\pm 0.094)	0.307(\pm 0.083)	15.7
DBSCAN	HD	0.574(\pm 0.030)	0.392(\pm 0.011)	0.384(\pm 0.006)	0.137(\pm 0.088)	0.051(\pm 0.085)	0.000(\pm 0.000)	0.117(\pm 0.013)	19.7
	t-SNE	0.597(\pm 0.047)	0.283(\pm 0.222)	0.629(\pm 0.092)	0.682(\pm 0.187)	0.721(\pm 0.108)	0.396(\pm 0.286)	0.719(\pm 0.022)	12.4
	UMAP	0.654(\pm 0.112)	0.440(\pm 0.058)	0.629(\pm 0.101)	0.524(\pm 0.261)	0.481(\pm 0.223)	0.477(\pm 0.106)	0.639(\pm 0.073)	12.9
CDP	HD	0.577(\pm 0.080)	0.492(\pm 0.117)	0.429(\pm 0.028)	0.397(\pm 0.010)	0.198(\pm 0.021)	NA	-0.005(\pm 0.004)	19.0
	t-SNE	0.370(\pm 0.138)	0.344(\pm 0.158)	0.380(\pm 0.148)	0.738(\pm 0.080)	0.871(\pm 0.039)	NA	0.344(\pm 0.010)	14.6
	UMAP	0.392(\pm 0.141)	0.377(\pm 0.162)	0.417(\pm 0.155)	0.742(\pm 0.017)	0.880(\pm 0.020)	NA	0.318(\pm 0.050)	13.6
LGC	HD	0.568(\pm 0.000)	0.529(\pm 0.063)	0.786(\pm 0.025)	0.594(\pm 0.033)	0.512(\pm 0.147)	NA	0.040(\pm 0.017)	15.7
	t-SNE	0.631(\pm 0.054)	0.783(\pm 0.007)	0.710(\pm 0.042)	0.692(\pm 0.074)	0.877(\pm 0.037)	NA	0.481(\pm 0.024)	9.0
	UMAP	0.663(\pm 0.038)	0.560(\pm 0.039)	0.808(\pm 0.007)	0.529(\pm 0.074)	0.840(\pm 0.045)	NA	0.284(\pm 0.041)	11.6
Meanshift	HD	0.561(\pm 0.042)	0.571(\pm 0.088)	0.727(\pm 0.054)	0.521(\pm 0.099)	0.653(\pm 0.080)	0.005(\pm 0.012)	0.115(\pm 0.021)	15.4
	t-SNE	0.735(\pm 0.075)	0.594(\pm 0.074)	0.704(\pm 0.039)	0.533(\pm 0.116)	0.726(\pm 0.170)	0.599(\pm 0.049)	0.507(\pm 0.058)	9.6
	UMAP	0.659(\pm 0.085)	0.635(\pm 0.111)	0.721(\pm 0.135)	0.698(\pm 0.096)	0.764(\pm 0.114)	0.662(\pm 0.073)	0.283(\pm 0.013)	8.7
Ncut	HD	0.366(\pm 0.155)	0.316(\pm 0.146)	0.419(\pm 0.192)	0.000(\pm 0.000)	0.311(\pm 0.100)	NA	0.002(\pm 0.005)	22.1
	t-SNE	0.353(\pm 0.173)	0.329(\pm 0.177)	0.306(\pm 0.163)	0.686(\pm 0.103)	0.736(\pm 0.126)	NA	0.337(\pm 0.055)	17.7
	UMAP	0.353(\pm 0.178)	0.317(\pm 0.173)	0.325(\pm 0.178)	0.638(\pm 0.087)	0.741(\pm 0.124)	NA	0.266(\pm 0.077)	19.0
DensityCut	HD	0.568(\pm 0.000)	0.564(\pm 0.104)	0.302(\pm 0.069)	0.699(\pm 0.017)	0.831(\pm 0.020)	0.305(\pm 0.000)	0.555(\pm 0.083)	12.3
	t-SNE	0.568(\pm 0.000)	0.781(\pm 0.012)	0.622(\pm 0.154)	0.818(\pm 0.005)	0.908(\pm 0.001)	0.806(\pm0.020)	0.680(\pm 0.005)	5.6
	UMAP	0.597(\pm 0.049)	0.797(\pm0.018)	0.816(\pm0.011)	0.791(\pm 0.015)	0.919(\pm 0.000)	0.670(\pm 0.018)	0.502(\pm 0.000)	4.3
RCC	HD	0.812(\pm 0.168)	0.538(\pm 0.116)	0.293(\pm 0.049)	0.790(\pm 0.059)	0.463(\pm 0.132)	0.000(\pm 0.236)	0.101(\pm 0.007)	15.1
	t-SNE	0.738(\pm 0.082)	0.492(\pm 0.091)	0.398(\pm 0.081)	0.168(\pm 0.019)	0.320(\pm 0.017)	0.100(\pm 0.037)	0.311(\pm 0.062)	15.9
	UMAP	0.806(\pm 0.169)	0.574(\pm 0.099)	0.616(\pm 0.158)	0.445(\pm 0.049)	0.558(\pm 0.028)	0.528(\pm 0.057)	0.203(\pm 0.014)	13.3
CDC	t-SNE	0.816(\pm0.043)	0.733(\pm 0.022)	0.729(\pm 0.016)	0.805(\pm 0.001)	0.905(\pm 0.005)	0.782(\pm 0.006)	0.769(\pm0.058)	3.0
	UMAP	0.735(\pm 0.010)	0.770(\pm 0.006)	0.796(\pm 0.012)	0.820(\pm0.005)	0.930(\pm0.001)	0.651(\pm 0.000)	0.635(\pm 0.062)	3.4

R4-Table 5. The average F1-score (\pm standard deviation) of the top 20 clustering results obtained by multiple clustering algorithms on seven high-dimensional real-world datasets. (NA means not applicable)

		Iris	Seeds	Wine	PenDigits	Digits	MNIST-train	Baron-mouse1	Rank
K-means	HD	0.476(\pm 0.172)	0.438(\pm 0.175)	0.499(\pm 0.200)	0.579(\pm 0.051)	0.648(\pm 0.069)	0.371(\pm 0.017)	0.357(\pm 0.051)	19.4
	t-SNE	0.452(\pm 0.196)	0.427(\pm 0.192)	0.428(\pm 0.184)	0.628(\pm 0.063)	0.740(\pm 0.112)	0.596(\pm 0.062)	0.428(\pm 0.100)	18.0
	UMAP	0.449(\pm 0.193)	0.434(\pm 0.189)	0.438(\pm 0.196)	0.698(\pm 0.081)	0.776(\pm 0.111)	0.667(\pm 0.092)	0.421(\pm 0.116)	16.6
DBSCAN	HD	0.747(\pm 0.009)	0.603(\pm 0.050)	0.604(\pm 0.009)	0.283(\pm 0.065)	0.218(\pm 0.062)	0.000(\pm 0.000)	0.396(\pm 0.051)	19.1
	t-SNE	0.750(\pm 0.010)	0.365(\pm 0.276)	0.745(\pm 0.053)	0.716(\pm 0.158)	0.752(\pm 0.098)	0.480(\pm 0.234)	0.807(\pm 0.100)	12.3
	UMAP	0.787(\pm 0.056)	0.633(\pm 0.028)	0.754(\pm 0.064)	0.591(\pm 0.210)	0.559(\pm 0.182)	0.552(\pm 0.087)	0.755(\pm 0.050)	12.1
CDP	HD	0.693(\pm 0.081)	0.609(\pm 0.121)	0.651(\pm 0.022)	0.481(\pm 0.008)	0.328(\pm 0.015)	NA	0.414(\pm 0.005)	19.0
	t-SNE	0.476(\pm 0.156)	0.461(\pm 0.166)	0.498(\pm 0.159)	0.763(\pm 0.068)	0.883(\pm 0.036)	NA	0.460(\pm 0.002)	14.9
	UMAP	0.495(\pm 0.152)	0.486(\pm 0.167)	0.530(\pm 0.154)	0.765(\pm 0.016)	0.891(\pm 0.019)	NA	0.428(\pm 0.083)	14.3
LGC	HD	0.746(\pm 0.000)	0.714(\pm 0.020)	0.858(\pm 0.017)	0.633(\pm 0.030)	0.566(\pm 0.113)	NA	0.438(\pm 0.021)	13.1
	t-SNE	0.746(\pm 0.022)	0.854(\pm 0.005)	0.804(\pm 0.029)	0.716(\pm 0.071)	0.888(\pm 0.034)	NA	0.577(\pm 0.019)	9.6
	UMAP	0.775(\pm 0.023)	0.680(\pm 0.032)	0.873(\pm 0.005)	0.557(\pm 0.073)	0.854(\pm 0.042)	NA	0.406(\pm 0.063)	13.1
Meanshift	HD	0.725(\pm 0.060)	0.715(\pm 0.055)	0.810(\pm 0.037)	0.580(\pm 0.081)	0.693(\pm 0.067)	0.186(\pm 0.009)	0.415(\pm 0.040)	14.3
	t-SNE	0.816(\pm 0.054)	0.703(\pm 0.064)	0.803(\pm 0.026)	0.578(\pm 0.102)	0.752(\pm 0.157)	0.636(\pm 0.044)	0.683(\pm 0.070)	10.6
	UMAP	0.782(\pm 0.044)	0.752(\pm 0.077)	0.821(\pm 0.079)	0.730(\pm 0.084)	0.791(\pm 0.099)	0.697(\pm 0.070)	0.391(\pm 0.016)	9.3
Ncut	HD	0.469(\pm 0.171)	0.431(\pm 0.163)	0.529(\pm 0.180)	0.182(\pm 0.000)	0.419(\pm 0.078)	NA	0.430(\pm 0.005)	20.7
	t-SNE	0.450(\pm 0.188)	0.430(\pm 0.191)	0.416(\pm 0.175)	0.714(\pm 0.091)	0.758(\pm 0.118)	NA	0.452(\pm 0.077)	18.3
	UMAP	0.448(\pm 0.194)	0.420(\pm 0.189)	0.425(\pm 0.194)	0.669(\pm 0.084)	0.762(\pm 0.117)	NA	0.397(\pm 0.110)	20.3
DensityCut	HD	0.746(\pm 0.000)	0.720(\pm 0.047)	0.560(\pm 0.025)	0.730(\pm 0.015)	0.848(\pm 0.018)	0.417(\pm 0.000)	0.708(\pm 0.051)	10.4
	t-SNE	0.746(\pm 0.000)	0.853(\pm 0.008)	0.773(\pm 0.077)	0.834(\pm 0.004)	0.917(\pm 0.000)	0.824(\pm0.024)	0.784(\pm 0.003)	5.0
	UMAP	0.756(\pm 0.022)	0.864(\pm0.012)	0.878(\pm0.007)	0.811(\pm 0.014)	0.927(\pm 0.000)	0.712(\pm 0.020)	0.679(\pm 0.000)	4.3
RCC	HD	0.880(\pm0.103)	0.699(\pm 0.070)	0.572(\pm 0.072)	0.811(\pm 0.051)	0.542(\pm 0.108)	0.182(\pm 0.183)	0.761(\pm 0.107)	10.4
	t-SNE	0.818(\pm 0.064)	0.626(\pm 0.098)	0.550(\pm 0.097)	0.193(\pm 0.022)	0.352(\pm 0.018)	0.119(\pm 0.033)	0.416(\pm 0.072)	17.3
	UMAP	0.857(\pm 0.137)	0.682(\pm 0.091)	0.711(\pm 0.141)	0.473(\pm 0.049)	0.586(\pm 0.027)	0.583(\pm 0.048)	0.287(\pm 0.019)	15.0
CDC	t-SNE	0.872(\pm 0.031)	0.821(\pm 0.016)	0.818(\pm 0.011)	0.821(\pm 0.001)	0.914(\pm 0.005)	0.801(\pm 0.006)	0.840(\pm0.072)	3.3
	UMAP	0.823(\pm 0.009)	0.845(\pm 0.005)	0.865(\pm 0.008)	0.837(\pm0.004)	0.937(\pm0.001)	0.695(\pm 0.000)	0.746(\pm 0.048)	3.4

R4-Fig. 1. The clustering results of CDC combined with t-SNE and UMAP on seven high-dimensional real-world datasets.

R4-Fig. 2. Comparison of nine clustering algorithms in ACC and running time on PenDigits and Digits datasets.

R4-Fig. 3. Comparison of nine clustering algorithms in ACC and running time on MNIST-train and Baron-mouse1 datasets.

According to your suggestion, we have revised the manuscript (line 287-312, page 15-16) as well as the Supplementary Information (Supplement Figure 9-11; Supplement Table 2-6).

Comment 2

Mean-shift (Chen YZ, Mean-shift, PAMI, 1995) is a widely used density-based clustering algorithm based on density model-seeking. This 'model-seeking' approach is complementary to the 'boundary-seeking' used by CDC. The authors need to add mean-

shift to discussions and comparisons. It seems that the 'model-seeking' approach is an easier and more elegant solution to density-based clustering as CDC is sensitive to the definition of the boundary points.

Thank you for your comment and suggestion. We have added Meanshift as a baseline on seven real-world datasets and the results are presented in the response of Comment 1. **CDC outperforms Meanshift in accuracy and has close time efficiency.** The discussions about the parameter sensitivity of CDC is presented in the responses of Comment 3 and 7.

Comment 3

Compared to Mean-shift, which has only one parameter, the kernel density estimation bandwidth, and densityCut, which only has one parameter, the number of neighbors, CDC needs both the number of neighbors and also the threshold to define boundary points. As the authors discussed, CDC is sensitive to these parameters. The authors need to show that the added complexity improves performance compared to the simpler methods.

Thank you for your valuable comment. **In terms of the number of parameters, CDC is more complex than Meanshift but same to DensityCut.** Meanwhile, according to the experimental results, CDC outperforms the clustering baselines (e.g., Meanshift, DensityCut, DBSCAN, etc), and only has one sensitive parameter (i.e., T_{DCM}).

Meanshift is definitely an elegant and powerful algorithm, which has only one parameter. However, the bandwidth is difficult to specify intuitively when switching to a new dataset. Similar to the Eps in DBSCAN, it is a sensitive distance-based parameter having an unfixed value range varied with the data distribution. **Determining the most appropriate bandwidth requires a measurement in advance and intensive trials. While DensityCut actually has two parameters**, i.e., the number of nearest neighbors k , and the damping factor α in density refinement. CDC is similar to DensityCut in the number and type of the parameters. k is an insensitive parameter and T_{DCM} is sensitive to the clustering results of CDC. As shown from the experimental results, **CDC has higher accuracy and stronger robustness than Meanshift and DensityCut on the abovementioned seven real-world datasets.**

Comment 4

In Figure 1a, the authors compared CDC with baselines using synthetic datasets. Although useful, the problem is that these algorithms, e.g., DBSCAN, are very sensitive to parameters, and the performance is largely determined by the parameters used. For a fair comparison, the authors need to tune the parameters the same way as for CDC, or at least for all methods (including CDC) used the default setting.

Thank you for your comment. **We believe that the comparison in our experiment is fair, since we compared the optimal clustering results of each algorithm in Fig. 1 by exploring all potential parameter settings in a brute-force manner.** Specifically, we set the parameters of each algorithm by tuning the sizes and combinations of parameters

extensively. We varied the parameter with a fine-granularity interval at each time to obtain the clustering results with higher accuracy. We present the parameter settings corresponding to the optimal clustering results in R4-Fig. 4 and we have submitted a supplementary excel file that contains all parameter settings in this paper.

Fig. 1 Comparison with baselines on synthetic datasets

	DS1	DS2	DS3
K-means	C = 10, iteration = 200	C = 6, iteration = 200	C = 7, iteration = 200
CDP	Manually selected 10 cluster centers	Manually selected 6 cluster centers	Manually selected 7 cluster centers
DBSCAN	Eps = 0.3105, Minpts = 1	Eps = 3, Minpts = 1	Eps = 45, Minpts = 1
LGC	k = 40, IM = 10, c = 0.3	k = 15, IM = 17, c = 0.21	k = 60, IM = 8, c = 0.4
CDC	C = 30, T _{DCM} = 0.1	C = 8, T _{DCM} = 0.17	C = 11, T _{DCM} = 0.24

R4-Fig. 4. The parameter settings of five clustering algorithms with the highest accuracy on three synthetic datasets.

According to your suggestion, we have uploaded the Supplementary Parameter Settings.

Comment 5

Similarly, in Figure2, the authors showed that CDC performed favorably compared to other methods. It's hard to know if this is a parameter setting problem. Could the authors show the results of using different combinations of parameters?

Thank you for your comment. We believe the comparisons in our experiments are fair, since **we compared the global top 20 clustering results in accuracy for each algorithm through a two-step search of the parameter ranges and intervals**. Firstly, we roughly go through the whole range of parameters to determine the samples with relatively high accuracy. Then, we set a sub-range and fine-granularity interval around the samples, and further calculate the clustering accuracy obtained by all possible combinations. We select the top 20 clustering results from all parameter combinations, **since a number of top results can reveal the best performance and robustness of each algorithm**. We present an example of the parameter settings in R4-Fig. 5 and more details can be seen from the newly uploaded supplementary excel file. (e.g., “k=1:30, iterations=200” means that we set the k ranging from 1 to 30 and iterations as 200 for K-means on Iris dataset).

Fig. 2(f-l) Comparison with baselines on scRNA-seq datasets (k=1:30 means setting k from 1 to 30 with the interval of 1)

	Baron-mouse1	Baron-human1	Baron-human3	ALM
Kmeans-UMAP	C = 1.30, iteration = 200	C = 1.30, iteration = 200	C = 1.30, iteration = 200	C = 1.30, iteration = 200
CDP-UMAP	C = 1.30, rho_x:delta_y=0.4:0.6	C = 1.30, rho_x:delta_y=0.4:0.6	C = 1.30, rho_x:delta_y=0.4:0.6	C = 1.30, rho_x:delta_y=0.4:0.6
DBSCAN-UMAP	Eps = 0.3:0.16, Minpts = 1	Eps = 0.3:0.5:10, Minpts = 1	Eps = 0.5:0.3:10, Minpts = 1	Eps = 0.1:0.02:4, Minpts = 1
LGC-UMAP	k = 10:10:50, IM = 10, c = 0.1:0.05:0.5	k = 10:10:50, IM = 10, c = 0.1:0.05:0.5	k = 10:10:50, IM = 10, c = 0.1:0.05:0.5	k = 10:10:50, IM = 10, c = 0.1:0.05:0.5
flowSOM-PCA50	C = 1.30	C = 1.30	C = 1.30	C = 1.30
PhenoGraph-PCA50	k=10:2:48	k=10:2:48	k=10:3:67	k=10:5:105
CDC-UMAP	k = 5:1:20, T _{DCM} = 0.35:0.01:0.5	k = 10:1:30, T _{DCM} = 0.3:0.01:0.45	k = 10:2:20, T _{DCM} = 0.1:0.05:0.5	k = 20:2:60, T _{DCM} = 0.08:0.01:0.13

R4-Fig. 5. The parameter settings of all clustering algorithms on scRNA-seq datasets.

According to your suggestion, we have uploaded the Supplementary Parameter Settings.

Comment 6

Why the authors only reported ARI in Figure 3? Why not reporting FI-score as in Figure 1. Also, normalized mutual information is a widely used metric. In addition, why the authors

only used 4 datasets for the comparison in Supp. Fig. 2 instead of all the 13 datasets? Supp.Fig.2 is a very informative figure, but again, it's not clear how the parameters were chosen for CDC. The authors should do the same parameter selection for all methods for a fair comparison.

Thank you for your suggestion. According to your suggestion, we have reported six validity indexes and added the detailed parameter settings in the supplementary files. Due to limits of the Web-based platform of SciBet, we only used four scRNA-seq datasets in the comparison. More details are listed as follows:

1) Selection of the validity indexes

We have calculated the results of six validity indexes in all the experiments, including Accuracy (ACC), Normalized Mutual Information (NMI), Adjusted Rand Index (ARI), F1-score, Rand Index (RI) and Jaccard Index (JI). Limited by the length of manuscript, we only present ARI in the manuscript, and the results of other five indexes are detailed in the Supplementary Information and two newly uploaded Excel files, i.e., Supplementary Result 1-2 (R4-Fig. 6 gives an example). **The results of all the six validity indexes prove the effectiveness and robustness of CDC.** We also added the equations of six validity indexes used in Supplementary Note 3 as follows:

To evaluate the clustering performance quantitatively, we adopt six validity indexes Accuracy (ACC), Normalized Mutual Information (NMI), Adjusted Rand Index (ARI), F1-score, Rand Index (RI) and Jaccard Index (JI).

ACC refers to the accuracy rate of the clustering results compared with the true labels. We set the true label vector and the predicted label vector as $\mathbf{l} = (l_1, l_2, \dots, l_n) \in \mathbb{R}^n$ and $\mathbf{r} = (r_1, r_2, \dots, r_n) \in \mathbb{R}^n$ respectively, and the ACC can be defined as

$$ACC = \frac{\sum_{i=1}^n \delta(l_i, \text{map}(r_i))}{n}$$

where $\delta(\cdot)$ denote an indicator function

$$\delta(x, y) = \begin{cases} 1 & \text{if } x = y \\ 0 & \text{otherwise} \end{cases}$$

$\text{map}(\cdot)$ is a mapping function that maps each predicted label to one of the true cluster label. Commonly, the best mapping can be found by using the Kuhn-Munkres or Hungarian Algorithm.

NMI measures the agreement of the predict and true assignments, ignoring permutations. It can be defined as

$$NMI = \frac{\sum_{i=1}^{|L|} \sum_{j=1}^{|R|} |L_i \cap R_j| \log \frac{n |L_i \cap R_j|}{|L_i| |R_j|}}{\sqrt{\left(\sum_{i=1}^{|L|} |L_i| \log \frac{|L_i|}{n} \right) \left(\sum_{j=1}^{|R|} |R_j| \log \frac{|R_j|}{n} \right)}}$$

where L_i denote the point set of the i th cluster that predicted by the algorithm, while R_j denotes the j th cluster of the true labels.

We define the number of point pairs that belong to the same clusters in both of the true and predicted labels as TP , that belong to the different clusters in both of the true and predicted labels as TN , that belong to the same clusters in the true labels but not in the predicted labels as FP , that belong to the same clusters in the predicted labels but not in the true labels as FN . Then we have

$$ARI = \frac{2(TP \cdot TN - FP \cdot FN)}{2(TP \cdot TN - FP \cdot FN) + (FP + FN)(TP + TN + FP + FN)}$$

$$F1\text{-score} = \frac{2TP}{2TP + FP + FN}$$

$$RI = \frac{TP + TN}{TP + TN + FP + FN}$$

$$JI = \frac{TP}{TP + FP + FN}$$

	Accuracy	NMI	ARI	Fscore	JI	RI
Kmeans	0.6922	0.5174	0.5002	0.6793	0.5143	0.7478
	0.6083	0.6062	0.4610	0.6187	0.4480	0.7760
	0.5341	0.5749	0.3757	0.5492	0.3785	0.7494
	0.4635	0.6225	0.3756	0.4873	0.3221	0.7906
	0.4404	0.6138	0.3658	0.4740	0.3106	0.7898
	0.4915	0.5834	0.3459	0.4923	0.3265	0.7643
	0.4185	0.6161	0.3422	0.4381	0.2805	0.7892
	0.4161	0.5990	0.3343	0.4414	0.2832	0.7823
	0.4075	0.6077	0.3209	0.4153	0.2621	0.7842
	0.3954	0.5906	0.2987	0.3928	0.2444	0.7785
	0.4611	0.5659	0.2972	0.4472	0.2880	0.7512
	0.5122	0.4605	0.2886	0.5257	0.3566	0.6696
	0.4027	0.5864	0.2766	0.3655	0.2236	0.7746
	0.3881	0.5787	0.2608	0.3480	0.2106	0.7710
	0.3832	0.5815	0.2531	0.3333	0.2000	0.7714
	0.3564	0.5800	0.2473	0.3256	0.1945	0.7705
	0.3309	0.5735	0.2370	0.3199	0.1904	0.7662
	0.3297	0.5828	0.2135	0.2836	0.1652	0.7641
	0.3029	0.5672	0.2097	0.2829	0.1647	0.7621
	0.3078	0.5568	0.1901	0.2586	0.1485	0.7582

R4-Fig. 6. Top 20 of the six validity indexes of K-means in Supplementary Result 1.

2) Selection of the single-cell datasets

We have compared CDC with the supervised method SciBet on four scRNA-seq datasets in Supp. Fig. 2 **by utilizing the Web-based platform provided by the authors (<http://scibet.cancer-pku.cn/>)**. SciBet is a deep learning classification model that requires extensive model training in advance to learn the features of different types of cells. **Since the online platform only provided limited scRNA-seq datasets that have been trained for validation, we have to select from the provided datasets.** We tested all 13 datasets on the platform, but only the selected four are applicable.

3) Parameter settings

We have adjusted the combinations and intervals extensively to obtain a high accuracy (see the response of the Comment 5). All the parameter settings are available in the supplementary excel file, i.e., Supplementary Result 1-2.

According to your suggestion, we have revised the Supplementary Information

(Supplement Note 3; Supplementary Figure 4) and uploaded the Supplementary Result 1-2 and Supplementary Parameter Settings.

Comment 7

The claim that CDC is insensitive to k is not generally true, as demonstrated in Figure 4d. Thank you. According to our experimental results, we insist upon our opinion that T_{DCM} is a sensitive parameter but k is relatively insensitive to the clustering results in a certain range (i.e., $k \in [0, 0.1n]$, $T_{DCM} \in [0, 0.5]$, since the clustering qualities obtained make no sense when k is greater than $0.1n$ or T_{DCM} surpass 0.5). To make the observation more convincing, we added two ARI curve charts of k under a group of fixed T_{DCM} in Fig. 3 of the revision, and redesigned the method of sensitivity analysis and enhanced the discussions.

The sensitivity of a parameter depicts the stability of the performance as it varies. We present the ARI curves under different combinations of k and T_{DCM} in R4-Fig. 7. As T_{DCM} changes, ARI goes up and down dramatically, and there is no a certain range of k keeping ARI in a stable level when T_{DCM} varies. While in a certain range of T_{DCM} , the ARI curves fluctuated between a stable range from 0.3 to 1 when k varies in $[0.15, 0.4]$. This pattern reveals that the clustering accuracy is unstable with the variation of T_{DCM} , but is steady in a certain range of k , which provides additional evidence for the observations of parameter sensitivity analysis.

R4-Fig. 7. ARI curves obtained by varying T_{DCM} under fixed k and varying k under fixed T_{DCM} on ELSDSR and MSLT datasets respectively, where the grey bands represent the ARI ranges when k and T_{DCM} are in the ranges of [10, 20] and [0.15, 0.40], and the curves falling in the bands were sampled with fixed intervals of 2 and 0.05.

Furthermore, to make the result of sensitivity analysis more reasonable, we redesigned our analysis method. We previously analyzed the sensitivity using Latin-Hypercube One-factor-At-a-Time (LH-OAT) method. However, **the sampling points generated by LH are insufficient to depict the complete range of the two parameters**, since LH divides the two-parameter space into $m \times m$ grid and only samples one points in each row or column. Meanwhile, the previous sensitivity indexes in Eq. (1) and (2) are affected by the proportion of the perturbation size to the whole range of corresponding parameter. Because the potential value range of k is from 0 to n and that of T_{DCM} is from 0 to 1, while the appropriate value ranges of them are $[0, 0.1n]$ and $[0, 0.5]$ respectively according to our experiments. To improve the accuracy and stability of the analysis approach, we have modified the original LH-OAT method and exerted perturbations with finer granularities.

$$S_k = \frac{1}{m} \sum_{i=1}^m \left| \frac{\text{ARI}(k_i + \Delta k, T_i) - \text{ARI}(k_i, T_i)}{\text{ARI}(k_i + \Delta k, T_i) + \text{ARI}(k_i, T_i)} / \frac{\Delta k}{2k_i + \Delta k} \right| \quad (1)$$

$$S_T = \frac{1}{m} \sum_{i=1}^m \left| \frac{\text{ARI}(k_i, T_i + \Delta T) - \text{ARI}(k_i, T_i)}{\text{ARI}(k_i, T_i + \Delta T) + \text{ARI}(k_i, T_i)} / \frac{\Delta T}{2T_i + \Delta T} \right| \quad (2)$$

The simulation results shown in R4-Table 5 demonstrate that **T_{DCM} is a sensitive parameter, while k is relatively insensitive**. With the increase of perturbation, both of the two sensitivity indexes grow. **k only reached the mild sensitive level when the perturbation is 20**, since the local centrality of most points is stable as k varies within a certain range. However, **T_{DCM} tended to reach the hypersensitive level when the perturbation is larger than 0.15**. As a continuous variable between 0 and 1, a small range of DCM may correspond to a considerable number of points. A slight adjustment of T_{DCM} may cause many points to be converted from boundary to internal points or vice versa. This conversion affects the recognition of boundary points and their binding force for internal points, which in turn influences the final clustering quality. Therefore, k can be specified with a coarse granularity in a certain range, whereas the tuning of T_{DCM} must be performed deliberately.

R4-Table 5. Sensitivity analysis of parameters in CDC using stratified sampling and random perturbation on six synthetic datasets. The blue shade represents the sensitivity levels, and the darker color indicates the stronger sensitivity.

Parameter	Perturbation	DS4 n=442	DS5 n=788	DS6 n=1000	DS7 n=1500	DS8 n=3000	DS9 n=6847
K	1	0.0389	0.0363	0.0308	0.0973	0.0338	0.0050
	5	0.0893	0.0719	0.0400	0.1187	0.0723	0.0174
	10	0.1790	0.1864	0.1036	0.1988	0.1678	0.0282
	15	0.2608	0.2341	0.1347	0.2403	0.2177	0.0320
	20	0.3584	0.2865	0.1744	0.2821	0.2734	0.0491
T_{DCM}	0.01	0.0580	0.1198	0.0844	0.2403	0.0916	0.0704
	0.05	0.1836	0.2577	0.1888	0.4276	0.5224	0.2525
	0.10	0.2622	0.3809	0.3209	0.6034	0.8385	0.3629
	0.15	0.3921	0.6968	0.4342	0.9683	0.9286	0.4495
	0.20	0.5428	0.7812	0.5021	0.9878	0.9813	0.5644

Sensitivity levels

For the analysis method, as shown in R4-Fig. 8, we firstly divided the value range of the two parameters using 10 equal intervals respectively, which generated a 10 by 10 grid with cells of the same size. Then we randomly selected one sampling point in each cell and exert a positive or negative perturbation to each point. We specified five fixed perturbations for each parameter. Only one parameter was varied at a time and the other was fixed. Finally, we calculated the sensitivity indexes that measures the relative change of the clustering quality before and after perturbations. Each group of experiment was conducted five times to avoid randomness in the simulations.

Adjusted Rand Index (ARI) was used as the objective function, since it can assess the clustering quality strictly. To ensure that ARI is positive ranging from 0 to 1, we conducted the sampling in the range of k from 0 to $0.1n$ and that of T_{DCM} from 0 to 0.5. Because the clustering qualities obtained by most of sampling points make no sense when k is greater than $0.1n$ or T_{DCM} surpass 0.5 (e.g., all the points are identified as internal points that form a single cluster), which would generate excess invalid results (ARI = 0) that affect the effectiveness of the sensitivity evolution. The sensitivity indexes S_k , S_T of k and T_{DCM} are defined as:

$$S_k = \frac{1}{100} \sum_{i=1}^{100} \left| \frac{ARI(k_i + \Delta k, T_i) - ARI(k_i, T_i)}{ARI(k_i + \Delta k, T_i) + ARI(k_i, T_i)} \right| \quad (3)$$

$$S_T = \frac{1}{100} \sum_{i=1}^{100} \left| \frac{ARI(k_i, T_i + \Delta T) - ARI(k_i, T_i)}{ARI(k_i, T_i + \Delta T) + ARI(k_i, T_i)} \right| \quad (4)$$

R4-Fig. 8. Illustration of sensitivity analysis using stratified sampling and random perturbation. (a) Sampling points generated by stratified sampling. **(b)** Operating random perturbations that could be either positive or negative to each sampling point.

According to your suggestion, we have revised the manuscript (line 247-251, page 13; line 269-275, page 14; line 350-362, page 18; line 364-369, 375-378, page 19) as well as the Supplementary Information (Supplement Note 6; Supplement Figure 2).

Comment 8

Figure 6 legend, 'blue represents intra-cluster triangles ...' should be 'inter-cluster'?

Thank you for your correction, we have revised the previous Fig. 6 as Fig. 4 and modified 'blue represents intra-cluster triangles ...' to 'blue represents cross-cluster triangles ...' in line 399-400, page 21.

Reviewers' Comments:

Reviewer #1:

Remarks to the Author:

Thanks for the authors' exhaustive explanations, which solves my concerns to some extent. It would be better if the authors could further consider the following questions and suggestions.

1. The supplemented VDD, SDC, and OPP work unstably on the high dimensional datasets, which suggests that applying CDC in high dimensional spaces with the above three strategies is impractical. In other words, the performance of CDC relies heavily on the dimensional reduction techniques such as tSNE and UMAP. As a result, the application of the proposed CDC is limited since dimensional reduction techniques might fail for data with higher dimensionality.
2. The authors reported clustering results on several datasets compared with other methods in R1-Table 3. However, I notice the clustering NMI (0.00) of RCC (Robust Continuous Clustering) on MNIST is much lower than the results reported in the original paper (which is 0.893 AMI). Please explain this result or make sure the compared methods are properly conducted. Besides, most of the datasets for evaluation are quite small, which is not that representative.
3. In the sensitivity analysis, the author claims that the Adjusted Rand Index (ARI) is adopted as the objective function. I do not quite understand the reason to do so. Also, how are the sensitivity levels in R1-Table 1 computed? The authors could give more explanations on that.

In brief, I believe the idea behind CDC is interesting. But my major concern lies in the generality and stability of CDC since it relies heavily on dimensional reduction techniques and outlier detection strategy, etc.

Reviewer #2:

Remarks to the Author:

All of my previous comments have been addressed very well by the authors. I hence suggest that the paper should be accepted in its current form.

David B. Blumenthal

Reviewer #3:

Remarks to the Author:

Major comments

The previous set of comments noted that the evaluations for scRNA-seq and CyTOF data relied on small and outdated datasets, and did not compare against standard clustering analysis pipelines for scRNA-seq and CyTOF data. Unfortunately, the new analyses and datasets in the updated manuscript do not sufficiently address this comment. Crucially, the evaluations still do not compare against standard scRNA-seq pipelines (e.g. Seurat / Bioconductor), using typical full-sized datasets (up to 1 million cells). There has also been a misunderstanding in applying CyTOF clustering methods to scRNA-seq data. Unfortunately, these results do not indicate that this clustering method could be used for analyzing typical scRNA-seq or CyTOF data.

Specifically:

(i) The analyses still rely on old and outdated datasets. Most of the scRNA-seq datasets (line 134 and Supplementary Table 1) are extremely small by modern standards.

For example, the main dataset described on line 134 has 801 cells and is sourced from a paper from 2013. Modern scRNA-seq datasets can contain >1 million cells, so these datasets are not representative of typical scRNA-seq clustering tasks.

Similarly, the CyTOF datasets are relatively small and outdated. Newer CyTOF datasets can routinely contain 40 dimensions and 1 million cells.

(ii) The previous comments noted that the authors did not compare against standard analysis pipelines for scRNA-seq and CyTOF data. Specifically, the comments mentioned: (a) for scRNA-seq: dimension reduction to 50 principal components (PCs) on a subset of highly variable genes followed by graph-based clustering (for example, standard pipelines are available from Seurat or Bioconductor), and (b) for CyTOF: clustering in the full-dimensional space using FlowSOM or PhenoGraph.

The authors have included new analyses for the scRNA-seq data, but these represent a fundamental misunderstanding of scRNA-seq and CyTOF analysis pipelines. Specifically, for scRNA-seq, the authors have applied FlowSOM and PhenoGraph in 50-dimensional PCA space – i.e. this is incorrectly using the CyTOF clustering methods for scRNA-seq data, as well as ignoring any additional preprocessing steps from the Seurat / Bioconductor pipelines such as selecting highly variable genes. The evaluations do not compare against the standard pipelines, so these results do not provide a representative performance comparison.

For the CyTOF data, the authors also mention that they “removed the cells without manually gated population labels” (Supplementary Note 4, page 6). This is inconsistent with previous published evaluations of these CyTOF datasets, where clustering was performed on all cells (including unlabelled) and performance subsequently calculated on the subset of known cells. Therefore, the evaluations here represent a much easier clustering task, and an inconsistent comparison.

(iii) The runtime evaluations (line 207 onwards) do not present a comprehensive evaluation of runtimes on modern, full-sized scRNA-seq datasets (e.g. 1 million cells), which can be routinely handled by the standard Seurat / Bioconductor pipelines.

For example, in the rebuttal letter (Reviewer 4, Comment 4), the authors mention “Limited by the memory of a commodity desktop computer (memory overflow problem of the dimension reduction for PCA, t-SNE and UMAP), we could not further test CDC-UMAP in a scRNA-seq dataset with millions of cells and ten thousand of genes.” However, analyses on millions of cells and tens of thousands of genes are routinely performed with Seurat / Bioconductor, so this represents a fundamental limitation in the applicability of the method.

Similarly, the rebuttal letter (Reviewer 4, Comment 4) mentions “Comparing with the recent publication on Nature Communications, Li et al. (2020) (the maximum scRNA-seq dataset has 100,000 cells and 500 genes), we believe our experiments follow the modern standard in data size”. This represents a misunderstanding of the scale of scRNA-seq data, which typically has tens of thousands of genes (i.e. transcriptome-wide), not 500.

Reviewer #4:

Remarks to the Author:

I appreciate the authors for the efforts in addressing my comments. The major limitations of the algorithm include only processing 2-dimensional data and the sensitivity of the algorithms on the parameter T. The authors’ empirical approach for selecting the T parameter also requires the number of clusters as inputs, which is typically unknown for real datasets. Thus I suggest the authors make these points clear early, e.g., in the introductions for the interesting readers to quickly grab the strength and weakness of their algorithm. Also, the authors’ claim about the computational efficiency of their algorithm is a little misleading, e.g., as in Fig.2m because Phonograph is run on 50D data and CDC is run on 2D data.

As for scRNA-seq data analysis, from both our internal analyses and other people of the field, for many datasets, we almost surely can get better clustering results by increasing the

dimensionalities of UMAP from 2D to higher dimensions such as 5D. Also, the 'standard' pipeline of the field clusters the principal components instead of UMAPs. The number of principal components also has a big impact on the clustering results. I don't want that the authors give the information to the field that people should cluster the 2D UMAPs. Thus, I suggest that the authors acknowledge the limitations of the results in Figure 2, e.g., no comparisons from using higher-dimensional UMAPs or using different numbers of principal components, and the results are specific to the data used, may not generalize to other scRNA-seq data.

If the authors do think clustering on 2D UMAPs is the preferred way for scRNA-seq data analysis, I suggest that the authors do more comprehensive experiments, e.g., using different numbers of PCs and different dimensional UMAPs, and following the standard pipelines of the field, e.g., considering the gene/feature selection step and the potential batch-correction step. Also, it's essential to make the scripts and data available so the readers can easily reproduce the comparison results as the results may change the way people clustering scRNA-seq data.

The CDC algorithm explores the KNN graph to identify boundary points and share similarities with other approaches. To put the CDC algorithm into context in the introduction section, I suggest that the authors discuss the relevant literature on 'in-neighbors' and other concepts such as 'reversed knn' in the introduction.

Reviewer #1 (Remarks to the Author):

Thanks for the authors' exhaustive explanations, which solves my concerns to some extent. It would be better if the authors could further consider the following questions and suggestions.

1. The supplemented VDD, SDC, and OPP work unstably on the high dimensional datasets, which suggests that applying CDC in high dimensional spaces with the above three strategies is impractical. In other words, the performance of CDC relies heavily on the dimensional reduction techniques such as tSNE and UMAP. As a result, the application of the proposed CDC is limited since dimensional reduction techniques might fail for data with higher dimensionality.

In this revision, we generalized CDC to higher-dimensional space by expanding the original definition of DCM (i.e., variance of angles). Experiments validated the effectiveness of CDC to handle high-dimensional data. Actually, often times that real-world data are distributed with manifold structures (data dimension is lower than feature dimension). Local direction centrality should be applicable in space of dimension lower than feature dimension. UMAP has been widely used to embed the manifold data into a proper dimension with low loss of local information. In our experiments of several application scenarios (i.e., synthetic, scRNA-seq, CyTOF, corpus and other real-world datasets), CDC-UMAP had high and stable outcomes with promising time efficiency, it even outperformed some clustering baselines. Therefore, we believe that CDC-UMAP can handle data with high dimensionality. Currently, CDC has been generalized to space of any dimension, but we recommend to apply it to low UMAP dimensions for the sake of time efficiency.

The idea of DCM generalization to high-dimensional space: DCM calculation requires to map the KNNs onto the unit spherical surface drawn by their center point firstly, then subdivide the spherical surface and measure the corresponding angles of the subdivision units (New Supplementary Fig. 6). It is intuitive to understand the definition of DCM that measures the variance of angles formed by KNNs in 2D space. However, subdividing the hyperspherical surface and measuring the generalized angles are complex in high-dimensional space. Actually, the generalized angles are equivalent to the corresponding the volume of the subdivision units (e.g., arc length in 2D circle, area of the spherical triangle in 3D sphere). So, the angle measurement can be converted to the volumes of the subdivision units. To subdivide the hyperspherical surface, we utilize Qhull algorithm to construct the convex complex of KNNs. Then, we calculate the volumes of the simplices (facets of the complex) and further estimate the volumes of the subdivision units. *The detailed calculation process of DCM is presented in Line 565-616, Page 29-31, and the proofs of simplex volume calculation are presented in Supplementary Note 6.*

New Supplementary Fig. 6. Graphical illustration of expanding DCM with hyperspherical subdivisions for high-dimensional spaces.

To validate the effectiveness of the generalized CDC on high-dimensional datasets, we selected **nine real-world datasets**, i.e., Iris, Seeds, Breast-Cancer, Wine, PenDigits, Dermatology, Control, Digits, MNIST10k (details in Revised Supplementary Table 2), to compare with **ten generic clustering baselines**, i.e., K-means, DBSCAN, CDP, AGNES, MeanShift, Rcut, Ncut, densityCut, RCC, GCSED. We conduct CDC in 2, 3, 4, 5 UMAP dimensions (denoted as CDC-U2, CDC-U3, CDC-U4, CDC-U5). ACC, NMI and ARI are reported for clustering accuracy in Revised Supplementary Table 3-5. In general, CDC obtained the highest scores on eight datasets under different UMAP dimensions and have significant advantages than the baselines. CDC-U2, CDC-U3, CDC-U4, CDC-U5 are the top four in the average rank of ACC, NMI and ARI.

Revised Supplementary Table 2. Nine real-world datasets used in the experiment of dimension expansion.

Dataset	No. of samples	No. of dimensions	No. of classes
Iris	150	4	3
Seeds	210	7	3
Breast-Cancer	683	10	2
Wine	178	13	3
PenDigits	10,992	16	10
Dermatology	358	34	6
Control	600	60	6
Digits	5,620	8×8 (64)	10
MNIST10k	10,000	28×28 (784)	10

Revised Supplementary Table 3. Evaluation of CDC and other 10 algorithms on 9 high-dimensional real-world datasets. The highest ACC score of all clustering algorithms in each dataset is highlighted in bold. The rank scores of all algorithms are summarized in the last column.

	Iris	Seeds	Breast	Wine	PenDigits	Dermatology	Control	Digits	MNIST10k	Rank
K-means	0.8867	0.8905	0.9605	0.9438	0.6616	0.8134	0.5683	0.7975	0.5269	6.8
DBSCAN	0.6733	0.5810	0.7570	0.5956	0.3854	0.5056	0.5500	0.6480	0.1028	11.8
CDP	0.6533	0.8905	0.8199	0.8820	0.5987	0.7437	0.5350	0.3863	0.2956	10.3
AGNES	0.6600	0.6000	0.7672	0.5955	0.3856	0.5042	0.5333	0.2014	0.1027	12.9
MeanShift	0.6667	0.8524	0.9619	0.9045	0.7740	0.7939	0.5733	0.8103	0.1873	8.1
Rcut	0.6733	0.5810	0.5766	0.6011	0.5627	0.5014	0.5710	0.5599	0.5794	10.8
Ncut	0.6657	0.6685	0.8635	0.6685	0.7332	0.7019	0.7367	0.8114	0.5103	8.9
densityCut	0.6667	0.8952	0.7980	0.8933	0.8089	0.8212	0.6667	0.9084	0.7215	6.2
RCC	0.7733	0.7846	0.3104	0.9438	0.8094	0.8631	0.6667	0.9052	0.5865	6.6
GCSED	0.5333	0.8333	0.9385	0.9551	0.7273	0.8301	0.6433	0.9382	0.6512	6.6
CDC-U2	0.9733	0.9095	0.9605	0.9438	0.8728	0.9441	0.7683	0.9486	0.6229	2.7
CDC-U3	0.9733	0.9142	0.9605	0.9607	0.8307	0.9441	0.7933	0.9185	0.5610	3.1
CDC-U4	0.9600	0.8714	0.9678	0.9326	0.8768	0.9497	0.8667	0.9381	0.3941	3.8
CDC-U5	/	0.8571	0.9678	0.9326	0.8918	0.9525	0.8000	0.9412	0.3938	3.8

Revised Supplementary Table 4. Evaluation of CDC and other 10 algorithms on 9 high-dimensional real-world datasets. The highest NMI score of all clustering algorithms in each dataset is highlighted in bold. The rank scores of all algorithms are summarized in the last column.

	Iris	Seeds	Breast	Wine	PenDigits	Dermatology	Control	Digits	MNIST10k	Rank
K-means	0.7419	0.6743	0.7478	0.8155	0.6778	0.8467	0.7558	0.6826	0.4540	7.9
DBSCAN	0.7337	0.5115	0.4230	0.4945	0.5513	0.6175	0.6437	0.6650	0.1035	11.8
CDP	0.7107	0.6797	0.3866	0.7104	0.6867	0.7509	0.6577	0.4025	0.2033	11.0
AGNES	0.7201	0.5165	0.4230	0.5026	0.5506	0.4529	0.6580	0.1641	0.0877	12.4
MeanShift	0.7337	0.7142	0.7572	0.7729	0.8053	0.8208	0.7444	0.7844	0.2000	7.6
Rcut	0.7235	0.4486	0.5568	0.5114	0.5445	0.7509	0.6577	0.5392	0.5630	11.1
Ncut	0.7225	0.7150	0.7830	0.7350	0.7354	0.8548	0.7629	0.6830	0.5317	7.0
densityCut	0.7337	0.6797	0.5400	0.7262	0.8341	0.8507	0.7936	0.8367	0.6337	7.0
RCC	0.7067	0.6602	0.2922	0.8662	0.8667	0.9301	0.8520	0.9259	0.7609	5.8
GCSED	0.6478	0.6466	0.7176	0.8529	0.7764	0.8291	0.7178	0.9265	0.7042	7.6
CDC-U2	0.9144	0.7523	0.7531	0.8718	0.8735	0.9052	0.8011	0.9422	0.7246	2.9
CDC-U3	0.9144	0.7564	0.7270	0.8759	0.8698	0.9146	0.8338	0.9026	0.6760	3.4
CDC-U4	0.8565	0.7280	0.7830	0.7923	0.8983	0.9243	0.8863	0.9114	0.5967	3.2
CDC-U5	/	0.7038	0.7830	0.7955	0.8877	0.9347	0.8464	0.9020	0.5945	4.0

Revised Supplementary Table 5. Evaluation of CDC and other 10 algorithms on 9 high-dimensional real-world datasets. The highest ARI score of all clustering algorithms in each dataset is highlighted in bold. The rank scores of all algorithms are summarized in the last column.

	Iris	Seeds	Breast	Wine	PenDigits	Dermatology	Control	Digits	MNIST10k	Rank
K-means	0.7163	0.7049	0.8465	0.8368	0.5399	0.8155	0.5847	0.6755	0.3657	7.1
DBSCAN	0.5681	0.4116	0.6970	0.4007	0.2003	0.3740	0.5495	0.6106	0.0006	11.3
CDP	0.5490	0.7076	0.3847	0.6724	0.5193	0.6688	0.4867	0.1723	0.1193	10.9
AGNES	0.5584	0.4259	0.6970	0.4071	0.2003	0.2055	0.5420	0.0500	0.0012	11.9
MeanShift	0.5681	0.6973	0.8595	0.7938	0.7280	0.7289	0.5728	0.7543	0.0351	8.1
Rcut	0.5657	0.3338	0.3210	0.4036	0.3163	0.1953	0.3149	0.3095	0.3366	12.1
Ncut	0.5569	0.5666	0.8685	0.5666	0.6275	0.6949	0.6198	0.6823	0.3408	8.6
densityCut	0.5681	0.7158	0.5306	0.6990	0.7372	0.7620	0.6665	0.9093	0.6093	6.0
RCC	0.7212	0.6593	0.2324	0.8856	0.7905	0.8669	0.6824	0.9075	0.5305	5.8
GCSED	0.4403	0.5951	0.8058	0.8685	0.5630	0.8509	0.5012	0.9141	0.5391	7.3
CDC-U2	0.9222	0.7920	0.8669	0.8546	0.8241	0.8983	0.7070	0.9363	0.5468	2.7
CDC-U3	0.9222	0.7955	0.8624	0.8970	0.8104	0.9106	0.7048	0.8895	0.5037	3.3
CDC-U4	0.8857	0.7470	0.8742	0.8031	0.8442	0.9124	0.7931	0.8789	0.3005	3.7
CDC-U5	/	0.6998	0.8742	0.8025	0.8393	0.9173	0.7780	0.9074	0.2993	4.4

We further analyzed the efficiency of CDC on Control dataset (New Supplementary Fig. 8). As the dimension increases, the number of simplices (i.e., number of subdivision units) and running time of CDC grow exponentially, while the accuracy indicated by ARI present a stable range. For the sake of both high time efficiency and accuracy, CDC is recommended to be conducted in low UMAP dimensions.

New Supplementary Fig. 8. Number of simplices, running time and accuracy of CDC under different UMAP dimensions on Control dataset.

2. The authors reported clustering results on several datasets compared with other methods in R1-Table 3. However, I notice the clustering NMI (0.00) of RCC (Robust Continuous Clustering) on MNIST is much lower than the results reported in the original paper (which is 0.893 AMI). Please explain this result or make sure the compared methods are properly conducted. Besides, most of the datasets for evaluation are quite small, which is not that representative.

Thank you. **For each baseline, we have adopted the same standard and obtained the highest accuracy by adjusting input parameters exhaustively to make sure the comparisons be properly conducted.** The previous NMI of RCC we obtained on MNIST was 0.00, because we conducted RCC directly on the raw data but did not preprocess using normalization. In this version, we have normalized MNIST10k dataset before conducting RCC and extended the space of input parameters. Now, **the highest NMI can reach 0.761**. Since the validity metrics (AMI in the RCC paper and NMI in this paper) and datasets (MNIST10k we used is the test set of the entire MNIST used in the RCC paper) are different, the NMI value is still lower than 0.893.

According to your suggestions, **we added more datasets with larger data sizes and higher dimensions, where the maximum one has reached over 1 million cells along with more than 30k dimensions (expressed genes, protein marker dimensions)** (Response Table 1) in the scRNA-seq and CyTOF experiments. The real-world datasets we used previously are not large. It is because the environment to run some baselines

(e.g., CDP, Rcut, Ncut and RCC) are limited by the data size, and oversized data would cause memory overflow relevant problems in a commodity desktop computer. To achieve clustering for large scRNA-seq datasets, we transplanted the entire experiment onto an Inspur Tiansuo Service provided by the supercomputing center of our university. It has 8 Intel(R) Xeon(R) CPU E7-8880 v4 with 176 cores, 4TB DDR4 memory, 500GB SSD and 8TB HDD.

Response Table 1. The added scRNA-seq and CyTOF datasets used in cell type identification.

Dataset	No. of cells	No. of dimensions	No. of cell populations	Description	Platform
Muraro	2,122	18,915	9	Human pancreas	CEL-Seq2
Segerstolpe	2,133	22,757	13	Human pancreas	SMART-Seq2
Xin	1,449	33,889	4	Human pancreas	SMARTer
MIHPF	1,093,785	31,053	3/43	Mouse isocortex and hippocampal formation	10X

Dataset	No. of cells	No. of protein marker dimensions	No. of cell populations	Description	Platform
Levine	265,627	32	14	Human marrow	FlowRepository
Samusik	841,644	40	24	Mouse marrow	FlowRepository

3. In the sensitivity analysis, the author claims that the Adjusted Rand Index (ARI) is adopted as the objective function. I do not quite understand the reason to do so. Also, how are the sensitivity levels in R1-Table 1 computed? The authors could give more explanations on that.

Thank you. The principle of parameter sensitivity analysis is measuring the degree to which the dependent variable (clustering accuracy in this paper) is affected by the perturbation of the parameters (k and T_{DCM} in CDC). To analyze the fluctuation of clustering accuracy as k and T_{DCM} vary, we use ARI (Adjusted Rand Index) as the dependent variable (i.e., objective function we mentioned previously) to evaluate the clustering accuracy. Other validity metrics certainly can be also selected as the dependent variable (e.g., ACC, NMI, etc.), but ARI is more strictly sensitive to the variation of clustering accuracy. The sensitivity indexes of the two parameters are calculated as follows:

$$S_k = \frac{1}{100} \sum_{i=1}^{100} \left| \frac{\text{ARI}(k_i + \Delta k, T_i) - \text{ARI}(k_i, T_i)}{\text{ARI}(k_i + \Delta k, T_i) + \text{ARI}(k_i, T_i)} \right|$$

$$S_T = \frac{1}{100} \sum_{i=1}^{100} \left| \frac{\text{ARI}(k_i, T_i + \Delta T) - \text{ARI}(k_i, T_i)}{\text{ARI}(k_i, T_i + \Delta T) + \text{ARI}(k_i, T_i)} \right|$$

Specifically, we firstly divided the value range of the two parameters using 10 equal intervals respectively, which generated a 10 by 10 grid with cells of the same size. Then

we randomly selected one sampling point in each cell and exert a positive or negative perturbation to each point. We specified five fixed perturbations for each parameter. Only one parameter was varied at a time and the other was fixed. Finally, we calculated the sensitivity indexes that measures the relative change of the clustering accuracy before and after perturbations. Each group of experiment was conducted five times to avoid randomness in the simulations.

Supplementary Fig. 2. Illustration of sensitivity analysis using stratified sampling and random perturbation. (a) Sampling points generated by stratified sampling. (b) Operating random perturbations that could be either positive or negative to each sampling point.

In terms of the sensitivity levels, we partition value range of the sensitivity indexes ($[0, 1]$) into four same intervals to indicate different levels of parameter sensitivity. Theoretically, ARI is between -1 and 1, and it is lower than 0 when the clustering quality is extremely poor. Since we conducted the sampling in a proper range of the two parameters (k from 0 to $0.1n$ and T_{DCM} from 0 to 0.5), almost all ARIs range from 0 to 1. In this case, the two sensitivity indexes are also between 0 and 1. Hence, we divide their value range evenly into four intervals and each interval corresponds a sensitivity level (i.e., Insensitive: 0-0.25, Mild sensitive: 0.25-0.5, Sensitive: 0.5-0.75, Hypersensitive: 0.75-1.00).

According to your suggestion, we have revised in Line 329-335, Page 17 as follow:
 Parameter sensitivity analysis aims to measure the degree to which the dependent variable is affected by the perturbation of the parameters, which facilitates to control the granularity of parameter tuning. In this paper, the clustering accuracy is the dependent variable, and we assessed the impact of the two input parameters of CDC, i.e., k and T_{DCM} (see Methods) on it. We conducted sensitivity analysis using stratified sampling and random perturbation, which is modified from Latin-Hypercube One-factor-At-a-Time (LH-OAT) (54) in hydrology.

In brief, I believe the idea behind CDC is interesting. But my major concern lies in the generality and stability of CDC since it relies heavily on dimensional reduction techniques

and outlier detection strategy, etc.

Thank you. **We have generalized CDC to high-dimensional space using Qhull algorithm and simplex volume calculation. The generality and stability of CDC were verified on the real-world, scRNA-seq, CyTOF and speaker corpus datasets.** The experimental results showed that CDC can yield high and robust clustering accuracy in different UMAP dimensions. Although dimension reduction may cause some degree of information loss and determine the upper bound of the clustering performance, there is no evidence suggested that “dimensional reduction techniques might fail for data with higher dimensionality”. **1) Dimension reduction is beneficial when the number of data features is large due to the curse of dimensionality.** UMAP can embed them into a space of proper dimension to reflect the local direction centrality due to the manifold distributions of many real-world data (see Response Fig. 1). For example, CDC-U2, CDC-U3, CDC-U4 and CDC-U5 obtained the higher scores on the real-world datasets than other clustering baselines. **2) UMAP only cannot guarantee high clustering accuracy, proper clustering methods are quite vital. Even in the same UMAP dimensions, CDC outperforms other baselines.** For example, CDC-U2 outperformed the SNN-based and classical methods conducted in 2D UMAP dimension as well as the biological pipelines on nine scRNA-seq datasets.

Response Fig. 1. An example of CDC clustering in 3D space and 2D UMAP space.

Two nearby 3D manifold clusters are shown in Response Fig. 1a, where the blue points denote the boundary points identified by CDC and the red refers to internal points. The detection of internal and boundary points is accurate for CDC, however, the boundary points is unable to constraint the internal connections in all 3D directions due to the manifold structure. We take the green internal point in Response Fig. 1a as an example, its reachable range (the green circle in Response Fig. 1c) is beyond the cluster to which it belongs from the view of X-Y plane. Since the cross-cluster connections have not been prevented by the boundary points, two clusters are misidentified as a whole in Response Fig. 1b. In comparison, we use UMAP to embed the 3D points to 2D space as shown in

Response Fig. 1d. In the real data dimension (2D) of the two manifold structures, the boundary points generate strong cages to bind the internal connections in all 2D directions, and the clustering result is accurate as presented in Response Fig. 1e.

Furthermore, noise elimination is a common problem in clustering analysis. Actually, it should be considered in preprocessing phase and is not the core of CDC in this paper. **Noise disposal can be a complement embedded in the procedure of CDC.** The noise elimination methods we currently used are KNN-based that can be combined with CDC seamlessly. These classical methods are widely-used in the field of data mining. The experimental results proved the validity of this combination. Users can also choose other methods as alternatives in this paper.

Reviewer #2 (Remarks to the Author):

All of my previous comments have been addressed very well by the authors. I hence suggest that the paper should be accepted in its current form.

David B. Blumenthal

Thank you for your positive comments.

Reviewer #3 (Remarks to the Author):

Major comments

The previous set of comments noted that the evaluations for scRNA-seq and CyTOF data relied on small and outdated datasets, and did not compare against standard clustering analysis pipelines for scRNA-seq and CyTOF data. Unfortunately, the new analyses and datasets in the updated manuscript do not sufficiently address this comment. Crucially, the evaluations still do not compare against standard scRNA-seq pipelines (e.g. Seurat / Bioconductor), using typical full-sized datasets (up to 1 million cells). There has also been a misunderstanding in applying CyTOF clustering methods to scRNA-seq data. Unfortunately, these results do not indicate that this clustering method could be used for analyzing typical scRNA-seq or CyTOF data.

Thanks for the comments and suggestions. As described in point-by-point responses below, we have extended our CDC method to higher dimensional space, and shown the performances on large-scale scRNA-seq or CyTOF datasets are comparable or better than current mainstream methods.

Specifically:

(i) The analyses still rely on old and outdated datasets. Most of the scRNA-seq datasets (line 134 and Supplementary Table 1) are extremely small by modern standards.

For example, the main dataset described on line 134 has 801 cells and is sourced from a paper from 2013. Modern scRNA-seq datasets can contain >1 million cells, so these datasets are not representative of typical scRNA-seq clustering tasks.

Similarly, the CyTOF datasets are relatively small and outdated. Newer CyTOF datasets can routinely contain 40 dimensions and 1 million cells.

Thank you. According to your suggestions, **we have added four new scRNA-seq and two CyTOF datasets, where the maximum one has reached over 1 million cells along with more than 30k dimensions (expressed genes, protein marker dimensions)** (Response Table 2). In our opinion, **data size is not the determining factor affecting the accuracy of algorithm, but the distribution pattern of data**. Meanwhile, performance comparisons among dozens of clustering algorithms on scRNA-seq dataset with over one million cells are rare, which are limited by the algorithm complexity (e.g., RCC, CDP, Ncut and etc.), experimental environment (memory size, software and etc.), and high computational cost. For instance, MATLAB has a constraint on the matrix size and R limits the number of non-zero elements. Due to this reason, Seurat software (v3.2.2) cannot be implemented to handle the MIHPF dataset with over one million cells in our experimental environment (see details in the response for question 3). Therefore, we selected nine small but representative scRNA-seq datasets from different platforms to evaluate the generality and stability of CDC by comparing with seven biological pipelines and seven classical

clustering methods. MIHPF with over one million cells was used to prove the ability and performance to handle large-scale scRNA-seq dataset.

Response Table 2. The added scRNA-seq and CyTOF datasets in cell type identification.

Dataset	No. of cells	No. of dimensions	No. of cell populations	Description	Platform
Muraro	2,122	18,915	9	Human pancreas	CEL-Seq2
Segerstolpe	2,133	22,757	13	Human pancreas	SMART-Seq2
Xin	1,449	33,889	4	Human pancreas	SMARTer
MIHPF	1,093,785	31,053	3/43	Mouse isocortex and hippocampal formation	10X

Dataset	No. of cells	No. of protein marker dimensions	No. of cell populations	Description	Platform
Levine	265,627	32	14	Human marrow	FlowRepository
Samusik	841,644	40	24	Mouse marrow	FlowRepository

(ii) The previous comments noted that the authors did not compare against standard analysis pipelines for scRNA-seq and CyTOF data. Specifically, the comments mentioned: (a) for scRNA-seq: dimension reduction to 50 principal components (PCs) on a subset of highly variable genes followed by graph-based clustering (for example, standard pipelines are available from Seurat or Bioconductor), and (b) for CyTOF: clustering in the full-dimensional space using FlowSOM or PhenoGraph.

The authors have included new analyses for the scRNA-seq data, but these represent a fundamental misunderstanding of scRNA-seq and CyTOF analysis pipelines. Specifically, for scRNA-seq, the authors have applied FlowSOM and PhenoGraph in 50-dimensional PCA space – i.e. this is incorrectly using the CyTOF clustering methods for scRNA-seq data, as well as ignoring any additional preprocessing steps from the Seurat / Bioconductor pipelines such as selecting highly variable genes. The evaluations do not compare against the standard pipelines, so these results do not provide a representative performance comparison.

According to your suggestions, **we collaborated with researchers in the field of bioinformatics and RNA biology, and redesigned the entire scRNA-seq experiment using standard preprocessing steps and analysis tools.** Specifically, **seven biological pipelines** (i.e., Seurat, monocle3, SC3, MetaCell, dropClust, SNN-Louvain and SNN-Walktrap) and **seven classical clustering methods** (i.e., AGNES, DIANA, hclust, DBCSAN, K-means, C-means, and CLARA) were investigated in comparing with CDC on **nine scRNA-seq datasets** (Supplementary Table 1). The biological pipelines follow their own default preprocessing steps, including no-expressed and low-expressed genes filtering, highly variable genes (HVG) selection and etc. PCA was used to select the first

50 principal components and UMAP further reduce the dimensions based on the 50-dimensional PCA data. SNN-based and the classical clustering baselines were conducted in both of PCA and UMAP space with different dimensions (2, 3, 4, 5, 10, 15, 20, 25, 30, 35, 40, 45, 50 dimensions for PCA and 2 dimensions for UMAP). CDC has been generalized to high-dimensional space and is conducted 2, 3, 4 and 5 UMAP dimensions. The details of preprocessing are illustrated in Supplementary Note 4.

According to the clustering accuracy reported by ARI in New Fig. 2e, **the performance of CDC-U2 is comparable and even better than the biological methods optimized for scRNA-seq clustering on the nine datasets.** It obtained promising max ARI scores (BH: 0.9542, BM: 0.9671, Muraro: 0.9302, Segerstolpe: 0.9729, Xin: 0.9709, AMB: 0.8881, ALM: 0.7528, VISp: 0.6249, TM: 0.8307) and have distinct advantage on the stability that the ARIs obtained in the parameter space are distributed closer to the max scores. Comparing with the classical clustering methods illustrated in New Supplementary Fig. 4, **CDC-U2 also has the most accurate and robust outcomes.** CDC under 2 to 5 UMAP dimensions dominates the top four places in both of accuracy and robustness on the first five datasets (BH, BM, Muraro, Segerstolpe and Xin), but the performances of CDC-U4 and CDC-U5 lack of competitiveness on the last four (AMB, ALM, VISp and TM). Those datasets are more challenging because they have a larger data volume and multi-level types of cells (e.g., class and subclass) with complex manifold distributions. For the hierarchical relationships, PCA and UMAP tend to capture high-level differences of gene expression, thus cannot separate all cell types in a fine granularity at one time. Nonetheless CDC equipped with UMAP outperforms the baselines in general. Often times that scRNA-seq data has a manifold distribution, i.e., data dimension is lower than the feature dimension. UMAP is able to embed the data to a proper dimension, in which the local direction centrality is applicable, that means the boundary points can bind the internal connections in all directions.

New Fig. 2e. ARI distributions of seven biological pipelines, K-means and CDC-U2 on nine scRNA-seq datasets, where the red points denote the max ARI scores and the yellow ones refer to the ARI scores with default parameter settings.

New Supplementary Fig. 4. ARI distributions of six classical methods and CDC under different UMAP dimensions on nine scRNA-seq datasets, where the blank bars mean that the methods are not applicable on the datasets.

A large-scale scRNA-seq dataset from the adult mouse isocortex and hippocampal formation (MIHPF) was collected to further validate the effectiveness of CDC. It contains 1,093,785 total cells along with 31,053 genes, which have been assigned to three classes (Glutamatergic neurons, GABAergic neurons and Non-neuronal) and can be further subdivided into 30, 7 and 6 subclasses respectively. The subclasses have been grouped into 8 neighborhoods, including 5 glutamatergic (DG/SUB/CA, L2/3 IT, L4/5/6 IT Car3, PT and NP/CT/L6b), 2 GABAergic (MGE and CGE), and one “other” neighborhood. From the heatmap in New Fig. 2f, we can find that **CDC-U2 obtained the clusters between class and subclass levels in line with the cell development.** The clusters are in accord with the proximity reflected in the transcriptomic taxonomy tree. As an unsupervised method, although CDC-U2 cannot obtain clusters in the same level at once, it extracted three neighborhoods (NP/CT/L6b, MGE and CGE) relatively completely. They are the bases for conducting further clustering to explore fine-grained subclasses. Meanwhile, most of the cells in the same subclasses were assigned to the same clusters except two small subclasses, CR and SMC-Peri (including 268 and 198 cells respectively). In comparison, K-means assigned cells of different classes to the same cluster (e.g., cluster 2 in Kmeans-PCA and cluster 1 in Kmeans-U2 with k=3, cluster 5 in Kmeans-PCA

and cluster 6 in Kmeans-U2 with k=43) as shown in New Supplementary Fig. 5. Therefore, CDC-U2 outperformed Kmeans-PCA and Kmeans-U2 in clustering accuracy at both the class and subclass levels (New Fig. 2g).

New Fig. 2f. Heatmap of the clustering result obtained by CDC-U2 with $k = 30$ and $T_{DCM} = 0.11$ on MIHPF dataset (part of clusters having less than 30 cells are not displayed), where the transcriptomic taxonomy tree and labels of classes and subclasses are from Yao et al., 2021.

References:

Yao, Z. et al. A taxonomy of transcriptomic cell types across the isocortex and hippocampal formation. *Cell* 184, 3222-3241 (2021).

New Supplementary Fig. 5. Heatmaps of the clustering results of Kmeans-PCA and Kmeans-U2 with $k = 3$ and $k = 43$, where the blue rectangles circle some clusters with wrong assignments. The errors lie in assigning two or more different cell type into one cluster.

New Fig. 2g. Comparison of accuracy between K-means-PCA, K-means-U2 and CDC-U2 on MIHPF dataset.

For the CyTOF data, the authors also mention that they “removed the cells without manually gated population labels” (Supplementary Note 4, page 6). This is inconsistent with previous published evaluations of these CyTOF datasets, where clustering was performed on all cells (including un-labelled) and performance subsequently calculated on the subset of known cells. Therefore, the evaluations here represent a much easier clustering task, and an inconsistent comparison.

According to your suggestion, **we have rerun the CyTOF experiment using two new and larger datasets**, i.e., Levine and Samusik (Response Table 2), by comparing with **16 clustering baselines** including dedicated algorithms for the cell population detection. The clustering was performed on all cells, including the ones without manually gated population labels, and the ARI score was subsequently calculated on the subset of known cells to evaluate the clustering accuracy. We followed the parameter settings in (Weber et al., 2016) and the default settings in software (ACCENSE) or online platforms (FLOCK). But the required number of clusters in some algorithms (flowClust, flowMeans, flowMerge, flowSOM, K-means, Rclusterpp) was specified as 40 in the original paper. We modified it according to the number of manually gated populations ($C = 14$ on Levine, $C = 24$ on Samusik). All the clustering algorithms were implemented on a commodity desktop computer with a 8-core Intel i7 processor and 64 GB RAM. The recorded running times include the processes of clustering and dimensionality reduction. Due to the computability issue, the running times of some methods are counted on the subsampling datasets. The detailed parameter settings can be seen in Supplementary Note 5 and are presented in Response Table 3.

Response Table 3. Detailed parameter settings of 17 clustering algorithms on two CyTOF datasets.

Method (Environment)	Dataset	Parameter setting	No. of cells
ACCENSE (ACCENSE 0.5.1)	Levine	z-score = TRUE, remove outliers = TRUE dimensionality reduction = Barnes-Hut-SNE no_dims = 2, perplexity = 30, theta = 0.5	All
	Samusik	as above	400,000
clusterX (R v4.0.3)	Levine	cytof_dimReduction: method = "tsne"	All
	Samusik	as above	All
densityCut (R v4.0.3)	Levine	Normalization = FALSE, k = 50, α = 0.9	All
	Samusik	Normalization = TRUE, k = 50, α = 0.9	All
DensVM (R v4.0.3)	Levine	cytof_dimReduction: method = "tsne"	All
	Samusik	as above	300,000
FLOCK (ImmPort Galaxy)	Levine	bins = 6, density = 3	All
	Samusik	as above	All
flowClust (R v4.0.3)	Levine	C = 14	All
	Samusik	C = 24	All
flowMeans (R v4.0.3)	Levine	Standardize = FALSE, C = 14	All
	Samusik	Standardize = FALSE, C = 24	All
flowMerge (R v4.0.3)	Levine	C = 14	NA
	Samusik	C = 24	10,000
flowPeaks (R v4.0.3)	Levine	all defaults	All
	Samusik	as above	All
flowSOM (R v4.0.3)	Levine	Scaled = FALSE, GridSize = 10, C = 14	All
	Samusik	Scaled = TRUE, GridSize = 10, C = 24	All
immunoClust (R v4.0.3)	Levine	classify.all = TRUE	All
	Samusik	as above	All
K-means (Matlab R2020b)	Levine	C = 14	All
	Samusik	C = 24	All
Meanshift (Matlab R2020b)	Levine	bandwidth = 5	All
	Samusik	bandwidth = 6	All
PhenoGraph (Python v3.7)	Levine	k = 30, metric = "euclidean"	All
	Samusik	as above	All
Rclusterpp (R v4.0.3)	Levine	C = 14, method = "ward"	All
	Samusik	C = 24, method = "ward"	All
SamSPECTRAL (R v4.0.3)	Levine	normal.sigma = 100, separation.factor = 1	All
	Samusik	as above	All
CDC-U2 (UMAP: Java v15.0.1 CDC: Matlab R2020b)	Levine	Normalization = FALSE, umap.n_neighbors = 50, n_components = 2, k = 60, T _{DCM} = 0.08	All
	Samusik	Normalization = TRUE, umap.n_neighbors = 30, n_components = 2, k = 35, T _{DCM} = 0.03	All

Revised Fig 2h-i. Performances of 17 clustering algorithms on Levine and Samusik datasets, including CDC-U2, ACCENSE, ClusterX, densityCut, DensVM, FLOCK, flowClust, flowMeans, flowMerge, flowPeaks, flowSOM, immunoClust, K-means, MeanShift, PhenoGraph, Rclusterpp, and SamSPECTRAL.

The ARI scores and running times are shown in Revised Fig 2h-i. **CDC-U2 the highest accuracy on Levine (ARI = 0.9618) and second on Samusik (ARI = 0.8148).** Meanwhile, it also can be found that **CDC-U2 has promising time efficiency above the average and is more efficient than densityCut and PhenoGraph on the entire datasets.** In comparison, ACCENSE, DensVM and flowMerge, require subsampling to make them computable due to the excessive running time on large data size.

References:

Weber, L. M. & Robinson, M. D. Comparison of Clustering Methods for High-Dimensional Single-Cell Flow and Mass Cytometry Data. *Cytom. Part A* 89, 1084-1096 (2016).

(iii) The runtime evaluations (line 207 onwards) do not present a comprehensive evaluation of runtimes on modern, full-sized scRNA-seq datasets (e.g. 1 million cells), which can be routinely handled by the standard Seurat / Bioconductor pipelines.

For example, in the rebuttal letter (Reviewer 4, Comment 4), the authors mention “Limited by the memory of a commodity desktop computer (memory overflow problem of the dimension reduction for PCA, t-SNE and UMAP), we could not further test CDC-UMAP in a scRNA-seq dataset with millions of cells and ten thousand of genes.” However, analyses on millions of cells and tens of thousands of genes are routinely performed with Seurat / Bioconductor, so this represents a fundamental limitation in the applicability of the method.

Similarly, the rebuttal letter (Reviewer 4, Comment 4) mentions “Comparing with the recent publication on Nature Communications, Li et al. (2020) (the maximum scRNA-seq dataset has 100,000 cells and 500 genes), we believe our experiments follow the modern standard in data size”. This represents a misunderstanding of the scale of scRNA-seq data, which typically has tens of thousands of genes (i.e. transcriptome-wide), not 500.

All of the previous experiments were implemented on a commodity desktop computer with a 8-core Intel i7 processor and 64 GB RAM. As we mentioned, a memory overflow problem occurred when conducting the dimension reduction techniques (i.e., PCA, t-SNE and UMAP) on scRNA-seq datasets having more than 100,000 cells and tens of thousands of genes. To settle down this problem, **we have conducted the entire scRNA-seq clustering experiment on an Inspur Tiansuo Service provided by supercomputing center of our university.** It has 8 Intel(R) Xeon(R) CPU E7-8880 v4 with 176 cores, 4TB DDR4 memory, 500GB SSD and 8TB HDD. We redesigned the experiments including reselecting the biological pipelines, adding new datasets and correcting the preprocessing steps.

However, **Seurat (v3.2.2) still cannot handle the one-million scRNA-seq dataset according to our experiment.** When we input the whole initial UMI (Unique Molecular Identifiers) matrix to the procedure, the *as.sparse.matrix* function reported an error during creating a Seurat object. But when we partitioned the raw data into pieces to input the algorithm, the *cbind2sparse* function reported the same error during merging the multiple Seurat objects. The reason for this error is that Seurat generated excess non-zero elements when creating the sparse matrix, whose number is beyond the size limit, i.e., 2,147,483,647 in R. Hence, we only compared Kmeans-PCA and Kmeans-U2 on the MIHPF dataset to evaluate its generality and stability. Results demonstrated CDC-U2 outperformed K-means both in class and subclass levels of cells.

Meanwhile, we are sorry to make you misunderstand what we said in the rebuttal letter *“Comparing with the recent publication on Nature Communications, Li et al. (2020) (the maximum scRNA-seq dataset has 100,000 cells and 500 genes), we believe our experiments follow the modern standard in data size”*. Actually, the real-world scRNA-seq datasets selected in that paper do have tens of thousands of genes, **but the four simulated datasets used for time efficiency evaluation in Fig. 2 only have 500 genes** (the original paper mentions *“Finally, we evaluated how these classifiers behaved in speed using 4 simulated datasets comprising 500 genes along with 1000, 10,000, 20,000 and 50,000 cells, respectively”*). Just like their experiment, we also used simulated datasets (the maximum one has 100,000 cells and 10,000 genes) previously to evaluate the running time of CDC. In the new version, we removed the time efficiency experiment in this section and added a new experiment to assess the trend of the running time of CDC as dimension increases in Supplementary Fig. 8.

Reviewer #4 (Remarks to the Author):

1) I appreciate the authors for the efforts in addressing my comments. The major limitations of the algorithm include only processing 2-dimensional data and the sensitivity of the algorithms on the parameter T . The authors' empirical approach for selecting the T parameter also requires the number of clusters as inputs, which is typically unknown for real datasets. Thus I suggest the authors make these points clear early, e.g., in the introductions for the interesting readers to quickly grab the strength and weakness of their algorithm. Also, the authors' claim about the computational efficiency of their algorithm is a little misleading, e.g., as in Fig.2m because Phonograph is run on 50D data and CDC is run on 2D data.

Thanks for the suggestions. **We have generalized CDC to high-dimensional space and evaluated its effectiveness through multiple experiments** (see our response to Reviewer #1 question 1). Besides, **the sensitivity of T_{DCM} would not limit the applications of CDC**. Parameter sensitivity problem is common in clustering algorithm (e.g., k in K-means, Eps in DBSCAN, bandwidth in MeanShift, etc.). According to the results of parameter sensitivity analysis, CDC is sensitive to T_{DCM} . Nonetheless, we design an adaptive method to determine T_{DCM} effectively in 2D space. While in high-dimensional space, a percentile ratio that measures the proportion of internal points in all points can be used to substitute the DCM threshold as discussed in Supplementary Note 4. In proper range of T_{DCM} , our algorithm CDC can outperform other baselines in clustering accuracy and robustness.

In terms of selecting T_{DCM} adaptively, a known number of clusters in advance can make the estimation more accurate, but it is not necessary. We determine T_{DCM} by sorting all DCMs and estimating the number of boundary points. Using the graph theory in the generated TIN, the number of boundary points can be calculated:

$$B = 2V - F - 2C$$

where B , V , F and C denote the number of boundary points, total points, intra-cluster triangles and clusters respectively. C is vague or unknown for real datasets, so we can set it as 1. It is because **C is usually significantly smaller than V and F , and has a trivial effect on the estimation of B** . Actually, the unknown C would cause some wrong divisions of internal and boundary points. Nonetheless, **CDC is robust to the wrong divisions to some extent**. If the determined boundary points are enough to avoid the cross-cluster connections, or a few internal points are misidentified that do not cut off the intra-cluster connections, the global assignments would not be affected (see sections of Validation of the adaptive methods and Discussion).

To make the claim of the required number of clusters more clearly, we have revised accordingly (Line 552-555, Page 28) as follows:

In terms of the number of clusters C , it is significantly smaller than V and F usually, which has a trivial effect on the estimation of B . Moreover, CDC is robustness to the DCM

threshold as mentioned in Discussion. Thus, C can be treated as 1, when it is vague or difficult to determine.

We believe **the evaluation of time efficiency in the previous experiment on the scRNA-seq and CyTOF datasets is fair**. Because we counted the **overall running time** for each method, including both the execution time of **dimension reduction** (PCA and UMAP) and **clustering processes** (PhenoGraph and CDC).

The idea of generalizing DCM to high-dimensional space: DCM calculation requires to map the KNNs onto the unit spherical surface drawn by their center point firstly, then subdivide the spherical surface and measure the corresponding angles of the subdivision units (New Supplementary Fig. 6). It is intuitive to understand the definition of DCM that measures the variance of angles formed by KNNs in 2D space. However, subdividing the hyperspherical surface and measuring the generalized angles are complex in high-dimensional space. Actually, the generalized angles are equivalent to the corresponding the volume of the subdivision units (e.g., arc length in 2D circle, area of the spherical triangle in 3D sphere). So, the angle measurement can be converted to the volumes of the subdivision units. To subdivide the hyperspherical surface, we utilize Qhull algorithm to construct the convex complex of KNNs. Then, we calculate the volumes of the simplices (facets of the complex) and further estimate the volumes of the subdivision units. *The detailed calculation process of DCM is presented in Line 565-616, Page 29-31, and the proofs of simplex volume calculation are presented in Supplementary Note 6.*

New Supplementary Fig. 6. Graphical illustration of expanding DCM with hyperspherical subdivisions for high-dimensional spaces.

To validate the effectiveness of the generalized CDC on high-dimensional datasets, we selected **nine real-world datasets**, i.e., Iris, Seeds, Breast-Cancer, Wine, PenDigits, Dermatology, Control, Digits, MNIST10k (details in New Supplementary Table 2), to compare with **ten generic clustering baselines**, i.e., K-means, DBSCAN, CDP, AGNES, MeanShift, Rcut, Ncut, densityCut, RCC, GCSED. We conduct CDC in 2, 3, 4, 5 UMAP dimensions (denoted as CDC-U2, CDC-U3, CDC-U4, CDC-U5). ACC, NMI and ARI are reported for clustering accuracy in New Supplementary Table 3-5. In general, CDC obtained the highest scores on eight datasets under different UMAP dimensions and have significant advantages than the baselines. CDC-U2, CDC-U3, CDC-U4, CDC-U5 are the top four in the average rank of ACC, NMI and ARI.

We further analyzed the efficiency of CDC on Control dataset (New Supplementary Fig. 6). As the dimension increases, the number of simplices (i.e., number of subdivision units) and running time of CDC grow exponentially, while the accuracy indicated by ARI present a stable range. For the sake of both high time efficiency and accuracy, CDC is recommended to be conducted in low UMAP dimensions.

New Supplementary Fig. 8. Number of simplices, running time and accuracy of CDC under different UMAP dimensions on Control dataset.

2) As for scRNA-seq data analysis, from both our internal analyses and other people of the field, for many datasets, we almost surely can get better clustering results by increasing the dimensionalities of UMAP from 2D to higher dimensions such as 5D. Also, the 'standard' pipeline of the field clusters the principal components instead of UMAPs. The number of principal components also has a big impact on the clustering results. I don't want that the authors give the information to the field that people should cluster the 2D UMAPs. Thus, I suggest that the authors acknowledge the limitations of the results in Figure 2, e.g., no comparisons from using higher-dimensional UMAPs or using different numbers of principal

components, and the results are specific to the data used, may not generalize to other scRNA-seq data.

If the authors do think clustering on 2D UMAPs is the preferred way for scRNA-seq data analysis, I suggest that the authors do more comprehensive experiments, e.g., using different numbers of PCs and different dimensional UMAPs, and following the standard pipelines of the field, e.g., considering the gene/feature selection step and the potential batch-correction step. Also, it's essential to make the scripts and data available so the readers can easily reproduce the comparison results as the results may change the way people clustering scRNA-seq data.

As you commented, there are two major deficiencies of the previous analysis on the scRNA-seq datasets: 1) CDC has not been conducted in space of higher UMAP dimensions, 2) the biological pipelines and classical clustering methods using different principal components have not been compared. The source of the deficiencies resides two aspects: 1) DCM (Direction Centrality Metric) in CDC cannot be calculated in high-dimensional space, 2) the previous experimental designs, including the selection of the standard pipelines, preprocessing steps and parameter settings, are not appropriate.

Currently, we have tackled the above problems. 1) **We have achieved DCM calculation in any dimension using Qhull algorithm and simplex volume calculation.** The validity of CDC was evaluated on nine high-dimensional datasets by comparing with ten generic clustering methods. 2) **The entire scRNA-seq clustering experiment has been redesigned.** Specifically, **seven biological pipelines** (i.e., Seurat, monocle3, SC3, MetaCell, dropClust, SNN-Louvain and SNN-Walktrap) and **seven classical clustering methods** (i.e., AGNES, DIANA, hclust, DBSCAN, K-means, C-means, and CLARA) were selected for comparing with CDC on **nine scRNA-seq datasets** (Supplementary Table 1). **The biological pipelines follow their own default preprocessing steps**, including no-expressed and low-expressed genes filtering, highly variable genes (HVG) selection and etc. **PCA was used to select the first 50 principal components** and UMAP further reduce the dimensions based on the 50-dimensional data. SNN-based and the classical clustering baselines were conducted in both of PCA and UMAP space of different dimensions (2, 3, 4, 5, 10, 15, 20, 25, 30, 35, 40, 45, 50 dimensions for PCA and 2 dimension for UMAP). CDC was performed under 2, 3, 4 and 5 UMAP dimensions. The detailed preprocessing and parameter settings are listed in Supplementary Note 4.

According to the clustering accuracy reported by ARI in New Fig. 2e, **the performance of CDC-U2 is comparable and even better than the biological methods optimized for scRNA-seq clustering on the nine datasets.** It obtained promising max ARI scores (BH: 0.9542, BM: 0.9671, Muraro: 0.9302, Segerstolpe: 0.9729, Xin: 0.9709, AMB: 0.8881, ALM: 0.7528, VISp: 0.6249, TM: 0.8307) and have distinct advantage on the stability that the ARIs obtained in the parameter space are distributed closer to the max scores. Comparing with the classical clustering methods illustrated in New Supplementary Fig. 4, **CDC-U2 also has the most accurate and robust outcomes.** CDC under 2 to 5 UMAP dimensions

dominates the top four places in both of accuracy and robustness on the first five datasets (BH, BM, Muraro, Segerstolpe and Xin), but the performances of CDC-U4 and CDC-U5 lack of competitiveness on the last four (AMB, ALM, VISp and TM). Those datasets are more challenging because they have a larger data volume and multi-level types of cells (e.g., class and subclass) with complex manifold distributions. For the hierarchical relationships, PCA and UMAP tend to capture high-level differences of gene expression, thus cannot separate all cell types in a fine granularity at one time. Nonetheless CDC equipped with UMAP outperforms the baselines in general. Often times that scRNA-seq data has a manifold distribution, i.e., data dimension is lower than the feature dimension. UMAP is able to embed the data to a proper dimension, in which the local direction centrality is applicable, that means the boundary points can bind the internal connections in all directions.

Although CDC-U2 obtained high and stable accuracy on the above nine scRNA-seq datasets, even so, 2D UMAP space may not always the most appropriate to represent the data distribution. Consequently, CDC may get the highest accuracy in other UMAP dimension (as we have discussed in Line 296-304, Page 15-16). CDC currently can be conducted in high-dimensional space, thus users are free to specify appropriate dimension for CDC according to their needs.

New Fig. 2e. ARI distributions of seven biological pipelines, K-means and CDC-U2 on nine scRNA-seq datasets, where the red points denote the max ARI scores and the yellow ones refer to the ARI scores with default parameter settings.

New Supplementary Fig. 4. ARI distributions of six classical methods and CDC under different UMAP dimensions on nine scRNA-seq datasets, where the blank bars mean that the methods are not applicable on the datasets.

A large-scale scRNA-seq dataset from the adult mouse isocortex and hippocampal formation (MIHPF) was collected to further validate the effectiveness of CDC. It contains 1,093,785 total cells along with 31,053 genes, which have been assigned to three classes (Glutamatergic neurons, GABAergic neurons and Non-neuronal) and can be further subdivided into 30, 7 and 6 subclasses respectively. The subclasses have been grouped into 8 neighborhoods, including 5 glutamatergic (DG/SUB/CA, L2/3 IT, L4/5/6 IT Car3, PT and NP/CT/L6b), 2 GABAergic (MGE and CGE), and one “other” neighborhood. From the heatmap in New Fig. 2f, we can find that **CDC-U2 obtained the clusters between class and subclass levels in line with the cell development.** The clusters are in accord with the proximity reflected in the transcriptomic taxonomy tree. As an unsupervised method, although CDC-U2 cannot obtain clusters in the same level at once, it extracted three neighborhoods (NP/CT/L6b, MGE and CGE) relatively completely. They are the bases for conducting further clustering to explore fine-grained subclasses. Meanwhile, most of the cells in the same subclasses were assigned to the same clusters except two small subclasses, CR and SMC-Peri (including 268 and 198 cells respectively). In comparison, K-means assigned cells of different classes to the same cluster (e.g., cluster 2 in Kmeans-PCA and cluster 1 in Kmeans-U2 with k=3, cluster 5 in Kmeans-PCA

and cluster 6 in Kmeans-U2 with k=43) as shown in New Supplementary Fig. 5. Therefore, CDC-U2 outperformed Kmeans-PCA and Kmeans-U2 in clustering accuracy at both the class and subclass levels (New Fig. 2g).

New Fig. 2f. Heatmap of the clustering result obtained by CDC-U2 with $k = 30$ and $T_{DCM} = 0.11$ on MIHPF dataset (part of clusters having less than 30 cells are not displayed), where the transcriptomic taxonomy tree and labels of classes and subclasses are from Yao et al., 2021.

Reference:

Yao, Z. et al. A taxonomy of transcriptomic cell types across the isocortex and hippocampal formation. Cell 184, 3222-3241 (2021).

*The transcriptomic taxonomy tree and labels of classes and subclasses are from Yao et al, 2021.

New Supplementary Fig. 5. Heatmaps of the clustering results of Kmeans-PCA and Kmeans-U2 with $k = 3$ and $k = 43$, where the blue rectangles circle some clusters with wrong assignments. The errors lie in assigning two or more different cell type into one cluster.

New Fig. 2g. Comparison of accuracy between K-means-PCA, K-means-U2 and CDC-U2 on MIHPF dataset.

3) The CDC algorithm explores the KNN graph to identify boundary points and share similarities with other approaches. To put the CDC algorithm into context in the introduction section, I suggest that the authors discuss the relevant literature on 'in-neighbors' and other concepts such as 'reversed knn' in the introduction.

According to your suggestion, we have revised accordingly (Line 52-62, Page 3) as follows: As a boundary-seeking approach, local gravitation clustering (LGC) (10) proposes two mean-shift-based metrics, centrality (CE) and coordination (CO), to measure the consistency between the local attractive forces and mean-shift directions of neighbors. Accordingly, it is capable to distinguish internal and boundary points of clusters, and forms clusters by connecting boundary and unlabeled points from internal points with a damping connecting capability. However, internal points in sparse clusters are difficult to detect since the mean shifts tend to move towards the dense regions. Density-based metrics such as Reverse K-Nearest Neighbors (RKNN) (11) have been also utilized to detect the boundary points of cluster. It queries the number of objects that consider a given point as the membership of their KNNs, but might fail to seek the boundaries with low-density densities.

Reviewers' Comments:

Reviewer #1:

Remarks to the Author:

Thanks for the through explanations. According to the responses and revisions, I feel the generality and stability of CDC is acceptable. The current form of the paper is sound to me. Here are a few more minor suggestions.

1. What is the time and memory complexity of the Qhull algorithm? Is it always practical when the dataset is considerably large? This is important as the proposed CDC relies on Qhull on high-dimensional data.
2. Since the previous comparison against RCC is not that 'fair', I suggest the authors check other baselines again to make sure that the evaluation is properly conducted.

Reviewer #4:

Remarks to the Author:

I appreciate the authors for the updated experiments. I can see the authors have put lots of time running these experiments. The results (Fig.2) presented in the paper are great. However, after reading the updated manuscript and from some quick experiments by myself. I still have the same concerns about the robustness of the algorithms on real data.

I checked the R code and wanted to do some analysis. Unfortunately, the CDC function has several parameters. Except for a data matrix, users also have to select at least two parameters, namely, k and ratio.

In practice, users may just want to use the default setting for analysis. With these extra unfamiliar parameters, new users may just turn away from the tool.

Anyway, I used the 'standard' PBMC data here (https://satijalab.org/seurat/articles/pbmc3k_tutorial.html) for testing. For this simple dataset, I expected CDC should work pretty well as the UMAP shows clear clusters. But unfortunately, it's not so obvious. I used a fixed $k=20$, and tried different ratio: 0.6, 0.7, 0.8, 0.9, 0.95. I used the 50 PCs as inputs. The outputs did not make much sense, e.g., for all ratios, almost all the cells were in cluster 0.

In addition, the CDC algorithm is not as efficient as advertised if all steps in the functions are considered.

There could be potential issues with bugs in code, parameter setting etc. However, the parameter setting for scRNA-seq data was not in github:

There are numerous cases where t-SNE and UMAP can make mistakes, especially for very large datasets. For example, there are frequent 'crowding' problems.

In summary, I still have concerns about the utility of the algorithm on real data such as scRNA-seq data. I'd really like to see what other people such as Lior Pachter think about this manuscript. He 'famously' argues that t-SNE/UMAP are just art, while the CDC algorithm in this paper, mostly run on UMAP results, performs better than competing algorithms such as Seurat using higher-dimensional data as inputs. I agree that clustering on t-SNE/UMAP may work on some datasets, as demonstrated in previous studies as in the Drop-seq paper. This is, in general, not the preferred way to go. It's typical to see very similar cell types that are mixed in tSNEs/UMAPs, e.g., different CD4 T cells.

I'm also curious why the authors selected the scRNA-seq data presented in this manuscript? The mouse retina neurons with a large number of discrete clusters could be a good test dataset.

Reviewers' comments:

Reviewer #1 (Remarks to the Author):

Thanks for the through explanations. According to the responses and revisions, I feel the generality and stability of CDC is acceptable. The current form of the paper is sound to me. Here are a few more minor suggestions.

Thank you for the positive comments.

1. What is the time and memory complexity of the Qhull algorithm? Is it always practical when the dataset is considerably large? This is important as the proposed CDC relies on Qhull on high-dimensional data.

The time and space complexity of Qhull are $O(nk \log k)$ and $O(k)$ respectively for $d \leq 3$; $O(n(k^{\lfloor \frac{d}{2} \rfloor} / \lfloor \frac{d}{2} \rfloor!))$ and $O(k^{\lfloor \frac{d}{2} \rfloor} / \lfloor \frac{d}{2} \rfloor!)$ for $d > 4$, where k is the number of nearest neighbors, and d is the data dimension (Klee et al., 1966, Barber et al., 1996). Hence, **Qhull will not increase the complexity of CDC algorithm in low-dimensional spaces**, and introduces computing overhead when the data dimension increases. According to our test, the runtime of CDC in 5D UMAP space of a scRNA-seq dataset with 10,000 cells is around 6s in MATLAB and 20s in R, **which is comparable with the commonly-used biological software Seurat**. So, **we recommend embedding the data into spaces of low dimensions (e.g., 2D-5D)** before conducting CDC. The time efficiency can be further improved by integrating the Qhull algorithm in C++ version.

References:

- [1] Klee, V. Convex polytopes and linear programming. In Proceedings of the IBM Scientific Computing Symposium: Combinatorial Problems. 123-158 (1966).
- [2] Barber, C. B., Dobkin, D. P. & Huhdanpaa, H. The Quickhull Algorithm for Convex Hulls. (1996). <https://www.cise.ufl.edu/~ungor/courses/fall06/papers/QuickHull.pdf>

2. Since the previous comparison against RCC is not that "fair", I suggest the authors check other baselines again to make sure that the evaluation is properly conducted.

Thank you for your constructive comments. RCC did not obtain high accuracy in the first and second rounds of revisions since the dataset MNIST was not preprocessed using normalization. Hence, **we had performed unified preprocessing for all baselines** in the last revision, including RCC (NMI has been improved to 0.761). The **experiment design has been double checked**, and we **guarantee that the current comparison is fair** since we used the same preprocessing and traversed parameter spaces exhaustively to search the optimal results for all baselines.

Reviewer #4 (Remarks to the Author):

I appreciate the authors for the updated experiments. I can see the authors have put lots of time running these experiments. The results (Fig.2) presented in the paper are great. However, after reading the updated manuscript and from some quick experiments by myself. I still have the same concerns about the robustness of the algorithms on real data.

I checked the R code and wanted to do some analysis. Unfortunately, the CDC function has several parameters. Except for a data matrix, users also have to select at least two parameters, namely, k and ratio.

In practice, users may just want to use the default setting for analysis. With these extra unfamiliar parameters, new users may just turn away from the tool.

Thank you for testing our code, as well as your invaluable comments. CDC has two input parameters as mentioned in the manuscript, i.e., *k* and *ratio*. We agree with you that setting appropriate parameters brings burdens on end users, especially for inexperienced users. According to your comments, we have **developed a toolkit that assembles six applications contained in our manuscript** (i.e., *scRNA-seq cluster*, *UCI benchmark test*, *synthetic data analysis*, *CytoF cluster*, *speaker recognition* and *face recognition*) by learning from Seurat, and have uploaded it to our Github. It integrates the functions of **data preprocessing, feature extraction, dimension reduction, clustering evaluation and comparison, visualization**, and necessary **default clustering parameter settings** together, making it much easier for users from different fields to utilize and customize it. The six applications are implemented using R, MATLAB and Python languages, and each application is put in a separate folder. The framework is shown in R-Fig. 1.

Specifically, the application of ‘scRNA-seq Cluster’ contains the full workflow for cluster analysis on scRNA-seq datasets:

Step 1: Convert the raw data to standard Seurat object.

Step 2: Preprocess the data using standard Seurat pipeline, including normalization, HVG selection, scaling and PCA&UMAP embedding.

Step 3: Perform CDC on the embedded data.

Step 4: Visualize and evaluate clustering results, as well as compare performances with integrated baselines (Seurat, SNN-Walktrap, SNN-Louvain, K-means).

R-Fig. 1 The framework of CDC toolkit for six applications.

We have provided the default arguments for CDC algorithm ($k = 30$, $\text{ratio} = 0.9$), as well as the detailed instructions for parameter setting in the code annotation. **Users only need to specify the file names and formats for running this application on their own datasets** (To be noted, in order to make a quick comparison, the parameter ranges of SNN-Walktrap and SNN-Louvain in file ‘Compare.R’ are narrowed and the traversing granularity are coarser than that in Supplementary Note 4). **We also wrote a “Getting Started” tutorial for this toolkit and made a screen-recording video to introduce how to run it.** Users can execute the demo code by opening and running the ‘main.R’ file easily, the screenshots for executing ‘scRNA-seq cluster’ application are presented in R-Fig. 2. Moreover, we put all the parameter settings of our experiments on synthetic, speaker corpus datasets and UCI benchmarks in Supplementary Table 7. For scRNA-seq and CyTOF datasets, their parameter settings can be found in Supplementary Notes 4 and 5. Users can specify them in the code to reproduce our experiments and make a comprehensive comparison.

R-Fig. 2. Screenshots of the 'scRNA-seq cluster' application in CDC toolkit.

Meanwhile, it should be noted that default parameter settings are empirical and can only produce rational results, but cannot guarantee optimal clustering results on all datasets, including both the mainstream

biological software and ours. For example, Seurat requires to specify parameters ‘dims’, ‘k.param’ and ‘resolution’ in the phase of clustering (i.e., the dimensions of reduction to use as input, k for the number of nearest neighbors in KNN, and value of the resolution parameter respectively). The optimal clustering result needs to be found by tuning the parameter settings and traversing the whole parameter space. Taking dataset pbmc3k as an example, the ARI of Seurat under defaults (dims = 1:10, resolution = 0.8) is 0.462, while the found optimal ARI is 0.784 (dims = 1:35, resolution = 0.3).

Anyway, I used the 'standard' PBMC data here (https://satijalab.org/seurat/articles/pbmc3k_tutorial.html) for testing. For this simple dataset, I expected CDC should work pretty well as the UMAP shows clear clusters. But unfortunately, it's not so obvious. I used a fixed k=20, and tried different ratio: 0.6, 0.7, 0.8, 0.9, 0.95. I used the 50 PCs as inputs. The outputs did not make much sense, e.g., for all ratios, almost all the cells were in cluster 0.

We are sorry that you did not obtain promising clustering result through the provided CDC code. **The main reason we analyzed is probably that there are some issues during the preprocessing that led to an inconsistent result with the standard pipeline. Considering that CDC is a generic clustering method, and multiple non-biological experiments for various application scenarios are also included in this manuscript, we only uploaded the core code of CDC in previous. After receiving your feedback, we realized that it might be a potential issue leading to the misunderstanding and incorrect use of the code.** We are sorry for the inconvenience.

In order to eliminate unnecessary misuses and make it easier for users from different fields to utilize our code, we made **great efforts to integrate the preprocessing codes for different applications with our core algorithm in recent three weeks.** Now, we have uploaded a new CDC toolkit to Github, which supports **six applications** contained in our manuscript (i.e., *scRNA-seq cluster*, *UCI benchmark test*, *synthetic data analysis*, *CyTOF cluster*, *speaker recognition* and *face recognition*). You can run each application to reproduce the results of our experiments.

In addition, we also used the scRNA-seq dataset *pbmc3k* you selected to testify our algorithm by comparing with four baselines, i.e., Seurat, SNN-Walktrap, SNN-Louvain, K-means. We specified the parameter space for each method as follows:

Method	Parameter	Setting
Seurat	dims	20, 25, 30, 35, 40, 45, 50
	resolution	0.1, 0.2, 0.3, 0.4, 0.5, 0.6, 0.7, 0.8, 0.9, 1.0
SNN-Walktrap	k	5, 10, 15, 20, 25, 30
SNN-Louvain	k	5, 10, 15, 20, 25, 30
K-means	k	2~50
CDC	n components (UMAP)	2, 3, 4, 5
	k	30, 40, 50
	ratio	0.55, 0.56, 0.57, 0.58, 0.59, 0.60, 0.61, 0.62, 0.63, 0.64, 0.65

We conducted SNN-Walktrap, SNN-Louvain, Kmeans both in 50D PCA (denoted as -PCA) and 2D UMAP spaces (denoted as -U2), while CDC in 2D-5D UMAP spaces (denoted as -U2, -U3, -U4 and -U5). As shown in R-Fig. 3a, our algorithm CDC obtained the highest ARI score 0.792 in 5D UMAP space. Meanwhile, CDC has good robustness as the parameter settings varying especially for U2, U3 and U5, and its average ARI scores outperformed that of Seurat, SNN-Walktrap-PCA, SNN-Walktrap-U2, SNN-Louvain-PCA, SNN-Louvain-U2, Kmeans-PCA, Kmeans-U2. We also performed CDC using the parameters you specified (k = 20, ratio = 0.6, 0.7, 0.8, 0.9, 0.95). The clustering results of CDC are shown in R-Fig. 3b-f. Cells are grouped into multiple clusters rather than “The outputs did not make much sense, e.g., for all ratios, almost all the cells were in cluster 0”. Therefore, the clustering results of CDC on dataset *pbmc3k* make sense through our test experiment.

R-Fig. 3 Clustering performances on *pbmc3k* dataset. (a) Clustering accuracies reported by ARI score of five algorithms conducted in PCA and UMAP spaces. (b-f) Clustering results generated using the parameters specified by Reviewer #4 (k = 20, ratio = 0.6, 0.7, 0.8, 0.9, 0.95).

In addition, the CDC algorithm is not as efficient as advertised if all steps in the functions are considered.

Thank you for your comments. In Response Fig. 4 of the last response letter, to purely compare the computational efficiency of the clustering algorithms in each method (e.g., SNN and Louvain algorithms in Seurat), we unified the preprocessing steps and used the same 2D UMAPs as inputs for each method. Therefore, that experiment only measured the runtimes of clustering algorithms, but not included that of preprocessing and UMAP embedding. While, we measured the overall runtime of CDC (including UMAP) in Fig. 2f and Fig. 5c of our manuscript, since the competitors have different preprocessing steps and dimension reduction techniques. In addition, the runtimes of CDC were measured in MATLAB for all previous experiments, and its computational efficiency is distinct in different programming languages. Actually, dozens of the baselines we used in this paper are natively implemented in different languages (MATLAB, R, Python, Java and etc.), and it is hard to unify. We can only ensure the same computational environment (a commodity desktop computer with an 8-core Intel i7 processor and 64 GB RAM) in the comparisons of time efficiency. The programming languages and parameter settings of the clustering methods are listed in the table of Supplementary Note 5.

To further clarify the computational efficiency of our algorithm, here, we tested the runtimes of SC3, Seurat v3 and CDC of R and MATLAB versions. We generated sub-datasets with different number of cells by sampling from the scRNA-seq dataset *TM*, and adopted the same preprocessing pipeline for all methods. The workflows of four cluster analysis methods for scRNA-seq dataset are presented in R-Fig. 4a. R-Fig. 4b measured the runtimes for clustering cells (not including preprocessing), and R-Fig. 4c measured that of the

entire workflows (including preprocessing). **To be noted, the runtimes of CDC in this experiment include that of UMAP.** The result indicates that the runtime of CDC (MATLAB) is almost the same with that of Seurat, and faster than other methods. Its computational efficiency is around 10-fold than that of CDC (R). In the same R environment, CDC runs slower than Seurat, but is around 3.4-fold faster than SC3 and 4.3-fold than SNN-Walktrap. It is worth mentioning that the runtimes of CDC (MATLAB) and CDC (R) in 5D UMAP space of scRNA-seq with 50k cells are less than one and seven minutes respectively, which are still faster than that of SC3 and SNN-Walktrap. But in fact, **the steps of preprocessing are the most time-consuming part.** It took seven minutes to preprocess the scRNA-seq dataset with 50k cells. Consequently, when considering the runtime of preprocessing, the gap of computational efficiency between Seurat and CDC (R) has narrowed significantly and could be acceptable.

R-Fig. 4 Computational efficiencies of four methods on sub-datasets with different number of cells sampled from scRNA-seq dataset *TM*. (a) The workflows of four cluster analysis methods on scRNA-seq dataset. (b, c) Runtimes of four methods (not including and including the preprocessing respectively) on scRNA-seq datasets with different number of cells.

CDC in MATLAB version runs much faster than that of R code, since the execution engine of MATLAB uses JIT compilation to accelerate the code. This architecture enables faster calling to built-in functions and indexing operations. In addition, many core MATLAB functions are implicitly multithreaded for better performance. Therefore, if a user expects a higher computational efficiency on large datasets, we recommend to utilize CDC in MATLAB version. MATLAB also provides toolboxes with the standard pipeline for scRNA-seq analysis. Meanwhile, we are developing a parallel version of CDC for performance acceleration upon distributed computing framework Apache Spark by utilizing the nature of KNN-based calculation, and the parallel version will be uploaded to our GitHub in future.

There could be potential issues with bugs in code, parameter setting etc. However, the parameter setting for scRNA-seq data was not in github:

As aforementioned, the bugs might be caused by inappropriate preprocessing or incorrect invocation of CDC. To help users to reproduce our experiments, and reduce the learning curves and parameter tuning costs, we have uploaded the newly developed CDC toolkit to our Github, which provides default values and necessary instructions for parameter settings. Meanwhile, all the parameter settings of the biological experiments in our paper can be found in the Supplementary Note 4 and 5.

There are numerous cases where t-SNE and UMAP can make mistakes, especially for very large datasets. For example, there are frequent 'crowding' problems.

In summary, I still have concerns about the utility of the algorithm on real data such as scRNA-seq data. I'd really like to see what other people such as Lior Pachter think about this manuscript. He 'famously' argues that t-SNE/UMAP are just art, while the CDC algorithm in this paper, mostly run on UMAP results, performs better than competing algorithms such as Seurat using higher-dimensional data as inputs. I agree that clustering on t-SNE/UMAP may work on some datasets, as demonstrated in previous studies as in the Drop-seq paper. This is, in general, not the preferred way to go. It's typical to see very similar cell types that are mixed in tSNEs/UMAPs, e.g., different CD4 T cells.

Thanks. We admit that t-SNE/UMAP may have their limitations, but it does not affect the feasibility of our algorithm. Clustering on high-dimensional PCs for scRNA-seq datasets is not the only solution. A “not the preferred way to go” may provide new potentials. According to your comments, we have discussed the potential impact of dimension reduction to CDC in Line 452-455, Page 23 in the manuscript.

- 1) **Dimension reduction is commonly used for preprocessing high-dimensional data.** As far as we know, monocle3 utilizes UMAP for dimension reduction; ACCENSE, clusterX, DensVM require to perform t-SNE before clustering. Our algorithm CDC can be equipped with any dimension reduction technique (PCA, t-SNE or PCA). For example, we used the t-SNEs as inputs for CDC on UCI benchmarks and obtained high clustering accuracies in the second round of revision.
- 2) ‘Crowding’ is a typical and common problem in dimension reduction. However, so far, we have not found there are cases in which CDC clustering fails on UMAPs through our experiments, and **we have verified that CDC equipped with UMAP obtained high accuracies and robust outcomes** (Fig. 2, 3 and 5 in the manuscript, and Supplementary Tables 2, 4 and 5). To be noted, we have specified a fixed parameter setting of random seed in our code, so that UMAP is able to output stable results. In addition, CDC does not have to be conducted in 2D UMAP space. **Users can specify the appropriate dimensions according to their needs.**
- 3) More importantly, **the cells might also be inseparable and mixed up in high-dimensional PCA space, when there is a ‘crowding’ problem in the UMAP results.** PCA is an essential step in scRNA-seq preprocessing. As a linear mapping technique, it cannot achieve complete divisibility of cells even in 50D space sometimes, especially for high-dimensional data that lies on non-linear manifolds (Maaten et al., 2008). While, UMAP can better achieve low-dimension representation for manifold data, and keep the divisibility between clusters, since it adopts spectral clustering to preprocess the data and preserves the global structure (McInnes et al., 2018). Thus, **it facilitates the direction centrality of our algorithm CDC to be better applicable** (see our response to Reviewer #1 question 4 in the last response letter).
- 4) UMAP only cannot guarantee high clustering accuracy, a proper clustering method is quite vital. **In the same UMAP dimension, CDC outperforms the baselines** (e.g., SNN-Walktrap, SNN-Louvain, K-means, C-means and other versatile methods) on experimental datasets (i.e., 10 scRNA-seq, 2 CyTOF, 2 corpuses, 9 UCI and one face image datasets). Moreover, CDC obtained more robust outcomes than Seurat and other clustering baselines that directly use high-dimensional features as inputs. We have also demonstrated the superiority of CDC in max ARI score on larger and more complex datasets (e.g., AMB, ALM, VISp, TM and

MIHPF in the manuscript) because of the capability for handling data with weak connectivity. Those experiments in this paper **have proved the advantages and value of our method**.

According to your comments, we selected two PBMC (Peripheral Blood Mononuclear Cells) benchmarks, *pbmc3k* and *SCINA*, and compared with biological and generic baselines to prove our points. The two benchmarks containing similar types of T cells including CD4 T cells that are mixed as shown in R-Fig. 5b. We specify the parameter spaces for each method as follow:

Method	Parameter	Setting
Seurat	dims	20, 25, 30, 35, 40, 45, 50
	resolution	0.1, 0.2, 0.3, 0.4, 0.5, 0.6, 0.7, 0.8, 0.9, 1.0
SNN-Walktrap	k	5, 10, 15, 20, 25, 30
SNN-Louvain	k	5, 10, 15, 20, 25, 30
K-means	k	2~50
CDC	n_components (UMAP)	2, 3, 4, 5
	k	30, 40, 50
	ratio	pbmc3k : 0.55, 0.56, 0.57, 0.58, 0.59, 0.60, 0.61, 0.62, 0.63, 0.64, 0.65 SCINA : 0.70, 0.71, 0.72, 0.73, 0.74, 0.75, 0.76, 0.77, 0.78, 0.79, 0.80

R-Fig. 5 Clustering performances on two PBMC datasets. (a) Clustering accuracies reported by ARI score of five algorithms conducted in PCA and UMAP spaces. (b) The true cell annotations of the two PBMC datasets.

Partial cells in PBMC datasets are mixed in 2D UMAP space as you mentioned (in R-Fig. 5b). However, the clustering accuracies of CDC in low-dimensional UMAP spaces are comparable to that of the baselines using 50 PCs as inputs, and even better than Seurat on *pbmc3k* (in R-Fig. 5a). In terms of the robustness, CDC performs relative stably in clustering accuracy as the parameters varying.

Accordingly, we have presented the above results in Supplementary Fig. 4, and added discussion about the potential impact of dimension reduction to CDC in Line 452-455, Page 23:

Utilizing dimension reduction techniques such as UMAP to embed the data to a proper dimension can broaden the application of CDC. Consequently, in some extreme cases, parameter setting should be more careful since dimension reduction may cause “crowding problem” and affect clustering accuracy.

References:

[1] van der Maaten, L. & Hinton, G. Visualizing data using t-SNE. *J. Mach. Learn. Res.* **9**, 2579-2605 (2008).
 [2] McInnes, L., Healy, J., Saul, N. & Großberger, L. UMAP: uniform manifold approximation and projection. *J. Open Source Softw.* **3**, 861 (2018).
 [3] pbmc3k: https://cf.10xgenomics.com/samples/cell/pbmc3k/pbmc3k_filtered_gene_bc_matrices.tar.gz
 [4] SCINA: <https://zenodo.org/record/3357167#.YhsmrN9ByUk>

I'm also curious why the authors selected the scRNA-seq data presented in this manuscript? The mouse retina neurons with a large number of discrete clusters could be a good test dataset.

In this manuscript, we selected nine typical scRNA-seq datasets published from different platforms to evaluate the generality and stability of CDC by comparing with seven biological pipelines and seven versatile clustering methods. We also used a scRNA-seq dataset with over one million cells to prove the scalability and upper limit to handle large-scale dataset. These datasets are representative, since they are collected from tissues of different species, and contain cells with different data distributions and true cell labels that can be adopted to calculate the evaluation metrics (e.g., ACC, NMI, ARI) for quantitative analysis. Besides, they have been **widely used as clustering benchmarks for the comparison of cell identification methods** (Abdelaal et al., 2019; Yao et al., 2021), which provides a fair reference for validating the effectiveness of our algorithm.

According to your comments, we collected four mouse retina scRNA-seq datasets from a PNAS paper (Heng et al., 2019) furtherly. From the comparison in R-Fig. 6a, we can find that CDC achieved comparable max ARI score with Seurat and the most stable outcomes among all the methods. CDC can also obtain promising clustering results using the default parameter settings as shown in R-Fig. 6b. Accordingly, we have presented the above results in Supplementary Fig. 5.

Method	Parameter	Setting
Seurat	dims	20, 25, 30, 35, 40, 45, 50
	resolution	0.1, 0.2, 0.3, 0.4, 0.5, 0.6, 0.7, 0.8, 0.9, 1.0
SNN-Walktrap	k	5, 10, 15, 20, 25, 30
SNN-Louvain	k	5, 10, 15, 20, 25, 30
K-means	k	2~50
	n_components (UMAP)	2, 3, 4, 5
CDC	k	30, 40, 50
	ratio	0.75, 0.77, 0.79, 0.81, 0.83, 0.85, 0.87, 0.89, 0.91, 0.93, 0.95

R-Fig. 6 Clustering performances on four mouse retina datasets. (a) Clustering accuracies reported by ARI score of five algorithms conducted in PCA and UMAP spaces. (b) Clustering results of CDC with default parameter settings ($k = 30$, ratio = 0.9).

References:

- [1] Abdelaal, T. et al. A comparison of automatic cell identification methods for single-cell RNA sequencing data. *Genome Biol.* **20**, 194 (2019).
- [2] Yao, Z. et al. A taxonomy of transcriptomic cell types across the isocortex and hippocampal formation. *Cell* **184**, 3222-3241 (2021).
- [3] Heng, J. et al. Hypoxia tolerance in the Norrin-deficient retina and the chronically hypoxic brain studied at single-cell resolution. *Proc. Natl. Acad. Sci. U. S. A.* **116**, 9103-9114 (2019).
- [4] WT_R1, WT_R2, NdpKO_R1, NdpKO_R2: GSE125708.

Reviewers' Comments:

Reviewer #1:

Remarks to the Author:

The authors response resolved my concerns. I feel the current form of the paper is acceptable.

Reviewer #4:

Remarks to the Author:

I really appreciate the authors for addressing my comments. I'm glad to see that the code in github has been updated.

Considering the PBMC results in R-Fig. 3, the results were still kind of poor, e.g., CD4/8/NK cells were in the same cluster, CD14/16 monocytes were in the same cluster. Only when the ratio parameter =0.6, NK and CD8+ T cells are separated. However, this time, CDC produced 18 clusters while the 'ground truth' has nine cell types. Some cell types were over-clustered, e.g., some clusters had only two cells.

As the authors demonstrated, for this simple dataset, CDC worked good only when 5D UMAP results were used. However, the authors did a granular parameter search, e.g., the ratio parameter was from 0.55 to 0.65, and k in {30, 40, 50}. To me, all these results also suggested that the CDC algorithm is sensitive to parameters. Moreover, these results demonstrated the limitations of clustering on 2D UMAPs.

```
I played with the pbmc data using the updated code
pbmc1 <- RunUMAP(pbmc, dims = 1:10, n.components=5)
c1 <- CDC(pbmc1@reductions[["umap"]])@cell.embeddings, 30, 0.65)
```

For K=30, I varied the ratio parameter from 0.55 to 0.65, the best ARI I could get was 0.6528 when ratio=0.6. But this time, CDC produced 20 clusters.

It may be good for the authors to make the exact scripts (also data) running all these experiments available for readers to reproduce the results.

As for clustering on tSNE/UMAP or clustering on high-dimensional space, Supplementary Fig. 11 of the paper "the specious art of single-cell genomics" shows that many digits were assigned to the wrong clusters detected from t-SNE/UMAP. For large datasets, the crowding problem is almost inevitable for t-SNE, as demonstrated by a recent paper: <https://arxiv.org/abs/2110.02573>

Although now CDC can also use high-dimensional data as inputs, however, there are still several problems, e.g., CDC is computational much more expensive for high-dimensional data (I tried CDC on 5D data and found it was much slower compared to running using 2D data as inputs). Moreover, the estimation of the knn directions in high-dimensional space may be bad because of the curse of dimensionalities.

I understand the authors have put lots of time on revising the manuscript. I'm sorry I still not convinced that tSNE/UMAP + CDC is good for clustering high-dimensional data, especially scRNA-seq data, although in some cases, this approach may help pick some clusters.

My comments may be biased based on my experience, so it could be good to have other reviewers with extensive experience on t-SNE/UMAP.

Reviewers' comments:

Reviewer #4 (Remarks to the Author):

Thanks for the invaluable suggestions for helping us improving our paper in previous reviews. However, we respectfully yet strongly disagree with your comments. The reasons are briefly summarized as follows:

- **The default preprocessing settings in our code were modified by Reviewer #4, and incurred reproductivity issue.** In the last round of review, Reviewer #4 suggested us providing default and detailed parameter settings in our code for audiences to reproduce the experimental results, and we have improved our code accordingly (as mentioned in previous response letter). However, the reviewer modified the default parameter settings when running our toolkit (*dims* in *RunUMAP* function is set as 1:50 in our code, but the reviewer modified it to 1:10, so that the obtained best accuracy is 0.6528 which is lower than our result 0.7920), thus obtained worse results than that we present in the response letter. It is a serious mistake and **the conclusions drawn based on the wrong use of our code make no sense** and such a clustering result could be also predictable, **since no algorithm can guarantee promising results under all possible parameter settings**, especially for *exploratory analysis methods such as clustering algorithm that need to specify a rational parameter range*.
- **Important concepts in cluster analysis are misused.** Commonly, **clustering quality is evaluated by accuracy metrics** (such as ARI, ACC and NMI), **rather than the number of identified clusters** (mentioned by the reviewer). Different parameter settings do produce different results, and a comprehensive evaluation should consider both recall and precision of all clusters. Therefore, **the effectiveness and robustness of the algorithm cannot be denied just because the number of identified clusters differs from that of the true clusters** (i.e., inseparable and over-clustered). Actually, the representative scRNA-seq clustering baselines, e.g., Seurat, SC3, monocle3 and SNN-based methods, also cannot guarantee a fixed number of output clusters as the parameters vary.
- **Biased comments.** Reviewer #4's negative comments on the approach of dimension reduction using tSNE/UMAP is empirical and subjective, which didn't reflect the facts of our experimental results. Through solid comparative experiments on different datatypes and multiple applications, we have demonstrated the validity of our algorithm. It is able to **obtain the highest accuracies on most experimental datasets**, commonly in the recommended range of parameters, and **achieve the most stable outputs with the default settings**. Since CDC can handle the potential weak connectivity in the raw data and embedded data using dimension reduction method, we did not meet the "crowding problem" mentioned by Reviewer #4. **We insist that the experimental results are authentic, reliable and reproducible.**

Below, we present a point-by-point response with detailed evidences to support our points:

1) Reviewer comments: *"I really appreciate the authors for addressing my comments. I'm glad to see that the code in github has been updated."*

Thank you for the positive comments.

2) Reviewer comments: *"I played with the pbmc data using the updated code*

```
pbmc1 <- RunUMAP(pbmc, dims = 1:10, n.components=5)
c1 <- CDC(pbmc1@reductions[["umap"]])@cell.embeddings, 30, 0.65)
```

For K=30, I varied the ratio parameter from 0.55 to 0.65, the best ARI I could get was 0.6528 when ratio=0.6. But this time, CDC produced 20 clusters.

It may be good for the authors to make the exact scripts (also data) running all these experiments available for readers to reproduce the results.”

Our response:

Improper parameter settings

The reviewer changed the default preprocessing setting in our code, thus did not reproduce our results. We have provided the optimal parameter settings of CDC on pbmc3k in the response letter, i.e., $k = 30, 40, 50$, ratio = 0.55~0.65. As shown in R-Fig. 3a in the last response letter, CDC can achieve high and stable accuracy in this parameter range using the *scRNA-seq Cluster* application in CDC Toolkit, and obtains ARI = 0.792 in 5D UMAP space, which outperforms Seurat and other baselines. We conduct UMAP based on 50 PCs and set dims = 1:50 as defaults in our code (Please check Line 28 in the “SeuratPreprocess.R” file, see Fig. 1). However, the reviewer changed the default dimensions of PCA to 10, i.e., “pbmc1 <- RunUMAP(pbmc, dims = 1:10, n.components=5)”, which is inconsistent with our setting. (Actually, we have clarified in the tutorial that the users could customized parameter settings in “main.R” as highlighted in Fig. 2 rather than modifying defaults in other files, such as “SeuratPreprocess.R”.)

```
library(argparse)
library(Seurat)
library(ggplot2)
library(readr)
library(SingleCellExperiment)

## This code aims to preprocess the raw scRNA-seq using Seurat, including normalization, HVG selection, scaling and dimension reduction.
## The intermediate results during preprocessing will be written in the data file.

SeuratPreprocess <- function(seuratSCE, filename, UMAP_Dim = 2){

  ## Normalization
  seuratSCE <- NormalizeData(seuratSCE)
  # write.csv(seuratSCE@assays$RNA@data, file=file.path(filename, "Seurat.GeneExprMat.AllGene.Norm.csv"))

  ## Find HVG
  seuratSCE <- FindVariableFeatures(seuratSCE)
  p <- VariableFeaturePlot(seuratSCE)
  ggsave(file.path(filename, "Seurat.HVG_selection.pdf"), p, width = 25, height = 20, units = "cm")
  write(seuratSCE@assays$RNA@var.features, file.path(filename, "Seurat.HVG.txt"))

  ## Scaling Data
  seuratSCE <- ScaleData(seuratSCE, features = rownames(seuratSCE))
  # write.csv(seuratSCE@assays$RNA@scale.data, file=file.path(filename, "Seurat.GeneExprMat.AllGene.Scaled.csv"))

  ## PCA & UMAP Embedding
  seuratSCE <- RunPCA(seuratSCE, features = variableFeatures(object = seuratSCE), npcs=50)
  seuratSCE <- RunUMAP(seuratSCE, n.components = UMAP_Dim, dims=1:50)
  write.csv(seuratSCE@reductions$pcacell.embeddings, file=file.path(filename, "Seurat.PCA.csv"))
  write.csv(seuratSCE@reductions$umapcell.embeddings, file=file.path(filename, "Seurat.UMAP.csv"))

  return(seuratSCE)
}
```

Fig. 1 The default parameter settings in “SeuratPreprocess.R” file.

```

1  ## Specify the filename, format and labels.
2  ## --Arguments--
3  ##   filename: filename of the scRNA-seq dataset (we provide nine sample datasets in this application, please decompress them in
4  ##   format: data format of the scRNA-seq dataset (we currently support two data formats, csv and 10X)
5  ##   labels: whether the file contains the true label of cells ('1' is Yes, '0' is No)
6  filename = 'Baron-Mouse'
7  format = 'csv'
8  labels = '1'
9
10 ## Generate Seurat Object from the raw data
11 source('BuildSeuratObject.R')
12 SeuratData <- BuildSeuratObject(filename, format, labels)
13
14 ## Preprocess scRNA-seq using standard Seurat pipeline
15 ## --Arguments--
16 ##   n_components: The dimension of UMAP space to embed into (Default: 2, Recommended: 2~5)
17 ##   n_neighbors: The number of neighboring points of UMAP (Default: 30, Recommended: 5~50)
18 ##   min_dist: It controls how tightly the UMAP embedding is allowed compress points together (Default: 0.3, Recommended: 0.1~1)
19 source('SeuratPreprocess.R')
20 n_components = 2
21 n_neighbors = 30
22 min_dist = 0.3
23 SeuratData <- SeuratPreprocess(SeuratData, filename, n_components, n_neighbors, min_dist)
24
25 ## Cluster the cells using CDC algorithm
26 ## --Arguments--
27 ##   k: k of KNN (Default: 30, Recommended: 5~50)
28 ##   ratio: percentile ratio of internal points (Default: 0.9, Recommended: 0.85~0.99, 0.55~0.65 for pbmc3k, 0.7~0.8 for SCINA)
29 source('CDC.R')
30 k = 50
31 ratio = 0.95
Idents(SeuratData) <- CDC(SeuratData@reductions[["umap"]]|cell.embeddings, k, ratio)
## Plot the clustering result
DimPlot(SeuratData, pt.size=1) + NoLegend()
## Evaluate the clustering accuracy using ARI metric
ARI <- mclust::adjustedRandIndex(Idents(SeuratData), SeuratData@meta.data[["cluster"]])
## Compare the clustering performance with Seurat, SNN-walktrap, SNN-Louvain, K-means
## Noted: Comparison is usually time consuming, if you just want to run CDC algorithm and don't
## want to compare with other algorithm by traversing different settings in the parameter
## spaces, please just comments out this part of the code
## --Arguments--
##   seurat_dim: dimension of reduction to use as input in Seurat (Default: seq(20, 50, 5))
##   seurat_resolution: resolution of Louvain algorithm in Seurat (Default: seq(0.1, 1, 0.1))
##   snnwalk_k: k of SNN graph in SNN-walktrap (Default: seq(5, 30, 5))
##   snnlouv_k: k of SNN graph in SNN-Louvain (Default: seq(5, 30, 5))
##   kmeans_k: k of k-means (Default: seq(2, 50))
##   CDC_k: k of KNN in CDC (Default: seq(30, 50, 10))
##   CDC_ratio: ratio of CDC (Default: seq(0.85, 0.99, 0.02))
seurat_dim = seq(20, 50, 5)
seurat_resolution = seq(0.1, 1, 0.1)
snnwalk_k = seq(5, 30, 5)
snnlouv_k = seq(5, 30, 5)
kmeans_k = seq(2, 50)
CDC_k = seq(30, 50, 10)
CDC_ratio = seq(0.85, 0.99, 0.02)
source('Compare.R')
box_res <- Compare(SeuratData, filename, seurat_dim, seurat_resolution, snnwalk_k, snnlouv_k, kmeans_k, CDC_k, CDC_ratio)
boxplot(box_res, ylab='ARI', col=c('#FF69B4', '#4682B4', '#3CB371', '#FF7F50', '#008B8B', '#FFD700', '#FF0000', '#1E90FF'))
## output the ARI score of CDC with the customized arguments
cat(paste0('ARI = ', round(ARI,4), ' (UMAP_Dim = ', n_components, ', k = ', k, ', ratio = ', ratio, ')'), '\n')

```

Fig. 2 The parameter settings that users can customize in “main.R” file is highlighted using red boxes (the default and recommended settings are also provided in code annotations).

```

## Specify the filename, format and labels.
## --Arguments--
## filename: filename of the scRNA-seq dataset (We provide nine sample datasets in this application, please decompress them in the correct
## format: data format of the scRNA-seq dataset (we currently support two data formats, csv and 10x)
## labels: whether the file contains the true label of cells ('1' is yes, '0' is No)
filename = 'pbmc3k'
format = '10x'
labels = '1'
## Generate Seurat object from the raw data
source('BuildSeuratObject.R')
seuratData <- BuildSeuratObject(filename, format, labels)
## Preprocess scRNA-seq using standard Seurat pipeline
## --Arguments--
## UMAP_Dim: The dimension of UMAP space to embed into. (Default: 2, Recommended: 2-5)
source('SeuratPreprocess.R')
UMAP_Dim = 5
SeuratData <- SeuratPreprocess(SeuratData, filename, UMAP_Dim)
## Cluster the cells using CDC algorithm
## --Arguments--
## k: k of KNN (Default: 30, Recommended: 30-50)
## ratio: percentile ratio of internal points (Default: 0.9, Recommended: 0.75-0.95, 0.55-0.65 for pbmc3k)
source('CDC.R')
k = 40
ratio = 0.62
Idents(SeuratData) <- CDC(SeuratData@reductions[["umap"]])@cell.embeddings, k, ratio)
## Plot the clustering result
DimPlot(SeuratData, pt.size=1) + NoLegend()
## Evaluate the clustering accuracy using ARI metric
ARI <- mclust::adjustedRandIndex(Idents(SeuratData), SeuratData@meta.data[["Cluster"]])
## Compare the clustering performance with Seurat, SNN-Walktrap, SNN-Louvain, K-means
## Noted: comparison is usually time consuming. If you just want to run CDC algorithm and don't
<

```

```

F:\NatCom\Toolkit\Toolkit\scRNA-seq Cluster/ >
> # Seurat_resolution = seq(0.1, 1, 0.1)
> # snnwalk_k = seq(5, 30, 5)
> # snnlouv_k = seq(5, 30, 5)
> # kmeans_k = seq(2, 50)
> # CDC_k = seq(30, 50, 10)
> # CDC_ratio = seq(0.75, 0.95, 0.02)
> #
> # source('Compare.R')
> # box_res <- Compare(SeuratData, filename, seurat_dim, seurat_resolution, snnwalk_k, snnlouv_k, kmeans_k, CDC_k, CDC_ratio)
> # boxplot(box_res, ylab='ARI', col=c('#FF69B4', '#4682B4', '#3CB371', '#FF7F50', '#008B8B', '#FFD700', '#FF0000', '#1E90FF'))
>
> ## Output the ARI score of CDC with the customized arguments
> cat(paste0('ARI: ', round(ARI,3), ' (n_components = ', UMAP_Dim, ', k = ', k, ', ratio = ', ratio, ')), '\n')
ARI = 0.792 (n_components = 5, k = 40, ratio = 0.62)
>

```

Fig. 3 Screenshot of the best ARI obtained by CDC with k=40 and ratio=0.62.

Based on our default preprocessing (dims = 1:50), by varying k from 30 to 50 and ratio from 0.55 to 0.65, we can get the ARIs on a Windows machine as shown in Table 1 and best ARIs on a macOS machine in Table 2 (To be noted the results obtained under different Operating Systems may varies due to the implementation of the dependency libraries, and we have mentioned it in the tutorial. The results presented in the paper are all obtained in Windows machine):

Table 1 ARIs of CDC in 5D UMAP space for pbmc3k dataset with different parameter settings on Windows

ratio	0.55	0.56	0.57	0.58	0.59	0.60	0.61	0.62	0.63	0.64	0.65
k = 30	0.678	0.697	0.679	0.705	0.706	0.706	0.760	0.760	0.760	0.760	0.759
k = 40	0.726	0.728	0.738	0.738	0.738	0.738	0.791	0.792	0.792	0.778	0.579
k = 50	0.743	0.742	0.742	0.742	0.731	0.718	0.718	0.718	0.779	0.779	0.778

Table 2 Best ARIs and parameter settings on MacOS using scRNA Cluster application in CDC Toolkit

Dimensions of UMAP	Parameter settings of CDC	ARI
n_components = 2	k = 50, ratio = 0.65	0.740
n_components = 3	k = 50, ratio = 0.65	0.761
n_components = 4	k = 50, ratio = 0.64	0.778
n_components = 5	k = 40, ratio = 0.64	0.758

The best ARI is 0.792 under k = 40 and ratio = 0.62 (see Fig. 3, it is consistent with the R-Fig. 3a in the last response letter), which is higher than the comment “the best ARI I could get was 0.6528 when ratio=0.6”.

3) Reviewer comments: *“Considering the PBMC results in R-Fig. 3, the results were still kind of poor, e.g., CD4/8/NK cells were in the same cluster, CD14/16 monocytes were in the same cluster. Only when the ratio parameter =0.6, NK and CD8+ T cells are separated. However, this time, CDC produced 18 clusters while the 'ground truth' has nine cell types. Some cell types were over-clustered, e.g., some clusters had only two cells.”*

Our response:

Clustering quality evaluation: global evaluation metrics vs. number of clusters

The comment *“Considering the PBMC results in R-Fig. 3, the results were still kind of poor”* is unfair. Because the results in R-Fig. 3b-f in the last response letter are based the parameter settings specified by Reviewer #4 (instead of our default settings) in the last round of review. This figure just aims to refute the previous comment “The outputs did not make much sense, e.g., for all ratios, almost all the cells were in cluster 0” under such parameter settings rather than demonstrating the accuracy advantages of CDC, and the results in R-Fig. 3b-f do support our point.

Furthermore, in cluster analysis, **the clustering quality is indicated by evaluation metrics, rather than the consistency between the number of identified and true clusters.** In practice, evaluation metrics (e.g., ARI, ACC and NMI) are adopted to assess the clustering quality, which consider recall and precision comprehensively. The inseparability and over-clustering problems are not the defects of our algorithm, but a common phenomenon for most of clustering methods due to the heterogeneous data distribution. For algorithms that do not take the number of clusters as a parameter, they have no control over how many output clusters. In this case, it is quite common that partial cell types are inseparable or over-clustered under some parameter settings. Even in the best case for clustering accuracy, it is not guaranteed that the number of identified clusters is the same as that of true clusters. We take **Seurat** as an example, and conduct it on pbmc3k dataset with $\text{dims} = 20$ and different resolutions. As shown in Fig. 4a, the number of clusters identified by Seurat vary from 4 to 31 as resolution parameter changes. When $\text{resolution} = 0.1$, the ARI is 0.765, but CD8 T, NK and other T cells are not separated and **are mixed in cluster 2** (in Fig. 4c). While for $\text{resolution} = 5$, Seurat produces 30 clusters, where B, Mono and CD4 T cells **are over-clustered**, and some clusters only contain a few cells (in Fig. 4d). For the heterogeneous data distribution, our algorithm CDC requires more boundary points to bind internal connections, which would produce fine-granularity clusters. Thus, the over-clustered problem occurs on pbmc3k dataset as the reviewer described.

Fig. 4 Clustering performance of Seurat on pbmc3k. (a) ARI and number of the identified clusters by Seurat under different resolutions. (b) Ground truth of pbmc3k dataset. (c-d) Clustering results of Seurat with $\text{dims} = 20$, $\text{resolution} = 0.1$ and 5 respectively.

4) Reviewer comments: “As the authors demonstrated, for this simple dataset, CDC worked good only when 5D UMAP results were used. However, the authors did a granular parameter search, e.g., the ratio parameter was from 0.55 to 0.65, and k in $\{30, 40, 50\}$. To me, all these results also suggested that the CDC algorithm is sensitive to parameters. Moreover, these results demonstrated the limitations of clustering on 2D UMAPs.”

Our response:

Parameter sensitivity

We have demonstrated the robustness and stability of our algorithm through comprehensive experiments on multiple datatypes and applications. As is mentioned above, the number of identified clusters cannot indicate the clustering quality, and it sensitively vary with the parameter settings. Thus, **parameter sensitivity analysis measures the degree to which the clustering accuracy is affected by the perturbation of the parameters, rather than the number of clusters** (see the section of “Parameter sensitivity analysis” in the manuscript). Meanwhile, there is nearly no algorithm that can achieve optimal accuracy with the defaults on all datasets. For example, Seurat obtains $\text{ARI} = 0.4617$ with the defaults ($\text{dims} = 10$, $\text{resolution} = 0.8$), while its best $\text{ARI} = 0.7841$ with $\text{dims} = 35$, $\text{resolution} = 0.3$. Hence, in this paper, we specify fixed parameter spaces for different algorithms respectively, and search their best and average clustering accuracies for a fair comparison.

In the specified parameter space, our algorithm CDC has good robustness to the parameter settings and its average ARI scores outperformed that of Seurat, SNN-Walktrap-PCA, SNN-Walktrap-U2, SNN-Louvain-PCA, SNN-Louvain-U2, Kmeans-PCA, Kmeans-U2 as shown in Fig. 5 (has been presented in the previous response letter).

Fig. 5 Clustering performances on (a) two PBMC and (b) four mouse retina datasets.

5) Reviewer comments: “It may be good for the authors to make the exact scripts (also data) running all these experiments available for readers to reproduce the results.”

Our response:

Detailed experimental parameter settings for result reproduction

Actually, all parameter settings of the experiments on synthetic, CyTOF, speaker corpus datasets and UCI benchmarks have been completely provided in Supplementary Table 7 in the previous revision. Besides, we have developed two code modules to facilitate users to use our algorithm and reproduce the results, i.e., *CDC Toolkit* (provided in the last revision, see Appendix 1) and *scRNA-seq Result Reproduction* (newly added in this revision, see Appendix 2). The code, demo videos and tutorial document have been uploaded to our GitHub (see Appendix 1.3). So that, readers can reproduce our experimental results according to our code annotations, supplementary information and tutorial documents presented in the following Table 3, Figs 6 and 7 (The complete parameter settings can be seen Supplementary Notes 4 and 5, Supplementary Table 7 of the manuscript, code annotation, toolkit tutorial, as well as the Appendix 2.1-2.17 of this response letter).

Table 3 Parameter settings of CDC for 13 scRNA-seq datasets

Function	Parameter	Parameter space
umap	n_neighbors	5~40 (interval = 5)
	min_dist	0.1~1.0 (interval = 0.1)

	n components	2, 3, 4, 5
	k	30, 40, 50
CDC	ratio	BH, BM, Muraro, Segerstolpe, Xin, ALM, VISp, TM, WT_R1, WT_R2, NdpKO R1, NdpKO R1: 0.85~0.99 (interval = 0.01) AMB: 0.95~0.99 (interval = 0.005)

```

a
1 % This application support CDC clustering for synthetic data analysis
2 % We have provide the 17 2D synthetic datasets
3 % If you want to input high-dimensional data (d>2), you should use UMAP to embed data into space of low dimensions
4
5 %% Specify the parameters
6 % k_num: k of KNN
7 % ratio: percentile ratio of internal points (Default: 0.9, Recommended: 0.70~0.95)
8 % The suggested parameter settings for all dataset are as follows:
9 % D51: [30, 0.70]; D52: [8, 0.67]; D53: [11, 0.92]; D54: [17, 0.77]; D55: [10, 0.70]; D56: [30, 0.80];
10 % D57: [30, 0.80]; D58: [32, 0.80]; D59: [30, 0.95]; D510: [30, 0.95]; D514: [30, 0.90]; D515: [30, 0.90];
11 % D516: [9, 0.60]; D517: [20, 0.70];
- k_num = 4;
- ratio = 0.75;
15 %% Input the data and labels
- data = testread('SyntheticDatasets/D52.txt');
- [n, m] = size(data);
- X = data(:, 1:2);
- label = data(:, 3);
21 %% Perform CDC algorithm
22 % The suggested parameter settings for all dataset are as follows:
23 % Leving_UMAP: k_num = 60; ratio = 0.9567;
24 % Sausvik_UMAP: k_num = 35; ratio = 0.956;
25 - k_num = 60;
26 - ratio = 0.9567;
27 - cluster = CDC(X, k_num, ratio);

b
40 %% --Arguments--
41 ## seurat_dim: dimension of reduction to use as input in seurat (default: seq(20, 50, 5))
## seurat_resolution: resolution of LogRaid algorithm in seurat (default: seq(0.1, 1, 0.1))
## snnwalk_k: k of SNN graph in SNN-walktrap (default: seq(5, 30, 5))
## snnlow_k: k of SNN graph in SNN-lowcost (default: seq(5, 30, 5))
## kmeans_k: k of K-means (default: seq(2, 50))
## CDC_k: k of KNN in CDC (default: seq(30, 50, 10))
## CDC_ratio: ratio of CDC (default: seq(0.75, 0.95, 0.02))
seurat_dim = seq(20, 50, 5)
seurat_resolution = seq(0.1, 1, 0.1)
snnwalk_k = seq(5, 30, 5)
snnlow_k = seq(5, 30, 5)
kmeans_k = seq(2, 50)
CDC_k = seq(30, 50, 10)
CDC_ratio = seq(0.75, 0.95, 0.02)

d
path <- getwd()
setwd(path)
## Read UCI data
## Iris, Seeds, wine and digits are TXT files, others are in csv format.
data <- read.table('./UCI benchmarks/wine.txt', header = FALSE, sep = '\t')
data <- read.csv('./UCI benchmarks/dermatology.csv', header = FALSE)
dat_mat <- as.matrix(data[,1:(ncol(data)-1)])
dat_mat <- dat_mat[,colMeans(dat_mat)>0]
dat_label <- unlist(data[,ncol(data)])
## Run CDC algorithm
14 %% --Arguments--
15 ## embedding_method: ("UMAP": use UMAP to reduce the dimensions; "None": do not reduce the dimensions)
## nps: The dimension of the space to embed into using UMAP (Default: 2, Recommended: 2-5)
## norm: normalize the data using max-min function (TRUE: YES; FALSE: NO)
## k: k of KNN (Default: 30, recommended: 10-50)
## ratio: percentile ratio of internal points (Default: 0.9, recommended: 0.70~0.95)
source("CDC.R")
res <- CDC(dat_mat, embedding_method = "UMAP", npc = 2, norm = TRUE, k = 30, ratio = 0.7)
clus <- as.integer(res)

```

Fig. 6 Parameter settings for (a) synthetic, (b) scRNA-seq, (c) CyTOF datasets and (d) UCI benchmarks in our code annotations.

Parameter settings on 2D synthetic datasets without noise				
DS1	DS2	DS3	DS4	DS5
k = 30, ratio = 0.70	k = 8, ratio = 0.67	k = 11, ratio = 0.92	k = 17, ratio = 0.77	k = 10, ratio = 0.70
DS6	DS7	DS8	DS9	DS14
k = 30, ratio = 0.80	k = 30, ratio = 0.80	k = 52, ratio = 0.80	k = 30, ratio = 0.95	k = 30, ratio = 0.90
DS15	DS16	DS17		
k = 30, ratio = 0.90	k = 9, ratio = 0.60	k = 20, ratio = 0.70		
Parameter settings on 2D synthetic datasets with noise (*T here is noise threshold for the noise elimination methods)				
	IDM	RKNN	LOF	
DS10	k = 20, T = 0.30, ratio = 0.90	k = 20, T = 0.38, ratio = 0.80	k = 20, T = 0.06, ratio = 0.90	
DS11	k = 20, T = 0.53, ratio = 0.90	k = 30, T = 0.20, ratio = 0.90	k = 30, T = 0.09, ratio = 0.90	
DS12	k = 30, T = 0.10, ratio = 0.90	k = 30, T = 0.35, ratio = 0.90	k = 30, T = 0.05, ratio = 0.90	
DS13	k = 30, T = 0.50, ratio = 0.70	k = 30, T = 0.35, ratio = 0.80	k = 25, T = 0.10, ratio = 0.60	
Parameter settings on speaker corpuses				
	ELSDSR	MSLT		
	n_components = 2, k = 5, ratio = 0.70	n_components = 2, k = 7, ratio = 0.53		
Parameter settings on UCI benchmarks (for obtaining the highest ARI score)				
Dataset	Parameter Setting	Evaluation Metric		
Iris	n_neighbors = 25, n_components = 2, k = 8, ratio = 0.75	ARI: 0.911, ACC: 0.947, NMI: 0.875		
	n_neighbors = 30, n_components = 3, k = 6, ratio = 0.90	ARI: 0.904, ACC: 0.967, NMI: 0.885		
	n_neighbors = 25, n_components = 4, k = 11, ratio = 0.90	ARI: 0.904, ACC: 0.967, NMI: 0.885		
Seeds	n_neighbors = 30, n_components = 2, k = 10, ratio = 0.88	ARI: 0.724, ACC: 0.895, NMI: 0.719		
	n_neighbors = 15, n_components = 3, k = 10, ratio = 0.93	ARI: 0.748, ACC: 0.905, NMI: 0.753		
	n_neighbors = 30, n_components = 4, k = 6, ratio = 0.88	ARI: 0.759, ACC: 0.910, NMI: 0.757		
Breast-Cancer	n_neighbors = 40, n_components = 5, k = 7, ratio = 0.82	ARI: 0.744, ACC: 0.905, NMI: 0.710		
	n_neighbors = 25, n_components = 2, k = 8, ratio = 0.975	ARI: 0.849, ACC: 0.955, NMI: 0.780		
	n_neighbors = 50, n_components = 3, k = 5, ratio = 0.95	ARI: 0.825, ACC: 0.955, NMI: 0.720		
Wine	n_neighbors = 50, n_components = 4, k = 15, ratio = 0.95	ARI: 0.825, ACC: 0.955, NMI: 0.720		
	n_neighbors = 25, n_components = 5, k = 11, ratio = 0.95	ARI: 0.850, ACC: 0.956, NMI: 0.780		
	n_neighbors = 40, n_components = 2, k = 8, ratio = 0.88	ARI: 0.898, ACC: 0.966, NMI: 0.866		
PenDigits	n_neighbors = 45, n_components = 3, k = 15, ratio = 0.95	ARI: 0.897, ACC: 0.966, NMI: 0.876		
	n_neighbors = 15, n_components = 4, k = 17, ratio = 0.94	ARI: 0.915, ACC: 0.972, NMI: 0.893		
	n_neighbors = 15, n_components = 5, k = 12, ratio = 0.77	ARI: 0.881, ACC: 0.955, NMI: 0.858		
Dermatology	n_neighbors = 20, n_components = 2, k = 20, ratio = 0.95	ARI: 0.823, ACC: 0.822, NMI: 0.867		
	n_neighbors = 10, n_components = 3, k = 13, ratio = 0.95	ARI: 0.824, ACC: 0.822, NMI: 0.867		
	n_neighbors = 10, n_components = 4, k = 20, ratio = 0.95	ARI: 0.819, ACC: 0.817, NMI: 0.861		
Control	n_neighbors = 20, n_components = 5, k = 10, ratio = 0.95	ARI: 0.824, ACC: 0.822, NMI: 0.866		
	n_neighbors = 25, n_components = 2, k = 15, ratio = 0.90	ARI: 0.870, ACC: 0.866, NMI: 0.934		
	n_neighbors = 25, n_components = 3, k = 10, ratio = 0.95	ARI: 0.870, ACC: 0.866, NMI: 0.934		
Control	n_neighbors = 50, n_components = 4, k = 10, ratio = 0.85	ARI: 0.871, ACC: 0.874, NMI: 0.928		
	n_neighbors = 15, n_components = 5, k = 10, ratio = 0.95	ARI: 0.870, ACC: 0.866, NMI: 0.934		
	n_neighbors = 25, n_components = 2, k = 12, ratio = 0.99	ARI: 0.682, ACC: 0.667, NMI: 0.852		
Control	n_neighbors = 25, n_components = 3, k = 27, ratio = 0.75	ARI: 0.631, ACC: 0.588, NMI: 0.818		
	n_neighbors = 20, n_components = 4, k = 10, ratio = 0.85	ARI: 0.632, ACC: 0.703, NMI: 0.764		
	n_neighbors = 10, n_components = 5, k = 15, ratio = 0.78	ARI: 0.630, ACC: 0.707, NMI: 0.764		

Fig. 7 Parameter settings of the experiments on synthetic, speaker corpus and UCI benchmarks.

- **CDC Toolkit** assembles six applications contained in our manuscript (scRNA-seq cluster, UCI benchmark test, synthetic data analysis, CyTOF cluster, speaker recognition and face recognition). It integrates the functions of data preprocessing, feature extraction, dimension reduction, clustering evaluation and comparison, visualization, and necessary default clustering parameter settings together, making it much easier for users from different fields to utilize and customize it. **This toolkit supports to conduct CDC in any dimensions.** Users can either reproduce our results in this toolkit, or **explore more datasets** for their own applications by specifying the parameter details. In general, in this toolkit, users have the freedom to customize the code settings.
- **scRNA-seq Result Reproduction** exclusively provides the **2D UMAP results preprocessed by Seurat** of 15 scRNA-seq datasets used in our paper for result quick reproduction (for higher dimensional, please use **CDC Toolkit**). It only requires the dataset name and type of running mode, and dispenses with specifying any algorithm/preprocessing parameters and spending much time preprocessing raw data with Seurat. This module supports two types of mode, i.e., **‘Best’** and **‘All’**. ‘Best’ mode only runs the algorithm with the best parameters of each scRNA-seq dataset directly, so that the users can check the consistence between the

obtained results and the best results presented in the manuscript (Fig.2 and Supplementary Table 2). While, ‘All’ mode goes through the entire parameter space in Supplementary Note 4 and achieves the exactly same results in Supplementary Table 2.

How to use CDC Toolkit

Here, we take ‘*scRNA-seq Cluster*’ application as an example. Before running the toolkit, you must name and organize the files as described in Data Format of our tutorial. We have provided nine sample datasets in .zip format. Please decompress them into the corresponding data folders named by the datasets before using them.

● **Specify the data file name**

Please open the ‘*main.R*’ file in the root directory. Then, specify the name and format of scRNA-seq dataset, and determine to read label file or not.

```
## Specify the filename, format and labels.
## --Arguments--
## --Arguments--
## filename: filename of the scRNA-seq dataset (We provide nine sample datasets in this application)
## format: data format of the scRNA-seq dataset (We currently support two data formats, csv and 10X)
## labels: whether the file contains the true label of cells ('1' is Yes, '0' is No)
filename = 'Baron-Mouse'
format = 'csv'
labels = '1'
```

● **Preprocess using Seurat pipeline**

Next, read the raw data files and convert them into a standard Seurat object. After that, the Seurat object should be preprocessed with the standard pipeline, including normalization, HVG selection, scaling and PCA&UMAP embedding. *n_components*, *n_neighbors*, *min_dist* of UMAP is set as 2, 30, 0.3 by default in Seurat. You can also customize them according to the settings in Appendix for subsequent CDC clustering.

```
## Generate Seurat Object from the raw data
source('BuildSeuratObject.R')
SeuratData <- BuildSeuratObject(filename, format, labels)

## Preprocess scRNA-seq using standard Seurat pipeline
## --Arguments--
## n_components: The dimension of UMAP space to embed into (Default: 2, Recommended: 2~5)
## n_neighbors: The number of neighboring points of UMAP (Default: 30, Recommended: 5~50)
## min_dist: It controls how tightly the UMAP embedding is allowed compress points together (Default: 0.3, Recommended: 0.1~1)
source('SeuratPreprocess.R')
n_components = 2
n_neighbors = 30
min_dist = 0.3
SeuratData <- SeuratPreprocess(SeuratData, filename, n_components, n_neighbors, min_dist)
```

● **Cluster the cells**

CDC has two parameters, *k* and *ratio*. We specify them as 30 and 0.9 by default respectively. You can also adjust them for searching best results in exploratory analysis. It should be noted that default parameter settings are empirical, and cannot guarantee optimal clustering results on all datasets. In order to obtain optimal clustering result, tuning the parameter settings and traversing parameter space are needed. The recommended range of 0.85 to 0.99 is obtained empirically through our experiments on scRNA-seq datasets. User can customize it according to the concrete application scenarios.

```
## Cluster the cells using CDC algorithm
## --Arguments--
## k: k of KNN (Default: 30, Recommended: 5~50)
## ratio: percentile ratio of internal points (Default: 0.9, Recommended: 0.85~0.99, 0.55~0.65 for pbmc3k, 0.7~0.8 for SCINA)
```

```

source('CDC.R')
k = 30
ratio = 0.9
Idents(SeuratData) <- CDC(SeuratData@reductions[["umap"]])@cell.embeddings, k, ratio)

## Plot the clustering result
DimPlot(SeuratData, pt.size=1) + NoLegend()

## Evaluate the clustering accuracy using ARI metric
ARI <- mclust::adjustedRandIndex(Idents(SeuratData), SeuratData@meta.data[["Cluster"]])

```

● Compare with baselines

We also integrate four clustering baselines for comparison, Seurat, SNN-Walktrap, SNN-Louvain, K-means. Seurat was performed on the 20~50 PCs; while SNN-Walktrap, SNN-Louvain, and K-means are conducted both on 50 PCs and low-dimensional UMAPs (depends on the argument *n_components*). You can also add other clustering algorithms in '*Compare.R*' file, and customize the parameter space.

```

## Compare the clustering performance with Seurat, SNN-Walktrap, SNN-Louvain, K-means
## Noted: Comparison is usually time consuming, if you just want to run CDC algorithm and don't
## want to compare with other algorithm by traversing different settings in the parameter
## spaces, please just comments out this part of the code
## --Arguments--
## seurat_dim: dimension of reduction to use as input in Seurat (Default: seq(20, 50, 5))
## seurat_resolution: resolution of Louvain algorithm in Seurat (Default: seq(0.1, 1, 0.1))
## snnwalk_k: k of SNN graph in SNN-Walktrap (Default: seq(5, 30, 5))
## snnlouv_k: k of SNN graph in SNN-Louvain (Default: seq(5, 30, 5))
## kmeans_k: k of K-means (Default: seq(2, 50))
## CDC_k: k of KNN in CDC (Default: seq(30, 50, 10))
## CDC_ratio: ratio of CDC (Default: seq(0.75, 0.95, 0.02))

seurat_dim = seq(20, 50, 5)
seurat_resolution = seq(0.1, 1, 0.1)
snnwalk_k = seq(5, 30, 5)
snnlouv_k = seq(5, 30, 5)
kmeans_k = seq(2, 50)
CDC_k = seq(30, 50, 10)
CDC_ratio = seq(0.85, 0.99, 0.02)

source('Compare.R')
box_res <- Compare(SeuratData, filename, seurat_dim, seurat_resolution, snnwalk_k, snnlouv_k, kmeans_k, CDC_k, CDC_ratio)
boxplot(box_res, ylab='ARI', col=c('#FF69B4', '#4682B4', '#3CB371', '#FF7F50', '#008B8B', '#FFD700', '#FF0000', '#1E90FF'))

```

How to use scRNA-seq Result Reproduction

Open the '*main.R*' file in the root directory of *scRNA Result Reproduction*, and then specify the data name and running mode and run it. The running results of all 15 scRNA-seq datasets are presented in the Appendix of this letter.

```

source('RunCDC.R')
## ---Specify the name of scRNA-seq dataset---
## We provide preprocessed UMAP results of 15 scRNA-seq datasets used in our manuscript:
## 'Baron-Human', 'Baron-Mouse', 'Muraro', 'Segerstolpe', 'Xin', 'AMB', 'ALM', 'VISp', 'TM'
## 'pbmc3k', 'SCINA', 'WT_R1', 'WT_R2', 'NdpKO_R1', 'NdpKO_R2'

## ---Specify the mode to reproduce our results---
## We provide two modes to reproduce our results: 'Best' and 'All'
## 'Best': This mode only runs the algorithm with the best parameters of each scRNA-seq dataset directly.
## 'All': This mode goes through the entire parameter space and can obtain the same results of our paper.

RunCDC('Baron-Human', 'All')

```

6) Reviewer comments: “Although now CDC can also use high-dimensional data as inputs, however, there are still several problems, e.g., CDC is computational much more expensive for high-dimensional data (I tried CDC on 5D data and found it was much slower compared to running using 2D data as inputs). Moreover, the estimation of the knn directions in high-dimensional space may be bad because of the curse of dimensionalities.”

Our response:

Time efficiency for high dimensionality

The computational complexity of DCM calculation increases as dimension grows, but we have pointed out that **the runtime of CDC in 3D-8D space is acceptable comparing with the overall preprocessing runtime** in the previous response. In fact, the steps of preprocessing are the most time-consuming part. It took about seven minutes for Seurat to preprocess a scRNA-seq dataset with 50k cells. Consequently, when considering the runtime of preprocessing, the gap of computational efficiency between Seurat and CDC (R) has narrowed significantly, see Fig. 8.

Fig. 8 Computational efficiencies of four methods on sub-datasets with different number of cells sampled from scRNA-seq dataset *TM*. (a) The workflows of four cluster analysis methods on scRNA-seq dataset. (b, c) Runtimes of four methods (not including and including the preprocessing respectively) on scRNA-seq datasets with different number of cells.

Meanwhile, we are currently developing a parallel version of CDC in Java based on High-Performance Computing (HPC) framework Apache Spark. For the dataset with 1 million samples, the time efficiency has been improved **157-fold** than the stand-alone CDC algorithm in Fig. 8. The runtime of parallel CDC was measured just in a pseudo-distributed computing environment (i.e., single-machine), its performance improvement would be much greater in a true distributed computing environment (i.e., multi-machine). CDC in high-dimensional space can be executed as fast as in 2D space. We will timely update the latest improvements of the HPC-version CDC (in Fig. 9).

Fig. 9 Runtimes of serial and parallel CDC algorithms.

pdh-coder Update main1.m 188e755 on 31 Mar 119 commits

- HPC-version** Add files via upload last month
- Toolkit Update main1.m last month
- pics Add files via upload last month
- Demo.mp4 First Commit 2 months ago
- README.md Update README.md last month
- Tutorial.pdf Add files via upload 2 months ago

ClusteringDirectionCentrality / HPC-version / Spark-based-CDC-Algorithms /

summer-chi Add files via upload 9b4c3c5 on 22 Mar History

- CyTOFDataSets Add files via upload last month
- GeoDataSets Add files via upload last month
- SyntheticDataSets Add files via upload last month
- src/main/java/com/yolo/CDC Add files via upload last month
- pom.xml Add files via upload last month

Fig. 10 Code of parallel CDC on our GitHub.

7) Reviewer comments: *“As for clustering on tSNE/UMAP or clustering on high-dimensional space, Supplementary Fig. 11 of the paper “the specious art of single-cell genomics” shows that many digits were assigned to the wrong clusters detected from t-SNE/UMAP. For large datasets, the crowding problem is almost inevitable for t-SNE, as demonstrated by a recent paper: <https://arxiv.org/abs/2110.02573>*

I understand the authors have put lots of time on revising the manuscript. I'm sorry I still not convinced that tSNE/UMAP + CDC is good for clustering high-dimensional data, especially scRNA-seq data, although in some cases, this approach may help pick some clusters.

My comments may be biased based on my experience, so it could be good to have other reviewers with extensive experience on t-SNE/UMAP.”

Our response:

Dimension reduction

As we discussed in the previous response, **due to the curse of dimension, dimension reduction is an essential preprocessing step before clustering for many classifiers (see Fig. 11), especially biological clustering baselines.** Monocle3 utilizes UMAP for dimension reduction; ACCENSE, clusterX, DensVM require to perform t-SNE before clustering. Reviewer #4 has admitted that *“My comments may be biased based on my experience”* against this type of methods, and which makes the reviewer didn’t comprehensively evaluate our work through our experimental results in a fair manner. Actually, we did not observe distinct “crowding” problem from the embedded data distributions of all scRNA-seq datasets in our experiments, and CDC obtained the highest max and average ARI on most of scRNA-seq datasets, which outperforms the algorithms conducted in high-dimensional PCA space. As a linear mapping technique, PCA cannot achieve complete divisibility of cells even in 50D space sometimes, especially for high-dimensional data that lies on non-linear manifolds (Maaten et al., 2008). While, UMAP can achieve better low-dimension representation for manifold data, and keep the divisibility between clusters, since it adopts spectral clustering to preprocess the data and preserves the global structure (McInnes et al., 2018). **Moreover, UMAP only cannot guarantee high clustering accuracy, a proper clustering method is quite vital.** In the same UMAP dimension, CDC outperforms the baselines (e.g., SNN-Walktrap, SNN-Louvain, K-means, C-means and other versatile methods). In general, **UMAP facilitates the local direction centrality of CDC to be applicable in a proper dimension,** and in turn separates different cell types mixed up with weak connectivity using reasonable parameter settings.

Fig. 11 The classifier performance in different data dimensions. As the dimension increases, the classifier’s performance increases until the optimal number of features is reached. Further increasing the dimension without increasing the number of training samples results in a decrease in classifier performance (<https://www.visiondummy.com/2014/04/curse-dimensionality-affect-classification/>).

In the previous round of review, Reviewer #4 suggested us testifying CDC-UMAP on different CD4 T cells, since a “crowding” problem may occur in low-dimensional UMAP space and the cell types might be difficult to separate. We have investigated two PBMC datasets (pbmc3k and SCINA) accordingly, on which CDC achieved higher and more stable accuracy than biological baselines (see Fig. 5a, b). Furthermore, the adaptability and accuracy advantage of our algorithm have been validated through comprehensive experiments on various cell types, hence **we insist that integrating dimension reduction technique with our CDC algorithm is promising for cluster analysis.**

References:

- [1] van der Maaten, L. & Hinton, G. Visualizing data using t-SNE. *J. Mach. Learn. Res.* **9**, 2579-2605 (2008).
- [2] McInnes, L., Healy, J., Saul, N. & Grobberger, L. UMAP: uniform manifold approximation and projection. *J. Open Source Softw.* **3**, 861 (2018).

Appendix

Appendix 1. CDC Toolkit

1.1. The framework of CDC Toolkit

We have developed a toolkit that assembles six applications contained in our manuscript (i.e., scRNA-seq cluster, UCI benchmark test, synthetic data analysis, CyTOF cluster, speaker recognition and face recognition) by learning from Seurat, and have uploaded it to our GitHub. It integrates the functions of data preprocessing, feature extraction, dimension reduction, clustering evaluation and comparison, visualization, and necessary default clustering parameter settings together, making it much easier for users from different fields to utilize and customize it. The six applications are implemented using R, MATLAB and Python languages, and each application is put in a separate folder. The framework is shown as follow

1.2. Six applications assembled in CDC Toolkit

1.3. Screenshots of our GitHub, tutorial and demo video for CDC Toolkit

Then, we open the main.R file in the RStudio.

1.4. How to reproduce scRNA-seq result using CDC Toolkit

If you want to reproduce the scRNA-seq results using *CDC Toolkit*, you can enter to the “*scRNA-seq Cluster*” application and open the “*main.R*” file. As shown in the following screenshot (using the dataset *Baron-Mouse* as an example), you can specify the dataset name, UMAP and CDC parameters in ①, ②, ③ respectively. The detailed parameter settings of each dataset have been listed in Appendix 2.

```
## Specify the filename, format and labels.
## --Arguments--
## filename: filename of the scRNA-seq dataset (We provide nine sample datasets in this application,
## format: data format of the scRNA-seq dataset (We currently support two data formats, csv and 10X)
## labels: whether the file contains the true label of cells ('1' is yes, '0' is no)
filename = "Baron-Mouse" ①
format = "csv"
labels = "1"
## Generate Seurat object from the raw data
source('BuildSeuratObject.R')
SeuratData <- BuildSeuratObject(filename, format, labels)
## Preprocess scRNA-seq using standard Seurat pipeline
## --Arguments--
## n_components: The dimension of UMAP space to embed into (Default: 2, Recommended: 2-5)
## n_neighbors: The number of neighboring points of UMAP (Default: 30, Recommended: 5-50)
## min_dist: It controls how tightly the UMAP embedding is allowed compress points together (Default:
source('SeuratPreprocess.R')
n_components = 2 ②
n_neighbors = 30
min_dist = 0.3
SeuratData <- SeuratPreprocess(SeuratData, filename, n_components, n_neighbors, min_dist)
## Cluster the cells using CDC algorithm
## --Arguments--
## k: k of KNN (Default: 30, Recommended: 5-50)
## ratio: percentile ratio of internal points (Default: 0.9, Recommended: 0.85-0.99, 0.55-0.65 for pb
source('CDC.R')
k = 50
ratio = 0.95 ③
Idents(SeuratData) <- CDC(SeuratData@reductions[["umap"]])@cell.embeddings, k, ratio)
```

Environment	History	Connections	Tutorial
Global Environment			
Data			
box_res	List of 8		
dat_mat	Large matrix (109730 elements, 878.4 kB)		
SeuratData	Large Seurat (307.1 MB)		
umap_mat	54865 obs. of 4 variables		
values			
ARI	0.968272980553528		
CDC_k	num [1:3] 30 40 50		
CDC_ratio	num [1:8] 0.85 0.87 0.89 0.91 0.93 0.95 0...		
dataname	"VISp"		
filename	"Baron-Mouse"		

R Graphics: Device 2 (ACTIVE)

文件 历史 重设大小

UMAP_2

UMAP_1

If you only want conduct CDC analysis, you can comment out the following code in the “*main.R*”. While, if you would like to perform a quick comparison with other baselines, the following code should be executed.

```
## Compare the clustering performance with Seurat, SNN-Walktrap, SNN-Louvain, K-means
## Noted: Comparison is usually time consuming, if you just want to run CDC algorithm and don't
## want to compare with other algorithm by traversing different settings in the parameter
## spaces, please just comments out this part of the code
## --Arguments--
## seurat_dim: dimension of reduction to use as input in Seurat (Default: seq(20, 50, 5))
## seurat_resolution: resolution of Louvain algorithm in Seurat (Default: seq(0.1, 1, 0.1))
## snnwalk_k: k of SNN graph in SNN-Walktrap (Default: seq(5, 30, 5))
## snnlouv_k: k of SNN graph in SNN-Louvain (Default: seq(5, 30, 5))
## kmeans_k: k of K-means (Default: seq(2, 50))
## CDC_k: k of KNN in CDC (Default: seq(30, 50, 10))
## CDC_ratio: ratio of CDC (Default: seq(0.75, 0.95, 0.02))
seurat_dim = seq(20, 50, 5)
seurat_resolution = seq(0.1, 1, 0.1)
snnwalk_k = seq(5, 30, 5)
snnlouv_k = seq(5, 30, 5)
kmeans_k = seq(2, 50)
CDC_k = seq(30, 50, 10)
CDC_ratio = seq(0.85, 0.99, 0.02)

source('Compare.R')
```

```
box_res <- Compare(SeuratData, filename, seurat_dim, seurat_resolution, snnwalk_k, snnlouv_k, kmeans_k, CDC_k, CDC_ratio)
boxplot(box_res, ylab='ARI', col=c('#FF69B4', '#4682B4', '#3CB371', '#FF7F50', '#008B8B', '#FFD700', '#FF0000', '#1E90FF'))
```

Then, the comparison results with the integrated four baselines will be presented in a box plot as follow:

To be noted, the results obtained under different Operating Systems (such as Windows and MacOS) may have slight difference due to the implementation of the dependency libraries, and we have mentioned it in our tutorial. The results presented in the paper are all obtained in Windows machine.

Appendix 2. scRNA-seq Result Reproduction and CyTOF Result Reproduction

2.1. Baron-Human

Parameter space: $n_components = 2$, $k = 30\sim 50$ (interval = 10), ratio = 0.85~0.99 (interval = 0.01)

The number of times CDC ran: 45

Overall runtime of CDC: 437.140 seconds

Average runtime of CDC: 9.714 seconds

Average ARI: 0.9183

Max ARI: 0.9532

The optimal parameter settings: $k = 30$, ratio = 0.88 (UMAP: $n_components = 2$, $n_neighbors = 40$, $min_dist = 0.7$)

2.2. Baron-Mouse

Parameter space: $n_components = 2$, $k = 30\sim 50$ (interval = 10), $ratio = 0.85\sim 0.99$ (interval = 0.01)

The number of times CDC ran: 45

Overall runtime of CDC: 47.720 seconds

Average runtime of CDC: 1.060 seconds

Average ARI: 0.9633

Max ARI: 0.9683

The optimal parameter settings: $k = 50$, $ratio = 0.95$ (UMAP: $n_components = 2$, $n_neighbors = 30$, $min_dist = 0.3$)

2.3. Muraro

Parameter space: $n_components = 2$, $k = 30\sim 50$ (interval = 10), $ratio = 0.85\sim 0.99$ (interval = 0.01)

The number of times CDC ran: 45

Overall runtime of CDC: 57.020 seconds

Average runtime of CDC: 1.267 seconds

Average ARI: 0.8407

Max ARI: 0.9235

The optimal parameter settings: $k = 30$, $ratio = 0.94$ (UMAP: $n_components = 2$, $n_neighbors = 30$, $min_dist = 0.7$)

2.4. Segerstolpe

Parameter space: $n_components = 2$, $k = 30\sim 50$ (interval = 10), $ratio = 0.85\sim 0.99$ (interval = 0.01)

The number of times CDC ran: 45

Overall runtime of CDC: 57.860 seconds

Average runtime of CDC: 1.286 seconds

Average ARI: 0.9634

Max ARI: 0.9734

The optimal parameter settings: $k = 40$, $ratio = 0.90$ (UMAP: $n_components = 2$, $n_neighbors = 30$, $min_dist = 0.3$)

2.5. Xin

Parameter space: $n_components = 2$, $k = 30\sim 50$ (interval = 10), ratio = 0.85~0.99 (interval = 0.01)

The number of times CDC ran: 45

Overall runtime of CDC: 31.460 seconds

Average runtime of CDC: 0.699 seconds

Average ARI: 0.9208

Max ARI: 0.9727

The optimal parameter settings: $k = 30$, ratio = 0.97 (UMAP: $n_components = 2$, $n_neighbors = 10$, $min_dist = 0.5$)

2.6. AMB

Parameter space: $n_components = 2$, $k = 30\sim 50$ (interval = 10), ratio = 0.95~0.99 (interval = 0.005)

The number of times CDC ran: 27

Overall runtime of CDC: 375.640 seconds

Average runtime of CDC: 13.913 seconds

Average ARI: 0.6081

Max ARI: 0.8873

The optimal parameter settings: $k = 40$, ratio = 0.99 (UMAP: $n_components = 2$, $n_neighbors = 40$, $min_dist = 1.0$)

2.7. ALM

Parameter space: $n_components = 2$, $k = 30\sim 50$ (interval = 10), $ratio = 0.85\sim 0.99$ (interval = 0.01)

The number of times CDC ran: 45

Overall runtime of CDC: 570.790 seconds

Average runtime of CDC: 12.684 seconds

Average ARI: 0.6250

Max ARI: 0.7186

The optimal parameter settings: $k = 40$, $ratio = 0.96$ (UMAP: $n_components = 2$, $n_neighbors = 25$, $min_dist = 0.8$)

2.8. VISp

Parameter space: $n_components = 2$, $k = 30\sim 50$ (interval = 10), $ratio = 0.85\sim 0.99$ (interval = 0.01)

The number of times CDC ran: 45

Overall runtime of CDC: 1067.170 seconds

Average runtime of CDC: 23.715 seconds

Average ARI: 0.4852

Max ARI: 0.8569

The optimal parameter settings: $k = 50$, $ratio = 0.93$ (UMAP: $n_components = 2$, $n_neighbors = 20$, $min_dist = 1.0$)

2.9. TM

Parameter space: $n_components = 2$, $k = 30\sim 50$ (interval = 10), $ratio = 0.85\sim 0.99$ (interval = 0.01)

The number of times CDC ran: 45

Overall runtime of CDC: 13870.450 seconds

Average runtime of CDC: 308.232 seconds

Average ARI: 0.8046

Max ARI: 0.8384

The optimal parameter settings: $k = 30$, $ratio = 0.97$ (UMAP: $n_components = 2$, $n_neighbors = 5$, $min_dist = 0.1$)

2.10. WT_R1

Parameter space: $n_components = 2$, $k = 30\sim 50$ (interval = 10), $ratio = 0.85\sim 0.99$ (interval = 0.01)

The number of times CDC ran: 45

Overall runtime of CDC: 350.140 seconds

Average runtime of CDC: 7.781 seconds

Average ARI: 0.8971

Max ARI: 0.9178

The optimal parameter settings: $k = 50$, $ratio = 0.96$ (UMAP: $n_components = 2$, $n_neighbors = 30$, $min_dist = 0.3$)

2.11. WT_R2

Parameter space: $n_components = 2$, $k = 30\sim 50$ (interval = 10), $ratio = 0.85\sim 0.99$ (interval = 0.01)

The number of times CDC ran: 45

Overall runtime of CDC: 337.980 seconds

Average runtime of CDC: 7.511 seconds

Average ARI: 0.8989

Max ARI: 0.9153

The optimal parameter settings: $k = 40$, $ratio = 0.98$ (UMAP: $n_components = 2$, $n_neighbors = 30$, $min_dist = 0.3$)

2.12. NdpKO_R1

Parameter space: $n_components = 2$, $k = 30\sim 50$ (interval = 10), $ratio = 0.85\sim 0.99$ (interval = 0.01)

The number of times CDC ran: 45

Overall runtime of CDC: 534.400 seconds

Average runtime of CDC: 11.876 seconds

Average ARI: 0.8393

Max ARI: 0.8678

The optimal parameter settings: $k = 30$, $ratio = 0.85$ (UMAP: $n_components = 2$, $n_neighbors = 30$, $min_dist = 0.3$)

2.13. NdpKO_R2

Parameter space: $n_components = 2$, $k = 30\sim 50$ (interval = 10), $ratio = 0.85\sim 0.99$ (interval = 0.01)

The number of times CDC ran: 45

Overall runtime of CDC: 347.080 seconds

Average runtime of CDC: 7.713 seconds

Average ARI: 0.8879

Max ARI: 0.9310

The optimal parameter settings: $k = 50$, $ratio = 0.98$ (UMAP: $n_components = 2$, $n_neighbors = 30$, $min_dist = 0.3$)

2.14. pbmc3k

Parameter space: $n_components = 2$, $k = 30\sim 50$ (interval = 10), $ratio = 0.55\sim 0.65$ (interval = 0.01)

The number of times CDC ran: 33

Overall runtime of CDC: 88.300 seconds

Average runtime of CDC: 2.676 seconds

Average ARI: 0.7005

Max ARI: 0.7563

The optimal parameter settings: $k = 40$, $ratio = 0.62$ (UMAP: $n_components = 2$, $n_neighbors = 30$, $min_dist = 0.3$)

2.15. SCINA

Parameter space: $n_components = 2$, $k = 30\sim 50$ (interval = 10), $ratio = 0.70\sim 0.80$ (interval = 0.01)

The number of times CDC ran: 33

Overall runtime of CDC: 1144.720 seconds

Average runtime of CDC: 34.688 seconds

Average ARI: 0.8346

Max ARI: 0.8679

The optimal parameter settings: $k = 50$, $ratio = 0.80$ (UMAP: $n_components = 2$, $n_neighbors = 30$, $min_dist = 0.3$)

2.16. MIHPF

To reproduce the first-round clustering on MIHPF with more than 1 million cells, we have provided an extra sub-folder named “*MIHPF*” in the application of “*scRNA-seq Cluster*” in the *CDC Toolkit*. It is implemented in MATLAB. Since the subsequent rounds of CDC clustering involved manual cluster selection (described in Line 246-249, Page 13 in the manuscript), we do not provide the detailed code and parameter settings.

Parameter space: $k = 30$, ratio = 0.99 (UMAP: $n_components = 2$, $n_neighbors = 50$, $min_dist = 0.5$)
ARI: 0.3419

2.17. Two CyTOF datasets (Levine and Samusik)

To reproduce the results of two CyTOF datasets, we have provided a “*CyTOF Cluster*” application in the *CDC Toolkit* previously. The code including the processes of clustering, evaluation and visualization. You can open the “*main.m*” file and set the parameters according to the code annotations.

Levine

Parameter space: $k = 60$, ratio = 0.9667 (UMAP: $n_components = 2$, $n_neighbors = 50$, $min_dist = 0.1$)

ARI: 0.9618

Samusik

Parameter space: $k = 27$, ratio = 0.9750 (UMAP: $n_components = 2$, $n_neighbors = 30$, $min_dist = 0.1$)

ARI: 0.8564

The MATLAB R2020b interface displays a script for clustering the Samusik dataset. The script is organized into three main sections: input, CDC algorithm execution, and evaluation/visualization.

```
1 %% Input the data and labels
data = textread('Samusik_UMAP.txt');
[n, m] = size(data);
X = data(:,1:2);
label = data(:,3);
nan_id = isnan(label);
label(nan_id) = [];
X(nan_id,:) = [];
10 %% Perform CDC algorithm
11 % Recommended Arguments
12 % Levine_UMAP: k_num = 60; Idcm = 0.08;
13 % Samusik_UMAP: k_num = 35; Idcm = 0.033;
k_num = 27;
ratio = 0.975;
cluster = CDC(X, k_num, ratio);
18 %% Evaluates the clustering accuracy and visualization
addpath ClusterEvaluation;
[accuracy, NMI, ARI, Fscore, JI, RI] = ClustEval(label, cluster);
plotcluster(X(1:10:length(X),:), cluster(1:10:length(X),:));
```

The workspace window shows the following variables and their values:

Name	Value
X	514386x2 double
RI	0.9639
res	81x3 double
ratio	0.9750
nan_id	84164x1 logical
n	841644
m	3
label	514386x1 double
k_num	27
JI	0.7817
Fscore	0.8775
data	841644x3 double
cluster	514386x1 double
ARI	0.8564

Figure 1 is a scatter plot showing the first 10 clusters of data points. The x and y axes both range from -15 to 15. The clusters are represented by different colors and are distributed across the plot area.

Reviewers' Comments:

Reviewer #2:

Remarks to the Author:

As asked by the handling editor (and since I had already reviewed the paper in a previous round and suggested acceptance), I will not provide a classical review of the paper but instead related to the ongoing discussion between the authors and Reviewer 4. Before commenting on the three summary replies on the first page of the revision summary, I will briefly explain my general take on the dispute:

- I tend to agree with almost all points raised by Reviewer 4 but also believe that they apply to virtually any method that is currently used for clustering scRNA-seq data (in particular: lacking robustness w.r.t. hyper-parameters and over-reliance on t-SNE or UMAP for dimensionality reduction). So the reviewer raises concerns which are warranted (in my opinion) but which really criticise an entire field.
- To almost all points raised by the reviewer, the authors essentially reply that the raised concerns also apply to the state of the art, which they claim to outperform w.r.t. several metrics. I think that these replies are largely correct but also believe that they do not resolve the raised concerns. The fact that everybody has the same problem does not justify the conclusion that the problem doesn't exist.
- Given these considerations, I do not believe that CDC constitutes a major breakthrough for clustering scRNA-seq data. It seems to perform better than the state of the art w.r.t. some evaluation metrics on some datasets, but has the same fundamental problems as existing approaches.
- That said, I still believe that CDC deserves to be published. The algorithm is extremely elegant, simple, and innovative. It doesn't happen often to come across such beautifully simple ideas. On top of this, the extensive empirical evaluation does show that CDC is competitive with state-of-the-art methods and even outperforms them in various settings. For the scRNA-seq application domain, these improvements might be insignificant in comparison to the larger challenges faced by the field. But for other application domains, they might still be significant.

First summary reply

Quote from revision summary

The default preprocessing settings in our code were modified by Reviewer #4, and incurred reproductivity issue. In the last round of review, Reviewer #4 suggested us providing default and detailed parameter settings in our code for audiences to reproduce the experimental results, and we have improved our code accordingly (as mentioned in previous response letter). However, the reviewer modified the default parameter settings when running our toolkit (`*dims*` in `*RunUMAP*` function is set as 1:50 in our code, but the reviewer modified it to 1:10, so that the obtained best accuracy is 0.6528 which is lower than our result 0.7920), thus obtained worse results than that we present in the response letter. It is a serious mistake and the conclusions drawn based on the wrong use of our code make no sense and such a clustering result could be also predictable, since no algorithm can guarantee promising results under all possible parameter settings, especially for exploratory analysis methods such as clustering algorithm that need to specify a rational parameter range.

My take the issue

I tend to agree with Reviewer 4. At least for clustering scRNA-seq data, sensitivity of result quality to parameter changes is problematic. If CDC was indeed successful at becoming a standard tool for clustering scRNA-seq data, most clinical scientists probably wouldn't bother to tune the parameters but instead use the defaults. But if slightly changing the defaults dramatically affects the quality, then there is little reason to believe that the defaults will be reasonable on all data.

Second summary reply

Quote from revision summary

Important concepts in cluster analysis are misused. Commonly, clustering quality is evaluated by accuracy metrics (such as ARI, ACC and NMI), rather than the number of identified clusters (mentioned by the reviewer). Different parameter settings do produce different results, and a comprehensive evaluation should consider both recall and precision of all clusters. Therefore, the effectiveness and robustness of the algorithm cannot be denied just because the number of identified clusters differs from that of the true clusters (i.e., inseparable and over-clustered). Actually, the representative scRNA-seq clustering baselines, e.g., Seurat, SC3, monocle3 and SNN-based methods, also cannot guarantee a fixed number of output clusters as the parameters vary.

My take on the issue

Again, I agree with the reviewer. Clinicians won't bother that typically ARI, ACC and NMI are used to evaluate cluster quality but might still lose the trust in the method if small parameter changes lead to very different numbers of clusters. It is of course true that also state-of-the-art approaches suffer from this problem. But this does not mean that the problem doesn't exist but simply that it is currently unsolved.

Third summary reply

Quote from revision summary

Biased comments. Reviewer #4's negative comments on the approach of dimension reduction using tSNE/UMAP is empirical and subjective, which didn't reflect the facts of our experimental results. Through solid comparative experiments on different datatypes and multiple applications, we have demonstrated the validity of our algorithm. It is able to obtain the highest accuracies on most experimental datasets, commonly in the recommended range of parameters, and achieve the most stable outputs with the default settings. Since CDC can handle the potential weak connectivity in the raw data and embedded data using dimension reduction method, we did not meet the "crowding problem" mentioned by Reviewer #4. We insist that the experimental results are authentic, reliable and reproducible.

My take on the issue

This reply is actually a bit offensive. It is based on Reviewer 4's sentence "My comments may be biased based on my experience [...]", which, however, is simply a polite phrase to make the critique sound less harsh. The critical view on using t-SNE or UMAP prior to clustering scRNA-seq data is rooted in two extremely convincing recently released preprints (Chari et al, <https://doi.org/10.1101/2021.08.25.457696>; Yang et al, <https://doi.org/10.48550/arXiv.2110.02573>), which are in fact referenced by the reviewer. The authors do not show that the concerns raised in these papers do not apply to their approach. For instance, for the preprint by Chari et al, this could have been done by computing an elephant-shaped embedding with PICASSO and then showing that UMAP+CDC produces better clusterings than PICASSO+CDC.

REVIEWERS' COMMENTS

Reviewer #2 (Remarks to the Author):

As asked by the handling editor (and since I had already reviewed the paper in a previous round and suggested acceptance), I will not provide a classical review of the paper but instead related to the ongoing discussion between the authors and Reviewer 4. Before commenting on the three summary replies on the first page of the revision summary, I will briefly explain my general take on the dispute:

- I tend to agree with almost all points raised by Reviewer 4 but also believe that they apply to virtually any method that is currently used for clustering scRNA-seq data (in particular: lacking robustness w.r.t. hyper-parameters and over-reliance on t-SNE or UMAP for dimensionality reduction). So the reviewer raises concerns which are warranted (in my opinion) but which really criticise an entire field.

- To almost all points raised by the reviewer, the authors essentially reply that the raised concerns also apply to the state of the art, which they claim to outperform w.r.t. several metrics. I think that these replies are largely correct but also believe that they do not resolve the raised concerns. The fact that everybody has the same problem does not justify the conclusion that the problem doesn't exist.

- Given these considerations, I do not believe that CDC constitutes a major breakthrough for clustering scRNA-seq data. It seems to perform better than the state-of-the-art w.r.t. some evaluation metrics on some datasets, but has the same fundamental problems as existing approaches.

- That said, I still believe that CDC deserves to be published. The algorithm is extremely elegant, simple, and innovative. It doesn't happen often to come across such beautifully simple ideas. On top of this, the extensive empirical evaluation does show that CDC is competitive with state-of-the-art methods and even outperforms them in various settings. For the scRNA-seq application domain, these improvements might be insignificant in comparison to the larger challenges faced by the field. But for other application domains, they might still be significant.

Thanks for your positive comments on our algorithm CDC, and your kindly helps and evaluable suggestions during the whole review process are sincerely appreciated.

First summary reply

Quote from revision summary

The default preprocessing settings in our code were modified by Reviewer #4, and incurred reproducibility issue. In the last round of review, Reviewer #4 suggested us providing default and detailed parameter settings in our code for audiences to reproduce the experimental results, and we have improved our code accordingly (as mentioned in previous response letter). However, the reviewer modified the default parameter settings when running our toolkit (*dims* in *RunUMAP*

function is set as 1:50 in our code, but the reviewer modified it to 1:10, so that the obtained best accuracy is 0.6528 which is lower than our result (0.7920), thus obtained worse results than that we present in the response letter. It is a serious mistake and the conclusions drawn based on the wrong use of our code make no sense and such a clustering result could be also predictable, since no algorithm can guarantee promising results under all possible parameter settings, especially for exploratory analysis

methods such as clustering algorithm that need to specify a rational parameter range.

My take the issue

I tend to agree with Reviewer 4. At least for clustering scRNA-seq data, sensitivity of result quality to parameter changes is problematic. If CDC was indeed successful at becoming a standard tool for clustering scRNA-seq data, most clinical scientists probably wouldn't bother to tune the parameters but instead use the defaults. But if slightly changing the defaults dramatically affects the quality, then there is little reason to believe that the defaults will be reasonable on all data.

Thank you. **We definitely agree that the parameter sensitivity matters for the users, and strategies to determine appropriate parameters adaptively are necessary.** In this response letter, **we have conducted further experiments to explore the relations between the changing trends of internal and external clustering evaluation indexes** on three representative scRNA-seq datasets under different parameter settings, and **discussed the potential by adopting internal evaluation indexes as the prior knowledge to guide the developed scRNA-seq cluster toolkit to set the parameters more adaptively.** We have also added a brief discussion about the parameter setting in the manuscript accordingly. Details are presented below:

In this paper, we analyzed the parameter sensitivity of the two CDC parameters (i.e., k and T_{DCM}), and came to the conclusion that k is an insensitive parameter and T_{DCM} is slightly sensitive. We propose an empirical method to determine k and a graph-based estimation method for T_{DCM} in 2D space. The effectiveness of the adaptive methods has been validated on synthetic and UCI datasets (Supplementary Fig. 12).

For the high-dimensional data, the empirical method can be inherited to determine k . While, the graph-based estimation method of T_{DCM} does not work in high dimensions. **We summarized a default range for *ratio*** (the conversion between *ratio* and T_{DCM} can be seen in Methods), i.e., **0.70-0.99**, according to our experiments. For most of datasets we used in this paper, CDC can achieve promising clustering accuracy and even obtain the highest accuracy in the default range of *ratio* (Fig. 2 and 5 in the manuscript).

Although the default range is applicable to most datasets, specifying *ratio* precisely relies on the data distribution. Commonly, clustering evaluation can divide into two categories, internal and external evaluation, where internal evaluation measures the cluster accuracy without true labels, by measuring the intra-cluster compactness and separation between clusters. Therefore, **the internal evaluation indexes can be used to guide the parameter settings when we do not have enough prior knowledge.**

Conventional internal evaluation indexes include Compactness, Separation, Dunn Validity Index (DVI), Davies-Bouldin Index (DBI), Silhouette Coefficient (SC), etc. Here, **we investigate DBI and SC on three scRNA-seq datasets, i.e., *pbmc3k*, *Xin* and *AMB*. The three datasets have different distributions in embedded UMAP space (in R-Fig. 1a). Clusters in *Xin* are distributed discretely. 1%-30% boundary points as the default settings (*ratio*: 0.70-0.99) can restrain the connections of internal points. While, the optimal *ratio* (0.55-0.65) of *pbmc3k* is lower than the defaults, since **multiple clusters are mixed up**, thereby requiring more boundary points to separate each other. For *AMB* dataset, cells in the same type are aggregated to multiple disconnected parts, thus high *ratio* (less boundary points) is required for high accuracy. Comparisons between the DBI, SC and ARI curves under different *ratio* are presented in R-Fig. 1b-c. As an external evaluation index, ARI can indicate the clustering accuracy by measuring the consistency between the predicted and true labels. In general, **the overall trends of DBI curves are converse with ARI curves, while SC has the similar tendency with ARI. Although the fluctuations are different in some details, we believe that they can be leveraged to set *ratio* of CDC more adaptively.****

R-Fig. 1. Internal evaluation for CDC on three scRNA-seq datasets. (a) 2D embedding results of *pbmc3k*, *Xin* and *AMB* datasets. (b) DBI and ARI curves by varying *ratio*. (c) SC and ARI curves by varying *ratio*.

Accordingly, we have added more discussion about the setting of *ratio* in CDC in Line 530-534, Page 27:

The parameter *ratio* has intuitive physical meaning and better stability (see Supplementary Table 6), which makes it easier to specify than T_{DCM} . According to our experiments, 70%~99% internal

points are the suggested default parameter range of *ratio* for promising clustering results. Nevertheless, when clusters are mixed up or connect to each other, more boundary points (lower *ratio*) are necessary to separate the close clusters.

Internal Evaluation Indexes

Davies-Bouldin Index (DBI) is defined as the ratio of the within cluster scatter to the between cluster separation:

$$DBI = \frac{1}{n} \sum_{i=1}^n \max_{j \neq i} \frac{\overline{CP}_i + \overline{CP}_j}{\|\omega_i - \omega_j\|}$$

where n denotes the total number of points, ω_i refers to the centroid of cluster i , and \overline{CP}_i means the average distance between ω_i and all points in cluster i . A lower DBI means better clustering quality.

Silhouette Coefficient (SC) provides a measure of how similar the point is to the assigned cluster as compared to other clusters. This is computed by calculating the silhouette value for each point, and then averaging the result across the entire dataset. For each point, its silhouette is defined as:

$$SC = \frac{b - a}{\max(a, b)}$$

where a is the mean intra-cluster distance and b is the mean inter-cluster distance to the closest cluster. The SC index ranges from -1 to 1, and the higher SC, the better clustering accuracy.

Second summary reply

Quote from revision summary

Important concepts in cluster analysis are misused. Commonly, clustering quality is evaluated by accuracy metrics (such as ARI, ACC and NMI), rather than the number of identified clusters (mentioned by the reviewer). Different parameter settings do produce different results, and a comprehensive evaluation should consider both recall and precision of all clusters. Therefore, the effectiveness and robustness of the algorithm cannot be denied just because the number of identified clusters differs from that of the true clusters (i.e., inseparable and over-clustered). Actually, the representative scRNA-seq clustering baselines, e.g., Seurat, SC3, monocle3 and SNN-based methods, also cannot guarantee a fixed number of output clusters as the parameters vary.

My take on the issue

Again, I agree with the reviewer. Clinicians won't bother that typically ARI, ACC and NMI are used to evaluate cluster quality but might still lose the trust in the method if small parameter changes lead to very different numbers of clusters. It is of course true that also state-of-the-art approaches suffer from this problem. But this does not mean that the problem doesn't exist but simply that it is currently unsolved.

Thank you. We agree with your comments. **For the methods that dispenses with inputting the number of clusters, they have no control over how many clusters they output and the number varies with their parameter settings.** It is an unsolved problem suffered by existing clustering methods, including our CDC. Actually, **the importance of the number of clusters is different for distinct clustering tasks.** In terms of the identification of cell types or other classification scenarios, the number of clusters matters. While in the field of data mining, e.g., hotspot detection of trajectory points, the number of clusters is not the most important.

Although we can't provide a technique-mature solution to this problem at current stage, **here we briefly discuss some possible ideas for determining appropriate parameter settings and the number of clusters.** In addition to the adaptive methods proposed in the paper, we consider to utilize internal evaluation to measure the clustering quality (see our response for question 1). This strategy can mitigate and ameliorate this problem when users have no prior knowledge. Besides, **a wizard can be added in the toolkit we developed, which may provide an effective approach to derive prior knowledge from the end user through interactions,** e.g., the possible value range of cluster number or cell types to be identified (which may provide domain contexts to imply the potential number of clusters). Meanwhile, a multi-stage clustering workflow can be used to process the initial clustering result of CDC. It aims to approach the input number of clusters by splitting the over-large clusters or merging small-size and adjacent clusters.

Third summary reply

Quote from revision summary

Biased comments. Reviewer #4's negative comments on the approach of dimension reduction using tSNE/UMAP is empirical and subjective, which didn't reflect the facts of our experimental results. Through solid comparative experiments on different datatypes and multiple applications, we have demonstrated the validity of our algorithm. It is able to obtain the highest accuracies on most experimental datasets, commonly in the recommended range of parameters, and achieve the most stable outputs with the default settings. Since CDC can handle the potential weak connectivity in the raw data and embedded data using dimension reduction method, we did not meet the "crowding problem" mentioned by Reviewer #4. We insist that the experimental results are authentic, reliable and reproducible.

My take on the issue

This reply is actually a bit offensive. It is based on Reviewer 4's sentence "My comments may be biased based on my experience [...]", which, however, is simply a polite phrase to make the critique sound less harsh. The critical view on using t-SNE or UMAP prior to clustering scRNA-seq data is rooted in two extremely convincing recently released preprints (Chari et al, <https://doi.org/10.1101/2021.08.25.457696>; Yang et al, <https://doi.org/10.48550/arXiv.2110.02573>), which are in fact referenced by the reviewer. The authors do not show that the concerns raised in these papers do not apply to their approach. For

instance, for the preprint by Chari et al, this could have been done by computing an elephant-shaped embedding with PICASSO and then showing that UMAP+CDC produces better clusterings than PICASSO+CDC.

We are sorry for the impolite reply. We agree the points that dimension reduction would lead to distance distortion and crowding problems (we have discussed “crowding problem” in Discussion). Nevertheless, according to our experiments, we find that CDC equipped with UMAP can obtain high and robust clustering results and even outperforms state-of-the-art baselines. That is because UMAP is able to preserve the intra-cluster compactness and separation between clusters simultaneously. This ability makes CDC more applicable and benefits clustering task. According to your suggestion, we have compared Picasso with t-SNE and UMAP on four mouse retina scRNA-seq datasets (Heng et al., 2019), which also demonstrates the separability of UMAP embedding and the effectiveness of CDC-UMAP. Furthermore, we are now developing a new dimension reduction method to address the aforementioned problems as well as time efficiency issue, and the preliminary experiments verify its effectiveness.

R-Fig. 2. 2D embeddings of four mouse retina scRNA-seq datasets using three dimension reduction techniques.

The embedding results of the three dimension reduction techniques are illustrated in R-Fig. 2. We adopt **KL divergence to assess whether embeddings capture important relative relationships between cells**, and use **the separability metric in Linear Discriminant Analysis (LDA) to evaluate the discrimination between different cell types**. The elephant-shaped embedding with Picasso and t-SNE can quantitatively represent the inferred relationships better than the respective UMAP embeddings (R-Fig. 3a). This result is consistent with the conclusion of the preprint (Chari et al., 2021). Nonetheless, **UMAP embeddings can keep higher separability of different cell types than Picasso and t-SNE** (R-Fig. 3b). **For clustering tasks, keeping the distinct separability between clusters is more important than preserving the relative relationships of data**. Consequently, CDC produces high and similar clustering quality on t-SNE and UMAP embeddings, which are significantly better than that of the elephant-shaped embeddings with Picasso (R-Fig. 3c). Considering the clustering accuracy and time efficiency (R-Fig. 3d) of the three techniques, we still believe **UMAP is a suitable preprocessing method for subsequent CDC clustering in actual application**.

R-Fig. 3. Comparison of three dimension reduction techniques. (a) KL divergence of the results of Picasso, t-SNE and UMAP. **(b)** Separability of the results of Picasso, t-SNE and UMAP. **(c)** ARI scores of CDC on the embedding results of Picasso, t-SNE and UMAP. **(d)** Runtimes of Picasso, t-SNE and UMAP under different number of cells.

As discussed in the two released preprints (Yang et al., 2021, Chari et al., 2021), existing dimension reduction methods tend to cause extensive distortions incurred on the global and local properties of data, thereby destroying the original shapes and relationships of patterns. **To tackle this problem, we currently devote ourselves to studying a new dimension reduction method**. This method

adopts **Fast Embedding and Anchor Projection (FEAP)**. Specifically, we propose a uniform sampling technique to select anchors from the raw data points while keeping the data distribution. In order to cope with the manifold structures, we utilize path-based distance to model the high-dimensional probabilities, and use a logarithm function to calculate low-dimensional probabilities for a more efficient learning process. Non-anchors would be fast embedded into the anchor space finally.

We have conducted a preliminary validation by comparing the performance of our method with t-SNE and UMAP on synthetic and UCI datasets. In general, **FEAP tend to maintain the relative relationships between the clusters and keep the original cluster shapes** (R-Fig. 4a). In terms of the time efficiency, **FEAP outperforms t-SNE and UMAP** (R-Fig. 4b), which benefits from the anchor sampling and stochastic gradient descent. We will continue to refine our dimension reduction method and we believe it will definitely generate sparks with CDC.

R-Fig. 4. Performance comparison of three dimension reduction methods. (a) 2D embeddings of t-SNE, UMAP and FEAP on three synthetic datasets. (b) Runtimes of t-SNE, UMAP and FEAP on synthetic and UCI datasets.

KL Divergence

We calculate the high-dimensional and low-dimensional probabilities as:

$$p_{j|i} = \frac{\exp(-\|x_i - x_j\|^2/2)}{\sum_{k \neq i} \exp(-\|x_i - x_k\|^2/2)}$$
$$q_{j|i} = \frac{\exp(-\|y_i - y_j\|^2/2)}{\sum_{k \neq i} \exp(-\|y_i - y_k\|^2/2)}$$

where x_i and y_i denote the high-dimensional and low-dimensional coordinates respectively, and $p_{j|i}$ and $q_{j|i}$ refer to the respective probabilities. The KL divergence is defined as:

$$\text{KL} = \sum_i \sum_j p_{j|i} \log \frac{p_{j|i}}{q_{j|i}}$$

Separability

Linear Discriminant Analysis (LDA) defines the separability as the ratio of the scatter within the clusters to inter-cluster scatter:

$$S = \frac{\sum_{i=1}^k |C_i| (\omega_i - \omega)^T (\omega_i - \omega)}{\sum_{i=1}^k \sum_{x \in C_i} (\omega_i - x)^T (\omega_i - x)}$$

where C_i denotes the i th cluster and k represents the number of clusters. ω and ω_i refer to the centroid of all data points and cluster C_i respectively.

References:

- [1] Heng, J. et al. Hypoxia tolerance in the Norrin-deficient retina and the chronically hypoxic brain studied at single-cell resolution. *Proc. Natl. Acad. Sci. U. S. A.* **116**, 9103-9114 (2019).
- [2] Chari, T., Banerjee, J. & Pachter, L. The Specious Art of Single-Cell Genomics. Preprint at <https://www.biorxiv.org/content/10.1101/2021.08.25.457696v3.full> (2021).
- [3] Yang, Z., Chen, Y. & Corander, J. T-SNE Is Not Optimized to Reveal Clusters in Data. Preprint at <https://arxiv.org/abs/2110.02573> (2021).